# Embodied Agent Interface: Benchmarking LLMs for Embodied Decision Making

**Manling Li**[1,2*], **Shiyu Zhao**[1*], **Qineng Wang**[1,2*], **Kangrui Wang**[1,2*], **Yu Zhou**[1*],
**Sanjana Srivastava**[1], **Cem Gokmen**[1], **Tony Lee**[1], **Li Erran Li**[3], **Ruohan Zhang**[1], **Weiyu Liu**[1],
**Percy Liang**[1], **Li Fei-Fei**[1], **Jiayuan Mao**[4], **Jiajun Wu**[1]

[1]Stanford University  [2]Northwestern University  [3]Amazon  [4]MIT

embodied-agent-interface.github.io

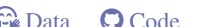 Data  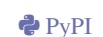 Code  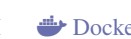 PyPI  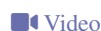 Docker  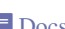 Video  Docs

## Abstract

We aim to evaluate Large Language Models (LLMs) for embodied decision making. While a significant body of work has been leveraging LLMs for decision making in embodied environments, we still lack a systematic understanding of their performance because they are usually applied in different domains, for different purposes, and built based on different inputs and outputs. Furthermore, existing evaluations tend to rely solely on a final success rate, making it difficult to pinpoint what ability is missing in LLMs and where the problem lies, which in turn blocks embodied agents from leveraging LLMs effectively and selectively. To address these limitations, we propose a generalized interface (EMBODIED AGENT INTERFACE) that supports the formalization of various types of tasks and input-output specifications of LLM-based modules. Specifically, it allows us to unify 1) a broad set of embodied decision-making tasks involving both state and temporally extended goals, 2) four commonly-used LLM-based modules for decision making: goal interpretation, subgoal decomposition, action sequencing, and transition modeling, and 3) a collection of fine-grained metrics that break down evaluation into error types, such as hallucination errors, affordance errors, and various types of planning errors. Overall, our benchmark offers a comprehensive assessment of LLMs' performance for different subtasks, pinpointing the strengths and weaknesses in LLM-powered embodied AI systems and providing insights into the effective and selective use of LLMs in embodied decision making.

## 1 Introduction

Large Language Models (LLMs) have emerged as powerful tools for building embodied decision-making agents capable of following human instructions (such as "*cleaning the refrigerator*", "*polishing furniture*") and achieving the specified goals through a sequence of actions in various digital and physical environments [1–3]. Despite many reports of their success, our understanding of LLMs' full capabilities and limitations in embodied decision-making remains limited. Existing evaluation methods fall short of providing comprehensive insights due to three key limitations: the lack of standardization in 1) embodied decision-making tasks, 2) modules that an LLM can interface with or be implemented for, and 3) fine-grained evaluation metrics beyond a single success rate. In this paper, we propose the EMBODIED AGENT INTERFACE to address these challenges.

(1) **Standardization of goal specifications**: We want embodied agents to achieve goals. However, the specification of goals and the criteria for agents' success evaluation vary significantly across different domains, even for similar tasks. For example, BEHAVIOR [4] focuses on achieving a state that satisfies certain *state goals* (e.g., "*not_stained*(fridge)" in Figure 1), while VirtualHome [5] uses

---
*Equal contribution.

38th Conference on Neural Information Processing Systems (NeurIPS 2024) Track on Datasets and Benchmarks.

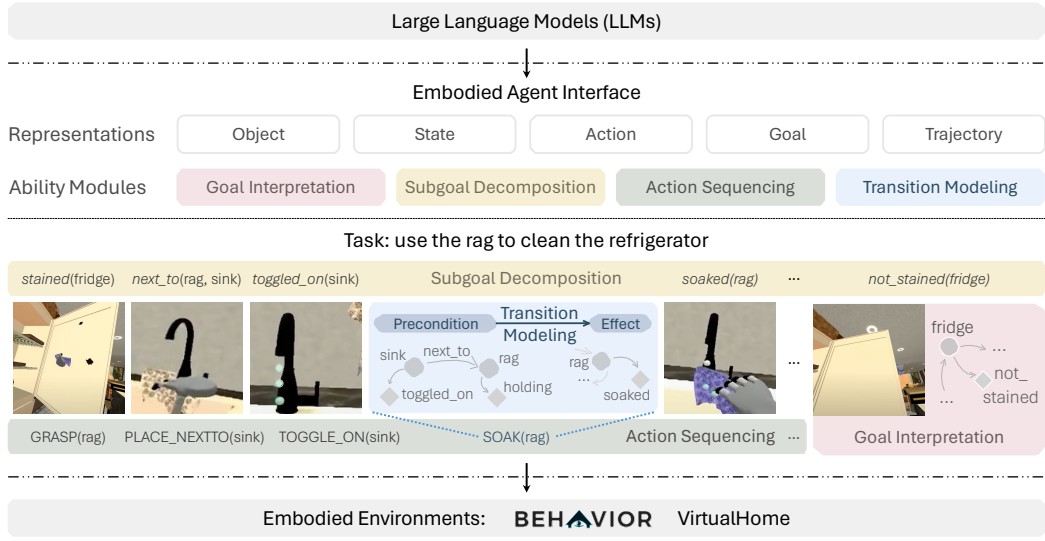

Figure 1: EMBODIED AGENT INTERFACE unifies a broad set of tasks involving both state and temporally extended goals and four LLM-based modules for decision-making.

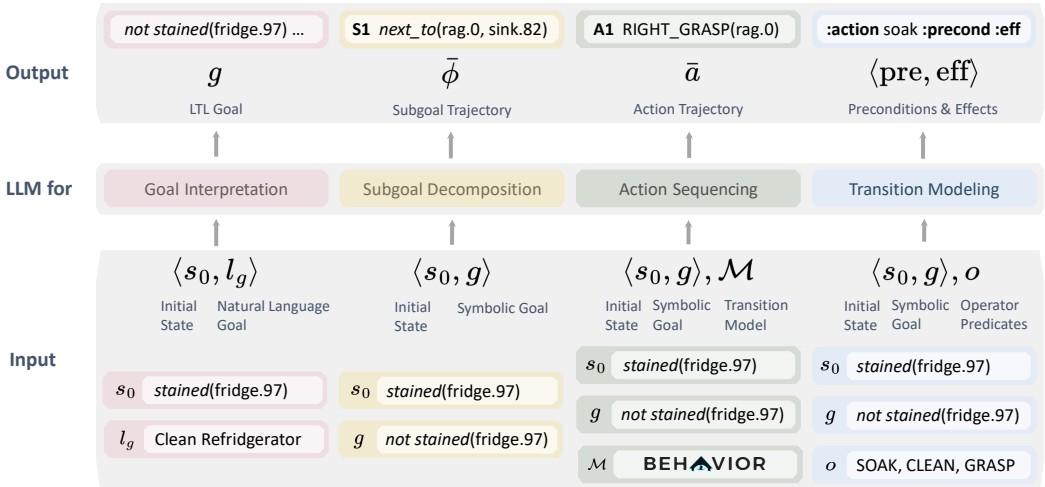

Figure 2: The input and output formulation of four ability modules.

temporally extended goals by imposing temporal order constraints on actions. We include an extended discussion in Appendix C.1. Our EMBODIED AGENT INTERFACE implements a general object-centric state and action representation, where object states, relations, and actions are represented in abstract language terms (see Figure 1). Our innovation is to describe goals as linear temporal logic (LTL) formulas, which define task-success criteria over trajectories. LTL affords the specification of both state-based and temporally extended goals and allows for alternative goal interpretations.

(2) **Standardization of modules and interfaces**: Existing LLM-based embodied agent frameworks often make different assumptions based on the availability of additional knowledge and external modules. For instance, Code as Policies [6] and SayCan [2] utilize LLMs for action sequencing given a given set of primitive skills, while LLM+P [7] uses LLMs for goal interpretation and generates plans using PDDL planners with given domain definitions; Ada [8] leverages LLMs to generate high-level planning domain definitions in PDDL and uses a low-level planner to generate control commands. Consequently, they have defined different input-output specifications for the LLM module, making comparisons and evaluations challenging. In EMBODIED AGENT INTERFACE, built on top of our object-centric and LTL-based task specification, we formalize four critical *ability modules* in LLM-based embodied decision making, as illustrated in Figure 1: *Goal Interpretation*, *Subgoal Decomposition*, *Action Sequencing*, and *Transition Modeling*. We formalize the input-output

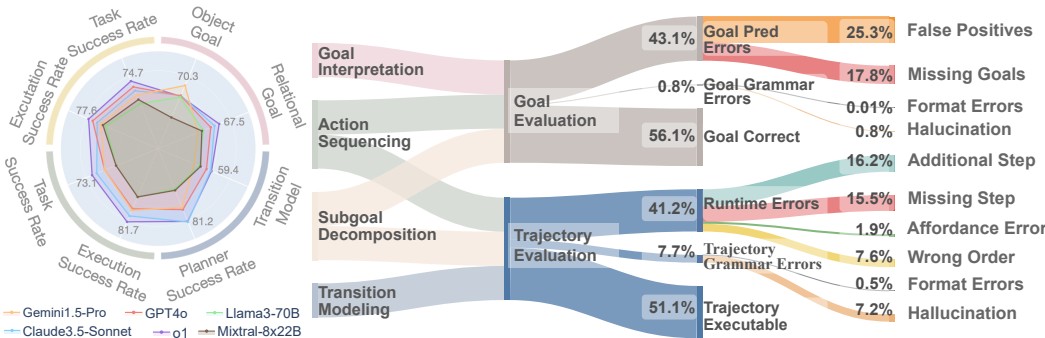

Figure 3: EMBODIED AGENT INTERFACE supports a collection of fine-grained metrics and provides automatic toolkits for error analysis and benchmarking different LLMs on various embodied decision-making tasks.

specifications that LLMs can use to interface with other modules in the environment. This modular interface automatically enables the integration of different LLM-based and external modules. Figure 2 shows the input and output formulation of four ability modules. Taking *Subgoal Decomposition* as an example, this module takes initial states (e.g., a fridge is stained initially) and a task goal (e.g., clean fridge), and asks LLMs to generate a subgoal trajectory (e.g., first the cloth is soaked, then it is held by the agent, then the agent is next to the fridge, and in the end, the fridge is clean). Formal definitions and notations can be found in Table 1.

(3) **Standardization of fine-grained evaluation metrics with broad coverage**: Current evaluations of LLMs for embodied decision-making have been overly simplified, usually focusing on the success rate of a single task. The recent work LOTA-Bench [9] aims to break down the evaluation but is limited to generating action sequences and does not support analysis of fine-grained planning errors. Our EMBODIED AGENT INTERFACE, leveraging object-centric and factorized representations of states and actions, implements a collection of fine-grained evaluation metrics, designed to automatically locate different types of errors such as hallucination errors, different types of planning errors (e.g., object affordance errors, wrong action orders). Figure 3 illustrates different types of errors made by GPT-4o on four different ability modules across two simulators. Specifically, we evaluate two aspects of each module: trajectory evaluation, which checks if the generated plan can be executed in simulators, and goal evaluation, which ensures the plan achieves correct outcomes. Goal evaluation applies to goal interpretation, action sequencing, and subgoal decomposition, while trajectory evaluation applies to action sequencing, subgoal decomposition, and transition modeling.

We implement EMBODIED AGENT INTERFACE on two embodied decision-making benchmarks: BEHAVIOR [4] and VirtualHome [5], and evaluated 18 different LLMs. Figure 3 visualizes the performance of 5 representative LLMs on different tasks in Behavior. Our key findings include:

- Most LLMs struggle to faithfully translate natural language instructions into grounded states (objects, object states, and relations) in the environment. They sometimes predict intermediate subgoals as part of the final goals, e.g., predicting the state *open*(freezer) for task "*drinking water*".
- Reasoning ability is a crucial aspect that LLMs should improve. Trajectory feasibility errors are common (45.2%), with a large portion of missing step (19.5%) and additional step (14.2%) errors, often due to overlooking preconditions. For instance, LLMs may ignore the agent's *sitting* or *lying* state and fail to include a *standup* action before executing other actions. They sometimes also fail to understand the need to *open* a *closed* object before *fetching* items from inside. Additional step errors frequently occur when LLMs output actions for previously achieved goals.
- Trajectory evaluation performance decreases as the trajectory sequence length increases; goal evaluation performance, which refers to evaluating if a plan can achieve task goals when executed, decreases when the environment becomes more complex, involving a larger variety of object and state features.
- LLM errors include not only hallucinations of nonexistent objects and actions but also a heavy reporting bias. They often ignore commonsense preconditions that are elided in the language. For example, "put the turkey on the table" should be interpreted as "put the turkey on a plate, and place the plate on the table."
- Subgoal decomposition, though designed to simplify planning, is as complex as action sequencing in abstract spaces, as LLMs must declaratively break goals into feasible steps.

Table 1: Summary of notations used in EMBODIED AGENT INTERFACE.

| | Notation | Symbol | Description |
|---|---|---|---|
| **Environment Representations** | Object | $u \in \mathcal{U}$ | An object, which has relational features $f$ |
| | State | $s = \langle \mathcal{U}, \mathcal{F} \rangle \in \mathcal{S}$ | A tuple of the universe of objects and relational features |
| | Action | $a = \langle name, args \rangle \in \mathcal{A}$ | A tuple of the action name and arguments |
| | Operator | $o = \langle name, vars \rangle \in \mathcal{O}$ | An action schema: a tuple of the name and a list of parameters. Each $o$ can be instantiated into an action $a$ |
| | Transition Model | $\mathcal{M} : \mathcal{S} \times \mathcal{A} \to \mathcal{S}$ | The deterministic transition function of the environment |
| | Natural Language Goal | $l_g$ | A sentence in English |
| | LTL Goal | $g$ | An LTL formula. Here, we only consider formulas containing a sequence of action items and a conjunction of propositions (for the final state): $g = a_1$ **then** $\ldots$ **then** $a_k$ **then** $(p_1 \wedge \ldots \wedge p_\ell)$. |
| | Action Trajectory | $\bar{a} = \{a_i\}_{i=1}^n$ | A sequence of $n$ actions |
| | Subgoal Trajectory | $\bar{\phi} = \{\phi_i\}_{i=1}^m$ | A sequence of LTL subgoals $\phi_i$ connected by "**then**" |
| | State-action Trajectory | $\bar{t} = \langle \{s_i\}_{i=0}^n, \{a_i\}_{i=1}^n \rangle$ | A sequence of state-action pairs. $\forall t. s_{t+1} = \mathcal{M}(s_t, a_t)$ |
| | Task | $\langle s_0, g, l_g \rangle$ | A tuple of the initial state and the LTL/Natural Language goals |
| **Abilities** | Goal Interpretation | $\mathcal{G} : \langle s_0, l_g \rangle \to g$ | Initial State & Natural Language Goal → LTL Goal |
| | Subgoal Decomposition | $\Phi : \langle s_0, g \rangle \to \bar{\phi}$ | Initial State & Goal → Subgoal Trajectory |
| | Action Sequencing | $\mathcal{Q} : \langle s_0, g \rangle, \mathcal{M} \to \bar{a}$ | Initial State & Goal & Transition Model → Action Trajectory |
| | Transition Modeling | $\mathcal{T} : \langle s_0, g \rangle, o \to \langle pre, eff \rangle$ | Initial State & Goal & Operator → Preconditions & Effects |

- We further provide quantitative analysis for the robustness of the modules through sensitivity analysis, pipeline-based versus modularized comparison, and replanning. These analyses aim to identify potential ways to integrate LLM-based and external modules.
- o1-preview significantly outperforms others, especially on the BEHAVIOR simulator (74.9% vs. 64.2%). It excels in goal interpretation on VirtualHome, as well as action sequencing, transition modeling, and subgoal decomposition on both BEHAVIOR and VirtualHome. Claude-3.5 Sonnet is strong in goal interpretation on BEHAVIOR and transition modeling on VirtualHome, while Mistral Large performs well in action sequencing on VirtualHome.

## 2 Embodied Agent Interface Based on LTL

Table 1 summarizes our EMBODIED AGENT INTERFACE. First, we define an **embodied decision-making problem representation** $\langle \mathcal{U}, \mathcal{S}, \mathcal{A}, g, \phi, \bar{a} \rangle$, which is a language-based, object-centric abstraction for embodied agent environments with *objects* ($o \in \mathcal{U}$), *states* ($s \in \mathcal{S}$), *actions* ($a \in \mathcal{A}$), *goal g*, *subgoal* $\phi$, and trajectories $\bar{a}$. Second, we formally define four **ability modules** $\langle \mathcal{G}, \Phi, \mathcal{Q}, \mathcal{T} \rangle$, including their standardized input-output specifications. They are fundamental and commonly-used modules that LLMs can be implemented for and interface with the *goal interpretation* module $\mathcal{G}$, the *action sequencing* module $\mathcal{Q}$, the *subgoal decomposition* module $\Phi$, and the *transition modeling* module $\mathcal{T}$. In this paper, we focus on object-centric modeling: states are described as relational features among entities in the environment, actions are defined functions that take entity names as inputs and can be executed in the environment, goals and subgoals are defined as linear-temporal logic (LTL) [10] formulas on states and actions. We define each component in detail as follows.

### 2.1 Representation for Objects, States and Actions

In EMBODIED AGENT INTERFACE, a state is represented as a tuple $s = \langle \mathcal{U}, \mathcal{F} \rangle$, where $\mathcal{U}$ is the universe of objects, assumed to be a fixed finite set. $\mathcal{F}$ is a set of relational Boolean features. Each $f \in \mathcal{F}$ is a table where each entry is associated with a tuple of objects $(o_1, \cdots, o_k)$. Each entry has the value of the feature in the state, and $k$ is the arity of the feature. Actions can be viewed as primitive functions that take objects as inputs, denoted as $\langle name, args \rangle$. Throughout the paper, we focus on tasks where states and actions are described in abstract language forms, including object states (e.g., *is-open*(cabinet1)), relations (e.g., *is-on*(rag0, window3)), and actions (e.g., *soak*(rag0)).

### 2.2 Representation for Goals, Subgoals, Action Sequences, and State-Action Trajectories

In EMBODIED AGENT INTERFACE, goals $g$, subgoals $\phi$, and action sequences $\bar{a}$ are modeled as linear temporal logic (LTL) formulas. This is motivated by two critical desiderata. First, we need an expressive and compact language to describe both state-based and temporally extended goals. Second, we need a unified interface between different LLM-based modules. LTL addresses both challenges. At a high level, an LTL formula can describe state constraints (e.g., a subgoal should be achieved),

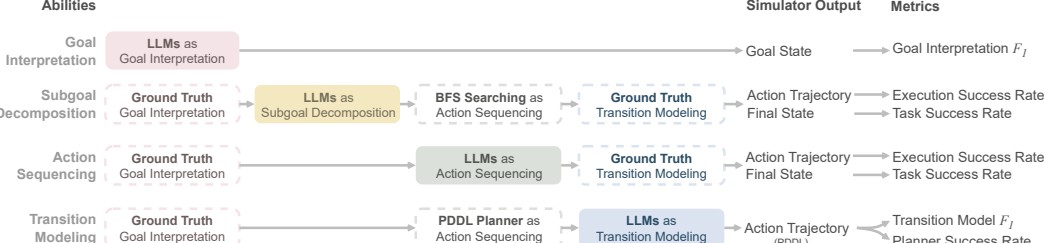

Figure 4: The overview of evaluation pipeline for four abilities. For each ability module, to provide a comprehensive evaluation for it, we isolate this single module to be handled by the LLMs while using existing data or tools for the other modules. Note that the pipeline consists of goal interpretation, action sequencing to achieve the goal, and transition modeling that predicts how each action operate the environment's state. Evaluating subgoal decomposition presents a challenge since it cannot be evaluated directly with no unified annotation strategy. To address this, we employ breadth-first search (BFS) to identify potential action sequences that accomplish each subgoal, allowing us to convert state trajectories into action sequences that can be executed in the simulator (Figure 21 in Appendix). Transition modeling evaluation poses another challenge, we first annotate transition models in PDDL for $F_1$ evaluation followed with a PDDL planner to validate the feasibility of supporting potential plans. We also conduct a pipeline-based vs modularized analysis, detailed in the Appendix G.

action constraints (e.g., a particular action should be executed), and possible temporal orders among them (e.g., all dishes should be cleaned before we cook). By combining temporal connectives (such as "eventually") and propositional logic connectives (such as "or"), we can also flexibly describe alternative goals or trajectories. As a byproduct, using a single description language for all inputs and outputs enables us to design a unified metric to measure accuracy, which we detail in Appendix C.1.

In EMBODIED AGENT INTERFACE, we use a fragment of the full linear temporal logic (LTL) formalism on finite trajectories. We allow two types of atomic propositions: state propositions (object properties and relations) and action propositions. Our LTL language contains Boolean conjunction $\wedge$, disjunction $\vee$, negation $\neg$, implication $\Rightarrow$, first-order logic quantifiers $\forall, \exists, \exists^{=n}$ (the equal quantifier: there are exactly $n$ objects satisfying a condition), and the temporal connective **then**.

An LTL formula is a trajectory classifier semantically: the function $eval(\phi, \bar{t})$ evaluates an LTL formula $\phi$ on a state-action sequence $\bar{t}$. We say that the state-action sequence satisfies $\phi$ if $eval(\phi, \bar{t}) = true$ (i.e., the goal $\phi$ is satisfied). For state formulas $\phi$ (formulas without **then**), we define $eval(\phi, \bar{t}) = \exists t.\phi(s_t)$ ("eventually" the goal is satisfied). For formulas connected by **then**, $eval(\phi_1 \textbf{ then } \phi_2, \bar{t}) = \exists k.\phi_1(\bar{t}_{\leq k}) \wedge \phi_2(\bar{t}_{>k})$ ($\phi_2$ is achieved after $\phi_1$), where $\bar{t}_{\leq k}$ and $\bar{t}_{>k}$ denote prefixes and suffixes. Currently, we have not implemented other temporal connectives such as "globally" and "until" but our overall framework can be extended to them. An LTL formula example of a subgoal plan for task *browse Internet* is: "*ontop*(character, chair) **then** *holds_rh*(character, mouse) $\wedge$ *holds_lh*(character, keyboard) **then** *facing*(character, computer)". We include LTL details in Appendix C.3.

## 2.3 Ability Module 1: Goal Interpretation $\mathcal{G} : \langle s_0, l_g \rangle \to g$

**Input-Output Specification.** The *goal interpretation* module takes the state $s_0$ and a natural language instruction $l_g$ as input, and generates an LTL goal $\hat{g}$, as a formal goal specification which a symbolic planner can conceivably take as input. In this paper, we only generate simple LTL goals formed by an ordered action sequence and a conjunction of propositions to be satisfied in the final state.

**Evaluation Metric.** An LTL goal can be evaluated by directly comparing it with the ground truth goal $g$. While we have restricted generated $\hat{g}$ to be simple LTL goals, we do not require the ground truth goal $g$ to be simple. Therefore, we additionally define $\mathcal{G}$ that takes the object universe $\mathcal{U}$ as input to translate $g$ to a set of simple LTL goals $g_0, g_1, \ldots, g_k$ where all $g_i$'s entail $g$. We describe our implementation in the Appendix. Given two simple LTL goals $g_i$ and $\hat{g}$, the accuracy of $\hat{g}$ can be computed as an $F_1$ set-matching score between them. Let $g = a_1 \overset{\text{then}}{\ldots} a_k \textbf{ then } (p_1 \wedge \ldots \wedge p_\ell)$. We define $set(g) = \{\{a_i\}_{i=1}^k\} \cup \{p_i\}_{i=1}^\ell$ (i.e., the action sequence $\{a_i\}$ is treated as a single element). The $F_1$ score between $g$ and $\hat{g}$ is defined as: $F_1(g, \hat{g}) = \max_{g_i \in \mathcal{G}(g, \mathcal{U})} F_1(set(g_i), set(\hat{g}))$.

## 2.4 Ability Module 2: Subgoal Decomposition $\Phi : \langle s_0, g \rangle \to \bar{\phi}$

**Input-Output Specification.** The *subgoal decomposition* module takes the task $\langle s_0, g \rangle$ as input and generates a sequence of subgoals $\bar{\phi} = \{\phi_i\}_{i=1}^k$, where each $\phi_i$ is an LTL formula. The entire

sequence $\bar{\phi}$ can also be represented as a single LTL formula. One may refer to Appendix D.3 for decomposition choice-making.

**Evaluation Metric.** To evaluate the subgoal decomposition module, we use a customized planner to refine it into an action sequence $\bar{a}$. This subgoal-action mapping function $\mathcal{AM}(\bar{\phi}, s_0)$ takes the LTL representation of $\bar{\phi}$ and $s_0$ and generates a state-action sequence $\bar{t}$. We implement this with a breadth-first search. Then, we use the same metrics in *action sequencing* for evaluation: trajectory feasibility and goal satisfaction. Since each $\phi$ can be grounded into different action sequences, we restrict the number of actions per subgoal to generate a finite set of possible action sequences $\bar{a}_i$ satisfying $\phi$. Then, we compute the metrics for each $\bar{a}_i$ and report the maximum score across all $\bar{a}_i$'s as the trajectory feasibility and the goal satisfaction scores for $\phi$.

### 2.5 Ability Module 3: Action Sequencing $\mathcal{Q} : \langle s_0, g \rangle, \mathcal{M} \to \bar{a}$

**Input-Output Specification.** The *action sequencing* module takes the task $\langle s_0, g \rangle$ as input, and the transition model $\mathcal{M}$, and generates an action sequence $\bar{a} = \{a_i\}_{i=1}^n$.

**Evaluation Metric.** We use two evaluation metrics for the action sequencing module. First, the *trajectory feasibility evaluation* focuses on evaluating whether the trajectory is executable (i.e., all actions are feasible). We will execute the trajectory $\bar{a}$ from $s_0$ in the simulator. When infeasible action presents, the execution may stop at an early step and we categorize the execution failure into missing steps, additional steps, wrong temporal order, and affordance errors.

Second, the *goal satisfaction evaluation* evaluates if the goal is satisfied after executing $\bar{a}$. Specifically, we obtain $T = \langle \{s_i\}_{i=0}^m, \{a_i\}_{i=1}^m \rangle$ by executing $\bar{a}$, and directly use the *eval(g, T)* function to check for goal satisfaction. We also evaluate the *partial goal satisfaction evaluation*, which is the percentage of "subgoals" in $g$ that are satisfied in $\bar{a}$. To compute this partial success rate, we again consider all simple LTL goals $g_i$ derived from $g$. Let $g_i = a_1 \overset{\text{then}}{\ldots} a_k$ **then** $(p_1 \wedge \ldots \wedge p_\ell)$. If there is a subsequence in $\bar{a}$ that is the same as $\{a_j\}_{j=1}^k$, we consider the action sequence successfully executed. Next, we evaluate all final state propositions $p_j$ and give models partial credits based on the number of propositions satisfied in $s_m$. Finally, $PartialSucc(\bar{a}, g) = \max_{g_i \in \mathcal{G}(g, \mathcal{U})} PartialSucc(\bar{a}, g_i)$.

### 2.6 Ability Module 4: Transition Modeling $\mathcal{T} : \langle s_0, g \rangle, o \to \langle pre, eff \rangle$

**Input-Output Specification.** The *transition modeling* module takes the task $\langle s_0, g \rangle$ and a set of operator definitions $\{o_i\}$ as input, and generates a PDDL operator definition [11] for each $o_i$. In this module, we aim to create a formal definition of actions in order to generate plans to solve the task. During evaluation, we first extract relevant operator definitions, $\{o_i\}$, based on the ground truth action trajectory $\bar{a}$ associated with each task, with details provided in Appendix C.3. Then, the LLM generates the preconditions and effects $\{\langle pre_i, eff_i \rangle\}$ for all operators $\{o_i\}$.

**Evaluation Metric.** The *transition modeling* module can be evaluated in two ways. First, the *logic matching score* for an operator $o_i$ compares the generated $pre_i$ and $eff_i$ against the ground truth operator definition annotated by human experts. This comparison uses a surface form matching score to produce an $F_1$-based score between two logic formulas. Intuitively, when both the LLM-generated $pre_i$ and ground truth $pre_i^{gt}$ are conjunctions of propositions, the $F_1$ score is computed as the set matching score between the sets of propositions. More complex logic formulas (e.g., $\forall x. \phi(x)$) are evaluated recursively, as detailed in Appendix C.3. The evaluation of effects is performed similarly.

Furthermore, the *planning success rate* assesses whether the preconditions and effects of different operators enable a viable plan. This is computed by running an external PDDL planner [12] based on generated operator definitions to achieve $g$ from the initial state $s_0$. For simplicity, we only state goals in $g$ (and ignore action subgoals). The planning success rate is 1 if the planner finds a plan.

## 3 Dataset Annotations and Benchmark Implementations

**Annotations.** Focusing on complex long-horizon tasks, we select BEHAVIOR (B) and VirtualHome (V) as our evaluation simulators based on their task length and scene complexity. We include a comparison of different simulators and detailed selection considerations in Appendix M.1. Table 2 shows our annotations. Apart from the goal and trajectory annotations, we introduce the Goal Action annotation to reflect necessary actions that do not have post effects, such as the goal action *touch* in the task "*pet the cat*", as detailed in Appendix M.3. In the subset of VirtualHome tasks we work on, 80.7% task categories include instructions with action steps longer than 10, and 33% of the instructions have step lengths of more than 10.

We select BEHAVIOR as another simulator for our evaluation due to its task complexity. BEHAVIOR BDDL goals may contain quantifiers, such as `(forpairs (?jar ?apple) (inside ?apple ?jar))`, which need to be translated into grounded goals of only atomic propositions, e.g., `and ((inside apple_1 jar_1) (inside apple_2 jar_2))`. There can be different grounded goals that satisfy the same BDDL goal, such as `((inside apple_2 jar_1) (inside apple_1 jar_2))`. We call them goal options. In general, one BDDL goal corresponds to a number of goal options. The average number of grounded goals for each task is 6.7, and there are 4, 164.4 goal options for each task on average. We show data distributions of goal options and other statistics in Appendix M.2.

Table 2: Simulator dataset statistics. New annotations collected in this paper are highlighted in color.

| | VirtualHome | BEHAVIOR |
|---|---|---|
| #task name | 26 | 100 |
| #task instruction | 338 | 100 |
| #goal | 801 | 673 |
|   - #state | 340 | 153 |
|   - #relation | 299 | 520 |
|   - #action | 162 | - |
| #trajectory | 338 | 100 |
|   - #step | 2960 | 1460 |
|   - avg. step | 8.76 | 14.6 |
| #transition model | 33 | 30 |
|   - #precondition | 99 | 84 |
|   - #effect | 57 | 51 |

**Implementation on simulators.** As BEHAVIOR does not have an action transition model layer, we implemented a symbolic simulator with an action transition model layer. Our implementation, EvalGibson, offers 30 actions that agents can use to change the states of objects. Implementation details are in Appendix N.1. We also revise the VirtualHome simulator to support accurate evaluation, as detailed in Appendix N.2. Evaluation settings for each large model are detailed in Appendix O.

## 4 Results

We evaluate 18 open-weight and proprietary LLMs on four embodied agent ability modules across two benchmark simulators: BEHAVIOR and VirtualHome. Table 3 gives an overview. Table 4, Table 5, Table 6, and Table 7 break down the analysis of four representative LLMs on four ability modules. Figure 5 shows examples of different types of error. We start with the overall analysis.

**Model Comparison.** Shown in Figure 3, the top performing models overall are o1-preview, Claude-3.5 Sonnet and Gemini 1.5 Pro, with o1-preview leading in all aspects except **object states** and Gemini 1.5 Pro leading in its **object state reasoning** ability. Among all open-weight models, the best performing models are Llama-3-70B and Mistral-Large-2402, while there is still a performance gap with commercial models.

**Ability Comparison.** o1-preview shows a clear advantage over other models, particularly on the BEHAVIOR simulator, where it achieves 74.9% compared to 64.2%. It leads in several areas, including goal interpretation on VirtualHome and both action sequencing and transition modeling on BEHAVIOR. Moreover, it outperforms in subgoal decomposition across both BEHAVIOR and VirtualHome simulators. In contrast, Claude-3.5 Sonnet shines in goal interpretation on BEHAVIOR and transition modeling on VirtualHome, while Mistral Large stands out in action sequencing on VirtualHome. Mixtral-8x22B shines in transition modeling among open-weight LLMs, and Llama-3-70B Instruct in goal interpretation.

We also observe a performance gap between different simulators. Models achieve significantly lower trajectory feasibility scores on BEHAVIOR compared to VirtualHome, but achieve higher scores on goal interpretation. This is because BEHAVIOR tasks have a much longer horizon (avg 14.6 steps) while VirtualHome goals have a larger state space to search (such as "*work*"), as detailed in Appendix L.2. It shows the inverse correlation between trajectory evaluation performance and sequence length, as well as between goal evaluation performance and environment complexity. We further perform a systematic analysis to discover the cofactors for the goal success rate, including the number of task goals, particularly node goals, the ground truth action length, and the task object length, with details in Appendix E.5.

**Object States vs Relationship.** Relational goals are generally harder to reason about compared to object-state goals. Spatial relations have a significantly lower recall in the goal interpretation task (Table 4) and a lower goal satisfaction rate (Table 5). Some non-spatial relations (e.g., *hold*) are even more difficult for LLM to predict than spatial relations, as shown in the transition modeling accuracy (Table 6): for example *holding*(toothbrush) should be a precondition for brushing teeth.

Table 3: Results (%) overview. *V*: VirtualHome, *B*: BEHAVIOR. Full results in Appendix E.

| Model | Goal Interpretation $F_1$ | | Action Sequencing Task SR | | Execution SR | | Subgoal Decomposition Task SR | | Execution SR | | Transition Modeling $F_1$ | | Planner SR | | Average Perf. Module SR | |
|---|---|---|---|---|---|---|---|---|---|---|---|---|---|---|---|---|
| | V | B | V | B | V | B | V | B | V | B | V | B | V | B | V | B |
| Claude-3 Haiku | 28.0 | 52.5 | 54.8 | 26.0 | 60.7 | 32.0 | 78.4 | 30.0 | 82.8 | 35.0 | 42.3 | 51.6 | 30.4 | 64.0 | 49.4 | 41.6 |
| Claude-3 Sonnet | 29.4 | 69.4 | 58.0 | 44.0 | 63.3 | 60.0 | 83.1 | 39.0 | 86.4 | 43.0 | 41.2 | 56.2 | 13.2 | 80.0 | 49.4 | 55.1 |
| Claude-3 Opus | 31.4 | 77.0 | 64.6 | 51.0 | 69.5 | 59.0 | 86.7 | 41.0 | 89.9 | 47.0 | 48.8 | 63.4 | 61.8 | 82.0 | 59.5 | 60.4 |
| Claude-3.5 Sonnet | 33.0 | **82.7** | 76.1 | 60.0 | 81.3 | 69.0 | 89.1 | 39.0 | 92.0 | 44.0 | **48.9** | 67.9 | 80.5 | 82.0 | **65.7** | 64.2 |
| Cohere Command R | 36.7 | 36.0 | 44.9 | 16.0 | 44.3 | 19.0 | 71.3 | 15.0 | 78.1 | 25.0 | 11.7 | 24.1 | 51.1 | 41.0 | 46.1 | 24.9 |
| Cohere Command R+ | 22.4 | 51.2 | 54.1 | 27.0 | 65.2 | 35.0 | 77.8 | 25.0 | 83.7 | 37.0 | 30.8 | 49.7 | 37.2 | 59.0 | 47.1 | 39.1 |
| Gemini 1.0 Pro | 23.8 | 60.0 | 45.6 | 27.0 | 56.7 | 32.0 | 70.4 | 24.0 | 84.6 | 33.0 | 41.8 | 45.8 | 11.8 | 16.0 | 41.7 | 35.5 |
| Gemini 1.5 Flash | 26.8 | 74.8 | 69.5 | 40.0 | 75.4 | 52.0 | 89.1 | 34.0 | 94.1 | 42.0 | 45.7 | 53.4 | 46.6 | 66.0 | 57.9 | 52.1 |
| Gemini 1.5 Pro | 36.2 | 79.6 | 76.7 | 42.0 | 83.6 | 54.0 | 87.0 | 31.0 | 91.1 | 37.0 | 34.1 | 45.8 | **91.9** | 39.0 | 65.7 | 48.8 |
| GPT-3.5-turbo | 22.7 | 50.4 | 24.9 | 16.0 | 40.7 | 20.0 | 69.2 | 24.0 | 81.4 | 36.0 | 30.0 | 42.1 | 0.7 | 41.0 | 33.0 | 33.0 |
| GPT-4-turbo | 33.2 | 77.2 | 60.0 | 38.0 | 65.2 | 45.0 | 85.5 | 38.0 | 94.1 | 47.0 | 42.9 | 44.2 | 56.1 | 46.0 | 57.1 | 49.6 |
| GPT-4o | 36.5 | 79.2 | 71.5 | 47.0 | 81.3 | 53.0 | 87.6 | 49.0 | 91.1 | 55.0 | 46.7 | 60.9 | 68.2 | 67.0 | 63.3 | 59.8 |
| Llama 3 8B Instruct | 22.6 | 28.3 | 21.3 | 10.0 | 23.6 | 16.0 | 48.8 | 22.0 | 58.0 | 29.0 | 12.9 | 35.0 | 28.7 | 29.0 | 28.4 | 23.1 |
| Llama 3 70B Instruct | 26.9 | 70.9 | 59.0 | 34.0 | 66.6 | 42.0 | 78.4 | 21.0 | 87.3 | 30.0 | 37.4 | 55.1 | 12.2 | 78.0 | 47.3 | 48.1 |
| Mistral Large | 26.8 | 74.3 | **78.4** | 33.0 | **84.6** | 50.0 | 84.3 | 31.0 | 92.0 | 38.0 | 36.1 | 49.5 | 31.1 | 77.0 | 55.8 | 50.4 |
| Mixtral 8x22B MoE | 26.6 | 54.7 | 63.3 | 30.0 | 67.9 | 40.0 | 80.5 | 28.0 | 90.2 | 33.0 | 42.0 | 52.4 | 37.5 | 55.0 | 52.5 | 41.6 |
| o1-mini | 31.2 | 76.4 | 71.5 | 56.0 | 76.4 | 65.0 | 79.3 | 31.0 | 84.6 | 39.0 | 41.5 | 56.4 | 69.0 | 77.0 | 59.3 | 57.5 |
| o1-preview | **42.7** | 81.6 | 65.2 | **81.0** | 72.5 | **91.0** | **89.4** | **60.0** | 93.2 | **62.0** | 48.0 | **69.5** | 72.4 | **89.0** | 64.4 | **74.9** |

Table 4: Logic form accuracy for *goal interpretation* (%). Full results in Table 9.

| Model | State Goal Precision | | Recall | | $F_1$ | | Relation Goal Precision | | Recall | | $F_1$ | | Action Goal Precision | | Recall | | $F_1$ | | Overall Precision | | Recall | | $F_1$ | |
|---|---|---|---|---|---|---|---|---|---|---|---|---|---|---|---|---|---|---|---|---|---|---|---|---|
| | V | B | V | B | V | B | V | B | V | B | V | B | V | B | V | B | V | B | V | B | V | B | V | B |
| Claude-3.5 Sonnet | 25.3 | 74.0 | **60.9** | 94.8 | 35.8 | 83.1 | 31.1 | **84.4** | **63.8** | 81.3 | 41.8 | **82.9** | 14.0 | - | **98.8** | - | 24.5 | - | 21.7 | **81.1** | **69.6** | 84.4 | 33.0 | **82.7** |
| Gemini 1.5 Pro | **47.2** | **94.0** | 47.5 | 92.8 | **47.3** | **93.4** | 42.0 | 74.4 | 7.2 | 76.7 | 12.4 | 75.6 | **24.1** | - | 81.4 | - | 37.2 | - | **33.6** | 78.8 | 39.3 | 80.4 | 36.2 | 79.6 |
| GPT-4o | 29.0 | 67.1 | 60.0 | 94.8 | 39.1 | 78.6 | 31.5 | 81.1 | 43.6 | 78.5 | 36.6 | 79.8 | 20.5 | - | 85.8 | - | 33.1 | - | 26.4 | 76.5 | 59.1 | 82.2 | 36.5 | 79.2 |
| Llama 3 70B | 23.9 | 69.5 | 61.2 | **95.4** | 34.3 | 80.4 | 22.6 | 70.0 | 37.5 | 73.3 | 28.2 | 71.6 | 11.2 | - | 88.8 | - | 19.8 | - | 17.5 | 64.7 | 58.0 | 78.3 | 26.9 | 70.9 |
| o1-mini | 26.3 | 63.8 | 58.6 | 90.8 | 36.3 | 74.9 | 30.4 | 77.3 | 39.9 | 76.5 | 34.5 | 76.9 | 13.5 | - | 56.8 | - | 21.8 | - | 22.4 | 73.3 | 51.3 | 79.8 | 31.2 | 76.4 |
| o1-preview | 28.2 | 66.8 | 60.3 | 94.8 | 38.5 | 78.4 | **44.9** | 82.9 | 62.4 | **82.7** | **52.2** | 82.8 | 26.0 | - | 81.5 | - | **39.5** | - | 31.8 | 78.1 | 65.4 | **85.4** | **42.7** | 81.6 |

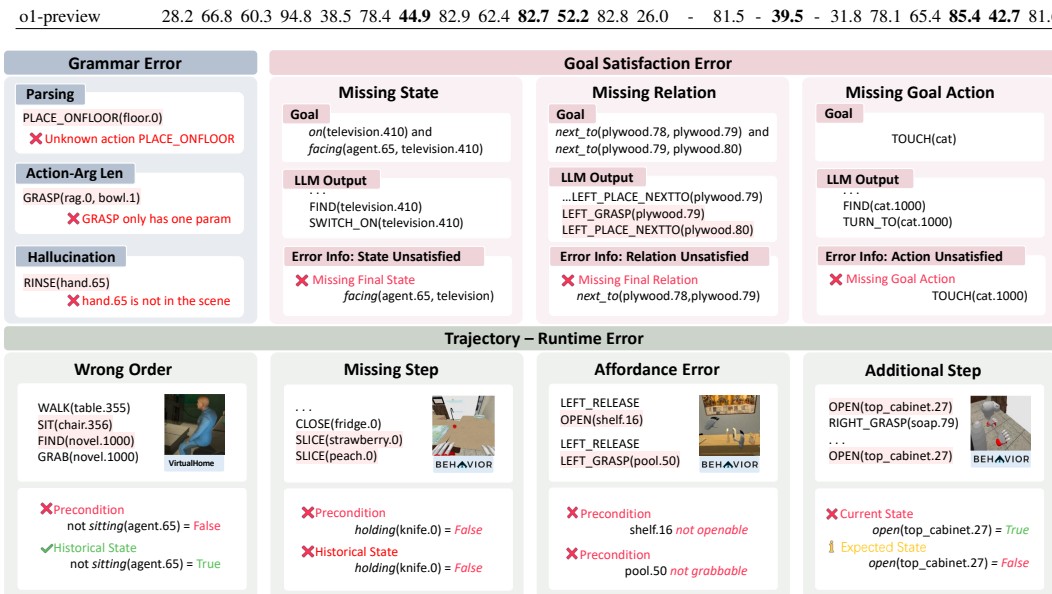

Figure 5: Examples of different types of errors in trajectory feasibility, logic form parsing (e.g., in subgoals decomposition and transition modeling), and goal satisfaction rates.

**Reporting Bias and Imprecise Physical Expressions.** Given the task "*serve a meal*", all LLMs predict the incorrect goal *ontop*(chicken, table) instead of *ontop*(chicken, plate), due to the commonly used natural language expression "*put the chicken on the table*". Also, for the task "*cleaning sneakers*", the goal state *onfloor*(gym_shoe, floor) is missing from all LLM predictions, as chat models ignore the *onfloor* spatial relationship as implicit for conversational language. However, such precise physical relationships are essential for embodied task planning.

Table 5: Goal satisfaction rates (%) for *action sequencing* and *subgoal decomposition*. Full results in Appendix E.2. Behavior does not include action goals.

| Model | Action Sequencing | | | | | | | | Subgoal Decomposition | | | | | | | |
|---|---|---|---|---|---|---|---|---|---|---|---|---|---|---|---|---|
| | State Goal | | Relation Goal | | Action Goal | | Total | | State Goal | | Relation Goal | | Action Goal | | Total | |
| | V | B | V | B | V | B | V | B | V | B | V | B | V | B | V | B |
| Claude-3.5 Sonnet | 87.8 | 63.0 | **83.3** | 62.4 | 60.8 | - | **79.9** | 62.6 | 92.9 | 41.0 | **88.6** | 39.5 | 87.0 | - | 90.1 | 39.9 |
| Claude-3 Opus | 57.2 | 45.0 | 77.8 | 53.0 | 54.7 | - | 62.7 | 50.8 | 92.4 | 43.0 | **88.6** | 41.6 | 83.3 | - | 89.1 | 42.0 |
| Gemini 1.5 Pro | 85.6 | 41.0 | 76.7 | 43.2 | **62.2** | - | 77.2 | 42.6 | 91.2 | 31.0 | 72.5 | 37.1 | 89.5 | - | 83.9 | 35.4 |
| GPT-4o | 87.1 | 49.0 | 76.1 | 45.5 | 56.1 | - | 76.2 | 46.5 | 92.1 | 50.0 | 84.2 | 53.2 | **93.2** | - | 89.4 | 52.3 |
| Llama 3 70B | 61.5 | 31.0 | 68.3 | 45.5 | 42.6 | - | 58.9 | 41.5 | **93.2** | 25.0 | 63.4 | 27.7 | 82.7 | - | 80.0 | 27.0 |
| o1-mini | **88.5** | 64.0 | 72.2 | 66.9 | 57.4 | - | 76.1 | 66.1 | 89.7 | 28.0 | 68.8 | 38.0 | 81.5 | - | 80.3 | 35.3 |
| o1-preview | 80.9 | **89.5** | 65.0 | **84.4** | 46.6 | - | 67.8 | **85.8** | 91.8 | **56.5** | 88.3 | **69.4** | 92.6 | - | **90.6** | **65.9** |

Table 6: Trajectory evaluation results (%) for *action sequencing* and *subgoal decomposition*. Full results in Appendix E.3.

| Model | Goal Evaluation | | | | Trajectory Evaluation | | | | | | | | | | | | | |
|---|---|---|---|---|---|---|---|---|---|---|---|---|---|---|---|---|---|---|
| | Task SR | | Execution SR | | Grammar Error (↓) | | | | | | Runtime Error (↓) | | | | | | | |
| | | | | | Parsing | | Hallucination | | Predicate-Arg Num | | Wrong Order | | Missing Step | | Affordance | | Additional Step | |
| | V | B | V | B | V | B | V | B | V | B | V | B | V | B | V | B | V | B |
| *Action Sequencing* | | | | | | | | | | | | | | | | | | |
| Claude-3.5 Sonnet | **76.7** | 60.0 | **81.3** | 69.0 | **0.0** | **0.0** | 2.0 | 0.0 | 0.3 | 0.0 | 0.3 | 5.0 | 14.4 | 25.0 | 1.6 | **1.0** | **1.3** | 2.0 |
| Claude-3 Opus | 64.9 | 51.0 | 69.5 | 59.0 | **0.0** | **0.0** | 17.0 | 0.0 | 0.3 | 0.0 | 0.3 | 3.0 | 12.8 | 35.0 | 0.3 | 3.0 | 2.3 | 2.0 |
| Gemini 1.5 Pro | **76.7** | 42.0 | 83.6 | 54.0 | **0.0** | **0.0** | **1.3** | 0.0 | 0.7 | 0.0 | 0.3 | 6.0 | 14.1 | 39.0 | **0.0** | **1.0** | 3.0 | 2.0 |
| GPT-4o | 71.5 | 47.0 | **81.3** | 53.0 | 1.0 | **0.0** | 2.0 | 0.0 | 0.7 | 0.0 | 0.3 | 9.0 | 15.1 | 36.0 | **0.0** | **1.0** | 2.3 | **0.0** |
| Llama 3 70B | 59.0 | 34.0 | 66.6 | 42.0 | **0.0** | **0.0** | 14.1 | 2.0 | 8.2 | 0.0 | 2.0 | 15.0 | **9.2** | 38.0 | **0.0** | 3.0 | 6.2 | 6.0 |
| o1-mini | 71.5 | 56.0 | 76.4 | 65.0 | 4.9 | **0.0** | 2.0 | 3.0 | **0.0** | **0.0** | 1.0 | 7.0 | 17.7 | 17.0 | 0.3 | 6.0 | 2.6 | 5.0 |
| o1-preview | 65.2 | **81.0** | 72.5 | **91.0** | 6.6 | **0.0** | 11.5 | **0.0** | **0.0** | **0.0** | **0.0** | **0.0** | 12.1 | **6.0** | 0.3 | 2.0 | 2.0 | 3.0 |
| *Subgoal Decomposition* | | | | | | | | | | | | | | | | | | |
| Claude-3.5 Sonnet | 89.1 | 39.0 | 92.0 | 44.0 | **0.0** | **0.0** | 1.8 | 1.0 | **0.0** | **0.0** | 1.5 | 11.0 | 2.7 | 44.0 | 2.1 | **0.0** | 24.6 | 4.0 |
| Claude-3 Opus | 87.0 | 39.0 | 89.9 | 47.0 | 0.3 | **0.0** | 3.3 | 3.0 | **0.0** | **0.0** | 1.2 | 5.0 | 3.0 | 45.0 | 2.4 | **0.0** | 16.0 | 5.0 |
| Gemini 1.5 Pro | 87.0 | 31.0 | 91.1 | 37.0 | **0.0** | 1.0 | **1.5** | **0.0** | 1.8 | 1.0 | **0.0** | **3.0** | 5.6 | 59.0 | **0.0** | **0.0** | 16.0 | 2.0 |
| GPT-4o | 88.8 | 48.0 | 90.2 | 55.0 | **0.0** | **0.0** | 6.2 | 3.0 | **0.0** | **0.0** | 1.2 | 5.0 | **2.4** | 37.0 | **0.0** | **0.0** | 15.7 | 0.0 |
| Llama 3 70B | 78.4 | 20.0 | 87.3 | 30.0 | **0.0** | 1.0 | 2.4 | 5.0 | 0.9 | 1.0 | 2.4 | 8.0 | 5.3 | 51.0 | 1.8 | 4.0 | 20.4 | 4.0 |
| o1-mini | 79.3 | 31.0 | 84.6 | 39.0 | **0.0** | **0.0** | **1.5** | 3.0 | 0.6 | 3.0 | 0.3 | 7.0 | 8.9 | 46.0 | 4.1 | 2.0 | 21.9 | **1.0** |
| o1-preview | **89.4** | **57.0** | **93.2** | **62.0** | **0.0** | 2.0 | **1.5** | 3.0 | **0.0** | **0.0** | 0.3 | 5.0 | 2.7 | **25.0** | 2.4 | 3.0 | **12.1** | 7.0 |

## 4.1 Ability Module Analysis

**Goal Interpretation.** LLMs generally have difficulties distinguishing intermediate subgoals and final goals. For example, in the VirtualHome task *Drink*, GPT-4o predicts some intermediate states as part of the final goal (e.g., *open*(freezer) and *inside*(water, glass)). Overall, we observe that LLMs tend to translate NL goals word-by-word into their symbolic correspondence, rather than grounding them in the environment state. More analyses are in Appendix E.1.

**Subgoal Decomposition and Action Sequencing on Trajectory Feasibility.** Most errors are runtime errors (rather than syntax errors). We illustrate examples in Figure 5. Overall, LLMs are more likely to make missing-step and additional-step errors than wrong-order or affordance errors. Missing-step errors occur when a precondition is not satisfied before the execution of an action (e.g., fetching an object without opening the box containing it). Additional steps form the most frequent errors, even for the most powerful models—it occurs when a goal has already been achieved but the model still predicts to execute an additional action to achieve it (e.g., opening a box twice). More analysis is in Appendix E.3.

**Subgoal Decomposition and Action Sequencing on Goal Satisfaction Rates.** Shown in Table 5, object goals (such as *toggled_on*) are generally easier to achieve than relational goals (such as *ontop*(agent, chair)). More analysis is provided in Appendix E.2.

**Transition Modeling.** Table 7 shows the overall performance of the logic form accuracy. For a systematic evaluation, we further categorize the tasks into five distinct ability categories requiring the transition modeling for different types of object states and relations (see Appendix F.3). Overall, we reveal significant variations in performance across different models; relational preconditions and effects are generally harder to predict than object-state ones. For instance, the Claude-3 Opus model excelled in object states (63% on VirtualHome), but its performance in spatial relations is weak. Additionally, in tasks that focus on object properties, models generally perform poorly in reasoning about object orientation (e.g., the agent should be facing the TV to watch it). We also provide a sensitivity analysis tool to visualize how different transition modeling errors result in downstream

Table 7: Logic form accuracy ($F_1$) and planner success rate (SR) for *transition modeling* (%). Full results in Appendix E.4.

| Model | Object States | | | | Object Orientation | | | | Object Affordance | | | | Spatial Relations | | | | Non-Spatial Relations | | | |
|---|---|---|---|---|---|---|---|---|---|---|---|---|---|---|---|---|---|---|---|---|
| | $F_1$ | | SR | | $F_1$ | | SR | | $F_1$ | | SR | | $F_1$ | | SR | | $F_1$ | | SR | |
| | V | B | V | B | V | B | V | B | V | B | V | B | V | B | V | B | V | B | V | B |
| Claude-3.5 Sonnet | 60.5 | **78.8** | 67.4 | **86.7** | **95.3** | - | 96.4 | - | 76.6 | - | 67.7 | - | **42.4** | **58.6** | **96.6** | 80.9 | 5.9 | 73.6 | **91.9** | 80.3 |
| Claude-3 Opus | **63.0** | 71.9 | 63.5 | 84.4 | 62.6 | - | 71.4 | - | 75.5 | - | 58.7 | - | 38.7 | 54.6 | 64.8 | 80.9 | 7.0 | 68.8 | 55.4 | 82.0 |
| Gemini 1.5 Pro | 18.8 | 55.9 | **94.4** | 35.6 | 90.9 | - | 89.3 | - | **77.7** | - | **95.8** | - | 38.7 | 35.9 | 89.0 | 40.4 | 7.8 | 52.8 | 83.8 | 39.3 |
| GPT-4o | 54.6 | 71.3 | 71.9 | 68.9 | 52.8 | - | 78.6 | - | 74.9 | - | 63.5 | - | 40.8 | 45.9 | 66.9 | 64.9 | 7.5 | 73.0 | 68.9 | 68.9 |
| Llama-3 70B | 32.5 | 66.3 | 10.1 | 68.9 | 56.6 | - | 3.6 | - | 57.0 | - | 6.6 | - | 27.0 | 47.2 | 15.2 | 77.7 | 3.0 | 58.9 | 18.9 | 85.2 |
| o1-mini | 59.0 | 41.3 | 63.5 | 77.8 | 56.3 | - | 82.1 | - | 58.5 | - | 59.3 | - | 32.5 | 53.1 | 75.9 | 77.7 | 4.5 | 67.5 | 71.6 | 75.4 |
| o1-preview | 58.5 | 78.3 | 69.1 | **86.7** | 78.4 | - | **100.0** | - | 77.5 | - | 67.1 | - | 38.8 | 56.3 | 76.6 | **89.4** | 11.8 | **83.5** | 78.4 | **90.2** |

planning errors (see Appendix F and E.4). We found that LLMs tend to overstate object states in effects while understating them in preconditions. Conversely, they overstate spatial relationships in preconditions and understate them in effects. As a result, in many cases, even if the downstream planner successfully generates a plan, it may not be feasible in the actual environment.

**Implications in Embodied Agent System Design.** We further investigate the potential integration of LLM-based ability modules and their robustness through **sensitive analysis** (Appendix F), **modularized vs pipeline-based** experiments (Appendix G), and **replanning** (Appendix H). We observe that trajectory feasibilities are similar, although with error accumulation from different module compositions, showing the potential of module composition. We have also compared different **prompting strategies** for embodied decision-making tasks, and summarize the best practices in Appendix I.

## 5   Related Work

Recent work in embodied decision making has been using LLMs to perform various tasks, and we include a comprehensive summary in Appendix P, see also Table 8 for a quick summary. LLMs can also be used to combine multiple of the above modules at once via chain-of-thought prompting or pipelined queries, such as goal interpretation with action sequencing [13–32], goal interpretation with subgoal decomposition [2, 27, 33], action sequencing with subgoal decomposition [27, 34, 18, 35], action sequencing with transition modeling [8, 28, 32, 36, 37, 13, 38]. Our work aims to standardize the interface between LLMs and various decision-making modules to support the seamless integration, modular evaluation, and fine-grained metrics, aiming to provide implications on using LLMs in embodied decision making more effectively and selectively. We provide additional related work on agent interfaces [39–43, 18, 44, 42, 45] and simulation benchmarks in Appendix P.

Table 8: Existing work in leveraging LLMs for embodied agents.

| Goal Interpretation | Subgoal Decomposition | Action Sequencing | Transition Modeling |
|---|---|---|---|
| [2, 7, 46, 47, 21, 48, 22–25, 27–32, 13–15, 49–53] | [2, 27, 34, 18, 33, 35, 54, 55] | [6, 8, 35, 56–59, 16, 43, 60, 17, 19, 61, 20, 42, 62, 14, 63–66, 3, 15, 45, 67–69] | [8, 28, 32, 36, 70, 37, 13, 38] |

## 6   Conclusions and Future Work

We propose a systematic evaluation framework EMBODIED AGENT INTERFACE to benchmark LLMs for embodied decision-making. It focuses on 1) standardizing goal specifications using LTL formulas, 2) unifying decision-making tasks through a standard interface and four fundamental ability modules, and 3) providing comprehensive fine-grained evaluation metrics and automatic error identification. We highlight the limitations of current LLMs in interpreting complex goals and different errors in reasoning, further attributing errors to various cofactors, including trajectory length, goal complexity, spatial relation goals, etc.

**Limitations and future work:** Our current evaluation is limited to states, actions, and goals that can be described in abstract language terms, with the input environment abstracted by relational graphs of objects. Future work should extend this to include sensory inputs and actuation outputs, possibly by extending the studied model class to include Vision-Language Models (VLMs), which we discuss further in Appendix K. Other aspects of extension include the integration of memory systems (episodic memory and state memory), geometric reasoning, and navigation.

## Acknowledgments and Disclosure of Funding

This work was in part supported by the Stanford Institute for Human-Centered Artificial Intelligence (HAI), NSF CCRI #2120095, AFOSR YIP FA9550-23-1-0127, ONR MURI N00014-22-1-2740, ONR YIP N00014-24-1-2117, Amazon, and Microsoft.

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

# Appendix

## Table of Contents

# A  Summary of Empirical Findings

1. **Goal Interpretation:**

   - Most LLMs still struggle to faithfully translate natural language instructions into grounded states (objects, object states, and relations) in the environment.
   - A common error is generating intermediate goals instead of final goals, e.g., predicting the state *open*(freezer) for the task "drinking water".
   - Another common error is omitting conversationally uncommon spatial relationship goals. For example, in the task "serving a meal", with ground truth goal condition *ontop*(chicken.0, plate.2) and *ontop*(plate.2, table.1), GPT-4o mistakenly predicts *ontop*(chicken.0, table.1), ignoring the crucial spatial relationship between the chicken, plate, and table.
   - Gemini 1.5 Pro achieves the highest overall goal interpretation performance (F1-score) in both VirtualHome and BEHAVIOR simulators, while Claude-3 Opus has the highest successful ground truth goal retrieval rate (Recall) in both simulators. For example, in the VirtualHome simulator, Gemini 1.5 Pro achieves an F1-score of 82.0%, and Claude-3 Opus achieves a Recall of 89.1%.
   - State-of-the-art proprietary LLMs make few to no grammar errors, while top open-source LLMs like Llama 3 70B Instruct suffer more from format/parsing errors and object/state hallucination. For instance, GPT-4o makes no parsing errors in both simulators, while Llama 3 8B makes parsing errors in 0.6% of cases in VirtualHome and 2.0% in BEHAVIOR.

2. **Action Sequencing:**

   - Reasoning ability is a crucial aspect that LLMs should improve. As shown in Fig 3 in the main paper, trajectory runtime errors are common (41.2%), with a large portion of missing step (15.5%) and additional step (16.2%) errors, often due to overlooking preconditions. For instance, LLMs may ignore the agent's *sitting* or *lying* state and fail to include a *standup* action before executing other actions. They sometimes also fail to understand the need to *open* a *closed* object before *fetching* items from inside. Additional step errors frequently occur when LLMs output actions for previously achieved goals.
   - In BEHAVIOR, o1-preview leads with the highest task success rate (81.0%) and execution success rate (91.0%), followed by o1-mini in second place (56.0%, 65.0%). The best non-o1-series model is GPT-4o (47.0%, 53.0%). Notably and interestingly, in VirtualHome, Mistral Large (73.4%,83.6%) and Gemini 1.5 Pro (73.1%, 83.3%) both outperform o1-preview (71.1%, 78.4%).
   - Better LLMs generally make fewer grammar errors compared to less advanced models. For example, Claude-3 Opus makes no parsing errors in both simulators, while GPT-3.5-turbo makes parsing errors in 4.0% of cases in BEHAVIOR.
   - The most common runtime errors are missing steps and wrong order in both simulators. For instance, in BEHAVIOR, GPT-4o encounters missing step errors in 36.0% of cases and wrong order errors in 9.0% of cases.
   - LLMs perform better in satisfying state goals than relation goals and struggle with complex action goals. For example, in VirtualHome, GPT-4o achieves a state goal success rate of 82.0% but a relation task success rate of 67.8%.
   - Task complexity, including the number of goals, state goals, relation goals, and action sequence length, adversely affects the task success rate. For instance, in BEHAVIOR, the success rate drops from around 60% for tasks with fewer than 5 goals to below 40% for tasks with more than 10 goals.

3. **Subgoal Decomposition:**

   - Subgoal decomposition is not strictly easier than action sequencing in abstract action spaces.
   - o1-preview demonstrates superior performance in both VirtualHome and BEHAVIOR simulators compared to other state-of-the-art (SOTA) LLMs, with success rates of 89.4% and 57.0%, respectively. In VirtualHome, Gemini 1.5 Flash and Claude-3.5 Sonnet also exhibit high performance with success rates of 89.1%.

- SOTA models generally avoid grammar errors but can hallucinate actions and objects. For example, GPT-4o tends to hallucinate the action *POUR* when dealing with the task "make coffee" in VirtualHome, which is not defined in the subgoal decomposition setting.

- The most common runtime errors differ between simulators: additional steps in VirtualHome and missing steps in BEHAVIOR. For instance, in VirtualHome, all LLMs are prone to produce additional step errors, even for SOTA LLMs like GPT-4o and Claude-3 Opus. This is mainly because, in the initial scene state, some of the goals have already been achieved, yet LLMs still prefer to plan the satisfied goals in their output.

- Stronger LLMs like o1-preview show higher accuracy in action task success rates in VirtualHome compared to weaker models like Llama 3 8B. However, achieving state and relation goals in BEHAVIOR is challenging due to more complex task representations and stricter precondition checks. For example, in BEHAVIOR, most state and relation goals are encapsulated within quantifiers, and quantifiers such as "forall" or "forpairs" tend to fail if even a single state or relation goal is not met.

- Overall LLM performance is lower in BEHAVIOR compared to VirtualHome due to complex task representations involving quantifiers like "forall" and "forpairs", which articulate complex temporal and spatial requirements. For instance, most tasks in BEHAVIOR have quantifiers with complex spatial or temporal requirements, while VirtualHome tasks have much easier goal definitions.

4. **Transition Modeling:**

- Models like Claude-3.5 Sonnet and o1-preview excel in specific categories like object orientation and non-spatial relations, suggesting that targeted training or specialized architectures enhance LLM capabilities in understanding different types of tasks in transition modeling. For example, Claude-3.5 Sonnet achieves an F1-score of 78.8% in object states in BEHAVIOR, while o1-preview achieves an F1-score of 83.5% in non-spatial relations in BEHAVIOR.

- Across various models, non-spatial relations consistently pose a challenge, highlighting a gap in the ability of LLMs to grasp complex relational dynamics. For instance, in VirtualHome, the best-performing model, o1-preview, only achieves an F1-score of 11.9% in non-spatial relations in VirtualHome.

- The effectiveness of planning relies heavily on the consistency of the predicted action space by LLMs; discrepancies between mixed predicted and ground truth actions lead to reduced planner success. For example, if we mix the action spaces of GPT-4o predictions and ground truth, using "plug_in" from GPT-4o prediction and "walk_toward" and "switch_on" from ground truth, the PDDL planner cannot find a feasible solution for the task.

5. **Sensitivity Analysis:**

- Specific actions like "plug_in" and "walk_towards" consistently show low success rates due to complex preconditions and spatial requirements. For instance, in VirtualHome, the success rate for "plug_in" is only 0.09, and for "walk_towards", it is 0.63.

- Complex interactions involving detailed object manipulation, such as "slice_carvingknife" and "place_inside", present notable challenges. For example, in BEHAVIOR, the success rate for "slice_carvingknife" is 0.00, and for "place_inside", it shows a rather low success rate in many tasks.

- Current training regimens may not fully capture the diversity of real-world interactions, especially in spatial and object-oriented tasks. This is evident from the generally lower success rates for actions involving complex spatial relationships and object interactions.

6. **Pipeline-Based vs. Modularized:**

- Both modularized and pipeline-based methods have similar trajectory executable rates. For example, in the pipeline of Goal Interpretation and Action Sequencing in BEHAVIOR, the modularized method has an execution success rate of 53.0% for GPT-4o, while the pipeline-based method has an execution success rate of 55.0%.

- Pipeline-based methods suffer from error accumulation due to the composition of two modules. For instance, in the pipeline of Goal Interpretation and Subgoal Decom-

position in BEHAVIOR, the task success rate for GPT-4o drops from 48.0% in the modularized method to 38.0% in the pipeline-based method.

- SOTA LLMs generally avoid grammar errors for both pipeline-based and modularized methods, unlike less advanced models. For example, GPT-4o makes no parsing errors in both methods, while Llama 3 8B makes parsing errors in 2.0% of cases in the pipeline-based method.

- All LLMs, regardless of their advancement, are prone to runtime errors, missing necessary steps in their generation process. For instance, in the pipeline of Goal Interpretation and Action Sequencing in BEHAVIOR, GPT-4o encounters missing step errors in 35.0% of cases in both modularized and pipeline-based methods.

7. **Replanning and Feedback:**

- Incorporating replanning based on feedback significantly improves the model's performance, demonstrating over a 10% increase in success rates. For example, with replanning, GPT-4o's task success rate increases from 47.0% to to 59.0%, and its execution success rate increases from 53.0% to 63.0% in BEHAVIOR .

- Replanning can sometimes result in the over-generation of actions, as indicated by an increased rate of additional steps errors. For instance, with replanning, GPT-4o's additional step error rate increases from 0.0% to 3.0% in BEHAVIOR .

These empirical findings, along with the provided examples, highlight the strengths and weaknesses of LLMs in embodied decision-making tasks across different ability modules and simulators. The insights gained from these experiments can guide future research and development efforts to address the identified challenges and improve the performance of LLM-based embodied agents. We present more examples in Appendix E to illustrate the specific areas where LLMs excel or struggle, providing a more concrete understanding of their capabilities and limitations in various scenarios.

## B    Embodied Agent Interface Design

We will introduce the additional details about the EMBODIED AGENT INTERFACE (EAI) in this section, including the motivation of the current design and its relationship with the Markov Decision Process.

### B.1    Motivation

Our research focus is **embodied decision making** capabilities of LLMs. EMBODIED AGENT INTERFACE (EAI) is a diagnostic benchmark by decomposing the LLM abilities involved in **embodied decision making**. Given the natural language instructions from humans (such as "*cleaning the refrigerator*", "*polishing furniture*"), LLMs serve as embodied agents to achieve the specified goals through a sequence of actions in various embodied environments.

The key difference between language models and embodied agent models is the ability to (1) interact with the environment, (2) be goal-driven, and (3) decision making to achieve the goal, As shown in Figure 6. While some prior works [9] have proposed benchmarks with simulators to validate the output plan with a success rate, they are in the high-level natural language planning space without connecting to objects and state changes in the embodied environment, as shown in Figure 8. We address the limitations of traditional evaluations on benchmarking embodied decision-making from three aspects: (1) "benchmarking": We propose **a broad coverage of evaluation and fine-grained metrics**. Our interface offers fine-grained metrics to automatically identify various error types (such as missing step, additional step, wrong temporal order, affordance error, etc), providing a comprehensive evaluation of LLM performance. (2) "embodied": We move from high-level natural language planning to lower-level object interactions in the embodied environment. We **standardize goal specifications as linear temporal logic (LTL) formulas based on object-centric representations**, extending goals from states to temporally dependent logical transitions. (3) "decision making": We **standardize interface and modules approach** by unifying a broad set of decision-making tasks involving states and temporally extended goals, four key LLM-based modules (goal interpretation, subgoal decomposition, action sequencing, and transition modeling), covering the fundamental abilities in the Markov Decision Process (detailed in Appendix B.3). Please see Section 1 in the

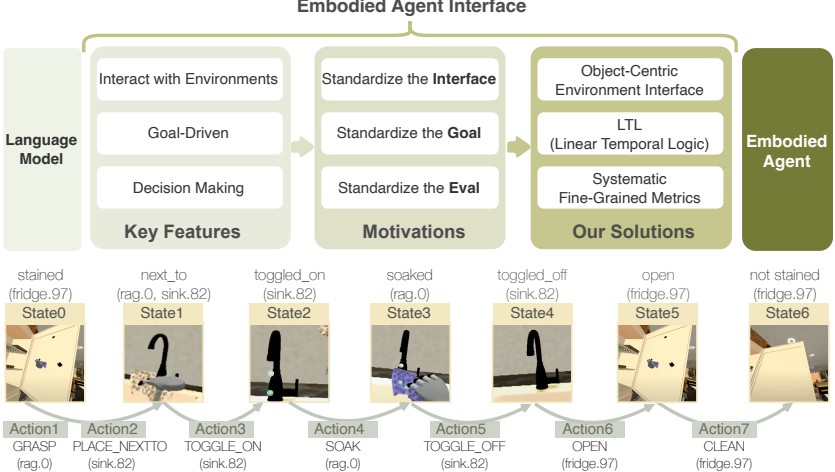

Example Task  Clean Refrigerator: use the rag to clean the refrigerator and ...

Figure 6: Compared to general language models, the embodied agent has three key new abilities, including interacting with environments, being goal-driven, and performing decision-making to achieve the goal. We believe a systematic evaluation for embodied decision-making should cover three aspects by standardizing the interface, the goal representation, and the evaluation metrics. Our EMBODIED AGENT INTERFACE addresses the limitations of traditional evaluations by focusing on goal-driven evaluation, standard interface, and modules, as well as broad coverage of evaluation and fine-grained metrics.

main paper for more details. Figure 7 summarizes the design of EMBODIED AGENT INTERFACE to connect LLMs with embodied environments.

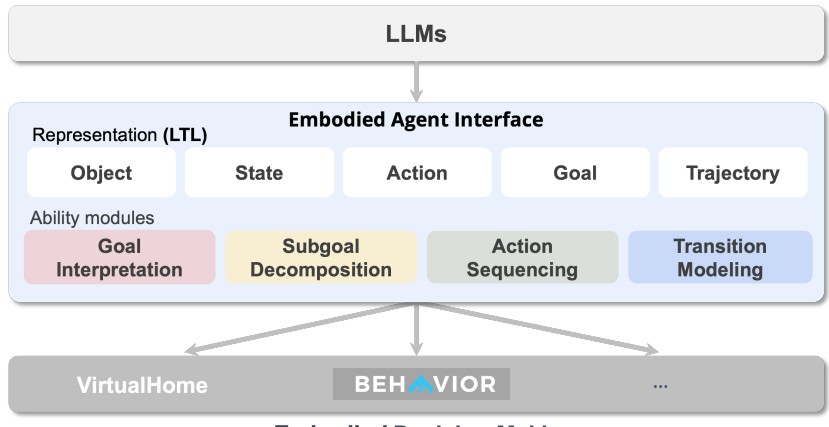

Figure 7: The EMBODIED AGENT INTERFACE aims to design a standard interface for LLMs to perform tasks in the embodied environment.

The evaluation is based on a comprehensive annotation of tasks, where each task contains the natural language task name, the natural language task instruction, the symbolic goal definition (including its LTL form), the symbolic action trajectory, the transition models involved in the task, as detailed in Figure 9 and Figure 10.

## B.2  Input and Output Details

As shown in Figure 11, the overall input of the interface consists of three main parts: (1) the task name and instruction, (2) the agent instructions, including in-context examples, and (3) the environment

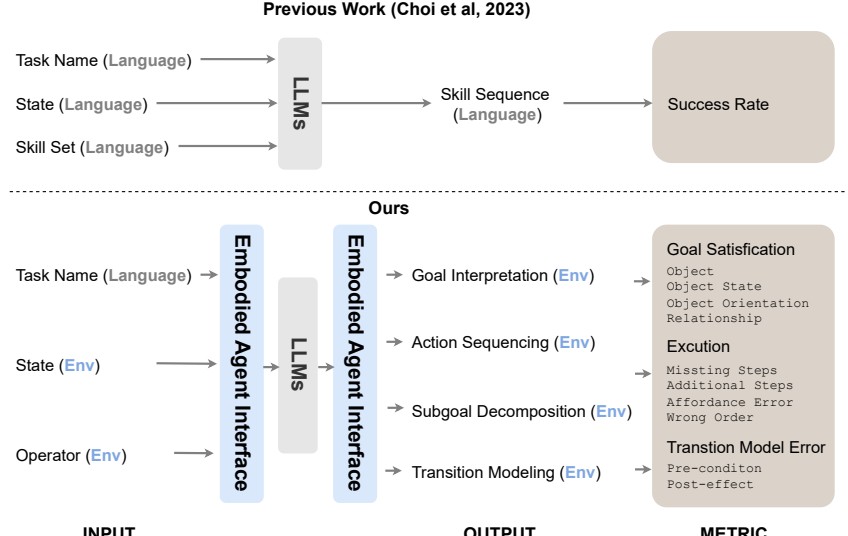

Figure 8: Comparison with existing benchmarks on LLMs for embodied decision making.

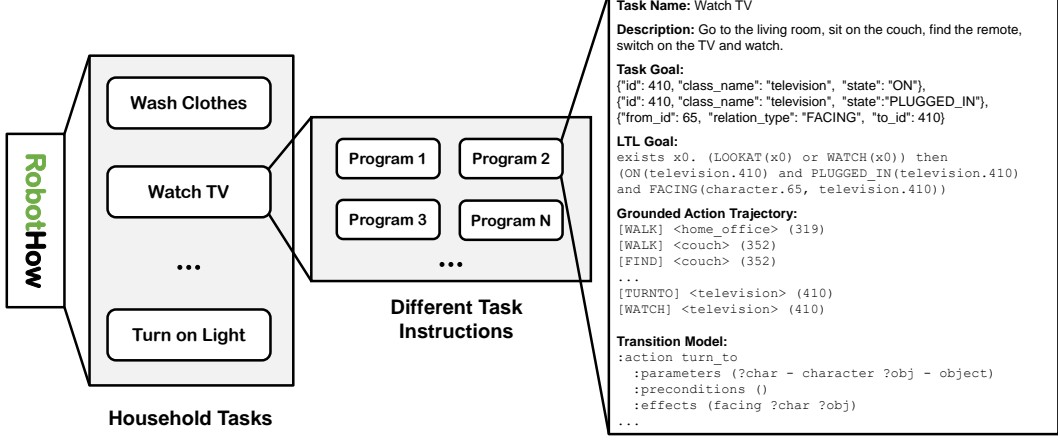

Figure 9: VirtualHome dataset structure example.

representation, which includes objects, their states, and relations. The detailed prompt templates are provided in Appendix I.

The input and output for each ability module differ, as illustrated in Figure 2. The mathematical formulation has been detailed in Section 2 of the main paper. **Goal Interpretation (ability module 1)** aims to ground the natural language instruction to the environment representations of objects, states, relations, and actions. For example, in Figure 2, the task instruction "*Use the rag to clean the trays, the bowl, and the refrigerator. When you are done, leave the rag next to the sink...*" can be grounded to specific objects with IDs, such as *fridge* (ID: 97), *tray* (ID: 1), *bowl* (ID: 1), *rag* (ID: 0), and *sink* (ID: 82). Note that a simple natural language description can be grounded into a set of multiple goal conditions (object state and relation).

The **Subgoal Decomposition (ability module 2)** generates a sequence of states, where each state can be a set of objects and their states. Here, we highlight the important states, such as the transitions between a sequence of *next_to*(rag.0, sink.82), *toggled_on*(sink.82), *soaked*(rag.0), *toggled_off*(sink.82), *open*(fridge.97), *not_stained*(fridge.97). To achieve these state transitions, we can use a high-level planner such as BFS to search for the **Action Sequences (ability module 3)** that achieve these state transitions. We obtain the following action se-

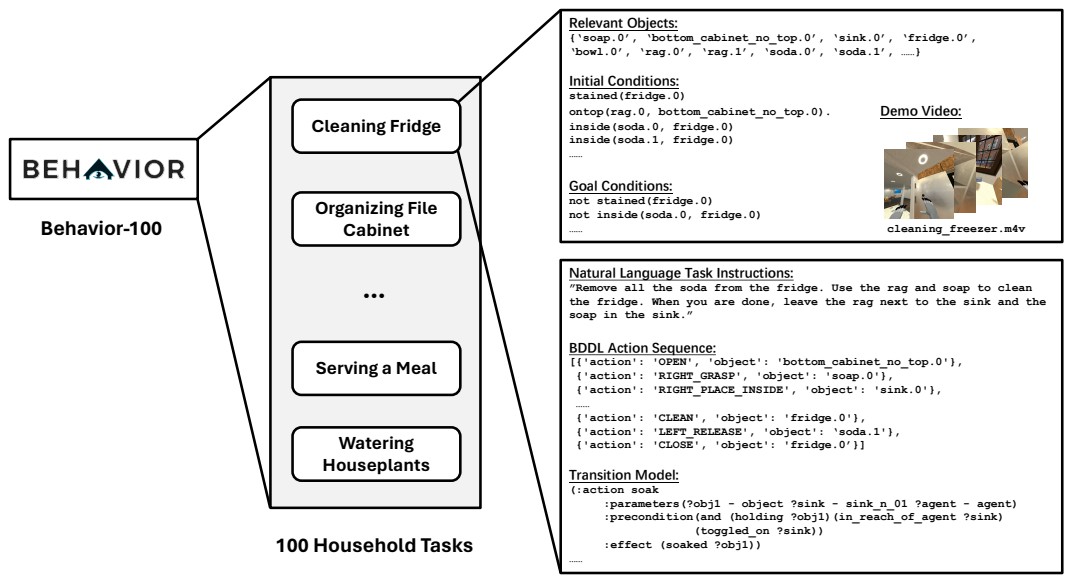

Figure 10: BEHAVIOR dataset structure example.

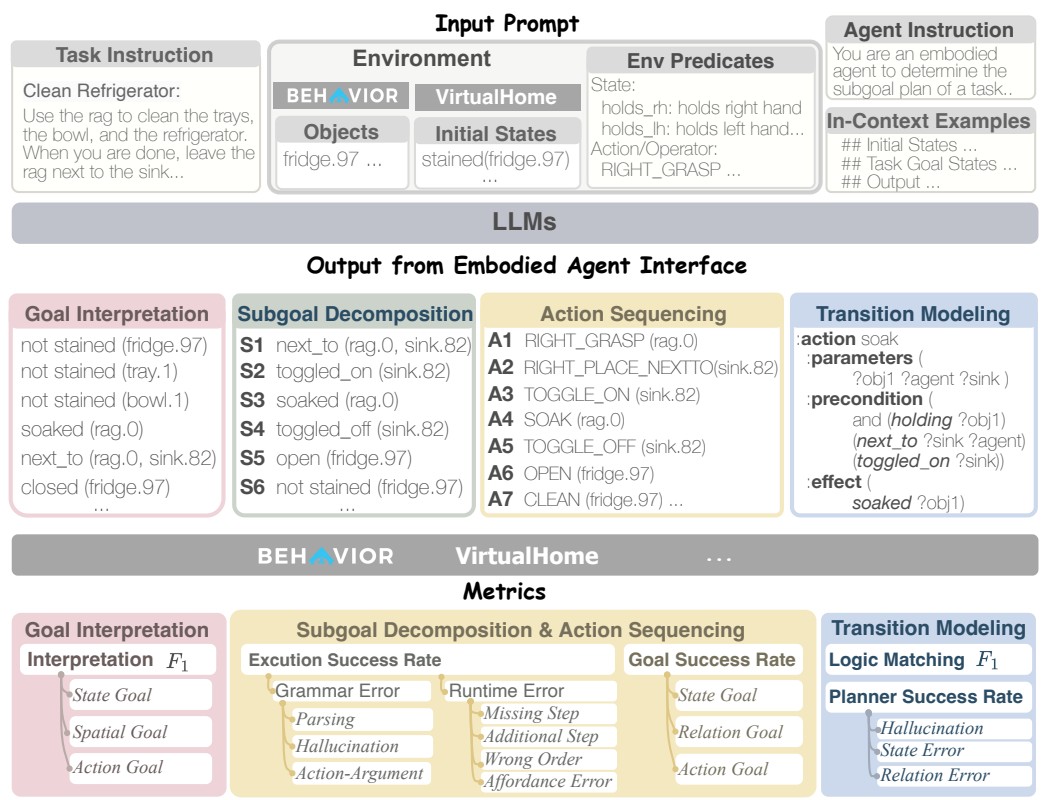

Figure 11: Example input and output for the ability modules.

quence: RIGHT_GRASP(rag.0), RIGHT_PLACE_NEXTTO(sink.82), TOGGLE_ON(sink.82), SOAK(rag.0), TOGGLE_OFF(sink.82), OPEN(fridge.97), CLEAN(fridge.97). Note that multi-

ple actions may be required to achieve a single one-step state transition. For example, to perform the state transition *next_to*(rag.0, sink.82) → *toggled_on*(sink.82), we need two actions RIGHT_GRASP(rag.0), RIGHT_PLACE_NEXTTO(sink.82). We show a successful execution of this piece of an action sequence in Figure 12.

**Transition Modeling (ability module 4)** is different from the previous modules. It serves as the low-level controller to guide the simulator in performing state transitions from preconditions to post-effects [71–74]. In Figure 2, the input is the operator name "*soak*", and the preconditions are three states: "*holding* (?obj1)", "*next_to* (?sink ?agent)", and "*toggled_on* (?sink)". The post effect after executing SOAK is "*soaked* (?obj1)".

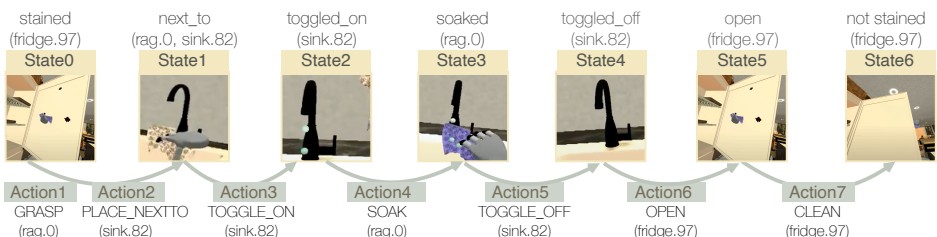

**Example Task** Clean Refrigerator: use the rag to clean the refrigerator and ...

Figure 12: An example of successful execution in BEHAVIOR .

## B.3 Grounding to Markov Decision Process

To support a wide range of tasks in various environments, we design the EMBODIED AGENT INTERFACE based on the Markov Decision Process (MDP) [75], a fundamental mathematical framework for robot learning to formalize sequential decision-making in embodied agents [76]. This allows us to create a structured approach to benchmark the robot's decision-making process.

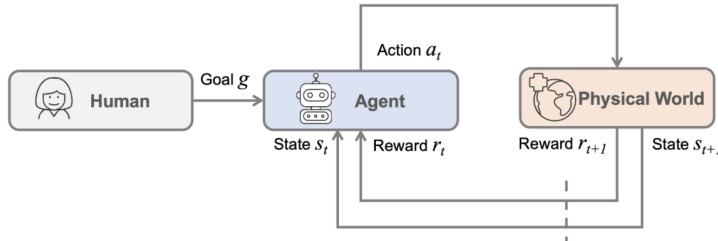

Figure 13: Embodied Decision Making is a Markov Decision Process.

An embodied agent takes natural language instructions from humans and achieves the specified goals through a sequence of physical state transitions. It is essentially a decision-making process to determine the actions based on the goal and the current state in the embodied environment. As a result, we formulate the MDP process as below to input natural language instructions and interact with the environment to achieve the specified goals.

**MDP Formulation for Embodied Agents.** As shown in Figure 13, the Markov Decision Process for an embodied agent can be defined by a tuple $\langle \mathcal{U}, \mathcal{S}, \mathcal{A}, \mathcal{M}, \mathcal{R}, g \rangle$, where:

$\mathcal{U}$ is the universe of objects in the environment, which are the fundamental entities that the agent interacts with. $\mathcal{S}$ is the state space, where each state $s \in \mathcal{S}$ is represented as a tuple $\langle \mathcal{U}, \mathcal{F} \rangle$. $\mathcal{F}$ is a set of relational Boolean features that capture the properties and relations among objects in the environment. $\mathcal{A}$ is the action space, which represents the set of actions the embodied agent can execute. Actions are represented as tuples $\langle name, args \rangle$, where *name* is the action name and *args* are the object arguments the action operates on. $\mathcal{M} : \mathcal{S} \times \mathcal{A} \to \mathcal{S}$ is the environmental transition model, which specifies the next state $s_{t+1}$ given the current state $s_t$ and action $a$. $\mathcal{R} : \mathcal{S} \times \mathcal{A} \times g \to \mathbb{R}$ is the reward function. It depends on the current state, action, and the goal specification $g$. For a

state $s$, action $a$, and goal $g$, $\mathcal{R}(s, a, g) = 1$ if $eval(g, s) = 1$ (i.e., the goal is satisfied in state $s$), and $\mathcal{R}(s, a, g) = 0$ otherwise. Here, $eval : g \times \mathcal{S} \rightarrow \{0, 1\}$ determines whether a state satisfies the goal specification. $g$ is the goal specification. The goal should be grounded in terms of the desired final states of objects and their interactions (relations and executed actions), capturing the intended outcome of the agent's actions. The input of the goal can be a natural language, such as "*cleaning the refrigerator*" or "*polishing furniture*". We denote natural language goal specification as $l_g$.

**Grounding Our Evaluation Protocol to the Fundamental Modules of MDP.** The embodied agent receives a natural language goal specification $l_g$, translates it to the environment objects and their states, relations, and actions as a goal specification $g$, and aims to achieve it through a sequence of state transitions. To abstract the embodied environment, we design the representation to contain *Object*, *State*, *Action*, and, based on that, *Goal* (as final states) and *Trajectory* (as temporally dependent sequences of actions/states). Our interface is built upon a LTL REPRESENTATION layer based on Linear Temporal Logic (LTL), which serves as a unified, expressive interface to communicate with robots in different environments (e.g., different simulators such as BEHAVIOR and VirtualHome).

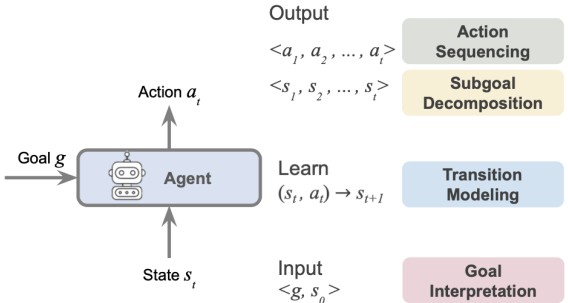

Figure 14: Our four ability modules are fundamental modules of the MDP process.

At each step, the agent observes the current state $s \in \mathcal{S}$, selects an action $a \in \mathcal{A}$ based on its policy $\pi : \mathcal{S} \times g \rightarrow \mathcal{A}$, and receives a reward $\mathcal{R}(s, a, g)$. The environment transitions to the next state according to $\mathcal{M}(s, a)$. As shown in Figure 14, according to MDP, it essentially focuses on four abilities:

- Input of Goals, which corresponds to **Goal Interpretation (ability module 1)**, translating the natural language goal to environment objects and their relations and actions.
- Output of Trajectories, where the output can be a sequence of actions or a sequence of states, which can be regarded as **Action Sequencing (ability module 2)** and **Subgoal Decomposition (ability module 3)**.
- The core part of the **Transition Model (ability module 4)** Learning, which is covered by the Transition Modeling (ability module 4).
- The goal-evaluation function *eval* and reward model can be reflected in the detailed, fine-grained evaluation metrics we provide.

In this way, the EMBODIED AGENT INTERFACE has a comprehensive coverage of the fundamental abilities and can provide a systematic evaluation of the foundational MDP process.

## B.4 The Relationship between Ability Modules

To identify the weaknesses and areas for improvement in LLMs for embodied decision-making, we need to evaluate each ability module individually and focus on detailed, fine-grained tasks. Rather than simply knowing that the final success rate is still insufficient, we aim to understand which abilities are already well-developed and how we can effectively integrate different ability modules to enhance overall performance. This includes exploring the integration between LLMs and external tools, as well as LLMs across different modules, enable us to guide embodied agents to use LLMs more selectively and effectively.

To achieve this, as shown in Figure 4, we design an evaluation protocol that isolates a single module to be handled by the LLM while using existing data or tools to serve as the other modules. This approach

shifts the focus from an end-to-end evaluation to an accurate assessment of each individual component. By doing so, we can probe the LLM's capabilities and limitations within each specific ability in detail, gaining a more nuanced understanding of its performance. This fine-grained evaluation allows us to identify the strengths and weaknesses of LLMs in each ability module, guiding future research efforts to address the identified challenges and improve the integration of LLMs in embodied decision-making tasks.

The Subgoal Decomposition and Action Sequencing modules are similar in that they both involve trajectory output and evaluate the ordering of decision-making. However, the fundamental distinction between them lies in the nature of their outputs. Action sequencing produces imperative actions, while subgoal decomposition generates declarative states, as illustrated in Figure 12.

Transition modeling can be considered as the low-level controller that governs the state transitions when executing an action. The hallmark of transition modeling is the ability to search a path to navigate from initial predicates to goal predicates using existing actions. Defining preconditions and post effects for each action enables this search and backtracking.

# C  LTL Representation and Implementation

## C.1  Why LTL

Our EMBODIED AGENT INTERFACE is built on top of the linear temporal logic (LTL) language. This is motivated by two critical desiderata of the interface. First, we need an expressive and compact language to describe task specifications. Classical choices such as first-order logic formulas on goal states or reward functions both have their limitations: goal state formulas only describe the requirements over the goal state but not any temporal ordering of how subgoals should be achieved. On the other hand, reward functions are general in specifying preferences over trajectories but they usually can not be represented in a compact way due to their numeric nature. Second, we need a unified interface between different modules of an embodied agent system, such as the inputs and outputs for goal interpreters, subgoal generators, etc. For example, BEHAVIOR [4] uses BEHAVIOR Domain Definition Language (BDDL) to represent goals as a target state with logic constraints, such as "`not(stained(fridge_97)), forall tray.n.01-tray.n.01 inside(tray.n.01,fridge_97) not(stained(tray.n.01)), ...`". In contrast, Virtual-Home [5] describes task goals in natural language with a different focus, such as "*take everything out of the fridge, throw anything outdated...*". Furthermore, different agents may follow different trajectories and have different criteria for achieving the same goal. For instance, BEHAVIOR focuses on state transitions to match final states with goals ("not *stained*(fridge)"), whereas VirtualHome evaluates execution success without verifying whether the goal states are satisfied. BEHAVIOR focuses on state transitions to match final states with goals("not *stained*(fridge)"), while VirtualHome only considers execution success rates without checking whether goal states are satisfied. This leads to significant differences in goal interpretation and object state representation across environments.

LTL provides an expressive and compact description language solution to these issues. At a high level, an LTL formula can describe basic state constraints (e.g., a subgoal should be achieved), action constraints (e.g., a particular action should be executed), and possible temporal orders and dependencies among them (e.g., all dishes should be cleaned before we start to cook). By combining temporal connectives such as "next" and propositional logic connectives, we can also flexibly describe alternative goals or subgoal sequences. Therefore, state goals, action sequences, pre-conditions and post-conditions of actions, subgoal sequences, or even sets of candidate subgoal sequences, can all be expressed in LTL in a compact way. As a byproduct, using a single description language for all inputs and outputs enables us to design a unified evaluation metric to measure the prediction accuracy, by measuring the similarity between two LTL formulas, which is detailed in later sections.

Figure 15 illustrates the complete process of subgoal decomposition using LTL in our EMBODIED AGENT INTERFACE. The process begins with describing the environment using LTL-like grammar. Once the prompt is crafted, a language model generates the corresponding output, which is then translated into a plain LTL formula. This formula is subsequently parsed into an LTL expression tree. Finally, a concrete subgoal path is sampled and converted into an action sequence. This action sequence is then executed in simulators to ensure two main criteria: (1) the subgoals are well-defined and executable, and (2) if executable, whether they meet the final state.

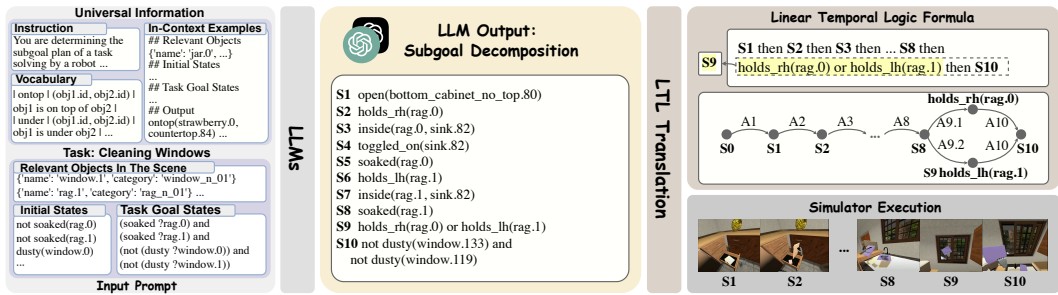

Figure 15: Pipeline of subgoal decomposition based on LTL in EMBODIED AGENT INTERFACE

## C.2    Comparision with Traditional LTL Representation

Compared with initial LTL, our adaptation introduces relational state representations, quantifiers (including a counting quantifier $\exists^{=n}$), and a custom "then" operator for temporal ordering in finite trajectories replacing typical "Next" and "Eventually" operators, making it more expressive for task planning. While these extensions enhance the framework, the use of logical connectives and recursive formula structure remains consistent with standard LTL.

## C.3    Syntax and Semantics of LTL Formulas

In EMBODIED AGENT INTERFACE, a state is represented as a tuple $s = \langle U, F \rangle$, where $U$ is the universe of entities, assumed to be a fixed finite set. $F$ is a set of relational Boolean features. Each feature $f \in F$ can be viewed as a table where each entry is associated with a tuple of entities $(o_1, \cdots, o_k)$. Each entry has the value of the feature in the state, and $k$ is the arity of the feature. For example, the feature $on(x, y)$ is a binary predicate. Actions can be viewed as primitive functions that take entities as inputs. For a physical robot, this corresponds to the available low-level controllers that our algorithm can interface with, such as moving and grasping.

**LTL syntax.** Our EMBODIED AGENT INTERFACE uses a fragment of the full linear temporal logic (LTL) formalism on finite trajectories. In particular, we consider the following two types of atomic propositions. The arguments to these propositions can be either object in the state (e.g., book1, cat1) or quantified variables (e.g., $x$).

(1) State propositions: Predicates that describe properties of object states and relations. For example, $ontop(\text{book1}, \text{chair1})$.
(2) Action propositions: Predicates that denote actions. For example, $touch(\text{cat})$.

An LTL formula $\phi$ is defined recursively as follows:

$$\phi ::= p \mid \neg\phi \mid \phi_1 \wedge \phi_2 \mid \phi_1 \vee \phi_2 \mid \phi_1 \Rightarrow \phi_2 \mid \forall x\, \phi(x) \mid \exists x\, \phi(x) \mid \exists^{=n} x\, \phi(x) \mid (\phi) \mid \phi_1 \text{ then } \phi_2$$

where $\phi_1$ and $\phi_2$ are LTL formulas, $p$ is an atomic proposition. $\neg$ (negation), $\wedge$ (and), $\vee$ (or), $\Rightarrow$ (implies) are logical connectives. $\forall, \exists$ and $\exists^{=n}$ are quantifiers. Note that, $\exists x$ means that there is at least one x such that $\phi(x)$ is satisfied, whereas $\exists^{=n} x$ means that there are exactly $n$ x's such that $\phi(x)$ is satisfied. **then** is a temporal connective, where $\phi_1$ **then** $\phi_2$ intuitively means $\phi_1$ should happen before $\phi_2$[†]. Note that the operator **then** is a combination of the "next" and the "eventually" operator in standard LTL formalism, and we do not include "globally" and "until," since the "then" operator is sufficient for describing all the task and input-output specifications in our system, although we can naturally extend our implementation to include them.

---

**LTL Grammar Definition**

```
?start: stmt
primitive: VARNAME "(" [args] ")"   # primitive format looks like varname(param)
object_name: VARNAME                # object_name can be an object name (eg. pants)
```

---

[†]The priority of LTL operators from highest to lowest is $() > \forall = \exists = \exists^{=n} > \neg > \wedge > \vee > \text{then}$.

```
            | VARNAMEWITHID          # or object name with ID (eg. pants.1000)
args: object_name ("," object_name)*  # definition of arguments

?stmt: then_stmt | primitive_stmt     # a formula is a Boolean stmt
then_stmt: or_stmt ("then" or_stmt)*  # connective priority order:
or_stmt: and_stmt ("or" and_stmt)*    # then < or < and < not < forall = forn = exists
and_stmt: primitive_stmt ("and" primitive_stmt)*
primitive_stmt: "not" primitive_stmt -> not_stmt
              | primitive
              | "(" stmt ")"
              | "forall" VARNAME "." "(" stmt ")" -> forall_stmt
              | "forn" VARNAME "." "(" stmt ")" -> forn_stmt
              | "exists" VARNAME "." "(" stmt ")" -> exists_stmt

%import common.WS
%ignore WS
VARNAME: /[a-zA-Z_]\w*/
VARNAMEWITHID: /[a-zA-Z_]\w*\.[0-9]+/
```

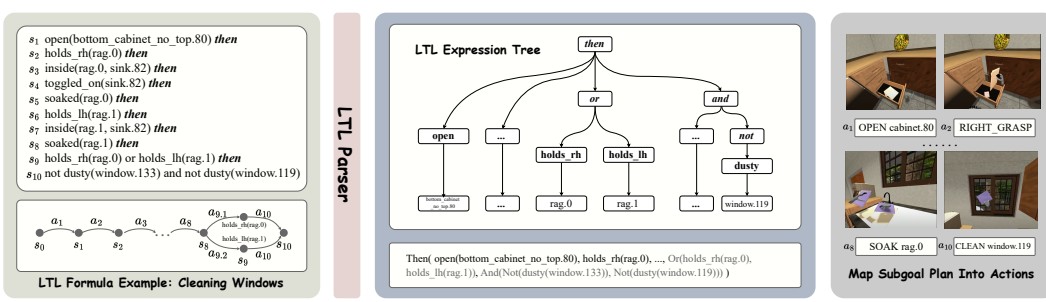

Figure 16: An example of LTL representation.

**LTL semantics**. An LTL formula can be viewed as a classifier over trajectories semantically: we can evaluate an LTL formulate $\phi$ based on a state-action sequence. If the evaluation returns true, we say the state-action sequence satisfies $\phi$. This can be directly used to evaluate whether a generated action sequence satisfies the task specification. The task of a planner would be to take an LTL formula as its specification and generate a state-action sequence that satisfies the formula.

Let a state-action trajectory $T$ be $[s_0, a_1, s_1, \ldots, a_n, s_n]$, $T_i = (s_i, a_i)$, and $U$ be the universe of entities in $T$. For a state-action pair, we can define the semantics of atomic propositions, logic connectives, and quantifiers. In particular, for atomic propositions $p$, $eval(p, (s_i, a_i))$ is true if $p$ is satisfied in $s_i$ (if $p$ is a state predicate) or $a_i = p$ (if $p$ is an action predicate). All logic connectives ($\neg$, $\wedge$, $\vee$, and $\Rightarrow$) and quantifiers ($\forall$ and $\exists$) follows their semantics in first-order logic. The for-n counting quantifier $\exists^{=n}$ has the semantics that: $eval(\exists^{=n}x.\phi(x), T_i) = \mathbb{1}[\sum_x eval(\phi(x), T_i) = n]$, where $\mathbb{1}[\cdot]$ is the indicator function. For compactness, if we apply a state-action formula $\phi$ on a trajectory $T$ instead of a concrete state-action pair $T_i$: $eval(\phi, T) = \exists k.eval(\phi, T_k)$. That is, $\phi$ is satisfied in at least one of the states in $T$.

The semantics of the operator **then** is defined as the following:

$$eval(\phi_1 \textbf{ then } \phi_2, T) = \exists k.\phi_1(T_{\leq k}) \wedge \phi_2(T_{>k}),$$

where $T_{\leq k}$ is the first $k$ state-action pairs in $T$ and $T_{>k}$ is the suffix sequence after $k$ steps. Intuitively, it means, there exists a segmentation of the trajectory $T$ such that $\phi_1$ is satisfied in the first half while $\phi_2$ is satisfied in the second half.

The LTL formula will be parsed into an LTL expression tree before the evaluation process, as demonstrated in Figure 16. In order to evaluate the function $eval(\phi, T)$ given the LTL formula and a state-action sequence, one needs to recursively evaluate components in $\phi$ based on their semantics. This is typically implemented with a dynamic programming algorithm over LTL formulas and subsequences of $T$.

# D  Fine-Grained Metrics and Automatic Error Detection

To evaluate each ability in the simulator, we design the evaluation pipeline of each ability and detailed in this section.

## D.1  Goal Interpretation: State Goal, Relation Goal and Action Goal

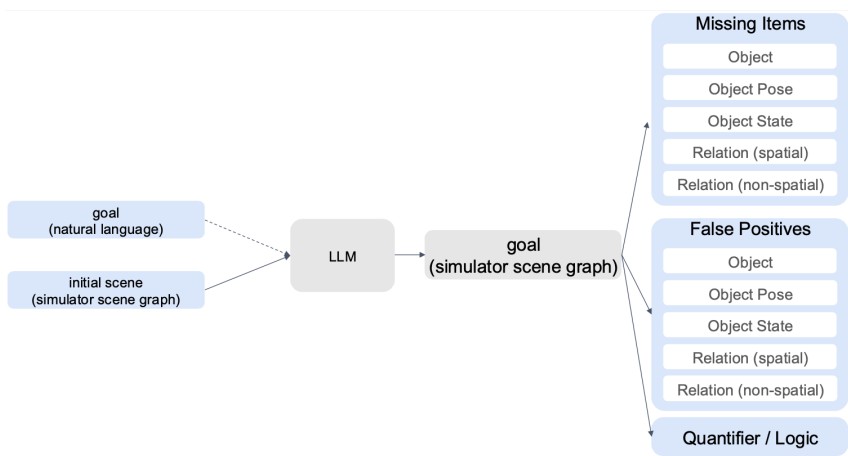

Figure 17: Evaluation pipeline of goal interpretation. We evaluate the LLM's output for the Goal Interpretation in three key dimensions: single-object states, object-object relations, and agent-action goals. For each dimension, we calculate precision, recall, and F1 score to measure the LLM's false positive predictions, likelihood of missing goals, and overall capability, respectively.

For the Goal Interpretation, the LLMs receive a natural language description of the overall goal, the initial world state $S_0$ (including relevant objects and their initial states), and the complete list of all possible actions and object states recognized by the simulator. Based on these inputs, the LLMs are expected to output a symbolic representation of the overall goal, denoted as $g$. This task is designed to assess the LLMs' ability to act as a translation layer between a human (who only interacts with the system using natural language) and the embodied agent (which only understands symbolic goals composed of simulator-recognizable states and relevant objects in the scene).

To evaluate the LLMs' output for the Goal Interpretation, we first filter for grammatically correct predicted goals while keeping track of structurally incoherent predictions and object/state hallucinations. Then, we evaluate the remaining grammatically correct predicted goals against ground truth goals in three key dimensions:

- **State Goal (or Node Goal)** with a focus on single-object states, e.g., *facing*(cat), *clean*(cup), *switched_on*(television).
- **Relation Goal (or Edge Goal)** with a focus on object-object relations, e.g., *next_to*(character, cat), *on*(cup, table), *on*(character, sofa).
- **Action Goal** with a focus on agent-action goals, e.g., *touch*(cat), *touch*(remote). Simulators such as VirtualHome contain action goals for some short tasks that have key actions but no post-effects to validate in the goal, such as the task of "*pat cat*". In contrast, other simulators like BEHAVIOR-100 do not include such action goals.

For each of the three goals, we calculate the following metrics:

- *Precision*: Measures the LLMs' false positive predictions, indicating the proportion of predicted goals that are correct.
- *Recall*: Measures the LLMs' likelihood to miss certain goals, indicating the proportion of ground truth goals that are correctly predicted.
- $F_1$ Score: A joint measure that combines precision and recall, representing the overall capability of the LLMs in each dimension.

These metrics provide a comprehensive evaluation of the LLMs' performance in interpreting natural language goals and translating them into symbolic representations that the embodied agent can understand and execute.

## D.2 Action Sequencing: Trajectory Error Detection for Missing Step, Additional Step, Wrong Temporal Order, Affordance Error

Action sequencing serves as an intuitive and pragmatic approach to evaluate the effectiveness of LLMs in the context of embodied agents. This task requires LLMs to generate a sequence of executable actions aimed at achieving predefined goals. The evaluation protocol for action sequencing focuses on assessing the LLMs' ability to produce accurate and executable action sequences within embodied environments. The process involves several key steps to ensure that the generated plans are both realistic and effective in achieving the specified goals.

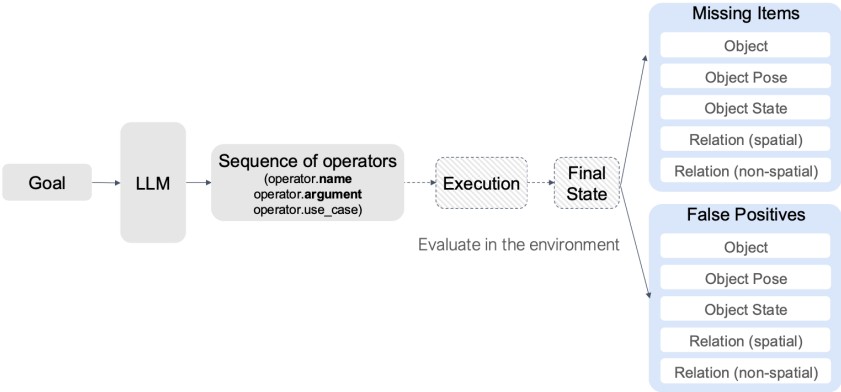

Figure 18: Evaluation pipeline of action sequencing.

**Input Instruction.**    In action sequencing tasks, LLMs are given an initial world state $s_0$, a goal $g$, and general environment information, such as the vocabulary of actions and predicates. The models are required to output a sequence of executable actions $\bar{a} = \{a_1, a_2, \cdots, a_n\}$, where each action $a_i$ includes an action name and an ordered list of object names as parameters.

The input includes the objects present in the scene, the relative initial state (we use the changes between the initial state and final state to decide a subset of relative objects and their states as the relative initial state), node goals, edge goals, and action goals.

The next step involves rephrasing the concrete goals into natural language, which helps improve the LLM's understanding and execution of the tasks. The rephrased goals are structured as node goals (specifying the state of an object in natural language, such as "*the television is switched on*"), edge goals (describing the relationship between objects in natural language, such as "*the agent is next to the television*"), action goals (describing the required actions in some tasks in natural language, such as "*the robot touches the remote control*").

The evaluation framework for this task is divided into two primary components: trajectory evaluation and goal achievement.

**Trajectory Evaluation.**    The trajectory evaluation assesses whether the action sequence $\bar{a}$ is executable. If $\bar{a}$ is found to be non-executable, errors are categorized into three main classes, each with a fine-grained set of subclasses:

   a. **Grammar Errors:**

- **Parsing Error**: Evaluates whether the output strictly adheres to specified format requirements.
- **Hallucination Error:** The format is correct, but the action names, object names, or predicate names are not in the environment vocabulary.

- *Action Name Hallucination* - Checks for the accuracy of action names supported by the environment.
- *Object Name Hallucination* - Ensures the correctness of object names supported by the environment.
- **Action-Argument Number Error** - Verifies that the action or argument parameters are incorrect, mainly if the length of parameters does not meet the specified requirements of the action.

b. **Runtime Errors:**

- **Affordance Error**: Determines if the properties of objects allow for the execution of the action. For example, *open*(shelf) is wrong as the shelf cannot be opened.
- **Additional Step**: Detects when an action is redundant given the current state. If objects are affordable for the action but the intended effect is already present in the current state, it indicates an unnecessary additional step. For example, *toggle_on*(light) cannot be executed when the *light* is already *toggled_on*.
- **Missing Step**: Identifies when a required precondition for an action is not met. If (1) properties match, (2) the effect has not been satisfied in the current state, then the execution error stems from an unsatisfied precondition. We then (3) check whether the precondition has never been satisfied in the historical states, which means that a step is missing to trigger the precondition. For example, *release*(book) cannot be executed when the precondition *grasped(book)* is NOT satisfied. If the book has never been grasped by the agent in the historical states, a necessary step *grasp*(book) is missing.
- **Wrong Order**: Determines if actions are executed out of sequence. If the precondition is wrong but there has been a precondition being satisfied in the historical states, then the error indicates a wrong order (i.e., the current step should be executed immediately after the relevant historical state). In the previous example, if *grasped(book)* has been previously satisfied (e.g., the book was picked up by the agent but then placed on the table), the order of actions is wrong, and *release*(book) should be promoted to the earlier steps when the book was still grasped.

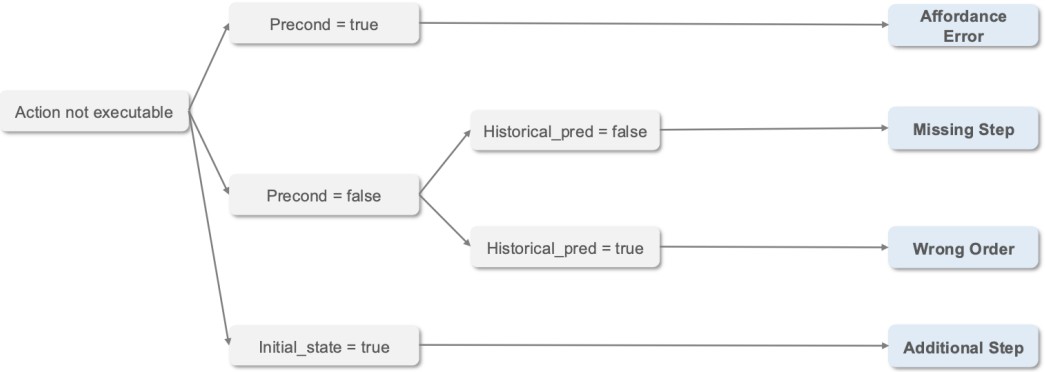

Figure 19: Automatic Error Categorization for Trajectory: (1) Check if the action is affordable based on the properties of objects in the current state. If not, return "Affordance Error". (2) If the action is affordable, check if the intended effect of the action is already satisfied in the current state. If it is, return "Additional Step". (3) If the effect is not redundant, check whether the precondition of the action is satisfied in the current state. If not, proceed to the next step. Otherwise, return "No Runtime Error". (4) If the precondition is not satisfied, check if it has ever been satisfied in any of the historical states. If the precondition has never been satisfied, return "Missing Step". (5) If the precondition has been satisfied in a historical state, return "Wrong Order".

**Trajectory Error Detection:**
**Missing Step, Additional Step, Wrong Temporal Order, Affordance Error**

```
def check_runtime_errors(action, current_state, historical_states):
    # Check affordance error
    if not is_affordable(action, current_state):
        return "Affordance Error"
```

```
    # Check if action effect is redundant (Additional Step)
    if is_effect_redundant(action, current_state):
        return "Additional Step"

    # Check if precondition for action is met
    if not is_precondition_satisfied(action, current_state):
        # Check if the effect has not been satisfied
        if not is_precondition_ever_satisfied(action, historical_states):
            # Check historical states for precondition satisfaction
            if not precondition_satisfied_in_history(action, historical_states):
                return "Missing Step"
            else:
                return "Wrong Order"

    return "No Runtime Error"
```

**Goal Satisfaction**  The success criteria are based on the accurate achievement of all specified goal conditions and the trajectory to achieve the goal. Both node goals, edge goals, and action goals are checked to ensure that the final state of the environment matches the desired outcomes.

If the action sequence $\bar{a}$ is executable, the subsequent evaluation examines whether executing $\bar{a}$ from $s_0$ satisfies the goal $g$. The simulator executes the action sequence $\bar{a}$ and collects the execution information and intermediate world states $s_i$ for each action $a_i \in \bar{a}$. The execution continues until either an action is not executable (marked as $a_t$) or all actions are executed successfully. The final world state is denoted as $s_t$.

Given a goal $g$, each goal condition is assigned a category (node goal, edge goal, or action goal) based on a simulator-based classification, similar to Appendix D.1. The satisfaction of each goal condition $g_i$ by $s_t$ is then checked using the simulator. The primary metric for this evaluation is the *Success Rate*, calculated as the ratio of tasks where $g$ is satisfied to the total number of tasks. Additionally, recall is calculated for all goals and different categories of goals to evaluate goal satisfaction.

### D.3 Subgoal Decomposition: Converting Subgoal Trajectory to Action Trajectory with BFS Searching

The subgoal decomposition module generates a sequence of declarative states in terms of temporal order. While there is no reference decomposition and multiple optional ways to decompose goals, our method avoids recursive translation between symbolic and natural language, which can add complexity. Additionally, decomposing goals purely at the final time step is often inefficient for search purposes. By using a temporal breakdown, the approach balances efficiency and effectiveness and is commonly employed in embodied agent systems such as Voxposer [77] and ReKap [78]. To validate the feasibility of the state transitions, we need to transform the state transitions into an actionable trajectory that can be executed and evaluated in the simulator. As depicted in Figure 4, we address this challenge by utilizing a customized planner to refine the subgoal decomposition output into an actionable sequence. As detailed in Section 2 in the main paper, this subgoal-action mapping function, denoted as $\mathcal{AM}(\bar{\phi}, s_0)$, takes the LTL representation of the subgoal sequence $\bar{\phi}$ and the initial state $s_0$ as inputs and generates a corresponding state-action sequence $\bar{t}$.

We implement this mapping function using a Breadth-First Search (BFS) algorithm. We show this process in Figure 20, where we find a sequence of actions that can transition the agent from the initial state to the desired subgoal states.

The BFS algorithm starts from the initial state $s_0$ and expands the search frontier by exploring all possible actions at each step. It maintains a queue of states to be visited and keeps track of the path from the initial state to each visited state. The search continues until a state satisfying the first subgoal is reached. The process is then repeated, using the reached subgoal state as the new initial state and the next subgoal as the target.

The subgoal-action mapping function $\mathcal{AM}(\bar{\phi}, s_0)$ takes the LTL representation of the subgoal sequence $\bar{\phi}$ and the initial state $s_0$ as inputs and returns a state-action sequence $\bar{t}$ that achieves the subgoals. The function can be described as follows:

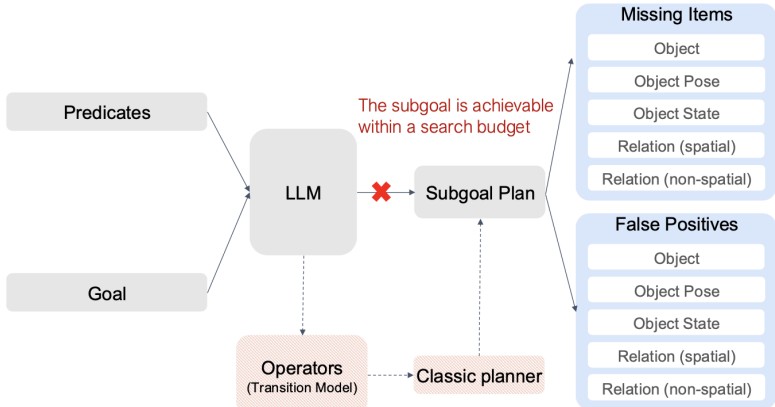

Figure 20: Evaluation pipeline of subgoal decomposition. The subgoal decomposition module generates a sequence of declarative states, which need to be transformed into an actionable trajectory for execution in the simulator.

---

**BFS Searching to Transform State Trajectory to Action Trajectory**

```
Initialize an empty state-action sequence t̄
Set the current state s_curr to the initial state s_0
For each subgoal φ_i in the subgoal sequence φ̄:
For each subgoal φ_i in the subgoal sequence φ̄:
    Perform BFS starting from s_curr to find a state s_goal that satisfies φ_i.
    Extract the path from s_curr to s_goal and append the corresponding state-action pairs
        to t̄.
    Set s_curr to s_goal.
Return the complete state-action sequence t̄.
```

---

By using the BFS algorithm to find action sequences that connect the subgoal states, we can effectively transform the declarative subgoal decomposition into an imperative action trajectory. This allows us to evaluate the subgoal decomposition module in the simulator and assess its effectiveness in guiding the agent towards the desired goal.

Note that the BFS algorithm guarantees finding the shortest path between subgoals if one exists, but it may be computationally expensive in large state spaces. In our current benchmark, due to limited pre-defined action space and state space (detailed in Appendix N.2 and Appendix N.1), we do not need to control the searching space. In practice, with larger action space and state space, heuristics or domain-specific knowledge can be incorporated to guide the search and improve efficiency.

By employing this approach, we can effectively evaluate the subgoal decomposition module within the simulator, ensuring that the generated subgoals can be successfully grounded and executed. This allows us to assess the quality of the subgoal decomposition and its impact on the overall decision-making process, even though the direct output of the module is declarative states rather than imperative actions.

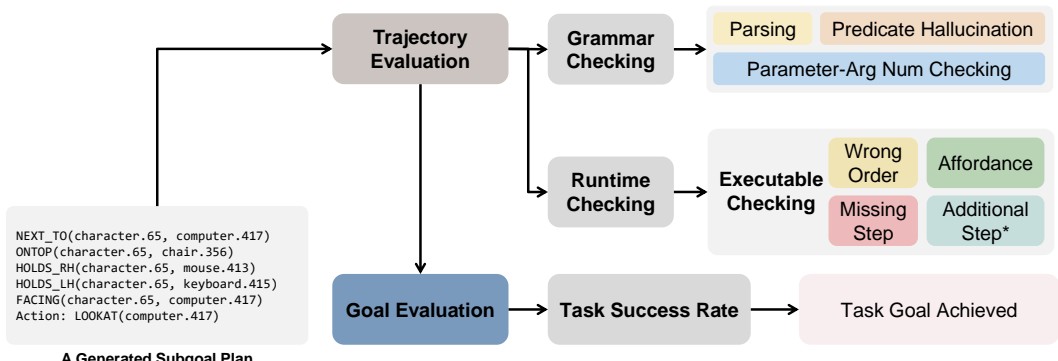

A Generated Subgoal Plan

Figure 21: The pipeline of evaluating a generated subgoal plan.

## D.4 Transition Modeling: Evaluating with PDDL Planners

The evaluation protocol for transition modeling aims to rigorously assess the ability of Large Language Models (LLMs) to predict accurate action preconditions and effects within embodied agent scenarios. This evaluation is critical for understanding the proficiency of LLMs in modeling the dynamics of physical interactions in simulated environments.

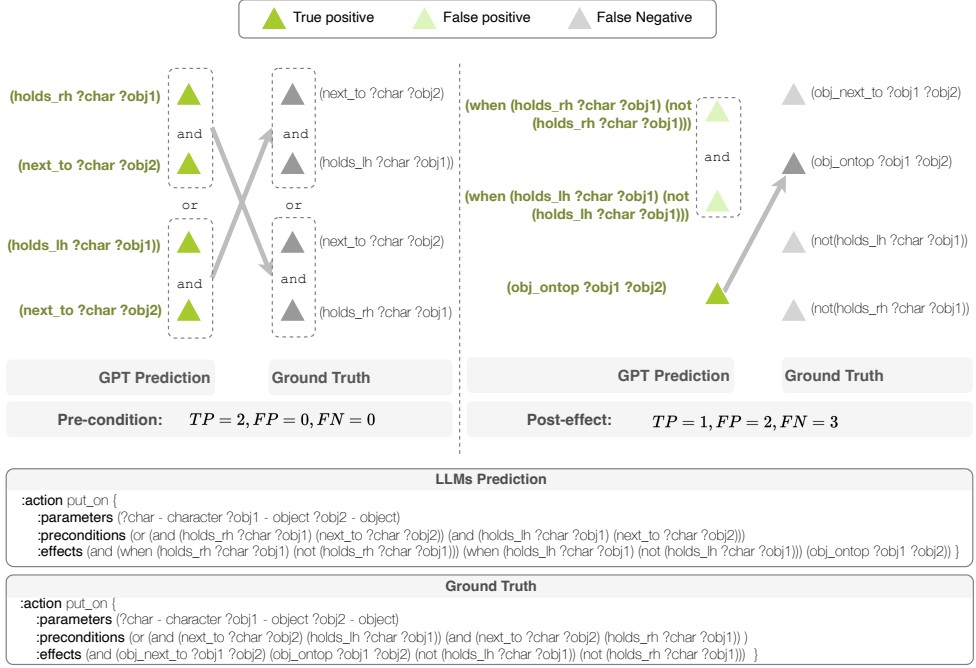

Figure 22: Transition Model evaluation metrics based on Bipartite Graph Matching for pre-conditions and post-effects.

The process begins with the preparation of input data, which includes goals, scripts, initial states, and final states provided in the dataset. Initially, predicates representing relations and properties within the domain are annotated. Based on these annotated predicates, PDDL (Planning Domain Definition Language) operators are defined. Subsequently, a comprehensive PDDL domain file is constructed, encapsulating the full set of predicates, including states, relations, and properties necessary for the planning tasks. Next, relevant objects are identified from the script, initial state, and final states. These objects are used to ground the goals into specific goals for each task. The problem file is formulated using the relevant objects, initial states, and grounded goals.

With the domain and problem files prepared, the LLM is tasked with predicting the transition model, that is, the action preconditions and effects. The inputs to the LLM include the predicate list, problem file, action name, and action parameters, and the output is the actions body predicted by LLM.

The output evaluation involves several key steps to assess the accuracy and reliability of the predicted transition model. First, the predicted preconditions and effects are extracted from the LLM-generated actions. These predicates are categorized to facilitate detailed analysis of the model's performance. The evaluation protocol uses several metrics to assess the performance of the LLMs. The logical scoring function evaluates the similarity between the predicted and gold action sequences by comparing the logical structure of preconditions and effects. The success rate by planner measures the proportion of correctly predicted actions that exists a feasible solution to achieve the desired goals under PDDL planner[79][‡]. Sensitivity analysis demonstrates how difficult it is for LLM to predict specific actions in different tasks.

---

**Logic Form Accuracy function(expr1, expr2)**

```
if expr1 and expr2 are literals:
    return expr1 == expr2
if type(expr1) != type(expr2):
    return 0
if isinstance(expr1, Not):
    return match_expressions(expr1.expression, expr2.expression)
if isinstance(expr1, When):
    if match_expressions(expr1.condition, expr2.condition)
    and match_expressions(expr1.consequence, expr2.consequence):
        return 1
    return 0
if isinstance(expr1, (Exists, Forall)):
    if quantifier are the same:
        return match_expressions(expr1.body, expr2.body)
    return 0
if isinstance(expr1, (And, Or)):
    adj_matrix = np.zeros((len(expr1.args), len(expr2.args)), dtype=int)
    for i in range(size1):
        for j in range(size2):
            adj_matrix[i, j] = match_expressions(sub1[i], sub2[j])
    match_result = maximum_bipartite_matching(sparse_matrix, perm_type='column')
    total_match = sum(adj_matrix[i, match_result[i]]
    for i in range(size1) if match_result[i] != -1)
    max_possible_match = min(size1, size2)
    return total_match / max_possible_match if max_possible_match > 0 else 0
```

---

**Logic Form Accuracy.** In PDDL, logical connectives such as *and*, *or*, *not*, *when*, *forall*, and *exists* are used to define preconditions and effects. Logical matching evaluates the similarity between the predicted and ground truth preconditions or effects by parsing them into clauses of disjunctive logical form and performing bipartite matching.

Given a set of predicted clauses $P = \{p_1, p_2, \ldots, p_n\}$ and a set of ground truth clauses $G = \{g_1, g_2, \ldots, g_m\}$, we define an adjacency matrix $A$ where $A[i, j] = \text{match}(p_i, g_j)$ represents the similarity between the $i$-th predicted clause and the $j$-th ground truth clause. The function $\text{match}(p, g)$ returns 1 if the clauses are identical, and 0 otherwise. If the ground truth clause $g_j$ is empty, $\text{match}(p, g) = 1$ if and only if $p_i$ is empty as well.

For clauses containing connectives, the clauses are recursively expanded and bipartite matching is conducted on the expanded nodes. If $p$ and $g$ are literals, the match function compares their equivalence directly. If either clause contains a connective, the clause is further decomposed and bipartite matching is applied to the sub-clauses.

The evaluation metrics for logical matching include precision, recall, and F1-score. True Positives (TP) are the matched clauses, False Positives (FP) are the unmatched clauses in the predicted set, and False Negatives (FN) are the unmatched clauses in the ground truth set.

The process begins by parsing the preconditions or effects into disjunctive logical form and constructing the adjacency matrix $A$ based on clause similarity. Bipartite matching is performed to find the optimal pairing of predicted and ground truth clauses. The matching process is recursive, expanding

---

[‡]https://github.com/ronuchit/pddlgym_planners

nested clauses and applying bipartite matching at each level. If a sub-clause does not match, the entire clause is considered a non-match.

Finally, the precision, recall, and F1-score are calculated using the formulas above, providing a robust measure of the LLM's accuracy in modeling logical relationships within PDDL tasks.

**Success Rate by External Planners.** The planner success rate evaluates the success of the LLM in generating correct and executable action sequences for each task category. The overall success rate across all categories is also reported to provide a comprehensive assessment.

Given a set of programs $P = \{p_1, p_2, \ldots, p_n\}$ and a corresponding set of categories $C = \{c_1, c_2, \ldots, c_k\}$, let $P_{c_j} \subseteq P$ denote the subset of programs belonging to category $c_j$. The success rate for category $c_j$ is defined as:

$$\text{SuccessRate}(c_j) = \frac{\sum_{p \in P_{c_j}} \text{IsSuccessful}(p)}{|P_{c_j}|}$$

where IsSuccessful($p$) is a binary function that returns 1 if the planner successfully generates an executable action sequence for program $p$, and 0 otherwise.

The overall success rate across all categories is given by:

$$\text{OverallSuccessRate} = \frac{\sum_{j=1}^{k} \sum_{p \in P_{c_j}} \text{IsSuccessful}(p)}{\sum_{j=1}^{k} |P_{c_j}|}$$

### D.5 Average Performance

To measure the overall performance of an LLM for all four ability modules, we use the following equation to calculate the accuracy:

$$\text{AveragePerf} = \frac{1}{4}(\text{Goal}_{F_1} + \text{Subgoal}_{\text{Task SR}} + \text{ActSeq}_{\text{Task SR}} + 0.5 * (\text{Trs}_{F_1} + \text{Trs}_{\text{Planner SR}}))$$

where $\text{Goal}_{F_1}$ represents the $F_1$ score of Goal Interpretation, $\text{Subgoal}_{\text{Task SR}}$ represents the task success rate of Subgoal Decomposition, $\text{ActSeq}_{\text{Task SR}}$ represents the task success rate of Action Sequencing, and $\text{Trs}_{F_1}$, $\text{Trs}_{\text{Planner SR}}$ represent $F_1$ score and planner success rate of Transition Modeling respectively.

## E Full Results with 18 models

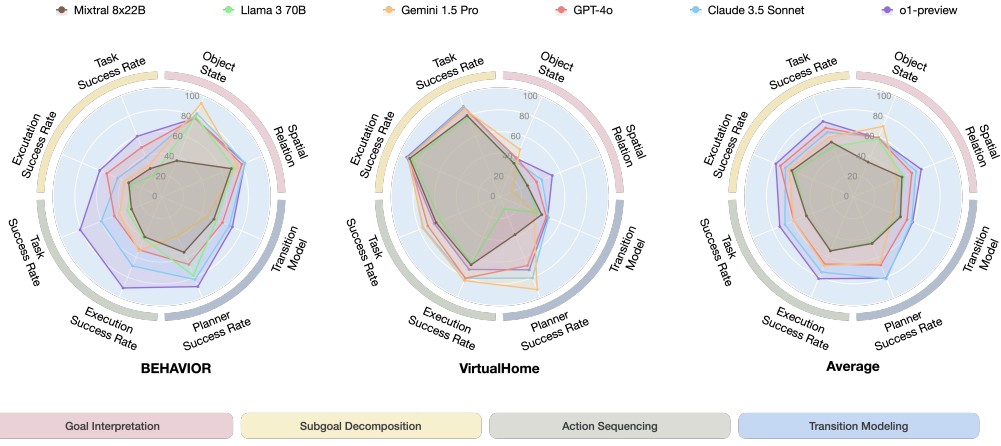

Figure 23: Performance Overview: Tasks in BEHAVIOR require longer sequences of actions and complex subgoal decomposition, while VirtualHome presents greater challenges in understanding environmental states.

In this section, we provide the full results and analysis for each ability module using 18 different models, and provide a further analysis on the potential coorelation with factors in Appendix E.5.

## E.1 Goal Interpretation

Table 9 examines the performance of various LLMs in translating natural language task instructions into actionable symbolic goals within two different simulators: VirtualHome (V) and BEHAVIOR (B). Additionally, Figures 24 and 25 show detailed qualitative error cases based on error type and goal type. Below are the detailed result analysis and error case studies.

### E.1.1 Result Analysis

Table 9 contains the quantitative benchmarking results of the Goal Interpretation ability module for both Behavior and VirtualHome simulators. The results are mainly broken down into three categories: State Goals or Unary Goals (which describe the goal state of just one object), Spatial Goals or Binary Goals (which describe the spatial relationship goal of two interacting objects), and Action Goals specific to the VirtualHome simulator (which describe the actions that an agent should finish to complete the task). Finally, we include an Overall section, where all goals' performances are combined with a micro average.

In Table 9, our evaluation metrics consist of precision, recall, and F1 score for each goal type and overall performance. In the case of the goal interpretation task, precision measures how unlikely a model is to make false-positive goal predictions given reasonable instructions and task information, whereas recall measures how unlikely a model is to leave out ground truth goals in their prediction. The F1 score combines these two metrics and reflects the overall model performance for each category.

Based on the results in Table 9, o1-preview achieves top overall goal interpretation performance (F1 score) in the VirtualHome simulator over other LLMs, whereas Claude-3.5 Sonnet achieves the highest performance (F1 score) in the BEHAVIOR simulator. Gemini 1.5 Pro also demonstrates very high performance in both simulators with very close precision, recall, and F1 scores compared to the top performer on each metric. Among open-source models, the top performer by a significant margin is the Llama-3 70B Instruct model, leading in overall F1 scores for both simulators.

For individual goal types, the performance trends are less uniform: o1-preview and Claude-3.5 Sonnet both demonstrate very strong spatial reasoning abilities, reaching top overall F1 scores in Behavior and VirtualHome's spatial goal categories, respectively. Gemini 1.5 Pro leads in terms of overall performance in VirtualHome action goals and Behavior state goals, while the Cohere Command R model leads in terms of VirtualHome state goals.

### E.1.2 Case Studies

In Figure 24, we show detailed examples of Behavior and VirtualHome goal interpretation errors broken down into two main categories: goal prediction errors and grammar errors.

Grammar errors can be further broken down into hallucination errors (such as object/state/action hallucination) and format errors (such as JSON parsing errors). In our empirical experiments, we find that given a reasonably clear prompt with in-context examples, commercial LLMs such as the GPT and Claude family models are extremely unlikely to display format errors, while open-source models such as Llama 3 8B and Mixtral 8x22B tend to make a small number of mistakes.

Goal prediction errors can be further broken down into missing goals and false positive goals, which can be quantitatively measured by recall and precision scores, respectively. Empirically, we find two kinds of errors to be common among both open-source and proprietary LLMs:

(1) As shown in the false positive goal error sections of Figure 24 (a) and (b), LLMs tend to output intermediate goal states in place of final goal predictions. This phenomenon can be intuitively explained by the LLMs' reliance on Chain-of-Thought Reasoning [80] for question answering. While our carefully designed system and user prompts restrict LLMs to directly output the predicted goals without any explicit intermediate explanation, models may still try to perform such intermediate reasoning steps in their structured output, thus leading to such errors.

(2) As shown in the missing goal error sections of Figure 24 (a) and (b), LLMs tend to leave out simple object spatial relationships in their output. For the task "serving a meal," with ground truth goal condition ONTOP(*chicken.0*, *plate.2*) and ONTOP(*plate.2*, *table.1*), GPT-4o (along with many other models) mistakenly predicts ONTOP(*chicken.0*, *table.1*). While the expression "chicken ontop of table" is acceptable in a conversational setting, it presents a completely wrong set of physical relationships between the objects chicken, plate, and table. This type of error is common in both Behavior and VirtualHome simulators for various models, highlighting a significant issue of imprecise spatial relationship description in applying LLMs for embodied planning and robotic control, where physical precision is crucial for task success.

---

**Takeaways of Goal Interpretation**

(1) **Top-performing Models:** o1-preview and Claude 3.5 Sonnet achieves top overall goal interpretation performance (F1-score) in VirtualHome and BEHAVIOR simulators, respectively. Whereas Gemini 1.5 Pro achieves the highest accuracy in VirtualHome .

(2) **Grammar Errors:** Current SOTA proprietary LLMs generally make very little to no grammar errors in goal interpretation, whereas top open-source LLMs including Llama 3 70B Instruct tend to suffer more from format/parsing errors and object/state hallucination.

(3) **Goal Prediction Errors:** Both open-source and proprietary LLMs suffer significantly from two common types of goal prediction errors:

  (a) Generating intermediate goals in place of final goals (when asked to directly output final goals), resulting in false positive goal predictions.

  (b) Omitting conversationally uncommon spatial relationship goals that are essential for precise robotic control, resulting in missing goals (reporting bias).

---

Table 9: All goal evaluation results (%) for goal interpretation

| Model Name | State | | | | | | Spatial | | | | | | Action | | | | | | Overall | | | | | |
|---|---|---|---|---|---|---|---|---|---|---|---|---|---|---|---|---|---|---|---|---|---|---|---|---|
| | *Precision* | | *Recall* | | *F1* | | *Precision* | | *Recall* | | *F1* | | *Precision* | | *Recall* | | *F1* | | *Precision* | | *Recall* | | *F1* | |
| | V | B | V | B | V | B | V | B | V | B | V | B | V | B | V | B | V | B | V | B | V | B | V | B |
| Claude-3 Haiku | 21.8 | 22.8 | 58.9 | 93.5 | 31.8 | 36.7 | 24.2 | 64.5 | 50.8 | 64.6 | 32.8 | 64.6 | 12.2 | - | 95.7 | - | 21.6 | - | 18.0 | 41.5 | 63.2 | 71.2 | 28.0 | 52.5 |
| Claude-3 Sonnet | 23.3 | 36.8 | 57.1 | 88.9 | 33.1 | 52.0 | 26.6 | 76.2 | 53.0 | 79.8 | 35.5 | 77.9 | 12.4 | - | 85.8 | - | 21.7 | - | 19.3 | 60.2 | 61.5 | 81.9 | 29.4 | 69.4 |
| Claude-3 Opus | 27.0 | 72.6 | 66.9 | 93.5 | 38.5 | 81.7 | 22.6 | 75.2 | 46.8 | 79.2 | 30.5 | 77.1 | 14.5 | - | 92.6 | - | 25.1 | - | 20.7 | 72.2 | 65.0 | 82.5 | 31.4 | 77.0 |
| Claude-3.5 Sonnet | 25.3 | 74.0 | 60.9 | 94.8 | 35.8 | 83.1 | 31.1 | **84.4** | **63.8** | 81.3 | 41.8 | **82.9** | 14.0 | - | **98.8** | - | 24.5 | - | 21.7 | **81.1** | **69.6** | 84.4 | 33.0 | **82.7** |
| Cohere Command R | **51.1** | 7.7 | **69.6** | 31.4 | **58.9** | 12.4 | 34.5 | 56.8 | 21.3 | 55.0 | 26.3 | 55.9 | 3.6 | - | 38.9 | - | 6.5 | - | 27.4 | 28.2 | 55.7 | 49.6 | 36.7 | 36.0 |
| Cohere Command R+ | 20.9 | 23.3 | 52.0 | 79.1 | 29.8 | 36.0 | 17.9 | 66.7 | 15.2 | 61.5 | 16.4 | 64.0 | 10.4 | - | 82.6 | - | 18.5 | - | 14.9 | 42.0 | 44.5 | 65.5 | 22.4 | 51.2 |
| Gemini 1.0 Pro | 25.3 | 27.4 | 57.9 | 81.1 | 34.9 | 41.0 | 17.0 | 75.2 | 20.6 | 70.4 | 18.6 | 72.7 | 9.9 | - | 68.7 | - | 17.2 | - | 16.2 | 51.0 | 45.2 | 72.8 | 23.8 | 60.0 |
| Gemini 1.5 Flash | 23.6 | 55.8 | 57.9 | 94.1 | 33.5 | 70.1 | 19.8 | 76.6 | 21.1 | 76.7 | 20.5 | 76.7 | 13.5 | - | 90.1 | - | 23.5 | - | 18.2 | 69.7 | 50.8 | 80.7 | 26.8 | 74.8 |
| Gemini 1.5 Pro | 47.2 | **94.0** | 47.5 | 92.8 | 47.3 | **93.4** | 42.0 | 74.4 | 7.2 | 76.7 | 12.4 | 75.6 | **24.1** | - | 81.4 | - | 37.2 | - | 33.6 | 78.8 | 39.3 | 80.4 | 36.2 | 79.6 |
| GPT-3.5-turbo | 22.4 | 52.0 | 50.0 | 66.7 | 30.9 | 58.5 | 8.5 | 51.5 | 18.8 | 46.9 | 11.7 | 49.1 | 15.2 | - | 60.5 | - | 24.4 | - | 15.7 | 49.5 | 40.5 | 51.4 | 22.7 | 50.4 |
| GPT-4-turbo | 28.6 | 70.4 | 58.5 | 86.9 | 38.4 | 77.8 | 24.7 | 77.5 | 32.9 | 76.4 | 28.2 | 76.9 | 19.0 | - | 82.1 | - | 30.9 | - | 24.0 | 75.6 | 53.8 | 78.8 | 33.2 | 77.2 |
| GPT-4o | 29.0 | 67.1 | 60.0 | 94.8 | 39.1 | 78.6 | 31.5 | 81.1 | 43.6 | 78.5 | 36.6 | 79.8 | 20.5 | - | 85.8 | - | 33.1 | - | 26.4 | 76.5 | 59.1 | 82.2 | 36.5 | 79.2 |
| Llama 3 8B Instruct | 21.7 | 17.3 | 54.4 | 80.4 | 31.0 | 28.4 | 14.0 | 51.4 | 7.4 | 20.8 | 9.7 | 29.6 | 11.1 | - | 79.4 | - | 19.4 | - | 15.5 | 24.1 | 41.9 | 34.3 | 22.6 | 28.3 |
| Llama 3 70B Instruct | 23.9 | 69.5 | 61.2 | **95.4** | 34.3 | 80.4 | 22.6 | 70.0 | 37.5 | 73.3 | 28.2 | 71.6 | 11.2 | - | 88.8 | - | 19.8 | - | 17.5 | 64.7 | 58.0 | 78.3 | 26.9 | 70.9 |
| Mistral Large | 23.6 | 63.5 | 59.1 | 92.2 | 32.8 | 75.2 | 23.7 | 75.1 | 40.3 | 76.2 | 29.8 | 75.6 | 11.2 | - | 84.0 | - | 19.7 | - | 17.5 | 69.6 | 57.1 | 79.8 | 26.8 | 74.3 |
| Mixtral 8x22B MoE | 23.6 | 22.9 | 56.9 | 83.7 | 33.4 | 36.0 | 22.2 | 70.7 | 36.3 | 67.7 | 27.5 | 69.2 | 11.2 | - | 94.8 | - | 20.0 | - | 17.4 | 44.4 | 56.2 | 71.3 | 26.6 | 54.7 |
| o1-mini | 26.3 | 63.8 | 58.6 | 90.8 | 36.3 | 74.9 | 30.4 | 77.3 | 39.9 | 76.5 | 34.5 | 76.9 | 13.5 | - | 56.8 | - | 21.8 | - | 22.4 | 73.3 | 51.3 | 79.8 | 31.2 | 76.4 |
| o1-preview | 28.2 | 66.8 | 60.3 | 94.8 | 38.5 | 78.4 | **44.9** | 82.9 | 62.4 | **82.7** | **52.2** | 82.8 | 26.0 | - | 81.5 | - | 39.5 | - | 31.8 | 78.1 | 65.4 | **85.4** | **42.7** | 81.6 |

## E.2 Subgoal Decomposition

### E.2.1 Result Analysis

Table 10 examines the performance of various LLMs in decomposing high-level goals into actionable subgoals within two different simulators: VirtualHome (V) and BEHAVIOR (B). Additionally, Table 11 investigates fine-grained goal satisfaction for each LLM. Below are the results and related error analysis.

**VirtualHome** **(1) Executable and Goal Success Rate.** The top-performing model is **o1-preview**, achieving the highest task success rate of **89.4%** in VirtualHome, closely followed by **Gemini 1.5**

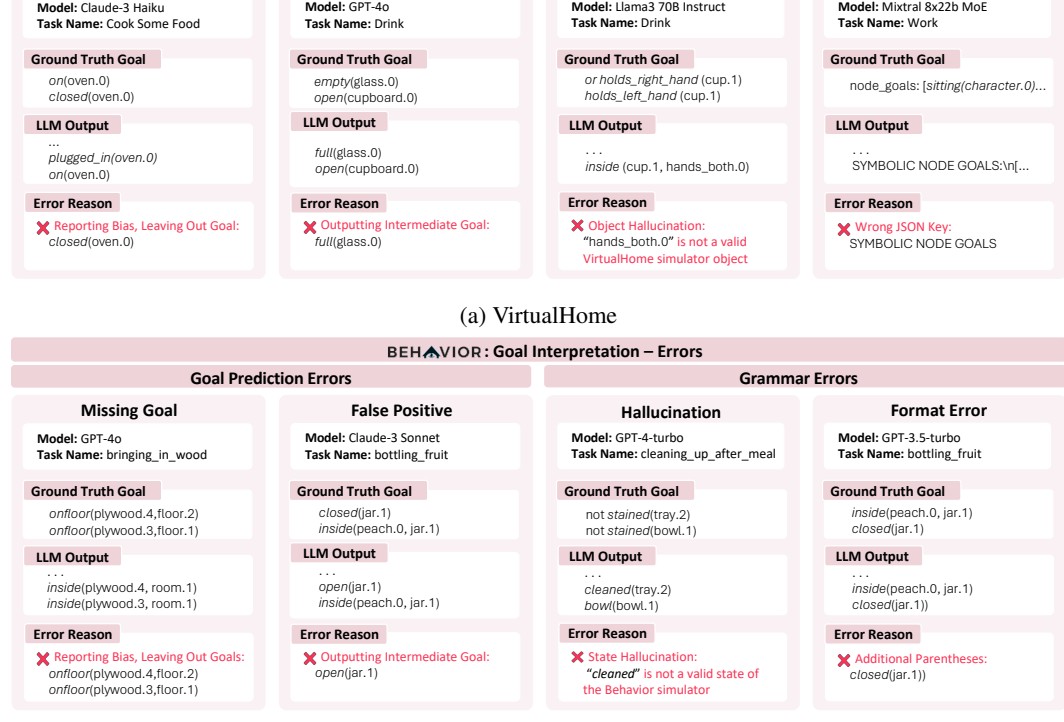

Figure 24: Goal evaluation error examples for goal interpretation.

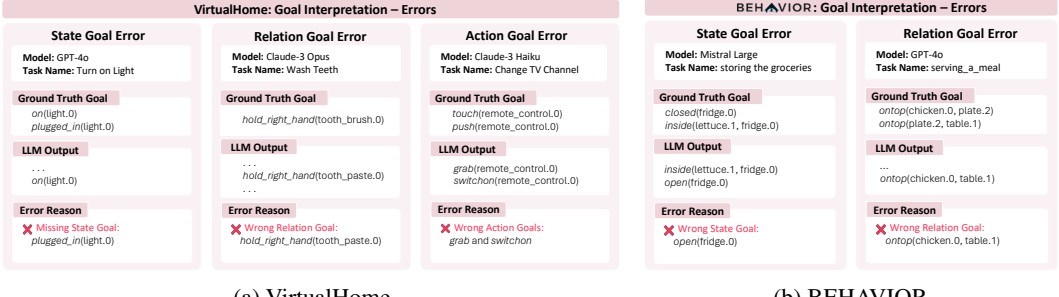

Figure 25: Goal satisfaction error examples for goal interpretation

**Flash** and **Claude-3.5 Sonnet**, both with success rates of **89.1%**. For execution success rate, both **GPT-4-turbo** and **Gemini 1.5 Flash** achieve the highest performance with rates of **94.1%**. Notably, o1-preview demonstrates superior performance overall, suggesting robust subgoal decomposition that aligns well with the VirtualHome simulator's requirements. Conversely, models like **Llama 3 8B**, with a success rate of **48.8%**, illustrate the challenges of effective subgoal decomposition.

**(2) Grammar Errors.** The occurrence of grammar errors is notably low across the board. Specifically, most SOTA LLMs make no errors in parsing and predicate-argument numbers. This indicates that current SOTA LLMs can follow the subgoal syntax. However, some models are prone to hallucinate non-existing predicates. Specifically, GPT-4o tends to hallucinate the action *POUR* when dealing with the task 'make coffee', which is not defined in the subgoal decomposition setting.

**(3) Runtime Errors.** In terms of runtime errors, compared with wrong temporal order and affordance errors, LLMs are prone to make missing steps and additional steps errors, while additional step errors are the most critical. (1) Wrong temporal order. Closed-sourced LLMs perform well in understanding the temporal requirement among subgoals, while open-sourced LLMs like Llama 3 8B will make more temporal order errors. For the wrong cases, LLMs tend to ignore the sitting or lying state of the agent and fail to call action *STANDUP* before they apply some actions requiring a standing state, yet this state has been achieved earlier. (2) Missing step. LLMs sometimes are prone to fail to satisfy some preconditions when applying actions. Among all LLMs, GPT-3.5-turbo performs worst in this error type. Specifically, it tends to ignore opening a closed object before fetching something inside it. (3) Affordance. Overall, Llama 3 8B performs much worse than other LLMs, meaning it cannot understand the semantics very well. (4) Additional steps. All LLMs are prone to produce additional step errors, even for SOTA LLMs like GPT-4o and Claude-3 Opus. This is mainly because in the initial scene state, some of the goals have already been achieved, yet LLMs still prefer to plan the satisfied goals in their output.

**(4) Goal Satisfaction Analysis.** In Table 11, we observe that LLMs perform well in understanding and satisfying state goals in VirtualHome. Models like **GPT-4-turbo**, **Gemini 1.5 Flash**, and **o1-preview** achieve state goal success rates of over **91%**. This is because state goals are usually isolated from other objects with fewer logical requirements. However, achieving relation and action goals is more challenging. For instance, **o1-preview** achieves the highest relation goal success rate of **88.3%** and the highest action goal success rate of **92.6%**, while weaker models like **Llama 3 8B** achieve less than **70%** in these categories. This suggests that stronger LLMs can better understand the semantics of actions and relations, whereas weaker LLMs struggle with the complexity of these goals. Generally, most SOTA LLMs are capable of achieving over **80%** of the goals defined in our annotated data, likely due to the relatively simple and straightforward goals in RobotHow.

**BEHAVIOR** **(1) Executable and Goal Success Rate. o1-preview** stands out with the highest task success rate at **57.0%** and an execution success rate of **62.0%** in BEHAVIOR, followed by **GPT-4o** with a task success rate of **49.0%** and an execution success rate of **55.0%**. Overall, all LLMs face challenges in achieving high performance in BEHAVIOR. This could be attributed to the more complex task representations in BEHAVIOR compared to VirtualHome. For example, most tasks in BEHAVIOR involve quantifiers like *forall* and *forpairs* with complex spatial or temporal requirements, whereas VirtualHome tasks have simpler goal definitions.

**(2) Grammar Errors.** Most LLMs are proficient in generating grammatically correct subgoal plans with no parsing errors and correct numbers of predicate parameters. There are a few exceptions, though. For instance, **Cohere Command R** has a parsing error rate of **23.0%** in BEHAVIOR, mostly due to illegal tokens or syntax for the LTL parser. Additionally, models like **Llama 3 8B** tend to hallucinate non-existing objects in the scene, leading to higher hallucination error rates.

**(3) Runtime Errors.** The most prevalent runtime errors in BEHAVIOR across various LLMs are due to missing steps. In fact, over half of the total cases involve such errors in the majority of LLMs. Even **o1-preview**, which has the lowest missing step error rate, encounters these errors in **25.0%** of cases. This can be attributed primarily to two reasons: First, the precondition checking in BEHAVIOR is stricter than that in VirtualHome . For instance, VirtualHome does not verify preconditions such as whether an agent is holding a cleaning tool before executing the *WASH* action. In contrast, BEHAVIOR requires such conditions to be satisfied before invoking a similar action like *CLEAN*. Secondly, the complexity of tasks in BEHAVIOR often leads LLMs to overlook seemingly trivial actions, such as opening a closed container before retrieving an item inside it. This issue persists even when a heads-up is included in the prompt. Interestingly, LLMs tend to make significantly fewer additional step errors in BEHAVIOR compared to VirtualHome . This is likely because the majority of task goals in BEHAVIOR are not satisfied in the initial state, making additional steps less frequent.

**(4) Goal Satisfaction Analysis.** From the statistics in Table 11, it is evident that LLMs do not perform as well in achieving state and relation goals in BEHAVIOR. The discrepancy arises because most state and relation goals are encapsulated within quantifiers. Consequently, quantifiers such as *forall* or *forpairs* tend to fail if even a single state or relation goal is not met. For example, **o1-preview** achieves a state goal success rate of **56.5%** and a relation goal success rate of **69.4%**, which are the highest among the models but still significantly lower than in VirtualHome. Additionally, since most quantifiers can have multiple solutions, it is challenging to obtain accurate statistics for state and

relation goals within quantifiers. Our evaluation metric considers quantifiers as a combined entity for both state and relation goals, contributing to the lower success rates observed.

Table 10: All trajectory evaluation results (%) for subgoal decomposition.

| Model | Goal Evaluation | | | | Trajectory Evaluation | | | | | | | | | | | | | |
| | Task SR | | Execution SR | | Grammar Error (↓) | | | | | | Runtime Error (↓) | | | | | | | |
| | | | | | Parsing | | Hallucination | | Predicate-Arg Num | | Wrong Order | | Missing Step | | Affordance | | Additional Step | |
| | V | B | V | B | V | B | V | B | V | B | V | B | V | B | V | B | V | B |
|---|---|---|---|---|---|---|---|---|---|---|---|---|---|---|---|---|---|---|
| Claude-3 Haiku | 78.4 | 30.0 | 82.8 | 35.0 | 0.3 | **0.0** | 2.4 | 1.0 | 1.8 | **0.0** | 1.8 | 3.0 | 2.7 | 58.0 | 8.3 | 3.0 | 20.4 | 3.0 |
| Claude-3 Sonnet | 83.1 | 39.0 | 86.4 | 43.0 | 0.0 | 0.0 | 1.8 | 2.0 | 0.0 | 2.0 | 0.6 | 3.0 | 2.7 | 51.0 | 8.6 | 1.0 | 33.7 | 3.0 |
| Claude-3 Opus | 87.0 | 41.0 | 90.0 | 47.0 | 0.3 | **0.0** | 3.6 | 3.0 | 0.0 | 0.0 | 1.2 | 5.0 | 3.0 | 45.0 | 2.4 | **0.0** | 16.0 | 6.0 |
| Claude-3.5 Sonnet | 89.1 | 39.0 | 92.0 | 44.0 | 0.0 | 0.0 | 1.8 | 1.0 | 0.0 | 0.0 | 1.5 | 11.0 | 2.7 | 44.0 | 2.1 | **0.0** | 24.6 | 4.0 |
| Gemini 1.0 Pro | 70.4 | 24.0 | 84.6 | 33.0 | 0.6 | 2.0 | 3.3 | 4.0 | 2.4 | **0.0** | 1.2 | 3.0 | 2.7 | 51.0 | 5.3 | 7.0 | **10.4** | 3.0 |
| Gemini 1.5 Flash | 89.1 | 34.0 | 94.1 | 42.0 | 0.0 | 2.0 | 1.5 | 1.0 | 0.0 | 0.0 | 0.6 | 2.0 | 3.9 | 53.0 | 0.0 | 0.0 | 13.3 | 3.0 |
| Gemini 1.5 Pro | 87.0 | 31.0 | 91.1 | 37.0 | 0.0 | 1.0 | 1.5 | 0.0 | 1.8 | 1.0 | 0.0 | 3.0 | 5.6 | 59.0 | 0.0 | 0.0 | 16.0 | 2.0 |
| GPT-3.5-turbo | 69.2 | 24.0 | 81.4 | 36.0 | 1.5 | 2.0 | 0.0 | 3.0 | 0.6 | **0.0** | 1.5 | 4.0 | 11.8 | 51.0 | 3.3 | 4.0 | 20.4 | 3.0 |
| GPT-4-turbo | 85.5 | 38.0 | **94.1** | 47.0 | 0.0 | 0.0 | 1.8 | 3.0 | 0.0 | 0.0 | 1.5 | 9.0 | **2.4** | 40.0 | 0.3 | 1.0 | 22.2 | 6.0 |
| GPT-4o | 88.8 | 49.0 | 90.2 | 55.0 | 0.0 | 0.0 | 6.2 | 3.0 | 0.0 | 0.0 | 1.2 | 6.0 | **2.4** | 36.0 | 0.0 | 0.0 | 15.7 | 5.0 |
| Cohere Command R | 71.3 | 15.0 | 79.6 | 25.0 | 2.1 | 23.0 | 3.9 | 10.0 | 0.9 | **0.0** | 1.5 | 0.0 | 6.2 | 37.0 | 5.9 | 5.0 | 14.5 | 4.0 |
| Cohere Command R+ | 79.0 | 25.0 | 83.7 | 37.0 | 1.5 | 2.0 | 4.5 | 4.0 | 2.1 | **0.0** | 0.9 | 4.0 | 7.7 | 52.0 | 2.7 | 1.0 | 16.0 | 6.0 |
| Mistral Large | 84.3 | 31.0 | 92.0 | 38.0 | 0.3 | 1.0 | 1.8 | 3.0 | 0.3 | **0.0** | 2.1 | 4.0 | 3.3 | 52.0 | 0.3 | 2.0 | 11.0 | 1.0 |
| Mixtral 8x22B MoE | 80.5 | 28.0 | 90.2 | 33.0 | 0.3 | **0.0** | 2.4 | 4.0 | 0.0 | **0.0** | 3.0 | 2.0 | 3.9 | 59.0 | 0.3 | 2.0 | 11.2 | **0.0** |
| Llama 3 8B | 48.8 | 21.0 | 58.0 | 29.0 | 0.6 | 2.0 | 2.4 | 11.0 | 0.6 | **0.0** | 6.8 | 6.0 | 5.0 | 44.0 | 26.6 | 8.0 | 18.3 | 7.0 |
| Llama 3 70B | 78.4 | 20.0 | 87.3 | 30.0 | 0.0 | 1.0 | 2.4 | 5.0 | 0.9 | 1.0 | 2.4 | 8.0 | 5.3 | 51.0 | 1.8 | 4.0 | 20.4 | 4.0 |
| o1-mini | 79.3 | 31.0 | 84.6 | 39.0 | 0.0 | 0.0 | 1.5 | 3.0 | 0.6 | 3.0 | 0.3 | 7.0 | 8.9 | 46.0 | 4.1 | 2.0 | 21.9 | 1.0 |
| o1-preview | **89.4** | **57.0** | 93.2 | **62.0** | 0.0 | 2.0 | 1.5 | 3.0 | 0.0 | **0.0** | 0.3 | 5.0 | 2.7 | **25.0** | 2.4 | 3.0 | 12.1 | 7.0 |

Table 11: All goal success results (%) for action sequencing and subgoal decomposition.

| Model | Action Sequencing | | | | | | | | Subgoal Decomposition | | | | | | | |
| | State Goal | | Relation Goal | | Action Goal | | Total | | State Goal | | Relation Goal | | Action Goal | | Total | |
| | V | B | V | B | V | B | V | B | V | B | V | B | V | B | V | B |
|---|---|---|---|---|---|---|---|---|---|---|---|---|---|---|---|---|
| Claude-3 Haiku | 71.2 | 27.0 | 53.9 | 38.7 | 41.9 | - | 58.9 | 35.5 | 89.4 | 26.0 | 82.2 | 34.8 | 71.6 | - | 83.1 | 32.4 |
| Claude-3 Sonnet | 80.2 | 41.0 | 68.3 | 59.8 | 38.5 | - | 66.5 | 54.6 | 89.1 | 37.0 | **89.3** | 49.8 | 83.3 | - | 88.0 | 46.3 |
| Claude-3 Opus | 57.2 | 45.0 | 77.8 | 53.0 | 54.7 | - | 62.7 | 50.8 | 92.4 | 43.0 | 88.6 | 41.6 | 83.3 | - | 89.1 | 42.0 |
| Claude-3.5 Sonnet | 87.8 | 63.0 | 83.3 | 62.4 | 60.8 | - | 79.9 | 62.6 | 92.9 | 41.0 | 88.6 | 39.5 | 87.0 | - | 90.1 | 39.9 |
| Gemini 1.0 Pro | 70.5 | 28.0 | 41.7 | 32.0 | 34.5 | - | 53.1 | 30.9 | 84.4 | 26.0 | 61.5 | 31.1 | 72.8 | - | 73.5 | 29.7 |
| Gemini 1.5 Flash | 86.0 | 34.0 | 73.3 | 50.0 | 47.3 | - | 72.8 | 45.6 | **93.5** | 44.0 | 88.3 | 36.0 | 92.0 | - | **91.3** | 38.2 |
| Gemini 1.5 Pro | 85.6 | 41.0 | 76.7 | 43.2 | 62.2 | - | 77.2 | 42.6 | 91.2 | 31.0 | 72.5 | 37.1 | 89.5 | - | 83.9 | 35.4 |
| GPT-3.5-turbo | 49.3 | 20.0 | 25.0 | 22.6 | 33.8 | - | 38.3 | 21.9 | 84.7 | 28.0 | 54.4 | 28.5 | 64.8 | - | 69.4 | 28.3 |
| GPT-4-turbo | 75.9 | 39.0 | 75.0 | 39.5 | 48.6 | - | 69.0 | 39.3 | **93.5** | 45.0 | 84.2 | 46.1 | 90.7 | - | 89.5 | 45.8 |
| GPT-4o | 87.1 | 49.0 | 76.1 | 45.5 | 56.1 | - | 76.2 | 46.5 | 92.1 | 50.0 | 84.2 | 53.2 | **93.2** | - | 89.4 | 52.3 |
| Cohere Command R | 46.8 | 20.0 | 38.9 | 25.9 | 48.6 | - | 44.9 | 24.3 | 85.3 | 20.0 | 67.4 | 21.4 | 60.5 | - | 73.6 | 21.0 |
| Cohere Command R+ | 68.7 | 28.0 | 54.4 | 32.0 | 50.7 | - | 60.1 | 30.9 | 89.4 | 34.0 | 66.8 | 29.6 | 75.9 | - | 78.3 | 30.8 |
| Mistral Large | 88.5 | 38.5 | **81.1** | 41.2 | **70.9** | - | 82.0 | 40.4 | 92.9 | 33.0 | 71.5 | 35.6 | 90.1 | - | 84.4 | 34.9 |
| Mixtral 8x22B MoE | 69.4 | 30.0 | 62.2 | 36.8 | 57.4 | - | 64.4 | 35.0 | 92.1 | 30.0 | 74.8 | 34.1 | 87.7 | - | 84.8 | 33.0 |
| Llama 3 8B | 34.2 | 16.0 | 16.7 | 23.7 | 11.5 | - | 23.4 | 21.6 | 68.8 | 21.0 | 54.7 | 23.6 | 50.0 | - | 59.8 | 22.9 |
| Llama 3 70B | 61.5 | 31.0 | 68.3 | 45.5 | 42.6 | - | 58.9 | 41.5 | 93.2 | 25.0 | 63.4 | 27.7 | 82.7 | - | 80.0 | 27.0 |
| o1-mini | **88.5** | 64.0 | 72.2 | 66.9 | 57.4 | - | 76.1 | 66.1 | 89.7 | 28.0 | 68.8 | 38.0 | 81.5 | - | 80.3 | 35.3 |
| o1-preview | 80.9 | **89.5** | 65.0 | **84.4** | 46.6 | - | 67.8 | **85.8** | 91.8 | **56.5** | 88.3 | **69.4** | 92.6 | - | 90.6 | **65.9** |

### E.2.2 Case Studies

In this section, we would like to investigate several error examples made in subgoal decomposition by LLMs. Specifically, we explore in both VirtualHome and BEHAVIOR .

**Grammar Errors**   Figure 26 illustrates parsing errors, predicate parameter length errors, and hallucination errors for both VirtualHome and BEHAVIOR . Specifically: (1) Parsing errors: the VirtualHome example duplicates '.' in an object name, whereas the BEHAVIOR example repeatedly outputs ')'. (2) Predicate parameter length errors: the VirtualHome example outputs two parameters for the action *GRAB*, while the BEHAVIOR example outputs three parameters for the state *nextto*. (3) Hallucination errors: the objects *kitchen.1* and *countertop.84* do not exist in the provided environment.

**Runtime Errors**   0Figure 27 demonstrates four types of runtime errors for both VirtualHome and BEHAVIOR : wrong order, missing step, affordance, and additional step errors. Specifically: (1) Wrong order errors: In the VirtualHome example, the agent needs to plug in *light.411* but cannot do so because both hands hold *novel.1000* and *spectacles.1001*, whereas they were previously empty. Similarly, in the BEHAVIOR example, the agent drops *soap.0* before cleaning *top_cabinet.25* despite having held the soap beforehand. (2) Missing step errors: In the VirtualHome scenario, the agent

is required to retrieve *food.1000*, but it is inside the closed *freezer.289*. The BEHAVIOR example presents the agent tasked with placing *vidalia_onion.67* on *countertop.26*, yet the onion is inside a closed container. (3) Affordance errors: For VirtualHome , the agent is asked to plug in *computer.417* and *mouse.413*, but these items lack plugs in VirtualHome . In the BEHAVIOR example, the agent is asked to hold *fridge.97* and slice *carving_knife.0*. However, the fridge is too large to hold, and the knife cannot be sliced. (4) Additional step errors: In VirtualHome , the agent is instructed to plug in *washing_machine.1001*, a step already fulfilled initially. Similarly, in BEHAVIOR , the agent is asked to open *carton.0*, which was already open in a previous state.

**Goal Satisfaction Errors** Figure 28 depicts missing state, and missing relation errors for both VirtualHome and BEHAVIOR , and includes an example of a missing goal action error for VirtualHome . Specifically: (1) Missing state errors: In the VirtualHome example, the generated subgoal plan fails to turn on *light.1002*, which is necessary for achieving the final goal. Similarly, in the BEHAVIOR example, the subgoal plan fails to open *window.76* and *window.81*, which are also required for the final goal. (2) In VirtualHome , after execution, the agent does not place *clothes_pants.1001* on top of *washing_machine.1000*, as required by the final goal. In the BEHAVIOR example, LLMs incorrectly instruct the agent to first put *plywood.78* next to *plywood.79*, and then put *plywood.79* next to *plywood.80*. Once the second state is achieved, the state *nextto(plywood.78, plywood.79)* is no longer satisfied because *plywood.79* has been moved. (3) Missing goal action error: In the VirtualHome example, the agent is required to perform the action *DRINK*, but this action is missing in the generated subgoal plan.

---

### Takeaways of Subgoal Decomposition

(1) **Top-performing Models:** o1-preview demonstrates superior performance in both VirtualHome and BEHAVIOR simulator compared with other SOTA LLMs, achieving the highest task success rates of 89.4% and 57.0%. Claude-3.5 Sonnet and Gemini 1.5 Flash also demonstrate high performance in VirtualHome with a success rate close to 90%.

(2) **Grammar and Runtime Errors:** State-of-the-art models generally avoid grammar errors but can hallucinate actions and objects. The most common runtime errors in VirtualHome are additional steps, whereas most of runtime errors in BEHAVIOR are missing steps.

(3) **Goal Satisfaction:** Stronger LLMs like o1-preview and GPT-4o show higher accuracy in action goal success rates in VirtualHome compared to weaker models like Llama 3 8B. However, achieving state and relation goals in BEHAVIOR is challenging due to more complex task representations and stricter precondition checks.

(4) **Impact of Task Complexity In Different Simulators:** Overall LLM performance is lower in BEHAVIOR compared with VirtualHome due to complex task representations involving quantifiers like *forall* and *forpairs*, which articulate complex temporal and spatial requirements.

---

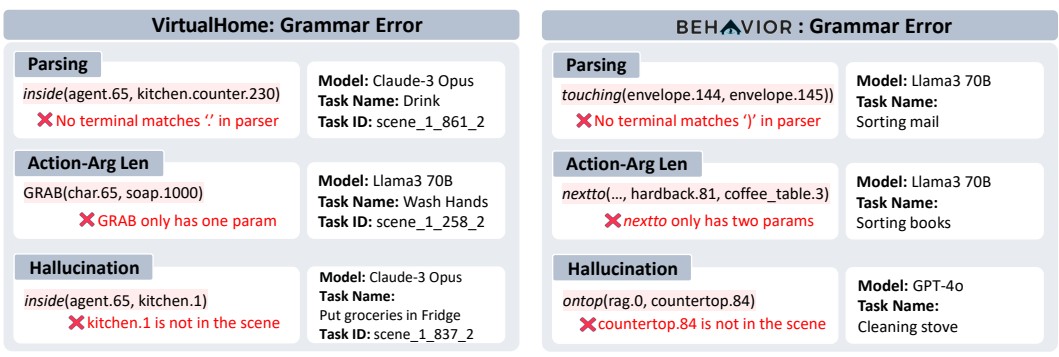

(a) VirtualHome          (b) BEHAVIOR

Figure 26: Parsing error examples for subgoal decomposition.

## E.3 Action Sequencing

The full results for the trajectory evaluation of action sequencing are presented in Table 12. Detailed results for goal satisfaction are shown in Table 11. The results from two simulators are provided separately, with *V* representing VirtualHome and *B* denoting BEHAVIOR .

**(1) Executable and Task Success Rate.** For VirtualHome , Mixtral Large achieved the best performance in both execution and task success rates, which is surprising as it outperforms several LLMs considered more powerful, such as o1-preview, Cluade-3 Opus, GPT-4o, and Gemini-1.5 Pro. Similarly, Cluade-3 Sonnet outperforms Cluade-3 Opus in execution success rate. We hypothesize that this is because most tasks in VirtualHome require only 3-5 actions, though they may involve a large number of objects in the scene. Therefore, it's crucial to capture key information and focus only on task-relevant objects. In such cases, some LLMs may "overthink" and act on irrelevant objects, while Mixtral Large take a more straightforward approach, effectively capturing the essential information.

For BEHAVIOR , the situation is the opposite: most tasks require long sequences of execution, with an average of 14.6 steps as shown in Figure 61. Additionally, the preconditions for the actions in BEHAVIOR are more complex and strict compared to VirtualHome , while the number of interactable objects in the scene is relatively small. This requires stronger long-term planning and commonsense reasoning abilities. As a result, LLMs that "think more" win, which explains why powerful, closed-source models achieved the much better performance, especially for o1-preview which uses reasoning tokens to think much more before generating responses. .

In general, the performance of LLMs in VirtualHome is significantly better than in BEHAVIOR , which is reasonable considering the task complexity. However, it is noteworthy that even the best LLMs only completed half of the tasks in BEHAVIOR which indicates there's still a long way to go before LLMs can be effectively applied to embodied agents tasks.

**(2) Grammar Errors.** The occurrence of grammar errors for different LLMs generally aligns with our overall understanding of their capabilities. That is, the larger the model, the fewer the grammar errors; newer editions tend to perform better than older ones; and LLMs trained with high-quality data excel. For instance, o1-preview, Claude-3 Opus, GPT-4o, and Mistral Large made fewer grammar errors compared to their other family members, indicating advancements in their

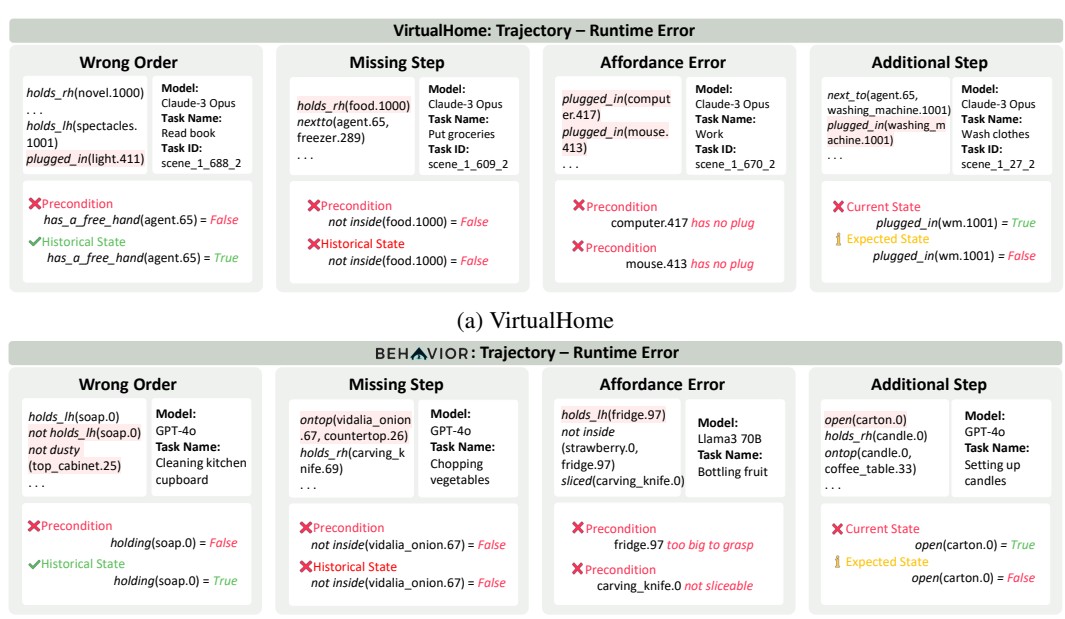

(a) VirtualHome

(b) BEHAVIOR

Figure 27: Trajectory runtime error examples for subgoal decomposition.

language understanding and generation capabilities. This trend holds true for both BEHAVIOR and VirtualHome tasks.

We studied the cases where LLMs made grammar errors, as shown in Figure 29. Common parsing errors include unfinished actions or action sequences, often due to LLMs mistakenly stopping generation early or producing overly long and incorrect action sequences that exceed the context window. Additionally, LLMs sometimes confuse the length of arguments. For example, although GPT-4o made very few grammar errors, it still confuses the parameters required for *rinse* and *wash* in VirtualHome , leading to action-arg length errors. Interestingly, *rinse* seems particularly challenging for GPT-4o, as it also made another hallucination error involving *rinse*.

We consider this metric a reflection of the LLMs' instruction-following and trustworthiness abilities. LLMs that can better follow instructions are likely to produce more grammatically correct outputs, which is crucial for applications requiring high levels of precision and reliability. This underscores the importance of continuous improvements in model size, training methodologies, and data quality to enhance the reliability and accuracy of language LLMs.

**(3) Runtime Errors.** The runtime error rate in VirtualHome is lower than BEHAVIOR , the reason of which has been previously discussed: 1) the action sequences required to accomplish tasks in BEHAVIOR are longer, and 2) the preconditions for executing an action are more complex and strict. For both simulators, the most frequent runtime errors are missing steps, as unsatisfied preconditions are the primary cause of failed action execution.

Figure 30 shows examples of errors made by LLMs. The reasons for making a *missing step* error are mainly due to the lack of common sense reasoning ability to satisfy the preconditions for a desired action, e.g., *getting to the sink*, *holding a soap* before *washing hands*, or *soaking the brush* before *cleaning a stain*. *Wrong temporal order* errors often arise from deficiencies in planning abilities, such as mistakenly *drinking* after *putting a cup back* or *grasping a tomato* before *slicing it*, demonstrating that LLMs still lack experience or common sense with daily life activities. Affordance errors occur due to a lack of understanding about properties of objects, like attempting to *type on a mouse* or not *assuming a strawberry will become different parts after slicing it*. For *additional step* errors, it may be an issue with in-context memory, where LLMs forget they have already opened a cabinet and attempt to open it again, or it could be related to reasoning about the common situation of the initial scene, like *the agent would be standing at the beginning*, therefore no need to *instruct it to stand up again*. In sum, the ability of commonsense reasoning is crucial for successfully predicting action sequences since our prompts do not provide detailed information about the environment and action instructions. LLMs must follow human conventions to reasonably guess the preconditions and post-effects of the actions.

For the runtime error metrics, o1-preview takes a significant lead. Other commonly acknowledged powerful LLMs, such as GPT-4-turbo, Gemini 1.5 Pro, and Claude-3 Opus, performed well in certain areas, they still struggled with other aspects of the metrics.

**(4) Goal Satisfaction.** In general, in VirtualHome , LLMs perform better at satisfying state goals than relation goals, while in BEHAVIOR , it is vice versa. The reason for this difference is that in VirtualHome , there are more objects in the scene, and objects can have more complex relations (like triple relations) compared to BEHAVIOR , where only binary relations exist. In BEHAVIOR , the

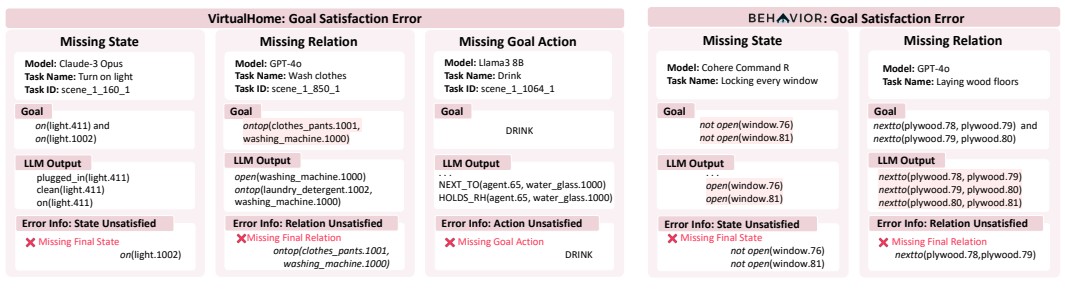

(a) VirtualHome      (b) BEHAVIOR

Figure 28: Goal satisfaction error examples for subgoal decompositions

preconditions for state actions are more complex, as shown in Tables 28 and 27. Therefore, it is easier for LLMs to achieve state goals in VirtualHome and relation goals in BEHAVIOR .

However, performance varies across different LLMs. In both simulators, o1-preview achieves the highest state goal satisfaction rates, which are higher than its relation goal satisfaction rates. Additionally, Claude-3 Sonnet and Mixtral Large performed exceptionally well in relation goals.

Figure 31 shows examples of errors that prevent LLMs from satisfying goals in a task. Generally, there are two cases where LLMs fail to satisfy a goal: 1) Missed goals: forgetting to generate actions to achieve the goal, e.g., forgetting to *TOGGLE_ON(laptop.1000)* to satisfy *on(laptop.1000)*, which could be due to deficiencies in instruction-following or issues with in-context memory. 2) Wrong actions: mistakenly corresponding goals to incorrect actions, e.g., corresponding *onfloors(plywood_78)* to *RIGHT_PLACE_NEXTTO(room_floor_living_room_0)* instead of *RIGHT_PLACE_FLOOR(room_floor_living_room_0)*, which could be a lack of reasoning ability.

---

**Takeaways of Action Sequencing**

1. Larger models did not always outperform smaller ones, especially for shorter and simpler tasks.

2. o1-preview takes the lead. Other LLMs (including powerful ones like GPT-4-turbo, Gemini 1.5 Pro, and Claude-3 Opus) consistently outperformed others across all runtime error metrics.

3. Generally, larger model sizes and higher quality training data correlated with lower grammar error rates.

4. Commonsense reasoning about object properties, agent states, and preconditions for actions remains a significant challenge for current LLMs, particularly when given limited context.

5. Errors in goal satisfaction can be attributed to: 1) Missed goals: Failing to generate actions to achieve the goal. 2) Wrong actions: Incorrectly mapping goals to inappropriate actions.

---

Table 12: Trajectory evaluation results (%) for *action sequencing*. Full results.

| Model | Goal Evaluation Task SR | | Execution SR | | Grammar Error (↓) Parsing | | Hallucination | | Predicate-Arg Num | | Runtime Error (↓) Wrong Order | | Missing Step | | Affordance | | Additional Step | |
|---|---|---|---|---|---|---|---|---|---|---|---|---|---|---|---|---|---|---|
| | V | B | V | B | V | B | V | B | V | B | V | B | V | B | V | B | V | B |
| Claude-3 Haiku | 54.8 | 26.0 | 60.7 | 32.0 | **0.0** | **0.0** | 6.6 | 6.0 | **0.0** | **0.0** | 0.7 | 7.0 | 30.5 | 54.0 | 1.6 | 1.0 | 1.6 | 1.0 |
| Claude-3 Sonnet | 58.4 | 44.0 | 63.3 | 57.0 | **0.0** | **0.0** | 4.9 | 1.0 | **0.0** | 7.9 | 4.6 | 11.0 | 26.9 | 19.0 | 0.3 | 11.0 | 3.0 | 2.0 |
| Claude-3 Opus | 64.9 | 51.0 | 69.5 | 59.0 | **0.0** | **0.0** | 17.0 | **0.0** | **0.0** | **0.0** | 0.3 | 3.0 | 12.8 | 35.0 | 0.3 | 3.0 | 2.3 | 2.0 |
| Claude-3.5 Sonnet | 76.7 | 60.0 | 81.3 | 69.0 | **0.0** | **0.0** | 2.0 | **0.0** | 0.3 | **0.0** | 0.3 | 5.0 | 14.4 | 25.0 | 1.6 | 1.0 | 1.3 | 2.0 |
| Gemini 1.0 Pro | 45.6 | 27.0 | 56.7 | 32.0 | **0.0** | 7.0 | 3.6 | 3.0 | 2.6 | 6.0 | 2.0 | 13.0 | 33.4 | 35.0 | 1.6 | 4.0 | 8.2 | 4.0 |
| Gemini 1.5 Flash | 69.5 | 40.0 | 75.4 | 52.0 | **0.0** | **0.0** | 2.0 | **0.0** | **0.0** | **0.0** | **0.0** | 5.0 | 22.3 | 42.0 | 0.3 | 1.0 | 1.6 | 2.0 |
| Gemini 1.5 Pro | 76.7 | 42.0 | 83.6 | 54.0 | **0.0** | **0.0** | 1.3 | **0.0** | 0.7 | **0.0** | 0.3 | 6.0 | 14.1 | 39.0 | **0.0** | 1.0 | 3.0 | 2.0 |
| GPT-3.5-turbo | 24.9 | 16.0 | 40.7 | 20.0 | 10.2 | 4.0 | 1.0 | 7.0 | 2.0 | 23.0 | **0.0** | 1.0 | 41.6 | 36.0 | 4.6 | 8.0 | 3.0 | 1.3 |
| GPT-4-turbo | 60.0 | 38.0 | 65.2 | 45.0 | **0.0** | **0.0** | 2.0 | **0.0** | 1.6 | **0.0** | **0.0** | 7.0 | 30.8 | 47.0 | 0.3 | 1.0 | 1.3 | **0.0** |
| GPT-4o | 71.5 | 47.0 | 81.3 | 53.0 | 1.0 | **0.0** | 2.0 | 1.0 | 0.7 | **0.0** | 0.3 | 9.0 | 15.1 | 36.0 | **0.0** | 1.0 | 2.3 | **0.0** |
| Cohere Command R | 44.9 | 16.0 | 44.3 | 19.0 | 2.0 | 5.0 | 23.3 | 13.0 | 2.6 | **0.0** | 2.0 | 8.0 | 17.0 | 43.0 | 9.8 | 12.0 | 3.6 | 4.0 |
| Cohere Command R+ | 54.1 | 27.0 | 65.2 | 35.0 | 6.9 | **0.0** | 12.5 | 1.0 | 2.0 | 15.0 | 0.3 | 10.0 | 11.8 | 39.0 | 1.6 | **0.0** | 3.9 | 15.0 |
| Mistral Large | **78.4** | 33.0 | **84.6** | 50.0 | **0.0** | **0.0** | 2.0 | **0.0** | 0.3 | **0.0** | **0.0** | 8.0 | 12.5 | 35.0 | 0.7 | 6.0 | 3.9 | 7.0 |
| Mixtral 8x22B MoE | 63.3 | 30.0 | 67.9 | 40.0 | **0.0** | 3.0 | 13.1 | 6.0 | 1.0 | **0.0** | 1.3 | 10.0 | 14.1 | 32.0 | 2.6 | 9.0 | 5.6 | 2.0 |
| Llama 3 8B | 21.3 | 10.0 | 23.6 | 16.0 | 1.0 | **0.0** | 34.1 | 15.0 | 8.2 | 9.0 | 1.0 | 6.0 | 30.8 | 44.0 | 1.3 | 9.0 | 2.6 | 5.0 |
| Llama 3 70B | 59.0 | 34.0 | 66.6 | 42.0 | **0.0** | **0.0** | 14.1 | 2.0 | 8.2 | **0.0** | 2.0 | 15.0 | **9.2** | 38.0 | **0.0** | 3.0 | 6.2 | 6.0 |
| o1-mini | 71.5 | 56.0 | 76.4 | 65.0 | 4.9 | **0.0** | 2.0 | 3.0 | **0.0** | **0.0** | 1.0 | 7.0 | 17.7 | 17.0 | 0.3 | 6.0 | 2.6 | 5.0 |
| o1-preview | 65.2 | **81.0** | 72.5 | **91.0** | 6.6 | **0.0** | 11.5 | **0.0** | **0.0** | **0.0** | **0.0** | **0.0** | 12.1 | **6.0** | 0.3 | 2.0 | 2.0 | 3.0 |

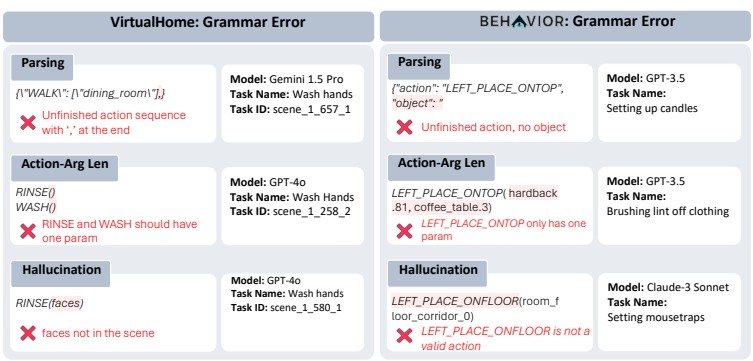

Figure 29: Grammar error examples for *action sequencing*.

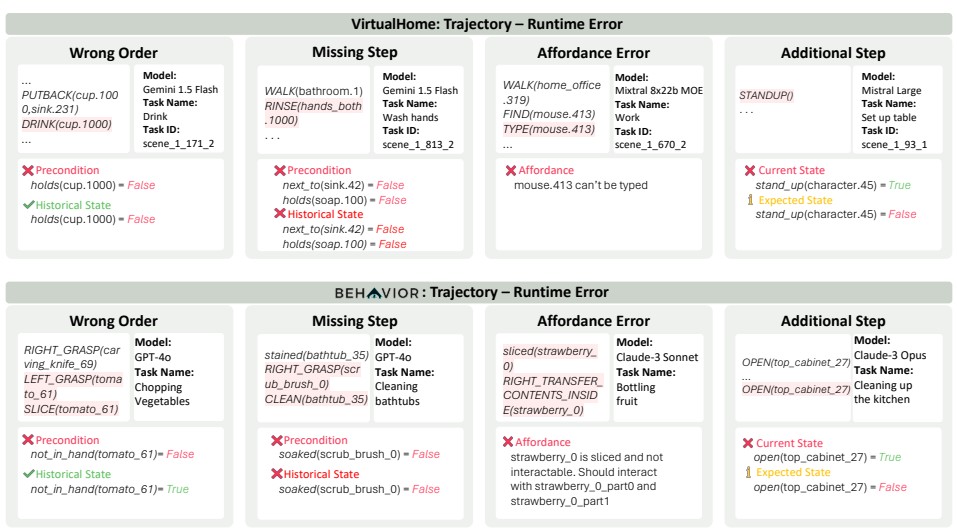

Figure 30: Trajectory runtime error examples for *action sequencing*.

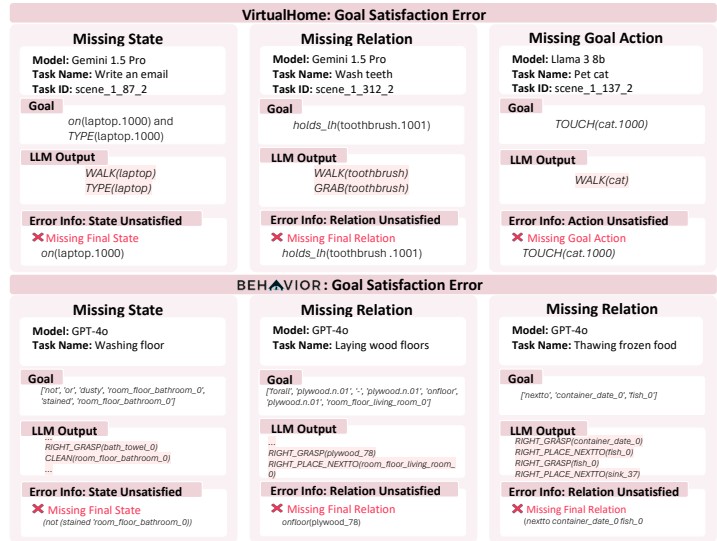

Figure 31: Goal satisfaction error examples for *action sequencing*.

Table 13: Full results of logic form accuracy for *transition modeling* in VirtualHome

| Model | Object States | | | Object Orientation | | | Object Affordance | | | Spatial Relations | | | Non-Spatial Relations | | |
|---|---|---|---|---|---|---|---|---|---|---|---|---|---|---|---|
| | *Precision* | *Recall* | *F1* | *Precision* | *Recall* | *F1* | *Precision* | *Recall* | *F1* | *Precision* | *Recall* | *F1* | *Precision* | *Recall* | *F1* |
| Claude-3 Haiku | 76.0 | 40.1 | 52.5 | 19.0 | 34.4 | 24.4 | 67.8 | 73.9 | 70.7 | 37.7 | 38.7 | 38.2 | 2.0 | 1.5 | 1.7 |
| Claude-3 Opus | **87.4** | **49.2** | **63.0** | 46.3 | **96.9** | 62.6 | 76.8 | 74.3 | 75.5 | 37.6 | 39.9 | 38.7 | 10.4 | **5.2** | 7.0 |
| Claude-3 Sonnet | 76.6 | 37.4 | 50.3 | 48.1 | 78.1 | 59.5 | 60.7 | 74.3 | 66.8 | 32.3 | 39.9 | 35.7 | 6.2 | 4.1 | 4.9 |
| Claude-3.5 Sonnet | 86.1 | 46.7 | 60.5 | **93.9** | **96.9** | **95.3** | 77.7 | **75.5** | 76.6 | 45.3 | 39.8 | 42.4 | 7.1 | 5.1 | 5.9 |
| Cohere Command R | 18.0 | 6.8 | 9.9 | 38.7 | 90.6 | 54.2 | 40.2 | 23.0 | 29.2 | 12.6 | 6.7 | 8.8 | 3.3 | 0.9 | 1.4 |
| Cohere Command R+ | 44.9 | 19.0 | 26.3 | 34.6 | 68.8 | 45.9 | 51.0 | 62.1 | 56.0 | 30.1 | 34.8 | 32.4 | 7.6 | 3.1 | 4.4 |
| Gemini 1.0 Pro | 68.4 | 12.3 | 20.4 | 16.3 | 62.5 | 27.9 | 55.3 | 20.1 | 29.6 | 45.0 | 16.5 | 24.3 | 7.7 | 2.5 | 3.8 |
| Gemini 1.5 Flash | 82.3 | 37.6 | 51.6 | 2.0 | 3.1 | 2.5 | 54.4 | 74.7 | 62.9 | **47.4** | **42.9** | **45.0** | 16.3 | **5.2** | 7.9 |
| Gemini 1.5 Pro | 45.3 | 11.9 | 18.8 | 88.2 | 93.8 | 90.9 | **79.9** | **75.5** | **77.7** | 42.2 | 35.8 | 38.7 | 15.5 | **5.2** | 7.8 |
| GPT-3.5-turbo | 63.5 | 21.9 | 32.5 | 11.4 | 15.6 | 13.2 | 57.2 | 53.1 | 54.9 | 35.2 | 21.7 | 26.8 | 1.7 | 0.3 | 0.6 |
| GPT-4-turbo | 79.3 | 44.2 | 56.7 | 10.1 | 31.3 | 15.3 | 65.9 | 71.0 | 68.4 | 31.8 | 34.2 | 32.9 | 3.8 | 1.0 | 1.6 |
| GPT-4o | 80.2 | 41.5 | 54.6 | 48.0 | 59.4 | 52.8 | 76.2 | 73.7 | 74.9 | 40.8 | 40.7 | 40.8 | 14.8 | 5.1 | 7.5 |
| Llama 3 8b | 30.8 | 13.7 | 18.9 | 0.0 | 0.0 | 0.0 | 1.6 | 3.2 | 2.1 | 15.5 | 18.2 | 16.8 | 0.0 | 0.0 | 0.0 |
| Llama 3 70b | 63.5 | 21.9 | 32.5 | 49.0 | 66.3 | 56.6 | 65.0 | 50.0 | 57.0 | 27.0 | 27.0 | 27.0 | 5.0 | 2.0 | 3.0 |
| Mistral Large | 30.0 | 8.0 | 13.0 | 48.0 | 88.0 | 62.0 | 72.0 | 29.0 | 41.0 | 35.0 | 18.0 | 24.0 | 3.0 | 1.0 | 1.0 |
| Mixtral 8x22B MoE | 72.0 | 33.0 | 45.0 | 43.0 | 83.0 | 57.0 | 64.0 | 74.0 | 69.0 | 40.0 | 38.0 | 39.0 | 12.0 | 4.0 | 6.0 |
| o1-mini | 82.5 | 45.9 | 59.0 | 51.3 | 62.5 | 56.3 | 59.8 | 57.1 | 58.5 | 32.1 | 32.8 | 32.5 | 5.0 | 4.1 | 4.5 |
| o1-preview | 83.0 | 45.1 | 58.5 | 69.0 | 90.6 | 78.4 | 84.7 | 71.4 | 77.5 | 39.8 | 37.8 | 38.8 | **17.1** | 9.0 | **11.8** |

## E.4 Transition Modeling

### E.4.1 Logic Form Accuracy

**VirtualHome** The results from VirtualHome (Table 13) illustrate the varied performance of models across five distinct categories, demonstrating their ability to predict complex logic and predicates for each action. The model Claude-3 Opus showcased superior performance in terms of all metrics in the object states category, achieving an F1 score of 63.0%, highlighting its effectiveness in interpreting complex object state transitions. Notably, Claude-3.5 Sonnet performed exceptionally well in object orientation, achieving the highest F1 score of 90.9%, which suggests its strong capability in understanding object orientations like 'facing'. Gemini 1.5 Pro also excels in the object affordance category on the precision, recall, and f1 score with an F1 score of 77.7%, demonstrating its well understanding of required object properties for different actions. However, across all models, the non-spatial relations category displayed generally low scores, with Gemini 1.5 Flash performing best at a modest F1 of 7.9%. This basically results from the complex logic and corner cases involved in some non-spatial actions. For example, when predicting action 'grab', few models consider 'not in a closed container' or 'both hands of the robot is holding something' as a precondition, and including

Table 14: Full results of logic form accuracy for *transition modeling* in BEHAVIOR

| Model | Object States | | | Spatial Relations | | | Non-Spatial Relations | | |
|---|---|---|---|---|---|---|---|---|---|
| | *Precision* | *Recall* | *F1* | *Precision* | *Recall* | *F1* | *Precision* | *Recall* | *F1* |
| Claude-3.5 Sonnet | 83.3 | **74.8** | **78.8** | **73.3** | **48.8** | **58.6** | 82.9 | 66.2 | 73.6 |
| Claude-3 Haiku | 64.1 | 55.2 | 59.3 | 54.7 | 37.4 | 44.4 | 63.3 | 51.4 | 56.7 |
| Claude-3 Opus | 74.6 | 69.4 | 71.9 | 70.4 | 44.6 | 54.6 | 68.5 | 69.1 | 68.8 |
| Claude-3 Sonnet | 66.2 | 68.7 | 67.5 | 62.8 | 39.8 | 48.7 | 68.8 | 52.0 | 59.2 |
| Cohere Command R | 59.7 | 43.9 | 50.6 | 29.1 | 11.6 | 16.6 | 27.2 | 15.3 | 19.6 |
| Cohere Command R+ | 58.0 | 58.4 | 58.2 | 54.2 | 33.6 | 41.5 | 53.0 | 56.6 | 54.7 |
| Gemini 1.0 Pro | 67.2 | 55.2 | 60.6 | 47.5 | 35.3 | 40.5 | 43.8 | 48.3 | 45.9 |
| Gemini 1.5 Flash | 73.9 | 57.2 | 64.5 | 54.5 | 40.7 | 46.6 | 60.7 | 53.8 | 57.0 |
| Gemini 1.5 Pro | 69.6 | 46.7 | 55.9 | 52.9 | 27.2 | 35.9 | 59.6 | 47.4 | 52.8 |
| GPT-3.5-turbo | 67.1 | 46.1 | 54.6 | 57.6 | 31.6 | 40.9 | 40.8 | 36.1 | 38.3 |
| GPT-4-turbo | 58.2 | 59.4 | 58.8 | 50.3 | 27.8 | 35.8 | 58.5 | 38.4 | 46.4 |
| GPT-4o | 73.1 | 69.6 | 71.3 | 63.9 | 35.8 | 45.9 | 84.7 | 64.2 | 73.0 |
| Llama 3 70b | 68.1 | 64.6 | 66.3 | 60.3 | 38.8 | 47.2 | 65.1 | 53.8 | 58.9 |
| Llama 3 8b | 40.3 | 32.4 | 35.9 | 29.6 | 22.7 | 25.7 | 48.9 | 43.9 | 46.2 |
| Mistral Large | 67.5 | 66.5 | 67.0 | 54.9 | 32.3 | 40.7 | 59.7 | 44.6 | 51.1 |
| Mixtral 8x22B MoE | 60.2 | 60.0 | 60.1 | 53.2 | 39.9 | 45.6 | 57.9 | 55.8 | 56.8 |
| o1-mini | 46.3 | 37.2 | 41.3 | 71.1 | 42.3 | 53.1 | 80.1 | 58.3 | 67.5 |
| o1-preview | **85.5** | 72.3 | 78.3 | 72.4 | 46.1 | 56.3 | **88.0** | **79.5** | **83.5** |

Table 15: Full results of planner success rate for *transition modeling* (%)

| Model | Object States | | Object Orientation | | Object Affordance | | Spatial Relations | | Non-Spatial Relations | |
|---|---|---|---|---|---|---|---|---|---|---|
| | *V* | *B* | *V* | *B* | *V* | *B* | *V* | *B* | *V* | *B* |
| Claude-3 Haiku | 13.5 | 68.9 | 3.6 | - | 19.8 | - | 46.9 | 62.8 | 73.0 | 62.3 |
| Claude-3 Opus | 63.5 | 84.4 | 71.4 | - | 58.7 | - | 64.8 | 80.9 | 55.4 | 82.0 |
| Claude-3 Sonnet | 11.2 | 80.0 | 3.6 | - | 10.8 | - | 20.0 | 79.8 | 13.5 | 80.3 |
| Claude-3.5 Sonnet | 67.4 | **86.7** | 96.4 | - | 67.8 | - | **96.6** | 80.8 | **91.9** | 80.3 |
| Cohere Command R | 44.6 | 48.9 | 82.1 | - | 40.1 | - | 62.6 | 38.3 | 58.3 | 39.3 |
| Cohere Command R+ | 36.5 | 77.8 | 46.4 | - | 35.3 | - | 40.7 | 57.4 | 31.1 | 47.5 |
| Gemini 1.0 Pro | 10.7 | 22.2 | 0.0 | - | 10.2 | - | 14.5 | 13.8 | 2.7 | 14.8 |
| Gemini 1.5 Flash | 34.8 | 55.6 | 7.1 | - | 46.7 | - | 61.4 | 68.1 | 60.8 | 70.5 |
| Gemini 1.5 Pro | **94.4** | 35.6 | 89.3 | - | **95.8** | - | 89.0 | 40.4 | 83.8 | 39.3 |
| GPT-3.5-turbo | 1.1 | 26.7 | 25.0 | - | 1.2 | - | 0.0 | 39.4 | 0.0 | 54.1 |
| GPT-4-turbo | 51.7 | 40.0 | 50.0 | - | 47.9 | - | 67.6 | 44.7 | 64.9 | 52.5 |
| GPT-4o | 71.9 | 68.9 | 78.6 | - | 63.5 | - | 66.9 | 64.9 | 68.9 | 68.9 |
| Llama 3 8b | 27.0 | 35.6 | 0.0 | - | 26.4 | - | 37.9 | 27.7 | 31.1 | 26.2 |
| Llama 3 70b | 10.1 | 68.9 | 3.6 | - | 6.6 | - | 15.2 | 77.7 | 18.9 | 85.2 |
| Mistral Large | 15.7 | 73.3 | 7.1 | - | 14.4 | - | 17.9 | 76.6 | 8.1 | 80.3 |
| Mixtral 8x22B MoE | 36.5 | 57.8 | 50.0 | - | 28.1 | - | 44.1 | 52.1 | 43.2 | 57.4 |
| o1-mini | 63.5 | 77.8 | 82.1 | - | 59.3 | - | 75.9 | 77.7 | 71.6 | 75.4 |
| o1-preview | 69.1 | **86.7** | **100.0** | - | 67.1 | - | 76.6 | **89.4** | 78.4 | **90.2** |

the logic 'when the robots hold something with one hand, hold the object with the other hand' in the effect is also too challenging for LLMs. This could point to a common challenge in modeling complex relationships with real-world scenario concerns, requiring perhaps a more nuanced understanding or a different approach in training.

**BEHAVIOR** In the BEHAVIOR environment (Table 14), the model Claude-3.5 Sonnet shows outstanding performance, particularly in object states with an F1 score of 78.8% and in spatial relations with an F1 of 58.6%. These results underscore its exceptional ability to handle both static state transitions of objects and spatial relations between the robot and the objects. In the Non-Spatial Relations category, the model o1-preview has significantly better performance than others with an F1 score of 83.5%.

**Overall Conclusions** Therefore, the consistently high performance of Claude-3.5 Sonnet models across both environments suggests that these models have potentially benefited from training regimens

or architectural features that enhance their understanding of object-oriented and relational aspects of embodied environments. There are also some divergences between the results of the two environments. In the non-spatial relation category, Gemini 1.5 Flash performs better in VirtualHome while o1-preview performs better in BEHAVIOR . The performance disparity between different environments also suggests that model training and optimization might need more customization toward specific types of embodied interactions. Overall, high-performing models in certain categories, such as Gemini 1.5 Pro in Object Orientation and Claude-3.5 Sonnet in Object States and Spatial Relations, highlight areas where specific models excel and can be leveraged for tasks requiring high accuracy in those domains. On the other hand, models like Llama 3 and Cohere Command R+ showed lower performance, which may indicate limitations in their training datasets or model architectures for these specific task requirements. Additionally, the generally lower scores in non-spatial Relations across most models suggest a gap in current modeling capabilities, offering a potential area for future research and model improvement.

### E.4.2 Planner Success Rate Analysis

This assessment highlights how well models predict feasible transitions and achieve objectives from initial states.

**Model Performance Highlights:** o1-preview stands out among all models, demonstrating robust success rates across various categories and different environments. For example, o1-preview achieves a 100% success rate in VirtualHome Object Orientation, and a 90.2% success rate in BEHAVIOR Non-Spatial Relations reflects the internal consistency of its generated operators.

**Spatial vs. Non-Spatial Relations:** The data reveals a notable disparity in performance between spatial and non-spatial Relations, with non-spatial Relations generally showing higher success rates. This suggests that even though LLM may not cover corner cases in non-spatial relation actions, models can still simplify it and get away with it compared with spatial relation actions, an insight that could direct future training enhancements.

**Challenges and Strategic Insights:** Despite successes, persistent challenges in categories like object orientation, where success rates are generally lower, highlight potential gaps in model training or architectural design. The variability in success rates across models points to significant differences in how they interpret and execute tasks within these dynamic environments. Even high-performing models struggle with precise spatial interactions, a critical area for tasks requiring detailed physical interactions. For example, GPT-4o struggles at non-spatial relation tasks and Claude-3 Opus is not doing well at object states and object affordance tasks.

**Future Directions:** The findings suggest valuable pathways for model improvements, particularly in enhancing relation understanding in physical task execution. Future research might focus on diversifying training scenarios that incorporate a broader range of spatial and non-spatial interactions and testing these models in more complex, real-world settings. Additionally, exploring hybrid models that combine various strengths of current LLMs could yield more capable and flexible systems for practical applications in embodied AI environments.

### E.4.3 Error categorization

In assessing the logical form accuracy of predicted PDDL action bodies by the LLM, we categorize errors into two main types: missing predicates, and additional predicates as illustrated in figure 32. Each category reflects specific discrepancies between the LLM outputs and the ground truth, affecting the precision, recall, and F1 scores of the model's predictions.

**Missing Predicates:** These errors occur when essential predicates are omitted in the LLM's output, leading to incomplete or incorrect action specifications. For instance, in the missing object affordance error, the LLM omits the '(lieable ?obj)' predicate required for the action LIE, replacing it with incorrect predicates like '(sittable ?obj)'. Apparently, to lie on some object, the object should be lieable instead of sittable. Such omissions can drastically affect the feasibility and correctness of the generated plan, as they fail to capture the necessary conditions for action execution.

**Additional Predicates:** This error type is characterized by the inclusion of unnecessary or incorrect predicates in the LLM output, which are not present in the ground truth. An example is seen in the additional object state, where the LLM predicts '(not (stained ?scrub_brush))' under effects for the

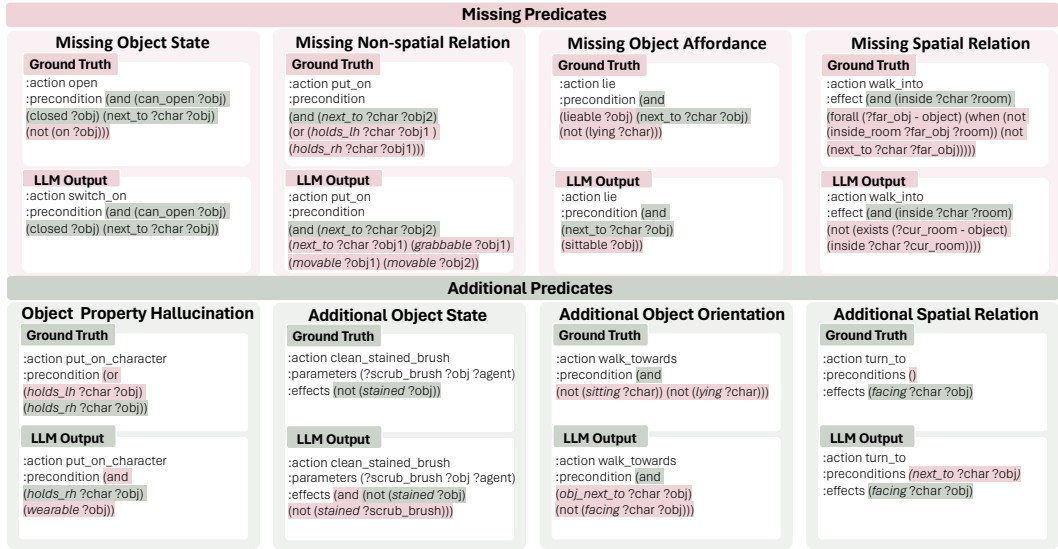

Figure 32: Transition modeling error examples

action CLEAN_STAINED_BRUSH. To clean something with a brush, the clean object should be the target object instead of the brush. These additional predicates can lead to over-constrained planning scenarios, potentially preventing the execution of valid actions.

Another type of additional predicate is property hallucination. Property hallucination errors arise when the LLM inaccurately attributes properties to objects that are not supported by the ground truth. In the depicted error, the LLM predicts that an object is '(wearable ?obj)', while 'wearable' is not in the vocabulary dictionary. This type of error not only introduces factual inaccuracies but also misleads the planning process by enforcing constraints that can never be true.

```
Transition Models all from the Ground Truth - EXCUTION SUCCESS

(:action plug_in
  :parameters (?char - character ?obj - object)
  :precondition (or (and (next_to ?char ?obj) (has_plug ?obj)
                (plugged_out ?obj)) (and (next_to ?char ?obj)
                (has_switch ?obj) (plugged_out ?obj)))
  :effect (and (plugged_in ?obj) (not (plugged_out ?obj)))
)

(:action walk_towards
  :parameters (?char - character ?obj - object)
  :precondition (and (not (sitting ?char)) (not (lying ?char))
                (next_to ?char ?obj) (forall (?far_obj - object)
                (when (not (obj_next_to ?far_obj ?obj))
                (not (next_to ?char ?far_obj)))) (forall (?close_obj - object)
                (when (obj_next_to ?close_obj ?obj)
                (next_to ?char ?close_obj))))
  :effect (and (next_to ?char ?obj))
)

(:action switch_on
  :parameters (?char - character ?obj - object)
  :precondition (and (has_switch ?obj) (off ?obj)
                (plugged_in ?obj) (next_to ?char ?obj))
  :effect (and (on ?obj) (not (off ?obj))))
)
```

Figure 33: If we use a high-level planner (PDDL planner) over Ground Truth transition models of *plug_in*, *walk_towards* and *switch_on*, the output action sequence can be executed successfully.

```
Transition Models all from LLMs - EXECUTION SUCCESS

(:action plug_in
  :parameters (?char - character ?obj - object)
  :precondition (and (has_plug ?obj) (plugged_out ?obj)
            (next_to ?char ?obj))
  :effect (and (plugged_in ?obj) (not (plugged_out ?obj)))
)

(:action walk_towards
  :parameters (?char - character ?obj - object)
  :precondition (and (inside ?char ?room) (inside ?obj ?room))
  :effect (next_to ?char ?obj)
)

(:action switch_on
  :parameters (?char - character ?obj - object)
  :precondition (and (has_switch ?obj) (off ?obj)
            (next_to ?char ?obj))
  :effect (and (on ?obj) (not (off ?obj)))
)
```

Figure 34: If we use a high-level planner (PDDL planner) over GPT-4o predicted transition models of *plug_in*, *walk_towards* and *switch_on*, the output action sequence can be executed successfully.

```
plug_in from LLMs and walk_towards and switch_on from Ground Truth - EXECUTION FAILURE

(:action plug_in
  :parameters (?char - character ?obj - object)
  :precondition (and (has_plug ?obj) (plugged_out ?obj)
            (next_to ?char ?obj))
  :effect (and (plugged_in ?obj) (not (plugged_out ?obj)))
)

(:action walk_towards
  :parameters (?char - character ?obj - object)
  :precondition (and (not (sitting ?char)) (not (lying ?char))
            (next_to ?char ?obj) (forall (?far_obj - object)
            (when (not (obj_next_to ?far_obj ?obj))
            (not (next_to ?char ?far_obj)))) (forall (?close_obj - object)
            (when (obj_next_to ?close_obj ?obj)
            (next_to ?char ?close_obj))))
  :effect (and (next_to ?char ?obj))
)

(:action switch_on
  :parameters (?char - character ?obj - object)
  :precondition (and (has_switch ?obj) (off ?obj)
            (plugged_in ?obj) (next_to ?char ?obj))
  :effect (and (on ?obj) (not (off ?obj)))
)
```

Figure 35: If we use a high-level planner (PDDL planner) over the mixture of transtion models, i.e., GPT-4o predicted transition models of *plug_in* with ground truth *walk_towards* and *switch_on*, the output action sequence cannot be executed successfully.

**Consistency of predicted action space**

We observe the consistency of LLM predicted action space, that is, the complexity of predicted actions is consistent. Such consistency facilitates the PDDL planner to find possible solutions from an initial state to a goal state. It is worth noticing that interleaving LLM-predicted actions and ground truth actions has a lower planner success rate than using only ground truth actions or only LLM-predicted actions. Figure 33 demonstrates the success case of PDDL planning using ground truth actions, and figure 34 shows that the actions predicted by GPT-4o also pass the PDDL planner test. However, if we mix the two action spaces, using *plug_in* from GPT-4o prediction and *walk_toward* and *switch_on* from ground truth, the PDDL planner cannot find a feasible solution for this task. This is because *plug_in* from GPT-4o prediction diverges with *switch_on* from ground truth. In ground truth action space, to *switch_on* an object, it needs to be *plugged_in* and has_switch. Ground truth *plug_in* can handle the cases when objects either *has_plug* or *has_switch*. In the GPT-4o predicted action space, although *plug_in* cannot handle the cases when objects *has_switch*, the LLM gets away with it by not

specifying *plug_in* in the preconditions of *switch_on*. Although the definition is not comprehensive, it can pass the PDDL planner test. However, when mixing the two action spaces together, *switch_on* requires precondition *plug_in*. But *plug_in* cannot handle the cases when objects only *has_switch* but not *has_plug*. Thus, the predicted action space maintains consistency, and it is crucial to provide LLM with the context action definitions when we ask LLM to predict a single action.

---

**Takeaways of Transition Modeling**

(1) **Specialized Performance**: Models such as o1-preview and Claude-3 Opus excel in specific categories like object states and object orientation, respectively, suggesting that targeted training or specialized architectures enhance LLM capabilities in understanding different types of tasks in transition modeling.

(2) **Difficulty with Non-Spatial Relations**: Across various models, non-spatial relations consistently pose a challenge, highlighting a gap in the ability of LLMs to grasp complex relational dynamics that are crucial for realistic scenario modeling.

(3) **Consistency in Action Space**: The effectiveness of planning relies heavily on the consistency of the predicted action space by LLMs; discrepancies between mixed predicted and ground truth actions lead to reduced planner success, emphasizing the need for coherent action definitions in LLM outputs.

---

### E.5 Correlection with Action Length and Goal Complexity

In this section, we analyze the factors that influence the goal success rates of GPT-4o's performance in action sequencing for BEHAVIOR . Generally, the goal success rate is influenced by task complexity, as shown in Figure 36. The number of goals within a task, the number of state goals, and the number of relation goals all adversely affect the success rates. The length of the ground-truth action sequence follows the same trend, although there are some fluctuations.

Success rates for different actions and predicates are provided in Figure 37. The predicate with the highest success rate is *open*, which corresponds to the action *OPEN* that has a 1.00 execution success rate. This high success rate is because GPT-4o can successfully reason about the preconditions for this action, as shown in Table 27.

The complexity of action preconditions also adversely affects the success rate. Actions with lower execution success rates often require tool usage, which is a challenge for LLMs. For example, *Soak* requires placing the object in a toggled-on sink, *Slice* requires holding a knife, and *Clean* requires a cleaning tool like a rag or a brush, and if cleaning a stain, the cleaning tool needs to be soaked.

Interestingly, the success rate of the same actions performed with the right hand is slightly higher than those performed with the left hand. The action with the lowest success rate is *SLICE*. There are only two cases in BEHAVIOR that require this action, and GPT-4o failed both.

---

**Takeaways of Error Factor Analysis**

1. Task complexity adversely affects goal success rates, including the number of total goals, state goals, relation goals, and ground-truth action sequence length.

2. Complexity of action preconditions adversely affects success rates, especially for actions requiring tool usage, e.g., *Soak*, *Slice*, and *Clean*.

3. Actions performed with the right hand have a slightly higher success rate than those with the left hand.

---

## F  Sensitivity Analysis

### F.1  Motivation and Problem Formulation

In the current setting of the transition modeling task, we use the entire space of LLM-predicted actions combined with the PDDL planner to check whether there exists a feasible solution to fulfill

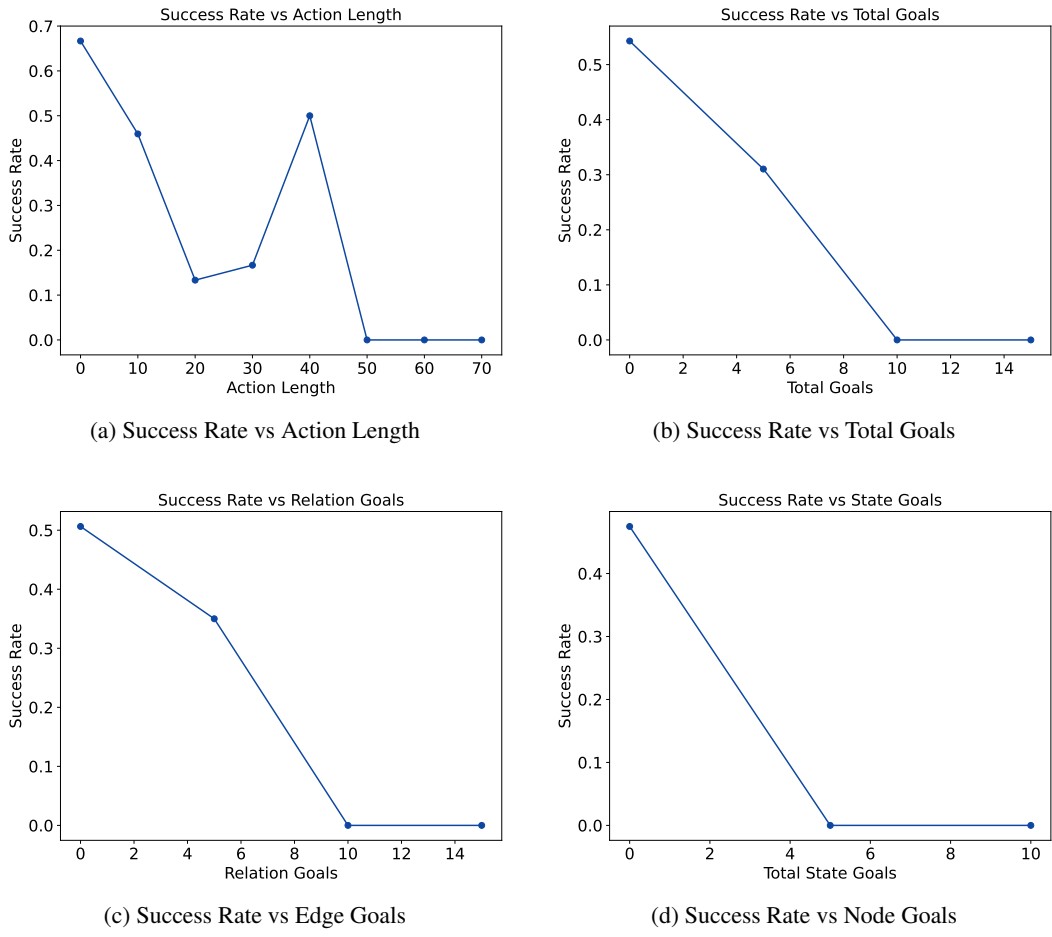

(a) Success Rate vs Action Length

(b) Success Rate vs Total Goals

(c) Success Rate vs Edge Goals

(d) Success Rate vs Node Goals

Figure 36: BEHAVIOR Success Rate as a Function of Different Cofactors

the goals. However, it still lacks finer-grid metrics to show which action fails if the PDDL planner fails to find a solution. Therefore, we conduct sensitivity analysis to examine how sensitive task success rates are to specific LLM-predicted actions. In the ground truth space, if after replacing a specific action with the LLM predicted one, the PDDL planner fails to find a solution, we call the task is *sensitive* to this action, and vice versa. This sensitivity analysis provides insights into the per-action impact of LLM predictions on the success rate of specific actions within each task category, highlighting areas where the LLM performs well and where it may require further improvement. Sensitivity analysis also highlights the actions that are crucial in different task categories, providing insights for fine-tuning and downstream tasks.

## F.2 Implementation Details

We start the sensitivity analysis by setting the action space as the ground truth space. Given the task, for each relevant action, we replace one ground truth action with the LLM-predicted counterpart and solve the task with a PDDL planner. Each task is categorized according to Appendix L.5 and we report the overall sensitivity analysis results and the results by task categories.

## F.3 Result Analysis

The sensitivity analysis conducted on a large language model's predictions for various tasks provides a detailed view of the model's robustness and areas necessitating improvement. This analysis evaluates the impact of replacing specific actions predicted by the model while maintaining other actions as

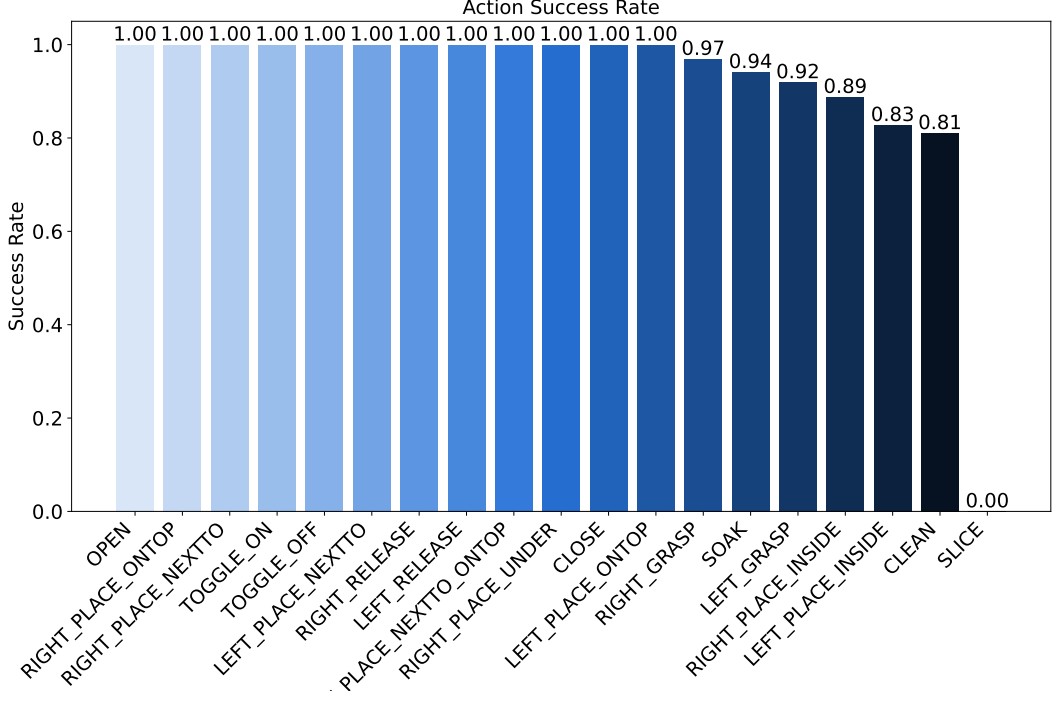

(a) Success Rate per Action

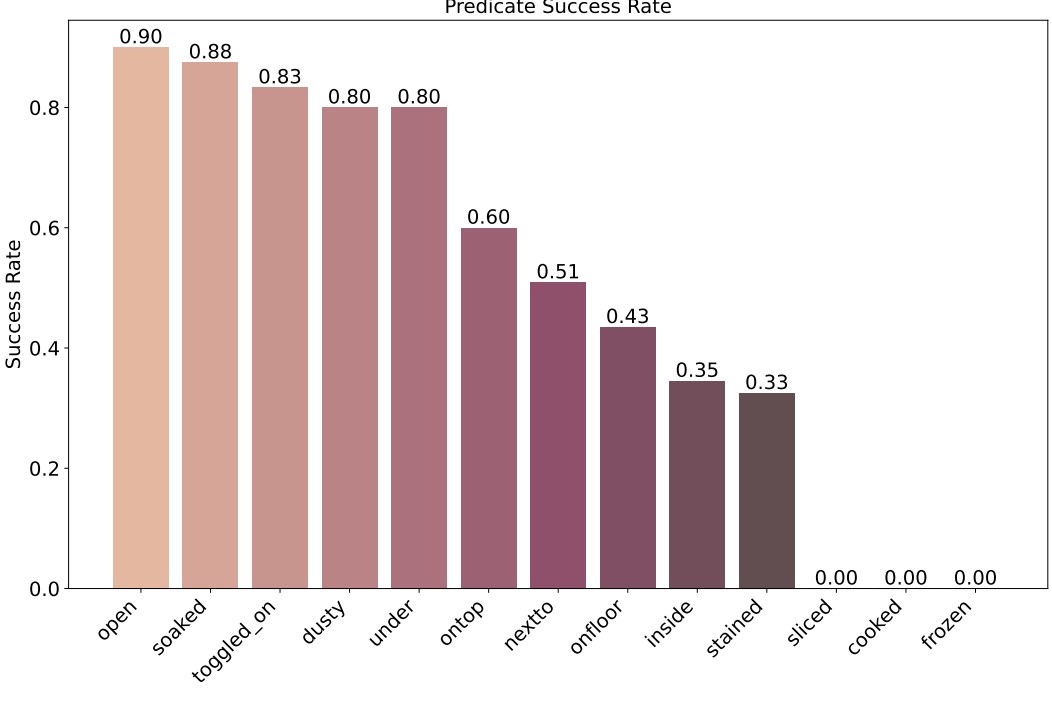

(b) Success Rate per Predicate

Figure 37: BEHAVIOR Success Rate Analysis for Actions and Predicates

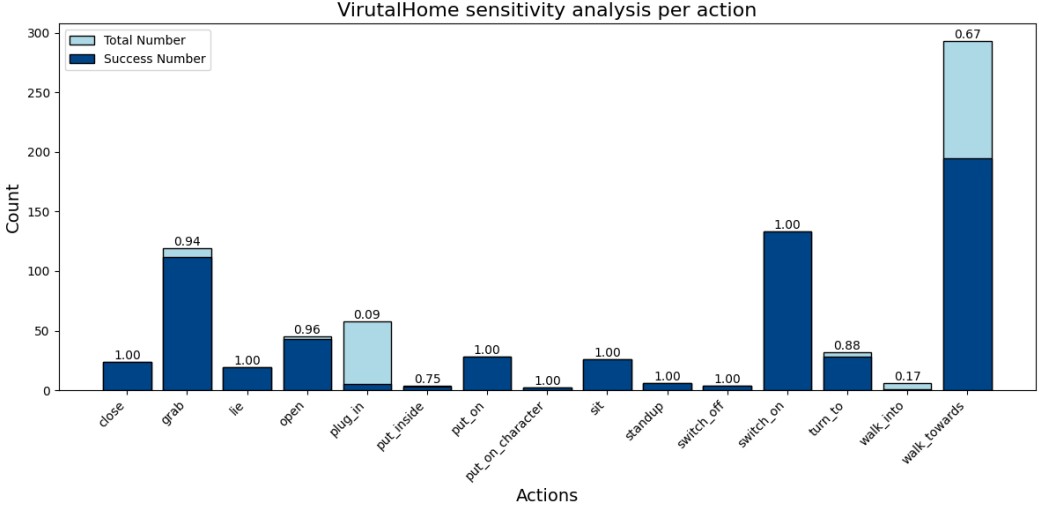

Figure 38: VirtualHome sensitivity analysis per action

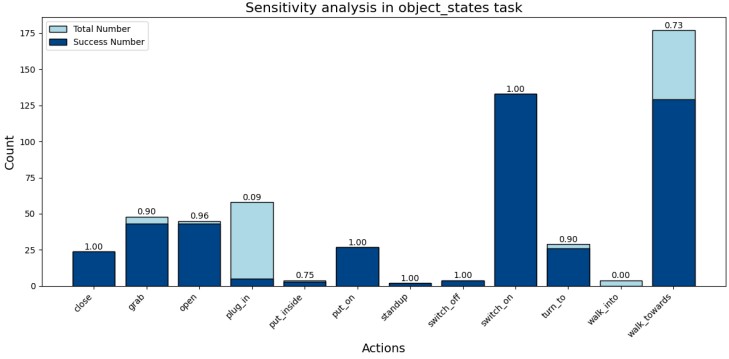

Figure 39: Sensitivity analysis for object states tasks in VirtualHome

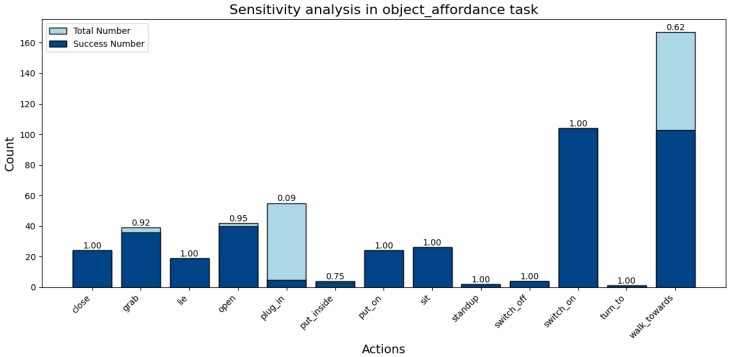

Figure 40: Sensitivity analysis for object affordance tasks in VirtualHome

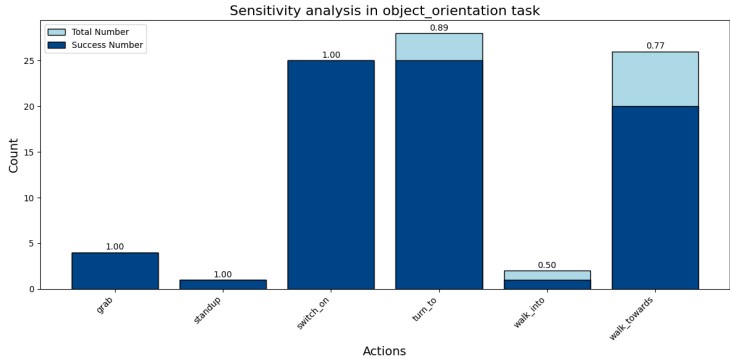

Figure 41: Sensitivity analysis for object orientation tasks in VirtualHome

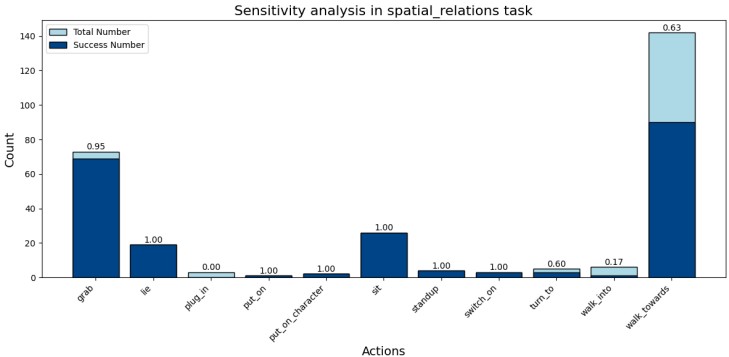

Figure 42: Sensitivity analysis for spatial relations tasks in VirtualHome

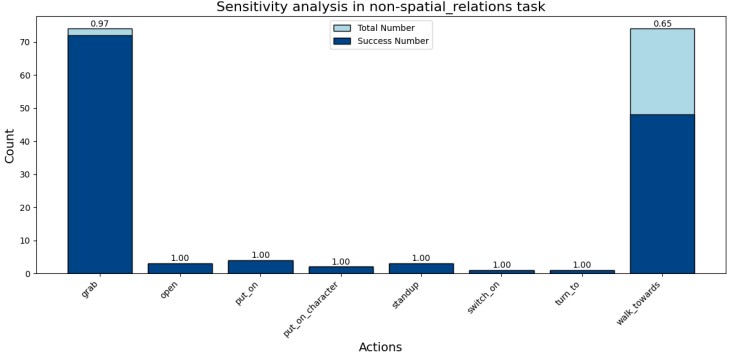

Figure 43: Sensitivity analysis for non-spatial relations tasks in VirtualHome

ground truth across five different task categories: non-spatial relations, object affordance, object orientation, object states, and spatial relations.

In the *non-spatial relations* task, the model exhibited a nearly perfect success rate for most actions except for "grab" and "walk_towards," with success rates of 0.97 and 0.65, respectively. The high success rates indicate a strong model performance in scenarios where spatial relationships are not a factor, although the lower rate for "walk_towards" suggests difficulties in predicting movement-related actions when the spatial context is absent.

For the *object affordance* task, there were notable variances, with "plug_in" and "put_inside" having significantly lower success rates of 0.09 and 0.75. This disparity suggests challenges in the model's understanding of actions involving complex interactions or manipulations, whereas actions directly associated with straightforward affordances like "close" and "grab" achieved perfect success.

The *object orientation* task showed high success rates for most actions, but "turn_to" and "walk_into" had lower success rates of 0.89 and 0.77. This indicates that the model is proficient with simple orientation changes yet slightly struggles with actions requiring precise orientation or movement toward specific objects.

In the *object states* task, "plug_in" displayed extremely low success rates of 0.09. The action "plug_in" likely presents conceptual challenges or ambiguities in interpretation, pointing to a critical need for enhancing the model's training on the diverse contexts and functionalities of objects.

The *spatial relations* task results highlight significant challenges, as seen with "walk_towards" and "walk_into" having success rates of 0.63 and 0.17. These outcomes underscore difficulties in handling spatially complex predictions and interactions within a spatial context.

From Figure 38, generally, we notice "plug_in" and "walk_towards" as the most challenging actions for LLM to predict. The preconditions of "plug_in" in VirtualHome contain two cases: either the device "has_plug", or the device "has_switch". While most LLMs are able to predict "has_plug", they can hardly catch the other case, which turns out to be quite common in the tasks. The difficulty of "walk_towards" lies in its complex constraints on spatial relations. Many LLMs either predict additional constraints or miss some necessary constraints, making the success rate of "walk_towards" low.

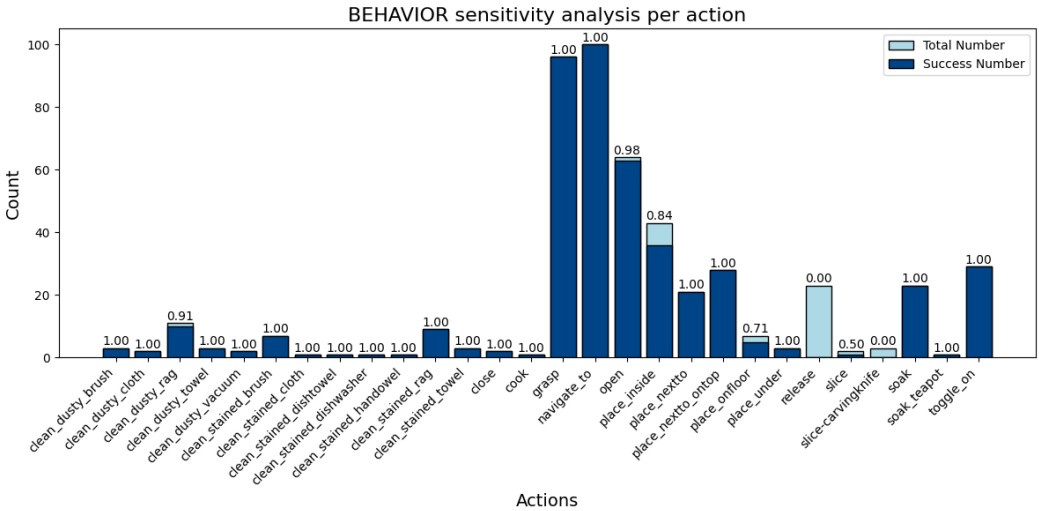

Figure 44: BEHAVIOR sensitivity analysis per action

The analysis conducted on the BEHAVIOR dataset for the tasks of non-spatial relations, object states, and spatial relations offers insights into the predictive capabilities of the LLM and delineates areas that necessitate improvements.

In the *non-spatial relations* task, the analysis revealed that the model performs exceptionally well for most actions, achieving perfect success rates. However, deviations are observed in the actions "release" and "slice_carvingknife" with success rates of 0.00 and 0.00, respectively. The low success

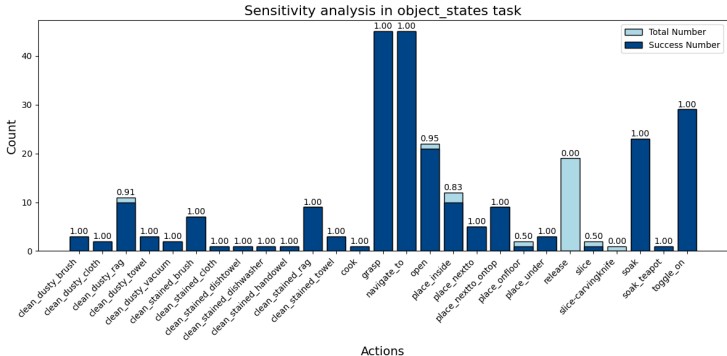

Figure 45: Sensitivity analysis for object states tasks in BEHAVIOR

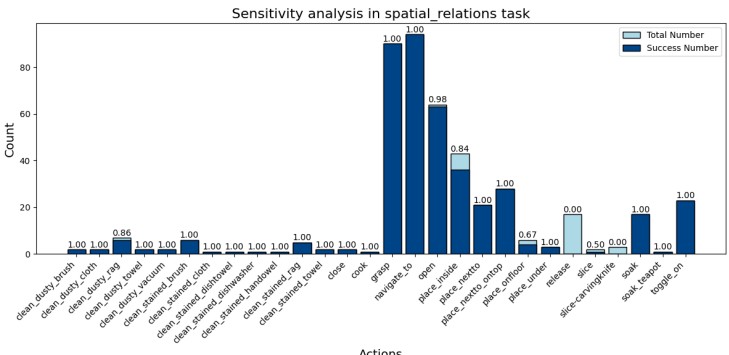

Figure 46: Sensitivity analysis for spatial relations tasks in BEHAVIOR

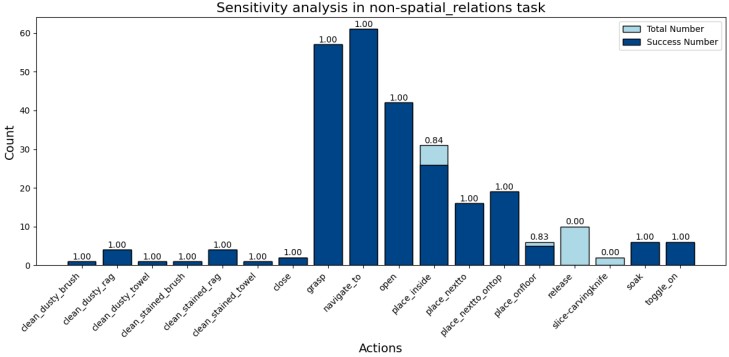

Figure 47: Sensitivity analysis for non-spatial relations tasks in BEHAVIOR

rate for "slice_carvingknife" suggests that the model struggles significantly with actions that involve complex interactions, including tool use and precise movements, potentially due to inadequate representation in the training data.

For the *object states* task, similar trends are evident where actions such as "grasp" and "release" exhibit lower success rates of 0.95 and 0.83, respectively, and "slice" again recorded a success rate of 0.00. These findings indicate that tasks requiring detailed manipulation of object states are challenging for the model, pointing to a gap in training on nuanced and precise operations.

The *spatial relations* task also highlights robust performances in most actions, yet "grasp" and "slice" show lower success rates of 0.84 and 0.00. The consistent underperformance on "slice" across different tasks underscores a significant deficiency in the model's ability to handle precise tool-based actions, while the challenges with "grasp" suggest difficulties in scenarios that demand fine motor skills or specific spatial awareness.

Across all categories in the BEHAVIOR dataset, as shown in figure 44, the model's proficiency in straightforward actions contrasts with its struggles in more complex or nuanced actions. For example, "place_inside" shows a rather low success rate in many tasks, probably because it requires an understanding of both spatial relations like "inside", and non-spatial relations like "holding". The agent also needs to be careful whether the robot currently is "handsfull". All of these decisions are challenging for current LLMs. Generally, figure 44 underscores a limitation in the model's current training, which may not sufficiently encompass the complexity of real-world interactions or the specificity required for precise task execution.

Overall, the model performs commendably on tasks involving basic manipulations but shows limitations in complex spatial judgments or detailed object interactions. To address these deficiencies, it is recommended to enhance the model's training with more diverse scenarios focusing on underperforming actions, fine-tune using specialized datasets, and conduct a thorough error analysis to precisely identify and rectify the sources of errors. This strategy will not only improve the model's performance but also expand its applicability across varied tasks, reinforcing its utility in complex real-world applications.

---

**Takeaways of Sensitivity Analysis**

(1) **Action-Specific Sensitivities**: The analysis reveals significant variations in the success rates for specific actions across tasks, with actions like "plug_in" and "walk_towards" consistently showing low success due to complex preconditions and spatial requirements. This underscores the need for LLMs to better understand and generate actions with multi-faceted constraints.

(2) **Challenges with Complex Interactions**: Complex interactions involving detailed object manipulation, such as "slice_carvingknife" and "place_inside," present notable challenges, highlighting deficiencies in the model's training on precise movements and tool use. Enhancing data representation for such scenarios in training sets is crucial for improving performance.

(3) **Robustness in Basic Tasks vs. Nuanced Deficiencies**: While the model performs well in straightforward tasks, it struggles with nuanced actions that require fine motor skills or advanced spatial awareness. This contrast points to the necessity of incorporating more complex and varied interaction scenarios in training to refine the model's capabilities in handling real-world complexities.

(4) **Training and Fine-Tuning Needs**: The consistent issues across different types of tasks suggest that the current training regimens may not fully capture the diversity of real-world interactions, especially in spatial and object-oriented tasks. A targeted approach to fine-tuning the model with specialized datasets that focus on underperforming actions could lead to significant improvements in model robustness and applicability in practical settings.

# G Pipeline-Based vs Modularized

## G.1 Motivation and Problem Formulation

**Motivation** In previous sections, we examined four modularized ability modules in EMBODIED AGENT INTERFACE. Despite covering all four abilities, it remains important to understand how LLMs address complete tasks when provided with natural goals instead of symbolic ones. This is crucial because, in most settings, only natural language goal descriptions are available, and symbolic goals are often missing. Natural language descriptions can be ambiguous. For instance, the natural language instruction *cook food* can be as simple as *cook the chicken then eat it*, whereas its symbolic goal should include detailed information such as `cooked(chicken) and ontop(chicken, plate) and ontop(plate, table)`. One advantage of our interface is the support for the flexible composition of primitive modules. This allows us to first have LLMs parse natural language goals into symbolic goals using the goal interpretation ($\mathcal{G}$) module before conducting further analysis. Therefore, we implement pipeline-based methods to further investigate such ability of LLMs.

**Problem Formulation** Given an initial state $s_0$ and a natural language goal $l_g$, our objective is to have LLMs generate a feasible action sequence $\bar{a}$ to achieve $l_g$ and to decompose $l_g$ into a subgoal trajectory $\bar{\phi}$.

## G.2 Implementation Details

We evaluate LLM abilities through pipeline-based methods in BEHAVIOR environment. Using the three methods—Goal Interpretation ($\mathcal{G}$), Action Sequencing ($\mathcal{Q}$), and Subgoal Decomposition ($\Phi$) provided in EMBODIED AGENT INTERFACE, we can establish two pipelines to obtain $\bar{a}$ and $\bar{\phi}$ respectively. Specifically, by leveraging the rule $\mathcal{G} : \langle s_0, l_g \rangle \to g$, we derive the symbolic goal $g$ corresponding to $l_g$. Then, we (1) apply the rule $\mathcal{Q} : \langle s_0, g \rangle \to \bar{a}$ to get the action sequence $\bar{a}$, and (2) apply the rule $\Phi : \langle s_0, g \rangle \to \bar{\phi}$ to derive the subgoal trajectory $\bar{\phi}$.

## G.3 Result Analysis

Table 16 presents a comparison of modularized methods and pipeline-based methods for pipelines $\mathcal{G} + \mathcal{Q}$ and $\mathcal{G} + \Phi$. Our observations indicate the following: (1) The trajectory executable rates are similar between both methods. (2) Pipeline-based methods suffer from error accumulation due to the composition of two modules.

These findings are consistent across both pipelines $\mathcal{G} + \mathcal{Q}$ and $\mathcal{G} + \Phi$. We attribute our observations to two main reasons: First, achieving an executable trajectory requires an understanding of semantics, which is largely related to the provided environment and vocabulary. Since these factors remain unchanged between modularized and pipeline-based methods, Language Models (LLMs) generate comparable executable trajectories for both. Second, pipeline-based methods tend to perform worse in terms of goal success rate. Errors made by the goal interpretation model mislead the LLMs in downstream modules, causing them to generate less accurate action and subgoal sequences aimed at the goal. Nevertheless, despite these challenges, LLMs are still capable of generating feasible and executable action sequences.

Furthermore, Table 16 demonstrates that while most state-of-the-art (SOTA) LLMs tend to avoid grammar errors. However, this finding does not hold for less advanced models, including Gemini 1.0 Pro, GPT-3.5-turbo, and some open-sourced LLMs. We attribute this to the fact that most SOTA LLMs are fine-tuned to follow human instructions more closely, whereas less advanced models either have smaller sizes or undergo less rigorous tuning strategies. Regarding runtime errors, all LLMs, regardless of their advancement, tend to miss the necessary steps in their generation process. This observation also aligns with our findings discussed in Section E.2.

(1) **Similar Trajectory Execution Rates:** Both modularized and pipeline-based methods have similar trajectory executable rates for pipelines $\mathcal{G} + \mathcal{Q}$ and $\mathcal{G} + \Phi$.

(2) **Error Accumulation in Pipelines:** Pipeline-based methods suffer from error accumulation, particularly due to the composition of two modules, which impacts their performance negatively.

(4) **Runtime Errors Persist:** All LLMs, regardless of their level of advancement, are prone to runtime errors, missing necessary steps in their generation process.

Table 16: Pipeline-based evaluation results for (1)$\mathcal{G} + \mathcal{Q}$ and (2) $\mathcal{G} + \Phi$ in BEHAVIOR . $\mathcal{G}$: Goal Interpretation. $\mathcal{Q}$: Action Sequencing. $\Phi$: Subgoal Decomposition. In this table, M means 'modularized', whereas P means 'pipeline-based'.

| | Goal Evaluation | | | | Trajectory Evaluation | | | | | | | | | | | | | |
| | Task SR | | Execution SR | | Grammar Error (↓) | | | | | | Runtime Error (↓) | | | | | | | |
| | | | | | Parsing | | Hallucination | | Predicate-Arg Num | | Wrong Order | | Missing Step | | Affordance | | Additional Step | |
| Model | M | P | M | P | M | P | M | P | M | P | M | P | M | P | M | P | M | P |
|---|---|---|---|---|---|---|---|---|---|---|---|---|---|---|---|---|---|---|
| *Goal Interpretation + Action Sequencing* | | | | | | | | | | | | | | | | | | |
| Claude-3 Haiku | 26.0 | 21.0 | 32.0 | 29.0 | 0.0 | 0.0 | 6.0 | 6.0 | 0.0 | 0.0 | 7.0 | 6.0 | 54.0 | 52.0 | 1.0 | 7.0 | 1.0 | 17.0 |
| Claude-3 Sonnet | 44.0 | 41.0 | 57.0 | 53.0 | 0.0 | 0.0 | 1.0 | 3.0 | 0.0 | 0.0 | 11.0 | 14.0 | 19.0 | 21.0 | 11.0 | 9.0 | 2.0 | 12.0 |
| Claude-3 Opus | **51.0** | **46.0** | 59.0 | 54.0 | 0.0 | 1.0 | 0.0 | 1.0 | 0.0 | 0.0 | 3.0 | 6.0 | 35.0 | 35.0 | 3.0 | 3.0 | 2.0 | 4.0 |
| Gemini 1.0 Pro | 27.0 | 26.0 | 32.0 | 35.0 | 7.0 | 5.0 | 3.0 | 3.0 | 6.0 | 6.0 | 13.0 | 14.0 | 35.0 | 38.0 | 4.0 | 2.0 | 4.0 | 11.0 |
| Gemini 1.5 Flash | 40.0 | 35.0 | 52.0 | 49.0 | 0.0 | 0.0 | 0.0 | 2.0 | 0.0 | 0.0 | 5.0 | 10.0 | 42.0 | 41.0 | 1.0 | 0.0 | 2.0 | 7.0 |
| Gemini 1.5 Pro | 42.0 | 37.0 | 54.0 | 55.0 | 0.0 | 1.0 | 0.0 | 1.0 | 0.0 | 0.0 | 6.0 | 7.0 | 39.0 | 35.0 | 1.0 | 1.0 | 2.0 | 0.0 |
| GPT-3.5-turbo | 16.0 | 14.0 | 20.0 | 32.0 | 4.0 | 1.0 | 7.0 | 3.0 | 23.0 | 15.0 | 1.0 | 5.0 | 36.0 | 39.0 | 8.0 | 6.0 | 1.0 | 3.0 |
| GPT-4-turbo | 38.0 | 32.0 | 45.0 | 47.0 | 0.0 | 1.0 | 0.0 | 1.0 | 0.0 | 0.0 | 7.0 | 9.0 | 47.0 | 41.0 | 1.0 | 1.0 | 0.0 | 0.0 |
| GPT-4o | 47.0 | 42.0 | 53.0 | 55.0 | 0.0 | 0.0 | 0.0 | 1.0 | 0.0 | 0.0 | 9.0 | 6.0 | 36.0 | 35.0 | 1.0 | 1.0 | 0.0 | 4.0 |
| Cohere Command R | 16.0 | 5.0 | 19.0 | 9.0 | 5.0 | 3.0 | 13.0 | 38.0 | 0.0 | 1.0 | 8.0 | 8.0 | 43.0 | 31.0 | 12.0 | 12.0 | 4.0 | 8.0 |
| Cohere Command R+ | 27.0 | 15.0 | 35.0 | 29.0 | 0.0 | 0.0 | 1.0 | 8.0 | 15.0 | 14.0 | 10.0 | 30.0 | 39.0 | 31.0 | 0.0 | 2.0 | 15.0 | 22.0 |
| Mistral Large | 33.0 | 31.0 | 50.0 | 38.0 | 0.0 | 0.0 | 1.0 | 3.0 | 0.0 | 0.0 | 8.0 | 14.0 | 35.0 | 37.0 | 6.0 | 8.0 | 7.0 | 5.0 |
| Mixtral 8x22B MoE | 30.0 | 26.0 | 40.0 | 36.0 | 3.0 | 3.0 | 6.0 | 13.0 | 0.0 | 0.0 | 10.0 | 14.0 | 32.0 | 21.0 | 9.0 | 13.0 | 2.0 | 15.0 |
| Llama 3 8B | 10.0 | 0.0 | 16.0 | 5.0 | 0.0 | 2.0 | 15.0 | 25.0 | 9.0 | 6.0 | 6.0 | 11.0 | 44.0 | 34.0 | 9.0 | 17.0 | 5.0 | 14.0 |
| Llama 3 70B | 34.0 | 26.0 | 42.0 | 40.0 | 0.0 | 1.0 | 2.0 | 3.0 | 0.0 | 0.0 | 15.0 | 18.0 | 38.0 | 35.0 | 3.0 | 5.0 | 6.0 | 9.0 |
| *Goal Interpretation + Subgoal Decomposition* | | | | | | | | | | | | | | | | | | |
| Claude-3 Haiku | 29.0 | 21.0 | 35.0 | 40.0 | 0.0 | 0.0 | 1.0 | 5.0 | 0.0 | 0.0 | 2.0 | 2.0 | 59.0 | 46.0 | 3.0 | 7.0 | 3.0 | 16.0 |
| Claude-3 Sonnet | 38.0 | 31.0 | 43.0 | 45.0 | 0.0 | 0.0 | 2.0 | 3.0 | 0.0 | 0.0 | 3.0 | 2.0 | 51.0 | 47.0 | 1.0 | 3.0 | 3.0 | 18.0 |
| Claude-3 Opus | 39.0 | 35.0 | 47.0 | 45.0 | 0.0 | 0.0 | 3.0 | 8.0 | 0.0 | 0.0 | 5.0 | 4.0 | 45.0 | 42.0 | 0.0 | 1.0 | 5.0 | 7.0 |
| Gemini 1.0 Pro | 23.0 | 14.0 | 33.0 | 30.0 | 2.0 | 0.0 | 4.0 | 10.0 | 0.0 | 1.0 | 3.0 | 1.0 | 51.0 | 45.0 | 7.0 | 13.0 | 3.0 | 17.0 |
| Gemini 1.5 Flash | 34.0 | 32.0 | 42.0 | 44.0 | 2.0 | 1.0 | 1.0 | 3.0 | 0.0 | 0.0 | 2.0 | 2.0 | 53.0 | 48.0 | 0.0 | 2.0 | 3.0 | 7.0 |
| Gemini 1.5 Pro | 31.0 | 26.0 | 37.0 | 38.0 | 0.0 | 1.0 | 1.0 | 3.0 | 0.0 | 0.0 | 3.0 | 2.0 | 59.0 | 56.0 | 0.0 | 0.0 | 2.0 | 1.0 |
| GPT-3.5-turbo | 24.0 | 14.0 | 36.0 | 27.0 | 2.0 | 0.0 | 3.0 | 12.0 | 0.0 | 22.0 | 3.0 | 1.0 | 52.0 | 32.0 | 4.0 | 6.0 | 3.0 | 5.0 |
| GPT-4-turbo | 37.0 | 37.0 | 47.0 | 49.0 | 0.0 | 0.0 | 3.0 | 4.0 | 0.0 | 0.0 | 9.0 | 8.0 | 40.0 | 37.0 | 1.0 | 2.0 | 6.0 | 6.0 |
| GPT-4o | 48.0 | 38.0 | 55.0 | 52.0 | 0.0 | 0.0 | 3.0 | 4.0 | 0.0 | 0.0 | 5.0 | 6.0 | 37.0 | 35.0 | 0.0 | 3.0 | 5.0 | 9.0 |
| Cohere Command R | 15.0 | 8.0 | 25.0 | 15.0 | 21.0 | 13.0 | 11.0 | 32.0 | 0.0 | 1.0 | 0.0 | 1.0 | 38.0 | 32.0 | 4.0 | 6.0 | 4.0 | 12.0 |
| Cohere Command R+ | 24.0 | 17.0 | 37.0 | 31.0 | 2.0 | 6.0 | 4.0 | 10.0 | 0.0 | 2.0 | 5.0 | 7.0 | 51.0 | 40.0 | 1.0 | 4.0 | 6.0 | 14.0 |
| Mistral Large | 30.0 | 22.0 | 38.0 | 29.0 | 1.0 | 1.0 | 3.0 | 12.0 | 0.0 | 1.0 | 4.0 | 5.0 | 52.0 | 50.0 | 2.0 | 2.0 | 1.0 | 5.0 |
| Mixtral 8x22B MoE | 27.0 | 22.0 | 33.0 | 29.0 | 0.0 | 0.0 | 4.0 | 9.0 | 0.0 | 2.0 | 2.0 | 2.0 | 59.0 | 45.0 | 2.0 | 13.0 | 0.0 | 17.0 |
| Llama 3 8B | 21.0 | 3.0 | 29.0 | 14.0 | 2.0 | 7.0 | 11.0 | 29.0 | 0.0 | 2.0 | 6.0 | 3.0 | 44.0 | 30.0 | 8.0 | 15.0 | 7.0 | 7.0 |
| Llama 3 70B | 20.0 | 19.0 | 30.0 | 31.0 | 1.0 | 1.0 | 5.0 | 22.0 | 1.0 | 1.0 | 8.0 | 7.0 | 51.0 | 35.0 | 4.0 | 3.0 | 4.0 | 7.0 |

# H   Replanning and Feedback

## H.1   Motivation and Problem Formulation

In our current setting, the agent has only one chance to generate plans or make predictions regardless of previous failures and warnings. However, in reality, it is an important ability for an agent to learn from feedback and re-plan based on the failure and warning. It demonstrates the agent's ability to make agile adjustments based on the simulator execution and environment. Practically, replanning and feedback also help to prevent the agent from making the same mistakes over and over again. Therefore, we regard the model's ability to re-plan from feedback as necessary.[9][81][82]

## H.2    Implementation Details

To evaluate the agent's ability to re-plan from the feedback, for each unsuccessful task, either failed in the execution (grammar error, runtime error, etc.), or failed in the goal satisfaction check (unsatisfied state goals, relation goals, or action goals), we construct the feedback as the error message and necessary information, and append them to the task prompt. We provide the previous agent's plan as *"At the [retry_cnt] retry, LLM predict the action sequence to be [predicted_action]"*, and detail the reason why it fails. For example, for runtime error missing step, the error message will be *Action [action] is not executable in the action sequence [actions]. It encounters an error: MISSING STEP. Missing step means that action [action] needs some other necessary action before its execution.".* Another error message example for unsatisfied goals is *Action sequence [actions] does not satisfy all the goals. Please check the action sequence and try again. Specifically, the following goals are not satisfied: Node goals not satisfied: [unsatisfied_node_goals] Edge goals not satisfied: [unsatisfied_edge_goals] Action goals not satisfied: [unsatisfied_action_goals].* We allow the agent to re-plan at most 3 times, and each time, we append all previous attempts and error messages to the feedback. We test this setting on the action sequencing task with model GPT-4o.

## H.3    Result Analysis

Table 17: Replanning evaluation results (%) for *action sequencing*

| Model | Goal Evaluation | | Trajectory Evaluation | | | | | | |
| | | | Grammar Error (↓) | | | Runtime Error (↓) | | | |
| | *Task SR* | *Execution SR* | *Parsing* | *Hallucination* | *Action-Arg Num* | *Wrong Order* | *Missing Step* | *Affordance* | *Additional Step* |
| GPT-4o | 65.2 | 71.8 | **0.0** | **1.3** | 0.7 | **0.0** | 25.3 | 1.0 | **0.3** |
| GPT-4o w/ replanning | **77.4** | **83.3** | **0.0** | **1.3** | **0.0** | **0.0** | **14.1** | **0.3** | 0.7 |

From table 17, we observe that with replanning, the model gains improvement by more than 10% on the success rate and executable success rate. Other metrics also gain improvement except for the additional steps error rate. A possible reason could be that the agent sometimes tries to iterate over previous attempts and make modifications based on those attempts. Such a strategy makes agents easily overstate some actions and cause additional step warnings.

---

**Takeaways of Replanning and Feedback**

(1) **Enhanced Adaptability through Replanning**: Incorporating the ability for models to replan based on feedback significantly improves their performance, demonstrating over a 10% increase in success rates. This shows that LLMs can effectively learn from their errors and adjust their action sequences to better align with the required task outcomes, enhancing overall adaptability. Replanning also helps to reduce the repetition of similar errors, showcasing the model's ability to evolve its strategy over successive iterations.

(2) **Risk of Over-Correction**: While replanning generally leads to improvements in task execution, it can sometimes result in the over-generation of actions, as indicated by an increased rate of additional step errors. This suggests that while the model is adept at adjusting to feedback, it may also benefit from more precise guidance to avoid introducing unnecessary actions in its revised plans.

---

## H.4    Replanning with Stochastic Actions

We also explore other failing cases, such as action failures in action sequencing. Specifically, we add a new experiment to simulate stochastic actions with a certain failure probability and allow replanning three times. The experiments are done on GPT-4o for action sequencing tasks in the BEHAVIOR simulator.

Table 18: Replanning for Stochastic Actions on BEHAVIOR (Exp.6)

| Fail Probability | Method | Execution SR (%) | Goal SR (%) |
|---|---|---|---|
| 0.05 | w/o Replanning | 60 | 50.0 |
| | w/ Replanning | 85 (↑25) | 65 (↑15) |
| 0.1 | w/o Replanning | 25 | 15.0 |
| | w/ Replanning | 70 (↑45) | 55 (↑40) |
| 0.2 | w/o Replanning | 10 | 5.0 |
| | w/ Replanning | 65 (↑55) | 45 (↑40) |

From Table 18, we show that (1) replanning can be helpful; (2) the gap between with and without replanning generally increases when the failure probability is larger.

# I  Prompt and Analysis

## I.1  Prompt of Goal Interpretation

We design the prompt template of goal interpretation as below:

---

**Prompt of Goal Interpretation**

```
# Behavior Goal Interpretation Prompt Template

SYSTEM_PROMPT = "For this task, please only output a parsable json string inside
    brackets. Please start your answer with { and end your answer with }. Don't
    include any notes or explanations with the output json string."

USER_PROMPT = "You are a helpful assistant for goal interpretation in an embodied
    environment. You should only output in json format. Your task is to understand
    natural language goals for a household robot, reason about the object states and
    relationships, and turn natural language goals into symbolic goal states in the
    designated format. The goals include: unary goals describing one object's own
    unary states, and binary goals describing object-object binary relationships.
    The input will be the goal's name, the goal's description, relevant objects as
    well as their possible unary states, and all initial unary and binary states.
    The output should be the symbolic version of the goal states.

    Relevant objects in the scene indicates those objects involved in the action
    execution initially. It will include the object name, and the object's all
    possible unary states (In goal conditions, each state can be set to true or not
    true). It follows the format: object id, possible unary
    states: ...(all possible unary states). Your proposed unary object states should
    be within the following set: {'Cooked', 'Open', 'Frozen', 'Dusty', 'Stained', '
    Sliced', 'Soaked', 'Toggled_On'}

    Relevant objects in the scene are:
    <object_in_scene>

    All initial states in the scene are:
    <all_initial_states>

    Symbolic goals format:

    Node goal states should be a set indicating the desired final goal states of
    single objects. Each goal in the list should be a list with two elements: the
    first element is the state name, which comes from the set {'Cooked', 'Open', '
    Frozen', 'Dusty', 'Stained', 'Sliced', 'Soaked', 'Toggled_On'}; the second
    element is the object name, which comes from the list of relevant objects in the
    scene provided above.

    Edge goal states should be a set indicating the desired binary relationships
    between two objects. Each goal state in the set is a list of three elements: the
    first element is the state name, which comes from the set {'NextTo', 'Inside',
    'OnFloor', 'Touching', 'Under'}, the second and third elements are the object
    names, with relationship as indicated by the first element.
```

---

```
        Here is an example of desired node and edge goal states:
        <in_context_example>

        Task Name and Goal Instructions:
        <instructions_str>

        Now using json format, output just the symbolic version of the goal states
         without any explanation. Output a single json object string, whose keys are '
         node goals' and 'edge goals', and values are your output of symbolic node goals
         and symbolic edge goals, respectively. That is, your output should be of the
         format: {'node goals': SYMBOLIC_NODE_GOALS, 'edge goals': SYMBOLIC_EDGE_GOALS}.
         Also, please strictly follow the aforementioned symbolic goal format."
```

Figure 48: Examples prompt of goal interpretation.

## I.2   Prompt of Subgoal Decomposition

We design the prompt template of subgoal decomposition as below:

**Prompt of Subgoal Decomposition**

```
# Behavior Subgoal Decomposition Prompt Template

SYSTEM_PROMPT = '''# Background Introduction
    You are determining the complete state transitions of a household task solving by
     a robot. The goal is to list all intermediate states (which is called subgoals
     as a whole) to achieve the final goal states from the initial states. The output
      consists of a list of boolean expression, which is a combination of the state
     predicates. Note that your output boolean expression list is in temporal order,
     therefore, it must be consistent and logical. In short, your task is to output
     the subgoal plan in the required format.

    # Data Vocabulary Introduction
    Below we introduce the detailed data vocabulary and their format that you can use
      to generate the subgoal plan.
    ## Available States
    The state is represented as a first-order predicate, which is a tuple of a
     predicate name and its arguments. Its formal definition looks like this "<
     PredicateName>(Params)", where <PredicateName> is the state name and each param
     should be ended with an id. An example is "inside(coin.1, jar.1)". Below is a
     list of available states and their descriptions.
    <available_behavior_states>
    ## Available Connectives
    The connectives are used to satisfy the complex conditions. They are used to
     combine the state predicates.
    <available_behavior_connectives>
    # Rules You Must Follow
    - The initial states are the states that are given at the beginning of the task.
    - Your output must be a list of boolean expressions that are in temporal order.
    - You must follow the data format and the available states and connectives
     defined above.
    - The output must be consistent, logical and as detailed as possible. View your
     output as a complete state transition from the initial states to the final goal
     states.

    - Please note that the robot can only hold one object in one hand. Also, the
     robot needs to have at least one hand free to perform any action other than put,
      place, take, hold.
    - Use holds_rh and holds_lh in your plan if necessary. For example, stained(shoe)
      cannot directly change to not stained(shoe), but needs intermediate states like
      [soaked(rag), holds_rh(rag), not stained(shoe)] or [holds_rh(detergent), not
     stained(shoe)].
    - Your output follows the temporal order line by line. If you think there is no
     temporal order requirement for certain states, you can use connective "and" to
     combine them. If you think some states are equivalent, you can use connective "
     or" to combine them.
    - Please use provided relevant objects well to help you achieve the final goal
     states. Note that inside(obj1, agent) is an invalid state, therefore you cannot
     output it in your plan.
    - Do not output redundant states. A redundant state means a state that is either
     not necessary or has been satisfied before without broken.
```

```
        - Your must strictly follow the json format like this {"output": [<your subgoal
          plan>]}, where <your subgoal plan> is a list of boolean expressions presented in
          the temporal order.
        - Start your output with "{" and end with "}". For each line of the output, DO
          NOT INCLUDE IRRELEVANT INFORMATION (like number of line, explanation, etc.).

        # Example: Task is bottling_fruit
        Below we provide an example for your better understanding.
        <in_context_example>'''

USER_PROMPT = '''Now, it is time for you to generate the subgoal plan for the
        following task.
        # Target Task: <task_name>
        ## Relevant objects in this scene
        <relevant_objects>
        ## Initial States
        <initial_states>
        ## Goal States
        <goal_states>
        ## Output: Based on initial states in this task, achieve final goal states
          logically and reasonably. It does not matter which state should be satisfied
          first, as long as all goal states can be satisfied at the end. Make sure your
          output follows the json format, and do not include irrelevant information, do
          not include any explanation. Output concrete states and do not use quantifiers
          like "forall" or "exists".'''
```

Figure 49: Examples prompt of subgoal decomposition.

## I.3 Prompt of Action Sequencing

We design the prompt template of action sequencing as below:

**Prompt of Action Sequencing**

```
SYSTEM_PROMPT = """For this task, please only output a parsable json string inside
    brackets. Please start your answer with [ and end your answer with ]. Don't
    include any notes or explanations with the output json string."""

USER_PROMPT="""

Problem:
You are designing instructions for a household robot.
The goal is to guide the robot to modify its environment from an initial state to a
    desired final state.
The input will be the initial environment state, the target environment state, the
    objects you can interact with in the environment.
The output should be a list of action commands so that after the robot executes the
    action commands sequentially, the environment will change from the initial state
    to the target state.

Data format: After # is the explanation.

Format of the states:
The environment state is a list starts with a uniary predicate or a binary prediate,
    followed by one or two obejcts.
You will be provided with multiple environment states as the initial state and the
    target state.
For example:
['inside', 'strawberry_0', 'fridge_97'] #strawberry_0 is inside fridge_97
['not', 'sliced', 'peach_0'] #peach_0 is not sliced
['ontop', 'jar_1', 'countertop_84'] #jar_1 is on top of countertop_84

Format of the action commands:
Action commands is a dictionary with the following format:
{{
        'action': 'action_name',
        'object': 'target_obj_name',
}}

or
```

```
{{
        'action': 'action_name',
        'object': 'target_obj_name1,target_obj_name2',
}}

The action_name must be one of the following:
LEFT_GRASP # the robot grasps the object with its left hand, to execute the action,
    the robot's left hand must be empty, e.g. {{'action': 'LEFT_GRASP', 'object': '
    apple_0'}}.
RIGHT_GRASP # the robot grasps the object with its right hand, to execute the action,
     the robot's right hand must be empty, e.g. {{'action': 'RIGHT_GRASP', 'object':
     'apple_0'}}.
LEFT_PLACE_ONTOP # the robot places the object in its left hand on top of the target
    object and release the object in its left hand, e.g. {{'action': '
    LEFT_PLACE_ONTOP', 'object': 'table_1'}}.
RIGHT_PLACE_ONTOP # the robot places the object in its right hand on top of the
    target object and release the object in its left hand, e.g. {{'action': '
    RIGHT_PLACE_ONTOP', 'object': 'table_1'}}.
LEFT_PLACE_INSIDE # the robot places the object in its left hand inside the target
    object and release the object in its left hand, to execute the action, the robot
    's left hand must hold an object, and the target object can't be closed e.g. {{'
    action': 'LEFT_PLACE_INSIDE', 'object': 'fridge_1'}}.
RIGHT_PLACE_INSIDE # the robot places the object in its right hand inside the target
    object and release the object in its left hand, to execute the action, the robot
    's right hand must hold an object, and the target object can't be closed, e.g.
    {{'action': 'RIGHT_PLACE_INSIDE', 'object': 'fridge_1'}}.
RIGHT_RELEASE # the robot directly releases the object in its right hand, to execute
    the action, the robot's left hand must hold an object, e.g. {{'action': '
    RIGHT_RELEASE', 'object': 'apple_0'}}.
LEFT_RELEASE # the robot directly releases the object in its left hand, to execute
    the action, the robot's right hand must hold an object, e.g. {{'action': '
    LEFT_RELEASE', 'object': 'apple_0'}}.
OPEN # the robot opens the target object, to execute the action, the target object
    should be openable and closed, also, toggle off the target object first if want
    to open it, e.g. {{'action': 'OPEN', 'object': 'fridge_1'}}.
CLOSE # the robot closes the target object, to execute the action, the target object
    should be openable and open, e.g. {{'action': 'CLOSE', 'object': 'fridge_1'}}.
COOK # the robot cooks the target object, to execute the action, the target object
    should be put in a pan, e.g. {{'action': 'COOK', 'object': 'apple_0'}}.
CLEAN # the robot cleans the target object, to execute the action, the robot should
    have a cleaning tool such as rag, the cleaning tool should be soaked if possible
    , or the target object should be put into a toggled on cleaner like a sink or a
    dishwasher, e.g. {{'action': 'CLEAN', 'object': 'window_0'}}.
FREEZE # the robot freezes the target object e.g. {{'action': 'FREEZE', 'object': '
    apple_0'}}.
UNFREEZE # the robot unfreezes the target object, e.g. {{'action': 'UNFREEZE', '
    object': 'apple_0'}}.
SLICE # the robot slices the target object, to execute the action, the robot should
    have a knife in hand, e.g. {{'action': 'SLICE', 'object': 'apple_0'}}.
SOAK # the robot soaks the target object, to execute the action, the target object
    must be put in a toggled on sink, e.g. {{'action': 'SOAK', 'object': 'rag_0'}}.
DRY # the robot dries the target object, e.g. {{'action': 'DRY', 'object': 'rag_0'}}.
TOGGLE_ON # the robot toggles on the target object, to execute the action, the target
     object must be closed if the target object is openable and open e.g. {{'action
    ': 'TOGGLE_ON', 'object': 'light_0'}}.
TOGGLE_OFF # the robot toggles off the target object, e.g. {{'action': 'TOGGLE_OFF',
    'object': 'light_0'}}.
LEFT_PLACE_NEXTTO # the robot places the object in its left hand next to the target
    object and release the object in its left hand, e.g. {{'action': '
    LEFT_PLACE_NEXTTO', 'object': 'table_1'}}.
RIGHT_PLACE_NEXTTO # the robot places the object in its right hand next to the target
     object and release the object in its right hand, e.g. {{'action': '
    RIGHT_PLACE_NEXTTO', 'object': 'table_1'}}.
LEFT_TRANSFER_CONTENTS_INSIDE # the robot transfers the contents in the object in its
     left hand inside the target object, e.g. {{'action': '
    LEFT_TRANSFER_CONTENTS_INSIDE', 'object': 'bow_1'}}.
RIGHT_TRANSFER_CONTENTS_INSIDE # the robot transfers the contents in the object in
    its right hand inside the target object, e.g. {{'action': '
    RIGHT_TRANSFER_CONTENTS_INSIDE', 'object': 'bow_1'}}.
LEFT_TRANSFER_CONTENTS_ONTOP # the robot transfers the contents in the object in its
    left hand on top of the target object, e.g. {{'action': '
    LEFT_TRANSFER_CONTENTS_ONTOP', 'object': 'table_1'}}.
RIGHT_TRANSFER_CONTENTS_ONTOP # the robot transfers the contents in the object in its
     right hand on top of the target object, e.g. {{'action': '
    RIGHT_TRANSFER_CONTENTS_ONTOP', 'object': 'table_1'}}.
LEFT_PLACE_NEXTTO_ONTOP # the robot places the object in its left hand next to target
     object 1 and on top of the target object 2 and release the object in its left
    hand, e.g. {{'action': 'LEFT_PLACE_NEXTTO_ONTOP', 'object': 'window_0, table_1
    '}}.
```

```
RIGHT_PLACE_NEXTTO_ONTOP # the robot places the object in its right hand next to
    object 1 and on top of the target object 2 and release the object in its right
    hand, e.g. {{'action': 'RIGHT_PLACE_NEXTTO_ONTOP', 'object': 'window_0, table_1
    '}}.
LEFT_PLACE_UNDER # the robot places the object in its left hand under the target
    object and release the object in its left hand, e.g. {{'action': '
    LEFT_PLACE_UNDER', 'object': 'table_1'}}.
RIGHT_PLACE_UNDER # the robot places the object in its right hand under the target
    object and release the object in its right hand, e.g. {{'action': '
    RIGHT_PLACE_UNDER', 'object': 'table_1'}}.

Format of the interactable objects:
Interactable object will contain multiple lines, each line is a dictionary with the
    following format:
{{
    'name': 'object_name',
    'category': 'object_category'
}}
object_name is the name of the object, which you must use in the action command,
    object_category is the category of the object, which provides a hint for you in
    interpreting initial and goal condtions.

Please pay specail attention:
1. The robot can only hold one object in each hand.
2. Action name must be one of the above action names, and the object name must be one
     of the object names listed in the interactable objects.
3. All PLACE actions will release the object in the robot's hand, you don't need to
    explicitly RELEASE the object after the PLACE action.
4. For LEFT_PLACE_NEXTTO_ONTOP and RIGHT_PLACE_NEXTTO_ONTOP, the action command are
    in the format of {{'action': 'action_name', 'object': 'obj_name1, obj_name2'}}
5. If you want to perform an action to an target object, you must make sure the
    target object is not inside a closed object.
6. For actions like OPEN, CLOSE, SLICE, COOK, CLEAN, SOAK, DRY, FREEZE, UNFREEZE,
    TOGGLE_ON, TOGGLE_OFF, at least one of the robot's hands must be empty, and the
    target object must have the corresponding property like they're openable,
    toggleable, etc.
7. For PLACE actions and RELEASE actions, the robot must hold an object in the
    corresponding hand.
8. Before slicing an object, the robot can only interact with the object (e.g.
    peach_0), after slicing the object, the robot can only interact with the sliced
    object (e.g. peach_0_part_0).

Examples: after# is the explanation.

Example 1:
Input:
initial environment state:
['stained', 'sink_7']
['stained', 'bathtub_4']
['not', 'soaked', 'rag_0']
['onfloor', 'rag_0', 'room_floor_bathroom_0']
['inside', 'rag_0', 'cabinet_1']
['not', 'open', 'cabinet_1']

target environment state:
['not', 'stained', 'bathtub_4']
['not', 'stained', 'sink_7']
['and', 'soaked', 'rag_0', 'inside', 'rag_0', 'bucket_0']

interactable objects:
{{'name': 'sink_7', 'category': 'sink.n.01'}}
{{'name': 'bathtub_4', 'category': 'bathtub.n.01'}}
{{'name': 'bucket_0', 'category': 'bucket.n.01'}}
{{'name': 'rag_0', 'category': 'rag.n.01'}}
{{'name': 'cabinet_1', 'category': 'cabinet.n.01'}}

Please output the list of action commands (in the given format) so that after the
    robot executes the action commands sequentially, the current environment state
    will change to target environment state. Usually, the robot needs to execute
    multiple action commands consecutively to achieve final state. Please output
    multiple action commands rather than just one. Only output the list of action
    commands with nothing else.

Output:
[
```

```
        {{
            'action': 'OPEN',
            'object': 'cabinet_1'
        }}, # you want to get the rag_0 from cabinet_1, should open it first
        {{
            'action': 'RIGHT_GRASP',
            'object': 'rag_0'
        }}, # you want to clean the sink_7 and bathtub_4, you found them stained, so you
         need to soak the rag_0 first
        {{
            'action': 'RIGHT_PLACE_INSIDE',
            'object': 'sink_7'
        }}, # to soak the rag_0, you need to place it inside the sink_7
        {{
            'action': 'TOGGLE_ON',
            'object': 'sink_7'
        }}, # to soak the rag_0, you need to toggle on the sink_7
        {{
            'action': 'SOAK',
            'object': 'rag_0'
        }}, # now you can soak the rag_0
        {{
            'action': 'TOGGLE_OFF',
            'object': 'sink_7'
        }}, # after soaking the rag_0, you need to toggle off the sink_7
        {{
            'action': 'LEFT_GRASP',
            'object': 'rag_0'
        }}, # now you can grasp soaked rag_0 to clean stain
        {{
            'action': 'CLEAN',
            'object': 'sink_7'
        }}, # now you clean the sink_7
        {{
            'action': 'CLEAN',
            'object': 'bathtub_4'
        }}, # now you clean the bathtub_4
        {{
            'action': 'LEFT_PLACE_INSIDE',
            'object': 'bucket_0'
        }} # after cleaning the sink_7, you need to place the rag_0 inside the bucket_0
]

Your task:
Input:
initial environment state:
{init_state}

target environment state:
{target_state}

interactable objects:
{obj_list}

Please output the list of action commands (in the given format) so that after the
    robot executes the action commands sequentially, the current environment state
    will change to target environment state. Usually, the robot needs to execute
    multiple action commands consecutively to achieve final state. Please output
    multiple action commands rather than just one. Only output the list of action
    commands with nothing else.

Output:
"""
```

Figure 50: Examples prompt of action sequencing.

## I.4   Prompt of Transition Modeling

We design the prompt template of transition modeling as below:

```
# Virtualhome Transition Modeling Prompt Template

SYSTEM_PROMPT = "You are a software engineer who will be writing action definitions
    for a household robot in the PDDL planning language given the problem file and
    predicates in domain file. For this task, please only output a parsable json
    string inside brackets. Please start your answer with { and end your answer with
     }. Don't include any notes or explanations with the output json string."

USER_PROMPT = "The following is predicates defined in this domain file. Pay attention
     to the types for each predicate.
(define (domain virtualhome)
(:requirements :typing)
    ;; types in virtualhome domain
    (:types
        object character  ; Define 'object' and 'character' as types
    )

    ;; Predicates defined on this domain. Note the types for each predicate.
    (:predicates
        (closed ?obj - object)  ; obj is closed
        (open ?obj - object)  ; obj is open
        (on ?obj - object)  ; obj is turned on, or it is activated
        (off ?obj - object)  ; obj is turned off, or it is deactivated
        (plugged_in ?obj - object)  ; obj is plugged in
        (plugged_out ?obj - object)  ; obj is unplugged
        (sitting ?char - character)  ; char is sitting, and this represents a state
    of a character
        (lying ?char - character)  ; char is lying
        (clean ?obj - object)  ; obj is clean
        (dirty ?obj - object)  ; obj is dirty
        (obj_ontop ?obj1 ?obj2 - object)  ; obj1 is on top of obj2
        (ontop ?char - character ?obj - object)  ; char is on obj
        (on_char ?obj - object ?char - character) ; obj is on char
        (inside_room ?obj ?room - object) ; obj is inside room
        (obj_inside ?obj1 ?obj2 - object)  ; obj1 is inside obj2
        (inside ?char - character ?obj - object)  ; char is inside obj
        (obj_next_to ?obj1 ?obj2 - object)  ; obj1 is close to or next to obj2
        (next_to ?char - character ?obj - object)  ; char is close to or next to obj
        (between ?obj1 ?obj2 ?obj3 - object)  ; obj1 is between obj2 and obj3
        (facing ?char - character ?obj - object)  ; char is facing obj
        (holds_rh ?char - character ?obj - object)  ; char is holding obj with right
    hand
        (holds_lh ?char - character ?obj - object)  ; char is holding obj with left
    hand
        (grabbable ?obj - object)  ; obj can be grabbed
        (cuttable ?obj - object)  ; obj can be cut
        (can_open ?obj - object)  ; obj can be opened
        (readable ?obj - object)  ; obj can be read
        (has_paper ?obj - object)  ; obj has paper
        (movable ?obj - object)  ; obj is movable
        (pourable ?obj - object)  ; obj can be poured from
        (cream ?obj - object)  ; obj is cream
        (has_switch ?obj - object)  ; obj has a switch
        (lookable ?obj - object)  ; obj can be looked at
        (has_plug ?obj - object)  ; obj has a plug
        (drinkable ?obj - object)  ; obj is drinkable
        (body_part ?obj - object)  ; obj is a body part
        (recipient ?obj - object)  ; obj is a recipient
        (containers ?obj - object)  ; obj is a container
        (cover_object ?obj - object)  ; obj is a cover object
        (surfaces ?obj - object)  ; obj has surfaces
        (sittable ?obj - object)  ; obj can be sat on
        (lieable ?obj - object)  ; obj can be lied on
        (person ?obj - object)  ; obj is a person
        (hangable ?obj - object)  ; obj can be hanged
        (clothes ?obj - object)  ; obj is clothes
        (eatable ?obj - object)  ; obj is eatable
        )
    ;; Actions to be predicted
)

Objective: Given the problem file of pddl, which defines objects in the task (:
    objects), initial conditions (:init) and goal conditions (:goal), write the body
     of PDDL actions (:precondition and :effect) given specific action names and
    parameters, so that after executing the actions in some order, the goal
    conditions can be reached from initial conditions.
```

```
Each PDDL action definition consists of four main components: action name, parameters
    , precondition, and effect. Here is the general format to follow:
(:action [action name]
  :parameters ([action parameters])
  :precondition ([action precondition])
  :effect ([action effect])
)

The :parameters is the list of variables on which the action operates. It lists
    variable names and variable types.

The :precondition is a first-order logic sentence specifying preconditions for an
    action. The precondition consists of predicates and 4 possible logical operators
    : or, and, not, exists!
1. The precondition should be structured in Disjunctive Normal Form (DNF), meaning an
    OR of ANDs.
2. The not operator should only be used within these conjunctions. For example, (or (
    and (predicate1 ?x) (predicate2 ?y)) (and (predicate3 ?x)))
3. Exists operator is followed by two parts, variable and body. It follows the format
    : exists (?x - variable type) (predicate1 ?x), which means there exists an
    object ?x of certain variable type, that predicate1 ?x satisfies.

The :effect lists the changes which the action imposes on the current state. The
    precondition consists of predicates and 6 possible logical operators: or, and,
    not, exists, when, forall.
1. The effects should generally be several effects connected by AND operators.
2. For each effect, if it is a conditional effect, use WHEN to check the conditions.
    The semantics of (when [condition] [effect]) are as follows: If [condition] is
    true before the action, then [effect] occurs afterwards.
3. If it is not a conditional effect, use predicates directly.
4. The NOT operator is used to negate a predicate, signifying that the condition will
    not hold after the action is executed.
5. Forall operator is followed by two parts, variable and body. It follows the format
    : forall (?x - variable type) (predicate1 ?x), which means there for all objects
    ?x of certain variable type, that predicate1 ?x satisfies.

6. An example of effect is (and (when (predicate1 ?x) (not (predicate2 ?y))) (
    predicate3 ?x))

Formally, the preconditions and effects are all clauses <Clause>.
<Clause> := (predicate ?x)
<Clause> := (and <Clause1> <Clause2> ...)
<Clause> := (or <Clause1> <Clause2> ...)
<Clause> := (not <Clause>)
<Clause> := (when <Clause1> <Clause2>)
<Clause> := (exists (?x - object type) <Clause>)
<Clause> := (forall (?x - object type) <Clause>)

In any case, the occurrence of a predicate should agree with its declaration in terms
    of number and types of arguments defined in DOMAIN FILE at the beginning.

Here is an example of the input problem file and unfinished action. Observe carefully
    how to think step by step to write the action body of hang_up_clothes:
Input:
Problem file:
(define (problem hang-clothes-problem)
  (:domain household)
  (:objects
    character - character
    shirt - object
    hanger - object
  ) ; This section declares the instances needed for the problem: character is an
    instance of a character; shirt is an instance of an object classified as clothes
    ; hanger is an object that is suitable for hanging clothes.
  (:init
    (clothes shirt)
    (hangable hanger)
    (holds_rh alice shirt)
    (next_to alice hanger)
  ) ; This section declares the initial conditions. (clothes shirt) and (hangable
    hanger) tells the properties of objects; (holds_rh alice shirt) indicates that
    Alice is holding the shirt in her right hand; (next_to alice hanger) means Alice
    is next to the hanger, ready to hang the shirt.
  (:goal
    (and
      (ontop shirt hanger)
    )
```

```
    ) ; This section declares the goal.  (ontop shirt hanger) is the goal, where the
        shirt should end up hanging on the hanger.
)
Action to be finished:
(:action hang_up_clothes
   :parameters (?char - character ?clothes - object ?hang_obj - object)
   :precondition ()
   :effect ()
)

Example output:
Given the objects in the problem file, and what typically needs to be true to perform
        an action like hanging up clothes: 1. clothes must indeed be a type of clothing
        . 2. hang_obj should be something on which clothes can be hung (hangable). 3.
        char should be holding the clothes, either in the right or left hand. 4. char
        needs to be next to the hanging object to hang the clothes. Besides, we need to
        write preconditions in Disjunctive Normal Form.
These insights guide us to write:
:precondition (or
                (and
                    (clothes ?clothes)  ; the object must be a piece of clothing
                    (hangable ?hang_obj)  ; the target must be an object suitable for
       hanging clothes
                    (holds_rh ?char ?clothes)  ; character is holding clothes in the
       right hand
                    (next_to ?char ?hang_obj)  ; character is next to the hanging
       object
                )
                (and
                    (clothes ?clothes)  ; the object must be a piece of clothing
                    (hangable ?hang_obj)  ; the target must be an object suitable for
       hanging clothes
                    (holds_lh ?char ?clothes)  ; character is holding clothes in the
       left hand
                    (next_to ?char ?hang_obj)  ; character is next to the hanging
       object
                )
             )
Effects describe how the world state changes due to the action. After hanging up
        clothes, you'd expect: 1. char is no longer holding the clothes. 2. clothes is
        now on the hang_obj.
These expectations convert into effects:
:effect (and
            (when (holds_rh ?char ?clothes)(not (holds_rh ?char ?clothes)))  ; if
       clothes are held in the right hand, they are no longer held
            (when (holds_lh ?char ?clothes)(not (holds_lh ?char ?clothes)))  ; if
       clothes are held in the left hand, they are no longer held
            (ontop ?clothes ?hang_obj)  ; clothes are now hanging on the object
          )

Combining these parts, the complete hang_up_clothes action becomes:
(:action hang_up_clothes
   :parameters (?char - character ?clothes - object ?hang_obj - object)
   :precondition (or
                    (and
                      (clothes ?clothes)
                      (hangable ?hang_obj)
                      (holds_rh ?char ?clothes)
                      (next_to ?char ?hang_obj)
                    )
                    (and
                      (clothes ?clothes)
                      (hangable ?hang_obj)
                      (holds_lh ?char ?clothes)
                      (next_to ?char ?hang_obj)
                    )
                  )
   :effect (and
            (when (holds_rh ?char ?clothes)(not (holds_rh ?char ?clothes)))
            (when (holds_lh ?char ?clothes)(not (holds_lh ?char ?clothes)))
            (ontop ?clothes ?hang_obj)
          )
)

Above is a good example of given predicates in domain file, problem file, action
        names and parameters, how to reason step by step and write the action body in
        PDDL. Pay attention to the usage of different connectives and their underlying
        logic.
```

```
Here are some other commonly used actions and their PDDL definition:
(:action put_to
    :parameters (?char - character ?obj - object ?dest - object)
    :precondition (or
      (and
          (hold_lh ?obj)          ; The character should hold either with left hand or
      right hand
          (next_to ?char ?dest) ; The character should be close to destination
      )
      (and
          (hold_rh ?obj)          ; The character should hold either with left hand or
      right hand
          (next_to ?char ?dest) ; The character should be close to destination
      )
    )
    :effect (obj_ontop ?obj ?dest)          ; The object is now on the destination
)
This case illustrates the use of OR to include all possible preconditions of an
    action.

(:action pick_and_place
    :parameters (?char - character ?obj - object ?dest - object)
    :precondition (and
        (grabbable ?obj)          ; The object must be grabbable
        (next_to ?char ?obj)     ; The character must be next to the object
        (not (obj_ontop ?obj ?dest)) ; Ensure the object is not already on the
    destination
    )
    :effect (and
        (obj_ontop ?obj ?dest)          ; The object is now on the destination
        (next_to ?char ?dest)          ; The character is now next to the destination
    )
)
This case illustrates a plain case with only AND operator.

(:action bow
    :parameters (?char - character ?target - character)
    :precondition (and
        (next_to ?char ?target)  ; The character must be next to the target to
    perform the bow
    )
    :effect ()
)
This case illustrates the action can have no effect (or no precondition.)

hint:
1. Don't enforce the use of WHEN everywhere.

2. You MUST only use predicates and object types exactly as they appear in the domain
    file at the beginning.

3. Use and only use the arguments provided in :parameters for each action. Don't
    propose additional arguments, unless you are using exists or forall.

4. It is possible that action has no precondition or effect.

5. The KEY of the task is to ensure after executing your proposed actions in some
    order, the intial state (:init) in problem file can reach the goals (:goal)!!!
    Pay attention to the initial state and final goals in problem file.

6. Preconditions and effects are <Clause> defined above. When there is only one
    predicate, do not use logic connectives.

For actions to be finished, write their preconditions and effects, and return in
    standard PDDL format:
(:action [action name]
  :parameters ([action parameters])
  :precondition ([action precondition])
  :effect ([action effect])
)
Concatenate all actions PDDL string into a single string. Output in json format where
    key is 'output' and value is your output string: {'output': YOUR OUTPUT STRING}

Input:
<problem_file>
<action_handlers>

Output:"
```

Figure 51: Examples prompt of transition modeling.

## I.5 Prompt of Environment Representation

We input the object-centric representation as the abstraction of the environment as below:



**Prompt of Environment Abstraction**

```
# Behavior Environment Prompt Template
AVAILABLE_STATES = '''The state is represented as a first-order predicate, which is a
    tuple of a predicate name and its arguments. Its formal definition looks like
    this "<PredicateName>(Params)", where <PredicateName> is the state name and each
    param should be ended with an id. An example is "inside(coin.1, jar.1)". Below
    is a list of available states and their descriptions.
    | State Name | Arguments | Description |
    | --- | --- | --- |
    | inside | (obj1.id, obj2.id) | obj1 is inside obj2. If we have state inside(A, B
    ), and you want to take A out of B while B is openable and stayed at "not open"
    state, please open B first. Also, inside(obj1, agent) is invalid.|
    | ontop | (obj1.id, obj2.id) | obj1 is on top of obj2 |
    | nextto | (obj1.id, obj2.id) | obj1 is next to obj2 |
    | under | (obj1.id, obj2.id) | obj1 is under obj2 |
    | onfloor | (obj1.id, floor2.id) | obj1 is on the floor2 |
    | touching | (obj1.id, obj2.id) | obj1 is touching or next to obj2 |
    | cooked | (obj1.id) | obj1 is cooked |
    | burnt | (obj1.id) | obj1 is burnt |
    | dusty | (obj1.id) | obj1 is dusty. If want to change dusty(obj1.id) to "not
    dusty(obj1.id)", there are two ways to do it, depending on task conditions. Here
    , all objects other than obj1 are types but not instances: 1. [inside(obj1.id,
    dishwasher or sink), toggledon(dishwasher or sink)] 2. holding other cleaning
    tool |
    | frozen | (obj1.id) | obj1 is frozen |
    | open | (obj1.id) | obj1 is open |
    | sliced | (obj1.id) | obj1 is sliced. If want to change "not sliced(obj1.id)" to
    "sliced(obj1.id)", one must have a slicer. |
    | soaked | (obj1.id) | obj1 is soaked |
    | stained | (obj1.id) | obj1 is stained. If want to change stained(obj1.id) to "
    not stained(obj1.id)", there are three ways to do it, depending on task
    conditions. Here, all objects other than obj1 are types but not instances: 1. [
    inside(obj1.id, sink), toggledon(sink)] 2. [soaked(cleaner)] 3. holding
    detergent. |
    | toggledon | (obj1.id) | obj1 is toggled on |
    | holds_rh | (obj1.id) | obj1 is in the right hand of the robot |
    | holds_lh | (obj1.id) | obj1 is in the left hand of the robot |'''

AVAILABLE_CONNECTIVES = '''The connectives are used to satisfy the complex conditions
    . They are used to combine the state predicates.
    | Connective Name | Arguments | Description |
    | --- | --- | --- |
    | and | exp1 and exp2 | evaluates to true if both exp1 and exp2 are true |
    | or | exp1 or exp2 | evaluates to true if either exp1 or exp2 is true |
    | not | not exp | evaluates to true if exp is false |
    | forall | forall(x, exp) | evaluates to true if exp is true for all x |
    | exists | exists(x, exp) | evaluates to true if exp is true for at least one x |
    | forpairs | forpairs(x, y, exp) | evaluates to true if exp is true for all pairs
     of x and y. For example, forpairs(watch, basket, inside(watch, basket)) means
    that for each watch and basket, the watch is inside the basket. |
    | forn | forn(n, x, exp) | evaluates to true if exp is true for exactly n times
    for x. For example, forn(2, jar_n_01, (not open(jar_n_01)) means that there are
    exactly two jars that are not open. |
    | fornpairs | fornpairs(n, x, y, exp) | evaluates to true if exp is true for
    exactly n times for pairs of x and y. For example, fornpairs(2, watch, basket,
    inside(watch, basket)) means that there are exactly two watches inside the
    basket. |'''

EXAMPLE_RELEVANT_OBJECTS = '''## Relevant objects in this scene
    {'name': 'strawberry.0', 'category': 'strawberry_n_01'}
    {'name': 'fridge.97', 'category': 'electric_refrigerator_n_01'}
    {'name': 'peach.0', 'category': 'peach_n_03'}
    {'name': 'countertop.84', 'category': 'countertop_n_01'}
    {'name': 'jar.0', 'category': 'jar_n_01'}
    {'name': 'jar.1', 'category': 'jar_n_01'}
    {'name': 'carving_knife.0', 'category': 'carving_knife_n_01'}
    {'name': 'bottom_cabinet_no_top.80', 'category': 'cabinet_n_01'}
```



```
        {'name': 'room_floor_kitchen.0', 'category': 'floor_n_01'}'''

INITIAL_STATES = '''inside(strawberry.0, fridge.97)
    inside(peach.0, fridge.97)
    not sliced(strawberry.0)
    not sliced(peach.0)
    ontop(jar.0, countertop.84)
    ontop(jar.1, countertop.84)
    ontop(carving_knife.0, countertop.84)
    onfloor(agent_n_01.1, room_floor_kitchen.0)'''

GOAL_STATES = '''exists(jar_n_01, (inside(strawberry.0, jar_n_01) and (not inside(
    peach.0, jar_n_01))))
    exists(jar_n_01, (inside(peach.0, jar_n_01) and (not inside(strawberry.0,
    jar_n_01))))
    forall(jar_n_01, (not open(jar_n_01)))
    sliced(strawberry.0)
    sliced(peach.0)'''
```

Figure 52: Examples prompt of object-centric representation for the embodied environment.

## I.6 Prompt Analysis and Learned Lessons

During our empirical experiments, we find many models struggle with outputting strictly formatted parsable output that can be accepted by our embodied agent interface, let alone the various intricate formats accepted by different simulators. In addition, many models are inclined to output accompanying explanations along with their answers despite explicit instructions forbidding such behavior.

To address these problems, we design structured and easy-to-follow code output formats that consist of JSON and Python list structures in place of long string output, leveraging popular LLMs' superior formal language (code) generation ability over natural language.

In order to standardize our evaluation, we use the same prompt for all models evaluated, following the convention established by [83–85]. More details about this design choice are in Appendix I.7.

Below are some of the other LLM prompting strategies we used in designing our evaluation prompts. Based on empirical qualitative experiments, prompts under these conditions generated the best results for our embodied agent interface:

---

**Takeaways of Prompting Techniques**

- **System Prompts** are preferred over user prompts for specifying output structure and overall output style as it is more strongly enforced.

- **Stop Sequences** are provided to prevent repetitive output and end-of-output explanations. This can improve format following and save token usage.

- **Shorter Strings** are encouraged to reduce grammar errors. For example, in the goal interpretation, we ask models to generate ["cond1", "and", "cond2", "or", "cond3"] instead of "cond1 and cond2 or cond3".

- **Start / End Tokens** are specified to discourage models from generating additional explanations and self-repetition of answers / prompt instructions.

---

## I.7 Further Consideration about Prompt Variability

To ensure fairness in evaluating large language models (LLMs), we adopted a model-agnostic prompt design. This approach prevents bias that could arise from using model-specific prompts. Through iterative empirical testing across all models, we ensured that each model could accurately interpret the prompts, with none exceeding a format error rate of 3.8%.

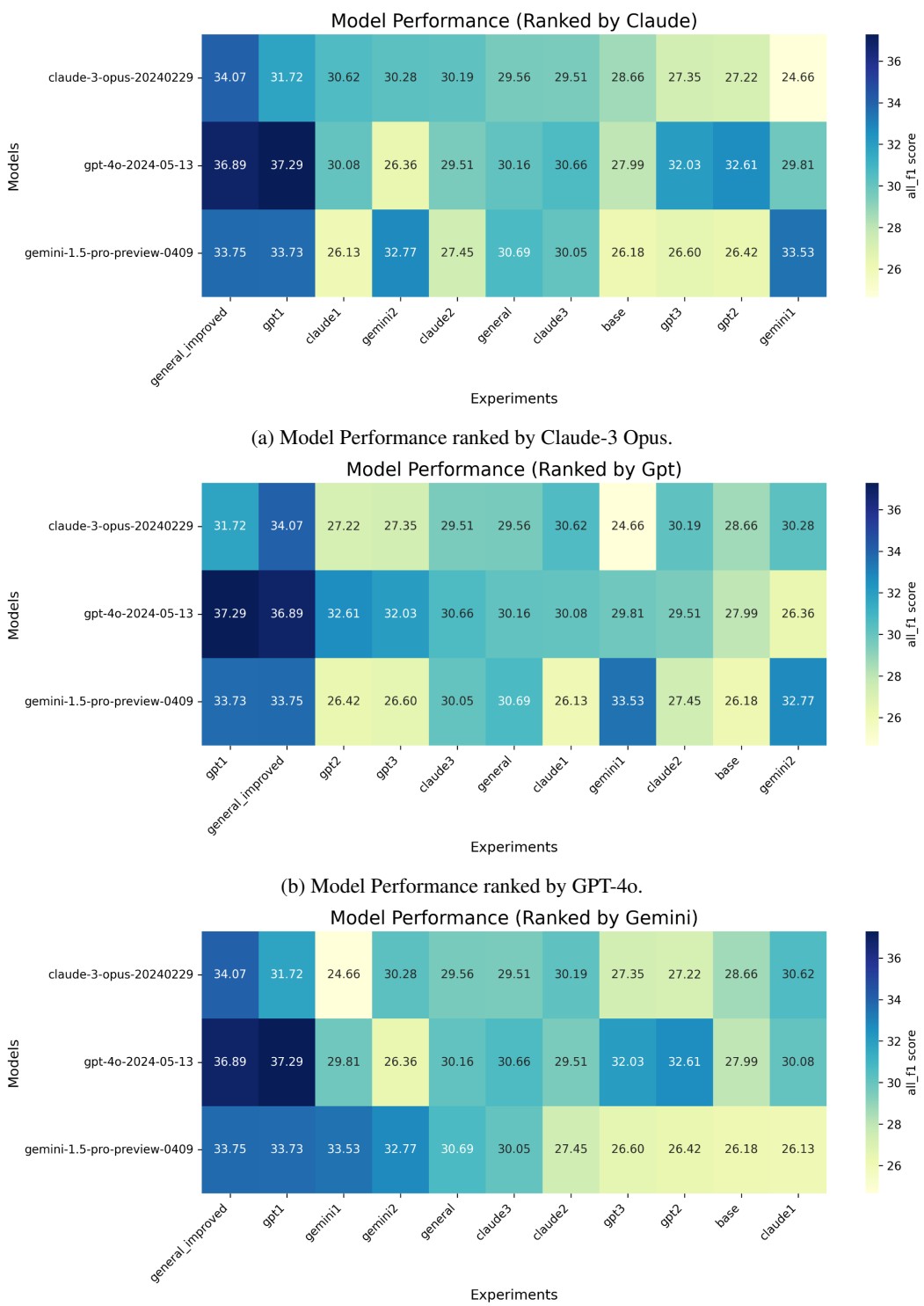

(a) Model Performance ranked by Claude-3 Opus.

(b) Model Performance ranked by GPT-4o.

(c) Model Performance ranked by Gemini-1.5 Pro.

Figure 53: Heatmap results showing minor differences between general and model-specific prompts across the three tested models.

As discussed in the above section, our prompt design aligns with established evaluation methods [83], ensuring a scalable and fair approach. Additionally, we conducted ablation studies demonstrating that model-specific prompt tuning has minimal effect on performance. For example, models like GPT-4o do not rely on explicit chain-of-thought prompts for reasoning tasks.

To further verify this, we tested model-specific prompt tuning on the VirtualHome goal interpretation task across three models. The results are visualized in heatmaps (Fig. 53). In our experiments, we test on "claude-3-opus-20240229", "gpt-4o-2024-05-13", "gemini-1.5-pro-preview-0409". We named the prompt in the following way:

- *base*, *general*, and *general_improved* are the three versions of prompts we used in previous benchmark development. We started from *base* and finalized the final version of *general_improved* prompt, based on which all our results are reported in the paper.
- *{model name}_{number}* are the model-specific prompts we tuned for each *{model name}* with *TOP_{number}* performance gain.

Our findings indicate that model-specific prompts do not lead to significant differences. For example, in the figures, when Claude performs best with certain prompts, these prompts are also among the top-performing prompts for other models. This suggests that prompt improvements can be generalized across large models, supporting our initial point.

## J    Human Performance Comparison

To evaluate the performance of Large Language Models (LLMs) relative to human capabilities, we conducted experiments comparing GPT-4 with human annotations across various tasks. The results are summarized in Table 19.

Table 19: Comparison of GPT-4 and Human Performance on Different Tasks

| Method | Goal Interpretation | Action Sequencing | | Subgoal Decomposition | | Transition Modeling | | Average Perf. |
|---|---|---|---|---|---|---|---|---|
| | $F_1$ | Task SR | Execution SR | Task SR | Execution SR | $F_1$ | Planner SR | Module SR |
| GPT-4o | 37.6 | 42.9 | 57.1 | **70.0** | **90.0** | 50.0 | **100.0** | 56.8 |
| Human Annotation | **80.6** | **57.1** | **85.7** | 60.0 | 80.0 | **52.9** | 66.7 | **64.4** |

Our findings indicate that while humans outperform GPT-4o in tasks such as Goal Interpretation and Action Sequencing Execution Success Rate, GPT-4o excels in Subgoal Decomposition and Transition Modeling. Specifically, GPT-4o achieves a higher Goal Success Rate and Execution Success Rate in Subgoal Decomposition, and a perfect Planner Success Rate in Transition Modeling. This suggests that LLMs like GPT-4o may excel in tasks requiring long-context reasoning, such as long-horizon logical reasoning and large-scale scene graph tracking, while humans perform better in tasks requiring nuanced understanding and execution.

## K    Further Discussion on Visual Information in Our Benchmark

### K.1    Integration of Visual Inputs in Long-Horizon Decision Making

While our primary focus is on evaluating long-horizon decision-making capabilities using text-based inputs and outputs, we acknowledge the critical role of visual information in practical embodied AI scenarios. To address concerns regarding the assumption that all elements can be described using text, we conducted additional experiments involving Vision-Language Models (VLMs) to explore their potential in planning tasks.

### K.1.1    Challenges with Existing VLMs

Current VLMs are not specifically designed to process scene-level information from complex environments that include multiple rooms and numerous objects, such as those found in the BEHAVIOR and

VirtualHome simulators. Recent works evaluating VLMs for embodied and online scenarios [86–91] primarily focus on perception tasks rather than long-horizon decision making.

### K.1.2 Comparative Experiments with LLMs and VLMs

To investigate the capabilities of VLMs in long-horizon decision-making, we conducted experiments comparing Large Language Models (LLMs) and VLMs under various settings on the BEHAVIOR dataset.

**Experiment Settings** We compare Llama 3 and LlaVA (with Llama 3 as the backbone model) for LLM and VLM comparison. Below are experimental settings.

- **Exp.0 (Baseline)**: Scene graph input to LLMs producing planning outputs.
- **Exp.1**: Image input to VLMs producing planning outputs without intermediate scene graphs (end-to-end approach).
- **Exp.2**: Image and scene graph inputs to VLMs producing planning outputs.

**Results and Analysis** The results of these experiments are summarized in Table 20.

Table 20: Performance Comparison between LLMs and VLMs on BEHAVIOR

| Model | Goal Interpretation ($F_1$, %) | Action Sequencing (Success Rate %) |
|---|---|---|
| Llama 3 | **31.5** | **11.1** |
| LLaVA (w/o scene graph) | 9.1 | 2.5 |
| LLaVA (w/ scene graph) | 25.8 | 11.0 |

These experiments reveal several key insights: (1) LLaVA shows significantly lower performance than Llama 3 when used end-to-end (Exp.1). This indicates challenges in effectively utilizing visual inputs for complex planning tasks. (2) Providing scene graphs as additional input to LLaVA (Exp.2) improves its performance but does not match the level of Llama 3 using scene graphs (Exp.0). This suggests that current VLMs may not fully leverage visual information for long-horizon reasoning. (3) End-to-end approaches with VLMs (Exp.1) entangle perception and decision-making errors, complicating the diagnosis of specific weaknesses in reasoning or planning abilities.

### K.1.3 Why LLMs Benchmarking is Useful

Many existing approaches leveraging foundation models include CodeAsPolicies [6], Voyager [35], VIMA [92], VoxPoser [77]. Therefore, we aim for **standardized evaluation** for LLMs, which can (1) guide **LLM researchers** for improvements to better support embodied AI; (2) provide **robotics researchers** with selection and modularization of different LLMs and their roles. On top of that, our experiments reveal a transferable pattern from LLMs to VLMs, shown as follows.

**Experiment Settings** Similar to Exp.0-Exp.2, we compare LLama 3 and LLaVA.

- **Exp.3:** Scene graph and broader context input to LLMs improving planning output.
- **Exp.4:** Images, scene graph, and broader input to VLMs to see if planning output is improved.

Table 21: Insights from LLMs (Exp.3) can be applied to VLMs (Exp.4)

| Model | Goal Interpretation ($F_1$, %) Improvement | Action Sequencing (SR%) Improvement |
|---|---|---|
| LLama 3 | $31.5 \rightarrow 31.9$ | $11.1 \rightarrow 13.9$ |
| LLaVA (w/ scene graph) | $25.8 \rightarrow 25.9$ | $11.0 \rightarrow 12.3$ |

**Result Analysis** Our experiments show that insights from LLM evaluations can be applied to Vision-Language Models (VLMs) to improve their performance in planning tasks. In Experiment 3, adding scene graphs and broader context to Llama led to improved goal interpretation ($F_1$ score increased from 31.5% to 31.9%) and action sequencing (success rate increased from 11.1% to 13.9%). Similarly, in Experiment 4, applying the same approach to LLaVA showed smaller but notable improvements, with the $F_1$ score increasing from 25.8% to 25.9% and the action sequencing success rate from 11.0% to 12.3%.

These findings suggest that techniques improving abstract planning in LLMs can also enhance VLM reasoning, helping the development of more capable robotic systems for real-world tasks.

### K.1.4 Implications for Future Research

Our findings highlight the need for further development of VLMs to enhance their long-horizon decision-making capabilities. Specifically:

1. **Improved Multimodal Integration**: VLMs require better mechanisms to integrate visual and textual information effectively for planning tasks.

2. **Modular Evaluation Approaches**: Decomposed evaluations that isolate perception and decision-making components can help accurately identify and address specific areas for improvement.

3. **Methodological Transfer**: Strategies that improve LLM performance, such as providing structured inputs or broader context, may be adapted to enhance VLM reasoning in embodied AI tasks.

### K.2 Impact of Perception and State Estimation Errors

From the above results, we find that perception errors can significantly affect the planning performance of models in embodied AI tasks. To assess this impact, we conducted experiments where scene graphs generated by VLMs were used as input to LLMs for planning. Specifically, we use VLM-generated scene graphs and then input to LLMs. We focus on subgoal decomposition since it is highly sensitive to perception performance.

### K.2.1 Experiment Setup (Exp.5)

- **Task**: Subgoal decomposition on the BEHAVIOR dataset.

- **Input**: Scene graphs generated by VLMs from images.

- **Models**: Different VLMs used for scene graph generation, followed by LLMs for planning.

### K.2.2 Results Analysis

The results are presented in Table 22.

Table 22: Effect of Perception Errors on Subgoal Decomposition

| VLM | Object Error Rate (%) | Predicate Error Rate (%) | Steps Generated Num | Success Rate (%) |
|---|---|---|---|---|
| Claude-3.5-Sonnet | **0.0** | **8.0** | 13 | **75.0** |
| ChatGPT-4 | **0.0** | 20.0 | 8 | 50.0 |
| LLaVA-Vicuna-13B | **0.0** | 83.0 | 6 | 12.5 |

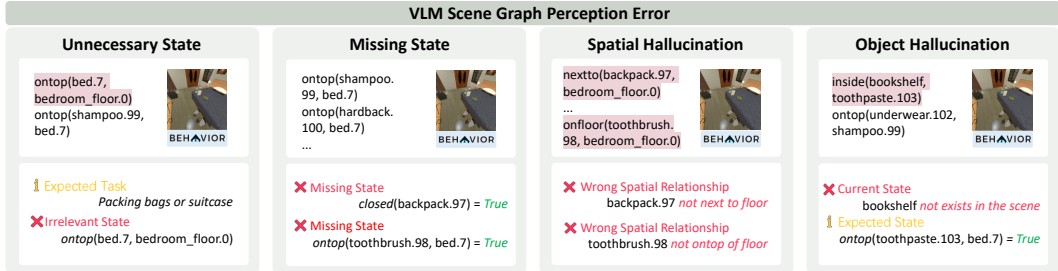

Figure 54: Possible VLM scene graph perception error types.

Figure 54 demonstrates four types of VLM perception errors: Unnecessary State, Missing State, Spatial Hallucination, and Object Hallucination. Here, we group missing state and spatial hallucination as predicate errors and refer to object hallucination as object error. From table 22, the experiments demonstrate that: (1) Higher predicate error rates in the scene graphs lead to lower success rates in subgoal decomposition. (2) Accurate perception of object relationships (predicates) is crucial for effective planning and task execution. (3) Claude-3.5 Sonnet is the strongest VLM among tested VLMs, with the highest performance at 75.0%.

### K.3 Assumptions on Scene Graphs in Our Benchmark

Scene graphs are widely used in robotics to represent the environment's semantic and relational structure [93–96]. They enable robots to reason about objects and their relationships, which is essential for complex task planning. Furthermore, generating relational graphs from real-world visual data is an active research area [93–95]. Inspired by this insight, we adopt scene graphs as perception results for our benchmark.

## L Dataset Statistics and Analysis

### L.1 Dataset Structure

We define a data structure containing a comprehensive annotation for the evaluation of four ability modules with different levels of planners. Each instance in the dataset represents a task goal, and each task contains the following data:

1. Natural language task name
2. Natural language task instruction
3. Symbolic goal definition (including its LTL form)
4. Symbolic action trajectory
5. The transition models involved in the task

For tasks in the BEHAVIOR environment, the dataset also includes accompanying VR human demonstration videos that showcase the execution of the ground truth action trajectories.

Please find our JSON data format in this link: https://huggingface.co/datasets/Inevitablevalor/EmbodiedAgentInterface:

---

**Data Format Examples for VirtualHome**

```
# Below is an example of VirtualHome
"1057_1": {
  "task_name": "Watch TV",
  "natural_language_description": "Go to the living room, sit on the couch, find the
    remote, switch on the TV and watch",
  "vh_goal": {
    "actions": [
      "LOOKAT|WATCH"
```

---

```
    ],
    "goal": [{
        "id": 410,
        "class_name": "television",
        "state": "ON"
    }, {
        "id": 410,
        "class_name": "television",
        "state": "PLUGGED_IN"
    }, {
        "from_id": 65,
        "relation_type": "FACING",
        "to_id": 410
    }]
},
"tl_goal": "(exists x0. ((LOOKAT(x0) or WATCH(x0))) then (ON(television.410) and
    PLUGGED_IN(television.410) and FACING(character.65, television.410)))",
"action_trajectory": [
    "[WALK] <home_office> (319)",
    "[WALK] <couch> (352)",
    "[FIND] <couch> (352)",
    "[SIT] <couch> (352)",
    "[FIND] <remote_control> (1000)",
    "[FIND] <television> (410)",
    "[SWITCHON] <television> (410)",
    "[TURNTO] <television> (410)",
    "[WATCH] <television> (410)"
],
"transition_model": <pddl_definition>
}
```

Figure 55: We release the task annotations in JSON format. Here is the data instance example for VirtualHome .

### Data Format Examples for BEHAVIOR

```
# Below is an example of BEHAVIOR
"cleaning_high_chair_0_Wainscott_0_int_0_2021-06-05_18-03-15": {
    "task_name": "cleaning_high_chair",
    "natural_description": "Clean the high chair.",
    "raw_bddl_goal": "(define (problem cleaning_high_chair_0)    (:domain igibson)\n\n
        (:objects\n         highchair.n.01_1 - highchair.n.01\n
        piece_of_cloth.n.01_1 - piece_of_cloth.n.01\n
        cabinet.n.01_1 - cabinet.n.01\n
        ...",
    "tl_goal": "not dusty(highchair.0)",
    "action_trajectory": [
        {
            "action": "OPEN",
            "object": "bottom_cabinet_no_top_80"
        },
        {
            "action": "RIGHT_GRASP",
            "object": "paper_towel_0"
        },
        {
            "action": "LEFT_GRASP",
            "object": "highchair_0"
        },
        {
            "action": "CLEAN",
            "object": "highchair_0"
        }
    ],
    "transition_model": <pddl_definition>,
    "demo": <demo_link>
}
```

Figure 56: We release the task annotations in JSON format. Here is the data instance example for BEHAVIOR .

As shown in Figure 9, we extend the RobotHow [5] data structure, where each task goal includes a natural language goal description, a VirtualHome action script, and the initial and final states of the scene. In addition, we offer precise symbolic goals for each task, which enhances the accuracy of evaluation by mitigating the impact of noisy final states. For instance, it is more precise to denote the final goal of the task "turn on light" as *on(light)*

rather than a noisier version like *facing(agent, light) and on(light)*. Moreover, we provide transition model annotations, which require accurate annotation of all the logical constraints of preconditions and post-effects.

For tasks in the BEHAVIOR environment, as shown in Figure 10, the dataset annotation focuses on the action sequences and transition models. The dataset also includes accompanying demo videos that showcase the execution of the ground truth action trajectories. For instance, the task "bottling fruit" includes a demo video showing the agent picking up peaches and placing them inside a jar, along with the corresponding action sequence and transition model annotations.

The designed data structure is general and can support a systematic evaluation and usage of the data. It enables the exploration of different integration methods for various modules and can flexibly support downstream applications. By providing a comprehensive set of annotations, including natural language descriptions, symbolic goals, action trajectories, and transition models, researchers can investigate the interplay between different components of embodied AI systems and develop novel approaches for integrating them effectively. Also, it is not limited to specific simulators and can be expanded to other environments. It has the potential to become a standardized data format for embodied AI tasks, facilitating the comparison and benchmarking of different methods across various domains. The consistent representation of goals, actions, and transitions allows for a unified evaluation framework and promotes the development of generalizable and transferable approaches.

The availability of natural language descriptions alongside symbolic representations enables the exploration of language grounding and the development of more intuitive human-robot interaction methods. For example, an agent can be trained to understand and execute natural language commands by grounding the language to the corresponding symbolic goals and actions.

## L.2 Data Statistics and Distribution

To understand the datasets used for evaluating long-horizon decision making capabilities for complex goals, we provide detailed statistics and distribution information in the following sections.

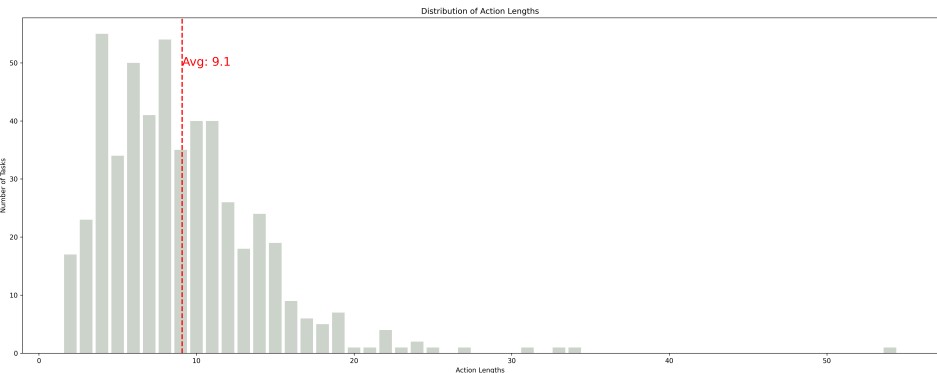

Figure 57: Action sequence length distribution in VirtualHome

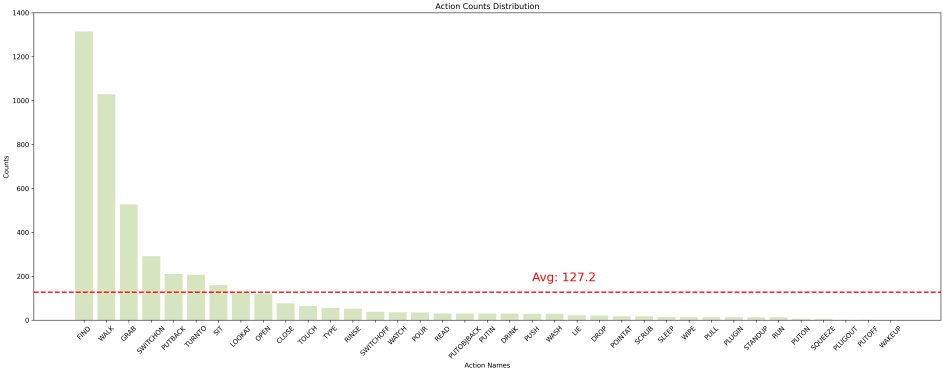

Figure 58: Action counts distribution in VirtualHome

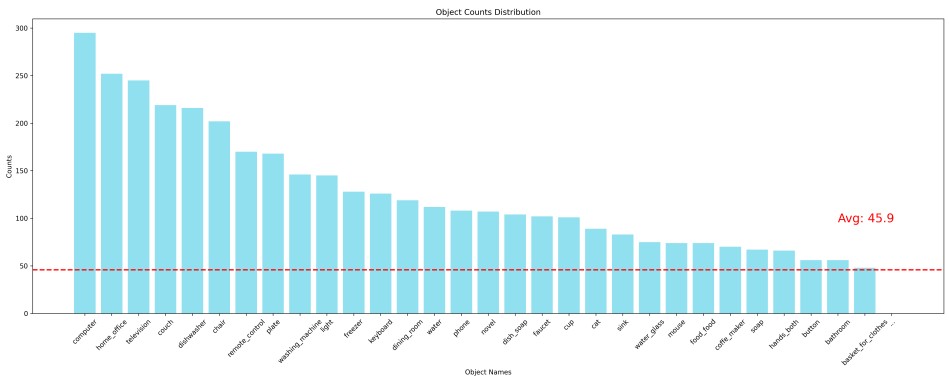

Figure 59: Object counts distribution in VirtualHome

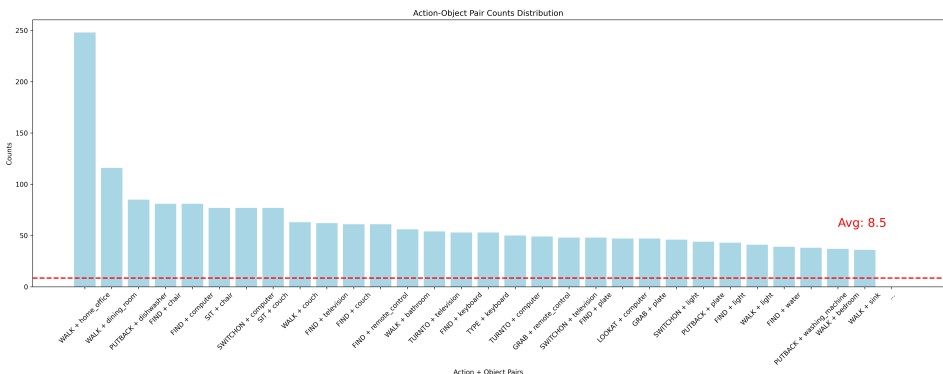

Figure 60: Action object pair counts distribution in VirtualHome

**VirtualHome** Table 2 in the main paper presents the statistics of our annotated RobotHow dataset. We identified a total of 801 goals, comprising 340 state goals, 299 relation goals, and 162 action goals. For the provided task instructions (referred to as 'trajectories' in the table), the average length of action steps is approximately 8.76, ranging from 2 to 54 steps. Notably, 21 out of 26 task categories include instructions with action steps exceeding 10 in length, and one-third of the instructions have step lengths of more than 10. This indicates the complexity of our annotated data.

**BEHAVIOR** As shown in Figure 61, the BEHAVIOR dataset has an average of 14.6 actions and 3.7 goals per task. The action length is determined by the number of actions in the ground truth action sequence. The number of goals is calculated by counting the expressions within the brackets of the outermost connective *and* in the BEHAVIOR goal, which is written in the Behavior Definition Description Language (BDDL) format. For example, for a goal expression such as `(and (contains ?hot_tub.n.02_1 ?chlorine.n.01_1) (filled ?hot_tub.n.02_1 ?water.n.06_1))`, we would consider it as having two goals: the first one is `(contains ?hot_tub.n.02_1 ?chlorine.n.01_1)`, and the second one is `(filled ?hot_tub.n.02_1 ?water.n.06_1)`.

### L.3 Goal Complexity Analysis

Goals in BEHAVIOR may contain quantifiers, such as `forpairs( (?jar.n.01 - jar.n.01) (?apple.n.01 - apple.n.01)) (inside ?apple.n.01 ?jar.n.01)`, which are referred to as BDDL goals. These BDDL goals can be translated into multiple grounded goals. For example, the BDDL goal `forpairs( (?jar.n.01 - jar.n.01) (?apple.n.01 - apple.n.01)) (inside ?apple.n.01 ?jar.n.01)` could be translated into the grounded goals `(and (inside apple.n.01_1 jar.n.01_1) (inside apple.n.01_2 jar.n.01_2))`. Different combinations of grounded goals can satisfy the same BDDL goal, which we refer to as goal options. For instance, `(and (inside apple.n.01_2 jar.n.01_1) (inside apple.n.01_1 jar.n.01_2))` also satisfies the same BDDL goal. Generally, one BDDL goal may have multiple goal options, each consisting of several grounded goals.

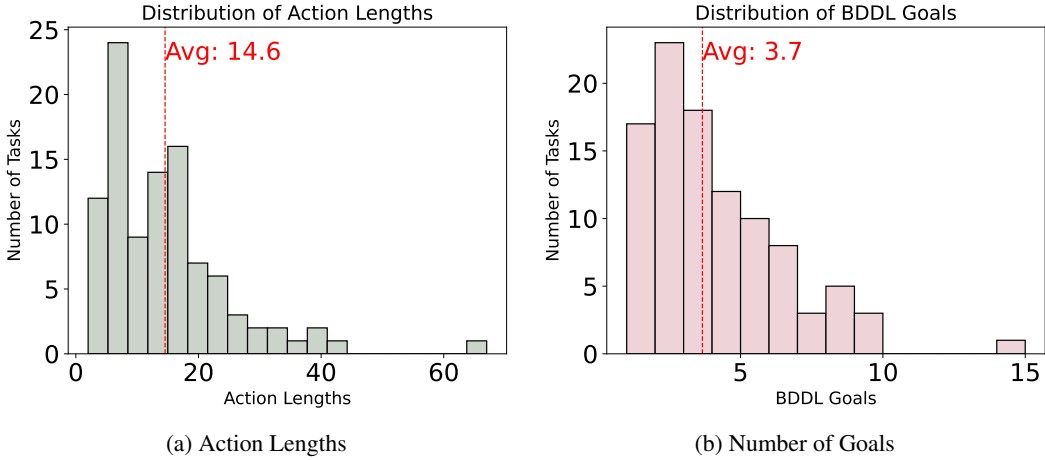

(a) Action Lengths       (b) Number of Goals

Figure 61: BEHAVIOR Task Complexity

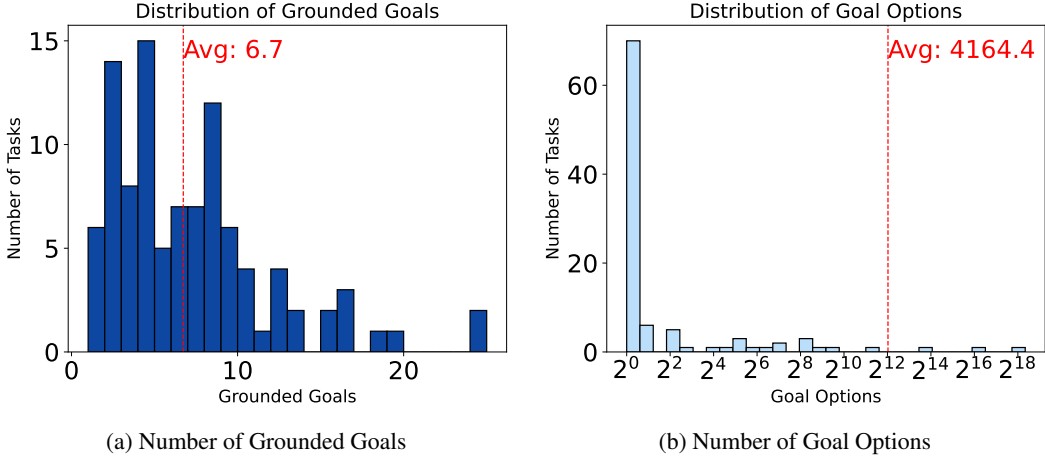

(a) Number of Grounded Goals       (b) Number of Goal Options

Figure 62: BEHAVIOR Goal Complexity

Figure 62 illustrates the distribution of the number of grounded goals and goal options in BEHAVIOR . The number of grounded goals is determined by sampling a goal option from the BDDL goal and counting the expressions within the brackets of the outermost connective *and*. For example, `(and (inside apple.n.01_2 jar.n.01_1) (inside apple.n.01_1 jar.n.01_2))` contains two grounded goals. The number of goal options is calculated by the different possible combinations of grounded goals that satisfy the BDDL goal. On average, each task in BEHAVIOR has 6.7 grounded goals and 4164.4 goal options.

## L.4 Task List

Our dataset's task selection criteria prioritize long-horizon tasks with complex goals, ensuring a challenging and comprehensive evaluation of embodied agents. For BEHAVIOR, we choose all the tasks with human demonstrations to ensure high-quality results. For VirtualHome, we select a typical scene and include all the tasks that can be executed within this scene. The combination of simulated tasks from VirtualHome and human-demonstrated tasks from BEHAVIOR allows us to evaluate the generalization capabilities of LLMs across different domains and levels of complexity. The task lists in Table 23 and Table 24 offer a clear overview of the selected tasks to understand the scope and variety of the evaluation benchmark.

The inclusion of a typical scene in VirtualHome enables a focused evaluation of an agent's performance within a specific environment. By providing all the executable tasks within this scene, researchers can thoroughly investigate the agent's ability to navigate, interact with objects, and complete goals in a constrained setting. This setup facilitates the analysis of the agent's behavior and helps identify strengths and weaknesses in its decision-making process. The selected tasks in VirtualHome cover a wide range of household activities, such as cooking, cleaning, and object manipulation. For instance, the task "cook some food" involves multiple steps,

Table 23: Task List on VirtualHome

| Task Name | Task Number | Task Name | Task Number |
|---|---|---|---|
| Wash hands | 22 | Pet cat | 27 |
| Drink | 28 | Work | 35 |
| Turn on light | 34 | Pick up phone | 28 |
| Go to toilet | 11 | Get some water | 1 |
| Wash teeth | 11 | Read book | 28 |
| Make coffee | 10 | Listen to music | 22 |
| Wash dishes by hand | 11 | Relax on sofa | 35 |
| Wash clothes | 21 | Wash dishes with dishwasher | 27 |
| Browse internet | 31 | Put groceries in Fridge | 23 |
| Set up table | 8 | Write an email | 21 |
| Take shower | 2 | Watch TV | 32 |
| Cook some food | 7 | Change TV channel | 29 |
| Go to sleep | 9 | | |
| Brush teeth | 5 | | |

including gathering ingredients, using kitchen appliances, and setting the table. Another example is the task "wash dishes by hand", which requires the agent to perform actions like picking up objects, wiping surfaces, and putting things back in their designated places. These tasks showcase the complexity and diversity of the scenarios.

In the BEHAVIOR dataset, the chosen tasks are based on real-world human demonstrations, ensuring the authenticity and practicality of the tasks. These tasks encompass various domains, such as object manipulation, tool use, and goal-directed behaviors. For example, the task "bottling fruit" requires the agent to grasp peaches, cut them, and place them inside a jar, demonstrating object manipulation skills. The inclusion of these human-demonstrated tasks in BEHAVIOR provides valuable insights into real-world challenges and helps to evaluate the performance of embodied agents in more realistic settings.

Table 24: Task List on BEHAVIOR . Length is the action trajectory length.

| Task Name | Length | Task Name | Length |
|---|---|---|---|
| assembling_gift_baskets | 32 | brushing_lint_off_clothing | 13 |
| boxing_books_up_for_storage | 16 | collecting_aluminum_cans | 12 |
| mopping_floors | 15 | preserving_food | 20 |
| re-shelving_library_books | 8 | polishing_silver | 6 |
| packing_boxes_for_household_move_or_trip | 20 | cleaning_freezer | 15 |
| installing_a_modem | 3 | cleaning_up_after_a_meal | 31 |
| putting_away_Christmas_decorations | 17 | setting_up_candles | 12 |
| cleaning_shoes | 10 | cleaning_sneakers | 30 |
| locking_every_window | 4 | washing_cars_or_other_vehicles | 8 |
| washing_dishes | 11 | putting_away_Halloween_decorations | 15 |
| collect_misplaced_items | 11 | cleaning_high_chair | 4 |
| washing_floor | 8 | sorting_mail | 12 |
| cleaning_bathrooms | 12 | cleaning_kitchen_cupboard | 18 |
| preparing_salad | 34 | putting_leftovers_away | 18 |
| preparing_a_shower_for_child | 8 | cleaning_stove | 14 |
| putting_dishes_away_after_cleaning | 17 | packing_child_s_bag | 13 |
| cleaning_windows | 13 | bottling_fruit | 21 |

| Task Name | Frequency | Task Name | Frequency |
|---|---|---|---|
| thawing_frozen_food | 20 | setting_mousetraps | 8 |
| washing_pots_and_pans | 24 | laying_wood_floors | 8 |
| putting_away_toys | 16 | cleaning_garage | 19 |
| watering_houseplants | 14 | installing_alarms | 6 |
| organizing_boxes_in_garage | 12 | putting_up_Christmas_decorations_inside | 18 |
| cleaning_bathtub | 8 | clearing_the_table_after_dinner | 14 |
| loading_the_dishwasher | 13 | filling_an_Easter_basket | 26 |
| cleaning_the_pool | 8 | opening_packages | 6 |
| packing_bags_or_suitcase | 15 | cleaning_up_refrigerator | 23 |
| making_tea | 13 | picking_up_trash | 10 |
| polishing_furniture | 4 | organizing_school_stuff | 15 |
| cleaning_cupboards | 24 | installing_a_printer | 3 |
| packing_food_for_work | 12 | storing_the_groceries | 23 |

## L.5 Task Categorization

To have a better insight into LLM's abilities in predicting transition modeling from different perspectives, we group all the predicates in PDDL into five categories: object affordance, object states, object orientation, spatial relations, and non-spatial relations.

In our approach, we categorize programs based on the most dominant predicate category, employing Inverse Document Frequency (IDF) to measure the significance of predicates within each program. Given a set of programs $P = \{p_1, p_2, \ldots, p_n\}$ and a corresponding set of predicates $T = \{t_1, t_2, \ldots, t_m\}$, each predicate $t_i$ is associated with a specific category $C_j$ within a predefined set of categories $C = \{c_1, c_2, \ldots, c_k\}$. Each action $a_i$ corresponds to a set of predicates $T'_{a_i} = \bigcup_{t_i \in \text{PDDL}(a_i)} t_i$ according to its PDDL definition $\text{PDDL}(a_i)$. Each program $p_i$ can be solved by a PDDL planner and has a corresponding action sequence $A_i$. The predicate set $T_i$ is defined as the unification of the predicate sets of all actions involved in action sequence $A_i$. Formally, $T_i = \bigcup_{a_j \in A_i} T'_{a_j} = \{t_{i1}, t_{i2}, \ldots, t_{il}\}$, where $l$ can vary between programs.

The task categorization process begins with the calculation of the IDF for each predicate across the corpus of programs. The IDF for a predicate $t$ is computed as follows:

$$\text{IDF}(t) = \log \left( \frac{|P|}{\text{df}(t)} \right)$$

where $|P|$ denotes the total number of programs and $\text{df}(t)$ is the number of programs that contain the predicate $t$. This metric quantifies the importance of a predicate relative to its frequency across all programs, thereby enhancing the significance of rarer predicates.

Following the computation of the IDF values, each program $p_i$ is scored for each category $c_j$ by summing the IDF values of the predicates belonging to $c_j$:

$$S(c_j, p_i) = \sum_{t \in p_i, t \in c_j} \text{IDF}(t)$$

Each program $p_i$ is assigned to its top $K$ scoring categories based on the accumulated IDF scores. This is formalized as follows:

$$\text{Categories}(p_i) = \text{top}_K \{(c_j, S(c_j, p_i)) \mid c_j \in C\}$$

where $\text{top}_K$ selects the set of $K$ categories with the highest scores for each program.

This scoring and categorization method emphasizes the content-specific importance of predicates within programs, assigning categories based on the most informative predicate category present in each program. This approach is particularly effective in distinguishing the dominant themes of programs when predicates are unevenly distributed across categories, thus providing a robust basis for categorization in large-scale and diverse datasets.

We categorize tasks with $K = 2$. Table 25 illustrates the number of tasks under each task category in VirtualHome and BEHAVIOR.

Table 25: Results on Task Categorization, showing task number in each task category.

|  | VirtualHome | BEHAVIOR |
|---|---|---|
| **Object States** | 178 | 45 |
| **Object Orientation** | 167 | - |
| **Object Affordance** | 28 | - |
| **Spatial Relations** | 145 | 94 |
| **Non-spatial Relations** | 74 | 61 |

# M  Annotation Details

## M.1  Simulator Comparison and Selection

The VirtualHome, BEHAVIOR, and AI2-THOR simulators each offer unique capabilities tailored to different aspects of embodied agents. VirtualHome is ideal for long, complex household tasks, supporting a wide range of scripted actions and detailed object states within home environments. BEHAVIOR provides a broad spectrum of human activities with high goal complexity, realistic physics, and large, dynamic state space, making it suitable for testing generalization across diverse scenarios. AI2-THOR focuses on photorealistic indoor environments, emphasizing detailed perception and manipulation tasks with rich real-time interactions and substantial state space.

In detail, we compare their key differences in terms of task length, goal complexity, task scenarios, actions, and state space:

**Task Length and Goal Complexity**

- VirtualHome is designed to simulate longer sequences of everyday activities, involving complex multi-step goals with many objects. It involves sequences of 10-30 high-level actions on average.

- **BEHAVIOR** has the longest task length and most complex multi-step trajectories. Its ranges from 50 to over 1000 low-level actions in length.

- **AI2-THOR** tasks are generally shorter, centered around navigation and basic object interactions with around 10 steps.

**Task Scenarios**

- **VirtualHome** specializes in simulating complete residential apartments/houses with furnished rooms to enable realistic household activity scenarios such as *making coffee* or *brush teeth*. It has around 20,000 unique household activity scenarios across 57 furnished residential environments.

- **BEHAVIOR-100** provides 100 different scripted human behavior scenarios across various indoor environments.

- **AI2-THOR** focuses more on navigation tasks and low-level object interactions in diverse 3D indoor scenes. It has over 120 unique room configurations across 8 different room types like bathrooms, living rooms, etc.

**Action Space and State Space**

- **VirtualHome**: Represents activities as high-level action sequences like "*<char0> [PutBack] <glass> (1) <table>*", enabling simulation of complex multi-step activities. It has around 300 unique high-level action types.

- **AI2-THOR**: Has a larger state space with many possible environment configurations, i.e., allowing over 30 different low-level interactions like open, pick-up, and toggle on/off across 200+ unique object types. It has an estimated state space of over $10^{25}$ possible environment configurations.

- **BEHAVIOR**: Provides low-level scripted actions like object grasp and movements. It provides over 100 unique low-level action types like grasps, movements, etc.

**Environments and Assets**

- **VirtualHome**: Has 57 fully furnished residential environments with over 3000 unique 3D object assets.

- **BEHAVIOR**: Provides 50 different indoor environments without physics simulation.
- **AI2-THOR**: Contains over 120 unique room configurations with over 1000 3D object assets.

Given our focus on embodied decision-making, which emphasizes the ability to navigate complex goals and extended action sequences, we have selected VirtualHome and BEHAVIOR for our research. VirtualHome is particularly suited for simulating long, intricate household tasks with its extensive high-level action sequences and richly detailed environments. BEHAVIOR complements this by offering a diverse set of human activities with detailed, low-level scripted actions, enabling comprehensive testing of decision-making capabilities across various scenarios. Together, these platforms provide a robust framework for evaluating and enhancing the decision-making abilities of embodied agents in complex, goal-oriented tasks.

## M.2 BEHAVIOR

For BEHAVIOR-100, we provide 2 additional sets of manual annotations: ground truth action sequences and natural language task descriptions. The annotation is done based on observing the real demonstrations [4].

### M.2.1 Annotating Action Sequence based on VR Human Demonstrations

The BEHAVIOR dataset consists of 100 tasks defined using the Behavior Domain Definition Language (BDDL), which is similar to the PDDL format. Each BDDL file has three parts: a list of objects, initial states, and goal states. To build the transition model for our simulator, we annotated the ground-truth action sequence for each task. The annotation process involves an automatic pipeline for objects' state tracking and segmentation from the demo, followed by manual effort for mapping state changes to action sequences.

### M.2.2 Complicated Goals with Quantifiers

BDDL goals may contain quantifiers, such as `forpairs( (?jar.n.01 - jar.n.01) (?apple.n.01 - apple.n.01)) (inside ?apple.n.01 ?jar.n.01)`, which need to be translated into grounded goals, e.g., `and ((inside apple.n.01_1 jar.n.01_1) (inside apple.n.01_2 jar.n.01_2))`. There can be different grounded goals that satisfy the same BDDL goal, which we refer to as goal options. For example, `((inside apple.n.01_2 jar.n.01_1) (inside apple.n.01_1 jar.n.01_2))` also satisfies the aforementioned BDDL goal. In general, one BDDL goal may have a number of goal options, and each goal option has a number of grounded atomic goals.

### M.2.3 Annotating Transition Models

**PDDL Problem File Annotation.** We devised an automatic pipeline to generate PDDL problem files from BDDL files defined for tasks in BEHAVIOR . Specifically, we would sample a solvable goal option from the BDDL goals, and select max _num grounded goals from this option as the goal for the PDDL problem file, then collect relevant objects and initial conditions for the grounded goals. This is done to ensure that the PDDL planner can solve each problem within the desired time frame.

**PDDL Domain File Annotation.** The PDDL domain file is modified from an existing PDDL domain file written for BEHAVIOR [§]. To make it compatible with our problem file and run in a PDDL planner, we modified the naming for object types, added new predicates, and changed the preconditions and post-effects for some operators.

## M.3 VirtualHome

We build our annotation of VirtualHome on top of RobotHow [5], which provides a categorized list of household tasks, each with a varying number of instructions. Each instruction includes a natural language goal description, a VirtualHome action script, and the initial and final states of the scene. However, RobotHow does not offer precise symbolic goals for each task, which limits the accuracy of evaluation due to noisy final states. For instance, it is more precise to denote the final goal of the task "*turn on light*" as *on(light)*, rather than a noisier version like *ontop(agent, chair) and on(light)*.

To enhance the reliability and standardization of evaluation outcomes, we manually annotated 338 task instructions across 30 task categories. Our annotation process involved two main steps:

---

[§]https://github.com/wmcclinton/downward/blob/main/behavior_full.pddl

### M.3.1 Wildcard Representations: Deriving Common Task-Related Final States

We identified common final states for all instructions within the same task category by using wildcards and object properties. This allows for multiple solutions for each task. For example, we use the property *clothes* to represent both *pants* and *shirts*. This step provides each task with a wildcard representation.

### M.3.2 Extracting and Annotating Abstract Goals

Based on the wildcard representations, we extracted all concrete goals and manually added any missing ones (which are usually few). We annotated the following three types of goals:

- State Goals: Represent object and agent states in the current scene, such as *plugged_in(TV)*.
- Relation Goals: Represent the spatial relationships that should be satisfied at the end of execution, such as *ontop(agent, chair)*.
- Action Goals: Represent actions that must be performed as part of the task instruction, especially those without post-effects. For example, in the task "*pat cat*", the agent is required to perform the action *touch* even though it has no post-effect in the final scene state.

### M.3.3 Grounding to Concrete Goals from Abstract Goals

In the VirtualHome simulator, each goal consists of an abstract goal and a concrete goal. An abstract goal is designed to be general and applicable to all potential trajectories. It may contain wildcards to represent similar objects, such as *book*, *notebook*, or *novel*. On the other hand, concrete goals are grounded in specific objects with their unique IDs, making them concrete and actionable.

The conversion between an abstract goal and a concrete goal involves mapping the wildcards to relevant objects in the scene. This process transforms the abstract goals into specific, tangible objectives that can be directly acted upon by the embodied agent. By replacing the wildcards with the appropriate object IDs, the abstract goals are instantiated into concrete goals that are tailored to the specific environment and the available objects. For example, an abstract goal like "pick up a *book*" may be converted into a concrete goal such as "pick up *book_1*" or "pick up *novel_2*", depending on the specific objects present in the scene and their corresponding IDs. This conversion process allows the embodied agent to have a clear understanding of the exact objects it needs to interact with to achieve the desired goal.

The mapping of wildcards to relevant objects is a crucial step in bridging the gap between the high-level, abstract goals and the low-level, executable actions. It enables the embodied agent to ground the goals in the context of the specific environment it operates in, making the goals more precise and achievable.

### M.3.4 Annotating Transition Models

In addition to the goals, we also annotate the transition model of preconditions and post-effects for each action in VirtualHome using standard PDDL formulations. Since VirtualHome does not provide an existing PDDL domain file, we define the predicate list and provide a PDDL implementation for each action.

The predicate list includes all relations (e.g., *inside*, *facing*), object states (e.g., *plugged_in*, *closed*), and object properties related to actions (e.g., *grabbable*, *has_switch*). The PDDL version of the actions follows the same preconditions and effects as their implementation in VirtualHome Github. The logic involves *and*, *or*, *not*, *when*, *exists*, and *forall*. Approximately 10 actions in VirtualHome simulate human body movement and have no effects on the environment. For example, the action *read* requires the robot to hold something readable but has no effects. Such actions would possibly be left out in the success rate by planner metrics because they would not contribute to the achievement of goals. However, we still provide their PDDL definitions and task the LLM to predict them, which is evaluated by the logic form accuracy metric.

### M.4 Quality Verification

To verify the quality of the annotations, we employed both automated and human evaluation methods:

- **Automated Verification**: We utilized the simulators (VirtualHome and BEHAVIOR) to automatically check the consistency and correctness of the annotations. This included verifying that the annotated actions and goals were executable and led to the desired outcomes within the simulation environments.
- **Human Evaluation**: In addition to automated checks, we performed human evaluations of the annotation quality. This involved assessing attributes such as action accuracy, action coverage, and overall human preference for the annotations.

## M.4.1 Annotation Quality Evaluation

The quality of the annotations was quantitatively evaluated using the following metrics:

Table 26: Annotation Quality Evaluation Metrics

| Attribute | Mean Score | Weighted MSE |
|---|---|---|
| Action Accuracy | 3.73 | 0.4062 |
| Action Coverage | 4.07 | 1.8438 |
| Human Preference | 4.27 | 0.8125 |

**Explanation of Metrics:**

- **Action Accuracy**: Measures how accurately the annotated actions correspond to the required actions for task completion.
- **Action Coverage**: Assesses the extent to which the annotations cover all necessary actions for task execution.
- **Human Preference**: Reflects the subjective preference of human evaluators for the quality and naturalness of the annotations.

To assess the variability of scores across tasks and evaluators, we calculated the Weighted Mean Squared Error (MSE):

- For tasks evaluated by multiple annotators, we computed the MSE per attribute by averaging the squared differences between individual scores and the mean score for that task.
- We then computed a weighted mean of these MSEs across all tasks, with weights based on the frequency of evaluations per task.

This method balances individual task variability with overall assessment consistency, accounting for varying numbers of evaluations per task.

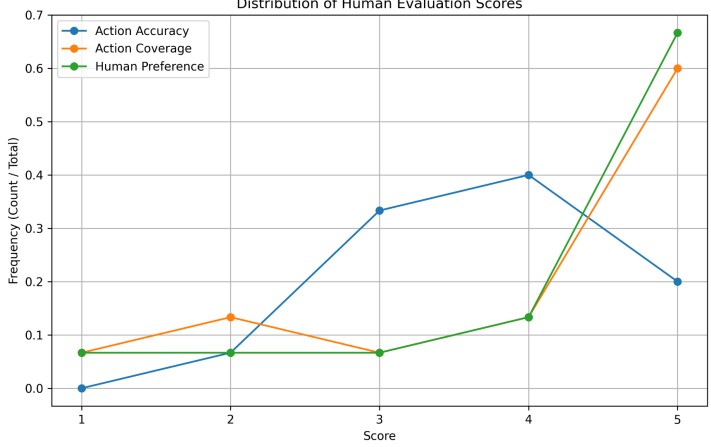

Figure 63: Score distribution visualization.

Our annotation process combines human expertise, iterative refinement, and automated checks to ensure a high-quality and reliable dataset. The evaluation metrics indicate that the annotations are of high quality, with mean scores above 3.7 out of 5 for all attributes and relatively low Weighted MSE values, demonstrating consistency among annotators. We also visualize the score distribution in Figure 63.

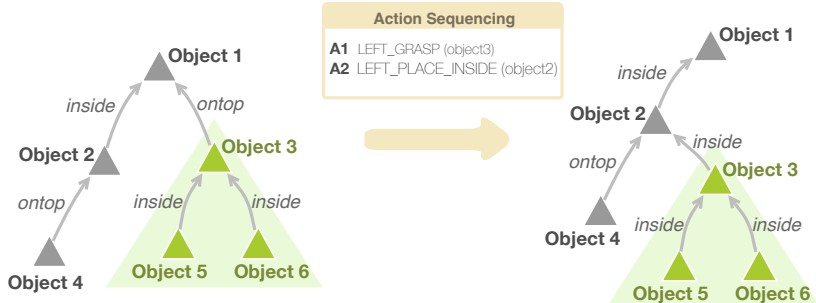

Figure 64: Implementation of Kinetics Tree for the Transition Model in BEHAVIOR .

# N Simulator Implementation Details

## N.1 BEHAVIOR Implementation Details

We developed an environment for evaluating LLMs in behavioral tasks, which includes two types of simulators. The first simulator is based on the iGibson framework [97], supporting visual rendering and physical simulations via PyBullet. The second simulator is symbolic, recording activities in a graph-based format. Both simulators share the same set of actions and object states.

### N.1.1 Building the Symbolic Simulator and Transition Model Implementation

Details about the arguments, preconditions, and post-effects of each action are presented in Table 27. An object's interactability is defined as (1) The object is not enclosed within another object, and (2) The object is within the agent's reach. However, as we currently allow auto-navigation for the agent, which automatically invokes the NAVIGATE_TO action, this reachability requirement is not necessary for LLMs to fulfill.

Table 27: BEHAVIOR Symbolic Simulator Implementation: Action Transition Models.

| Action Name | Arguments | Preconditions | Post Effects |
|---|---|---|---|
| NAVIGATE_TO | tar_obj1 | tar_obj1 is interactable. | The agent is next to tar_obj1. |
| LEFT_GRASP | tar_obj1 | tar_obj1 is interactable; One of the agent's hands is empty; tar_obj1 is small enough. | tar_obj1 is in the agent's left hand. |
| RIGHT_GRASP | tar_obj1 | tar_obj1 is interactable; The agent's right hand is empty; tar_obj1 is small enough. | tar_obj1 is in the agent's right hand. |
| LEFT_RELEASE | tar_obj1 | tar_obj1 is interactable; The agent's left hand holds tar_obj1. | tar_obj1 is released to the floor. |
| RIGHT_RELEASE | tar_obj1 | tar_obj1 is interactable; The agent's right hand holds tar_obj1. | tar_obj1 is released to the floor. |
| LEFT_PLACE_ONTOP | tar_obj1 | tar_obj1 is interactable; The agent's left hand holds an object. | The object in the agent's left hand is on top of tar_obj1. |
| RIGHT_PLACE_ONTOP | tar_obj1 | tar_obj1 is interactable; The agent's right hand holds an object. | The object in the agent's right hand is on top of tar_obj1. |

| Action Name | Arguments | Precondition | Post Effects |
| --- | --- | --- | --- |
| LEFT_PLACE_INSIDE | tar_obj1 | tar_obj1 is interactable; The agent's left hand holds an object; tar_obj1 is big enough and open if openable. | The object in the agent's left hand is inside tar_obj1. |
| RIGHT_PLACE_INSIDE | tar_obj1 | tar_obj1 is interactable; The agent's right hand holds an object; tar_obj1 is big enough and open if openable. | The object in the agent's right hand is inside tar_obj1. |
| LEFT_PLACE_NEXTTO | tar_obj1 | tar_obj1 is interactable; The agent's left hand holds an object. | The object in the agent's left hand is next to tar_obj1. |
| RIGHT_PLACE_NEXTTO | tar_obj1 | tar_obj1 is interactable; The agent's right hand holds an object. | The object in the agent's right hand is next to tar_obj1. |
| LEFT_PLACE_UNDER | tar_obj1 | tar_obj1 is interactable; The agent's left hand holds an object. | The object in the agent's left hand is under tar_obj1. |
| RIGHT_PLACE_UNDER | tar_obj1 | tar_obj1 is interactable; The agent's right hand holds an object. | The object in the agent's right hand is under tar_obj1. |
| LEFT_TRANSFER_ CONTENTS_INSIDE | tar_obj1 | tar_obj1 is interactable; The agent's left hand holds an object; tar_obj1 is big enough and open if openable. | The contents inside the object in the agent's left hand are in tar_obj1. |
| RIGHT_TRANSFER_ CONTENTS_INSIDE | tar_obj1 | tar_obj1 is interactable; The agent's right hand holds an object; tar_obj1 is big enough and open if openable. | The contents inside the object in the agent's right hand are in tar_obj1. |
| LEFT_TRANSFER_ CONTENTS_ONTOP | tar_obj1 | tar_obj1 is interactable; The agent's left hand holds an object. | The contents inside the object in the agent's left hand are on top of tar_obj1. |
| RIGHT_TRANSFER_ CONTENTS_ONTOP | tar_obj1 | tar_obj1 is interactable; The agent's right hand holds an object. | The contents inside the object in the agent's right hand are on top of tar_obj1. |
| LEFT_PLACE_ NEXTTO_ONTOP | tar_obj1, tar_obj2 | tar_obj1 and tar_obj2 are interactable; The agent's left hand holds an object. | The object in the agent's left hand is next to tar_obj1 and on top of tar_obj2. |
| RIGHT_PLACE_ NEXTTO_ONTOP | tar_obj1, tar_obj2 | tar_obj1 and tar_obj2 are interactable; The agent's right hand holds an object. | The object in the agent's right hand is next to tar_obj1 and on top of tar_obj2. |
| TOGGLE_ON | tar_obj1 | tar_obj1 is interactable; One of the agent's hands is empty; tar_obj1 is toggled off and closed. | tar_obj1 is toggled on. |
| TOGGLE_OFF | tar_obj1 | tar_obj1 is interactable; One of the agent's hands is empty; tar_obj1 is toggled on. | tar_obj1 is toggled off. |
| CLOSE | tar_obj1 | tar_obj1 is interactable; One of the agent's hands is empty; tar_obj1 is open. | tar_obj1 is closed. |

| Action Name | Arguments | Precondition | Post Effects |
|---|---|---|---|
| OPEN | tar_obj1 | tar_obj1 is interactable; One of the agent's hands is empty; tar_obj1 is closed and not toggled on. | tar_obj1 is open. |
| CLEAN | tar_obj1 | tar_obj1 is interactable; tar_obj1 is dusty or stained; the agent has a cleaning tool. | tar_obj1 is not dusty or stained (according to the tool). |
| DRY | tar_obj1 | tar_obj1 is interactable; tar_obj1 is soaked. | tar_obj1 is dry. |
| SLICE | tar_obj1 | tar_obj1 is interactable; tar_obj1 is not sliced; the agent has a knife. | tar_obj1 is sliced. |
| SOAK | tar_obj1 | tar_obj1 is interactable; One of the agent's hands is empty; tar_obj1 is dry and in a toggled on sink or a pot. | tar_obj1 is soaked. |
| FREEZE | tar_obj1 | tar_obj1 is interactable; One of the agent's hands is empty; tar_obj1 is unfrozen and in the fridge. | tar_obj1 is frozen. |
| UNFREEZE | tar_obj1 | tar_obj1 is interactable; tar_obj1 is frozen. | tar_obj1 is unfrozen. |
| COOK | tar_obj1 | tar_obj1 is interactable; One of the agent's hands is empty; tar_obj1 is not cooked; tar_obj1 is on/in the pan. | tar_obj1 is cooked. |

### N.1.2  BEHAVIOR State Space and Action Space

Our environment defines 15 object states, derived from a subset of iGibson's state definitions, and offers 30 actions that agents can use to interact with the objects in a task scene. The names of these actions and object states are provided in Table 28. Object states are categorized into unary states, describing conditions or attributes of an object, and binary states, representing physical relationships between two objects. Actions are divided into state actions, which modify the unary states of objects, and spatial actions, which modify the binary states between two objects or between an object and the agent.

### N.2  VirtualHome Implementation Details

### N.2.1  VirtualHome State Space and Action Space

We have developed an evaluation environment for LLMs using the VirtualHome simulator as our foundation. At the core of this environment is a MotionPlanner, which leverages the EnvironmentGraph from VirtualHome to record the state of the environment. Additionally, the MotionPlanner incorporates runtime error analysis for all 42 possible actions. To ensure accurate execution outcomes, we made necessary modifications to the VirtualHome simulator, aligning its action executors with expected results. The current state space and action space utilized in this environment are detailed in Table 29.

## O  Evaluation Settings of LLMs

### O.1  Decoding Parameters

To ensure standardization and consistency across all models, we utilized the same set of decoding parameters for all the LLMs evaluated in this study. Specifically, we used a temperature of zero for all models, as our goal was to use the arg max under the model's distribution. Furthermore, since several of the models only support temperature-based sampling and not other sampling methods, we limited ourselves to temperature scaling during

Table 28: BEHAVIOR State Space and Action Space.

| State Space | | Action Space | |
| --- | --- | --- | --- |
| **Unary States** | **Binary States** | **State Actions** | **Spatial Actions** |
| *Cooked* | *Inside* | TOGGLE_ON | LEFT_GRASP |
| *Dusty* | *OnFloor* | TOGGLE_OFF | RIGHT_GRASP |
| *Frozen* | *OnTop* | CLOSE | LEFT_RELEASE |
| *Open* | *Under* | OPEN | RIGHT_RELEASE |
| *Sliced* | *NextTo* | CLEAN | LEFT_PLACE_ONTOP |
| *Soaked* | | DRY | RIGHT_PLACE_ONTOP |
| *Stained* | | SLICE | LEFT_PLACE_NEXTTO |
| *ToggledOn* | | SOAK | RIGHT_PLACE_NEXTTO |
| *Slicer* | | FREEZE | LEFT_PLACE_INSIDE |
| *CleaningTool* | | UNFREEZE | RIGHT_PLACE_INSIDE |
| | | COOK | LEFT_PLACE_UNDER |
| | | | RIGHT_PLACE_UNDER |
| | | | LEFT_TRANSFER_CONTENTS_INSIDE |
| | | | RIGHT_TRANSFER_CONTENTS_INSIDE |
| | | | LEFT_TRANSFER_CONTENTS_ONTOP |
| | | | RIGHT_TRANSFER_CONTENTS_ONTOP |
| | | | LEFT_PLACE_NEXTTO_ONTOP |
| | | | RIGHT_PLACE_NEXTTO_ONTOP |
| | | | NAVIGATE_TO |

Table 29: VirtualHome State Space and Action Space.

| State Space | | Action Space | |
| --- | --- | --- | --- |
| **Unary States** | **Binary States** | **State Actions** | **Spatial Actions** |
| *Closed* | *On* | OPEN | TURN_TO |
| *Open* | *Inside* | SIT | PUT_BACK |
| *On* | *Between* | STANDUP | PUT_IN |
| *Off* | *Close* | SLEEP | POUR |
| *Sitting* | *Facing* | WAKEUP | PUT_ON |
| *Dirty* | *Holds_RH* | CLOSE | FIND |
| *Clean* | *Holds_LH* | DRINK | RUN |
| *Lying* | | GRAB | WALK |
| *Plugged_in* | | LOOKAT | POINT_AT |
| *Plugged_out* | | LOOKAT_SHORT | TOUCH |
| | | LOOKAT_LONG | WATCH |
| | | SWITCH_OFF | MOVE |
| | | SWITCH_ON | RELEASE |
| | | TYPE | DROP |
| | | PUSH | |
| | | PULL | |
| | | SQUEEZE | |
| | | WASH | |
| | | RINSE | |
| | | SCRUB | |
| | | EAT | |
| | | PLUG_IN | |
| | | PLUG_OUT | |
| | | CUT | |
| | | READ | |
| | | LIE | |

the sampling process. It is important to note that for a given prompt, the model's completion involves sampling, which introduces randomness in determining the specific completion decoded for each instance. However, in our scenarios, this is not a significant factor since we are decoding the $\arg\max$ through low-temperature decoding.

## O.2   Evaluation Cost

To perform a single run of all models on our benchmark, a total of 180 runs would be required. This involves evaluating each specific model on each specific simulator ability module. The total cost of this process amounts to approximately 192,280,000 tokens and 84,960 queries across all models, which results in an API cost of $1540.70. For open-source models, costs are based on the pricing of the Together AI API [¶].

## O.3   Model Cards

Due to space limitations, we use shorthand model names in the main paper. Here we provide the details of the models in Table 30.

Table 30: Model Cards for All Evaluated Large Language Models

| Model Name | Creator | Complete Model ID | Release | Hosting |
|---|---|---|---|---|
| Claude-3 Haiku | Anthropic | claude-3-haiku-20240307 | 03/07/24 | Anthropic |
| Claude-3 Sonnet | Anthropic | claude-3-sonnet-20240229 | 02/29/24 | Anthropic |
| Claude-3 Opus | Anthropic | claude-3-opus-20240229 | 02/29/24 | Anthropic |
| Claude-3.5 Sonnet | Anthropic | claude-3-5-sonnet-20240620 | 06/20/24 | Anthropic |
| Cohere Command R | Cohere | command-r | 03/11/24 | Cohere |
| Cohere Command R+ | Cohere | command-r-plus | 04/04/24 | Cohere |
| Gemini 1.0 Pro | Google | gemini-pro | 12/13/23 | GCP Vertex |
| Gemini 1.5 Flash | Google | gemini-1.5-flash-preview-0514 | 05/14/24 | GCP Vertex |
| Gemini 1.5 Pro | Google | gemini-1.5-pro-preview-0409 | 04/09/24 | GCP Vertex |
| GPT-3.5-turbo | OpenAI | gpt-3.5-turbo-0125 | 01/25/24 | OpenAI |
| GPT-4-turbo | OpenAI | gpt-4-turbo-2024-04-09 | 04/09/24 | OpenAI |
| GPT-4o | OpenAI | gpt-4o-2024-05-13 | 05/13/24 | OpenAI |
| Llama 3 8B Instruct | Meta | meta-llama-3-8b-instruct | 04/18/24 | TogetherAI |
| Llama 3 70B Instruct | Meta | meta-llama-3-70b-instruct | 04/18/24 | TogetherAI |
| Mistral Large | MistralAI | mistral-large-2402 | 02/26/24 | MistralAI |
| Mixtral 8x22B MoE | MistralAI | mixtral-8x22b-instruct-v0.1 | 04/17/24 | TogetherAI |
| o1-mini | OpenAI | o1-mini-2024-09-12 | 09/12/24 | OpenAI |
| o1-preview | OpenAI | o1-preview-2024-09-12 | 09/12/24 | OpenAI |

# P   Extensive Related Work

## P.1   LLMs for Embodied Planning

Existing works in embodied task and motion planning (TAMP) have used LLMs to perform varying tasks, serving different ability modules defined within our embodied agent interface. Due to the page limit, we only list a subset of such works and their categorization in the main paper. Here in Table 31 we provide an extended list of such works with detailed categorization for your reference.

---

[¶]https://www.together.ai/

Table 31: Categorization of Existing Embodied Agent Planning Works' Usage of Large Language Models: Each "LLMs" refers to the usage of LLMs to perform an ability module. For example, Ada [8] uses LLMs for Action Sequencing and Transition Modeling, while LLM+P [7] uses LLMs for Goal Interpretation.

| Existing Work | Ref. | Goal Interpretation | Action Sequencing | Subgoal Decomposition | Transition Modeling |
|---|---|---|---|---|---|
| SayCan | [2] | LLMs | LLMs | | |
| Ada | [8] | LLMs | | | LLMs |
| LLP+P | [7] | LLMs | | | |
| AutoTAMP | [18] | | LLMs | | LLMs |
| Code as Policies | [6] | LLMs | LLMs | LLMs | |
| Voyager | [35] | LLMs | LLMs | | |
| Demo2Code | [13] | LLMs | | LLMs | LLMs |
| LM as ZeroShot Planner | [3] | | LLMs | LLMs | |
| SayPlan | [15] | LLMs | LLMs | | LLMs |
| Text2Motion | [48] | | LLMs | | |
| LLMGROP | [46] | LLMs | LLMs | | |
| REFLECT | [66] | LLMs | LLMs | | |
| Generating Consistent PDDL Domains with LLMs | [32] | LLMs | | | LLMs |
| PlanSeqLearn | [64] | | LLMs | | |
| COWP | [70] | LLMs | LLMs | | LLMs |
| HumanAssisted Continual Robot Learning | [65] | | LLMs | | |
| DECKARD | [34] | LLMs | | | LLMs |
| MOSAIC | [63] | | LLMs | | |
| Interactive Task Planning with Language Models | [14] | | LLMs | LLMs | |
| RoCo | [62] | | LLMs | | |
| Cook2LTL | [42] | LLMs | | | |
| InnerMonologue | [21] | | LLMs | | |
| MLDT | [20] | | LLMs | | |
| Learning to Reason over Scene Graphs | [61] | LLMs | LLMs | | LLMs |
| GRID | [19] | LLMs | LLMs | | |
| LLMplanner | [33] | LLMs | LLMs | | |
| DELTA | [17] | | LLMs | | |
| Look Before You Leap | [60] | LLMs | LLMs | | |
| CAPE | [36] | LLMs | LLMs | | |
| HERACLEs | [43] | | LLMs | | |
| RoboTool | [16] | | LLMs | | LLMs |
| PROMST | [59] | | LLMs | | |
| LLM3 | [31] | LLMs | LLMs | | |
| Ghost in the Minecraft | [27] | | LLMs | | |
| PlanBench | [98] | LLMs | LLMs | | |
| TaPA | [30] | LLMs | LLMs | LLMs | |
| ChatGPT Robot Control | [29] | | LLMs | | |
| LLM World Models for Planning | [28] | LLMs | LLMs | | |
| DEPS | [81] | LLMs | LLMs | | |

| Existing Works | Ref. | Goal Interpretation | Action Sequencing | Subgoal Decomposition | Transition Modeling |
|---|---|---|---|---|---|
| Grounded Decoding | [25] | | LLMs | | |
| ProgPrompt | [37] | LLMs | LLMs | | |
| DROC | [24] | | LLMs | | LLMs |
| LMPC | [58] | LLMs | LLMs | | |
| GPTPDDL | [47] | | LLMs | | |
| RAP | [57] | | LLMs | | |
| LEAGUE++ | [38] | | LLMs | | LLMs |
| CoPAL | [23] | LLMs | LLMs | | |
| SayCanPay | [22] | LLMs | LLMs | | |
| LLMGenPlan | [56] | | LLMs | | |

## P.2  LTL Agent Interface

Initially proposed as a formal verification for computer programs [44], Linear Temporal Logic (LTL) expressions have since then been extensively used in robotics [39–41] as a formalism that enables the extraction of guarantees on robot performance given a robot model, a high-level description of robotic actions, and a class of admissible environments. Recent work in embodied agent planning has adopted LTL expressions for their agent interface due to their high expressiveness and formality, bridging the gap between natural language and robotic control language [42, 43, 18]. Among these works [42] and [43] directly translate natural language actions and task goals into LTL to perform planning tasks, while [18] utilizes an intermediary representation called Signal Temporal Logic (STL) derived from LTL for action and subgoal planning. Due to the compactness and expressiveness of LTL, our work leverages LTL as a unified interface between modules and agents to describe task specifications.

## P.3  Embodied Agent Benchmarks

Recent years have seen a proliferation of benchmarks for evaluating embodied simulation and decision-making, each focusing on different aspects of the problem. PlanBench [99] provides an extensible framework for evaluating LLMs on planning and reasoning about change but is limited to generating action sequences without considering other crucial aspects of embodied decision-making. Similarly, LoTa-Bench [9] benchmarks language-oriented task planners for embodied agents but focuses primarily on action sequencing and lacks support for fine-grained analysis of planning errors.

Several benchmarks are built on specific simulators. ALFRED [71] offers a benchmark for interpreting grounded instructions for everyday tasks, while VirtualHome [5] simulates household activities via programs. However, both lack a standardized interface for evaluating different LLM-based modules and don't provide detailed metrics for analyzing various types of errors in embodied decision-making tasks.

The HAZARD Challenge [100] evaluates embodied decision-making in dynamically changing environments, but primarily emphasizes social metrics such as the value of rescued objects and damage, rather than providing a comprehensive evaluation of language-based reasoning and planning. ACT-Thor [101] provides a controlled benchmark for embodied action understanding in simulated environments but focuses mainly on action recognition and prediction rather than the full spectrum of embodied decision-making tasks.

In contrast to these existing benchmarks, our Embodied Agent Interface provides a more comprehensive evaluation framework. It covers multiple decision-making modules (goal interpretation, subgoal decomposition, action sequencing, and transition modeling), offers standardized interfaces for modular evaluation of LLM capabilities, and includes fine-grained metrics for various types of errors. Moreover, by using LTL formulas for goal specification, our framework enables the unified representation and evaluation of complex, temporally extended goals, addressing a key limitation in many existing benchmarks.

# Q  Maintenance Plan

## Q.1  Dataset URLs, License, and Hosting Plan

Our dataset is also hosted on https://github.com/embodied-agent-interface/embodied-agent-interface/ which will provide long-term support for hosting the dataset.

Contact information for the authors is available on our website. We can also be reached through GitHub issues or via email for any inquiries or support related to the dataset and evaluation tool. We will maintain an erratum on the GitHub issues to host any approved corrections or updates suggested by the authors or the broader video research community.

## Q.2    Long-term Preservation and DOI

We provide a persistent dereferenceable identifier DOI: `https://datadryad.org/stash/share/bLEj8IHqyKc9c_Ff0M8tTSZMTxOLxIup5zUWn9yq_j8`.

## Q.3    URL of Croissant Metadata Record

We provide structured metadata (schema.org standards) in `https://huggingface.co/datasets/Inevitablevalor/EmbodiedAgentInterface`.

## Q.4    Author Statement

The authors bear all responsibility in case of violation of rights. All dataset annotations were collected by the authors and we are releasing the dataset under CC-BY-4.0.

## Q.5    URLs of Code and Re-productivity

We have made the codebase publicly available on GitHub (`https://github.com/embodied-agent-interface/embodied-agent-interface/`), along with detailed instructions. In this section, we provide the URLs for the VirtualHome evaluator, BEHAVIOR evaluator, and LLM implementations. Further details regarding these implementations can be found in Appendix N.

### Q.5.1    Code for VirtualHome Evaluator and Computation Resources

We provide the code for the VirtualHome evaluator at `https://github.com/embodied-agent-interface/embodied-agent-interface/`, along with detailed usage instructions. This code facilitates goal interpretation, subgoal decomposition, action sequencing, and transition modeling.

To use our evaluator, ensure that Docker is installed in your environment. For annotating the RobotHow dataset, we also offer code for human interactive annotation of goals within VirtualHome , with usage instructions available in the repository.

The experiment of subgoal decomposition is run on a Windows 11 system with an AMD Ryzen 7 5800H processor and 16GB of memory. For goal interpretation, action sequencing, and transition modeling, the experiments are run on a Linux system with 8 Intel(R) Xeon(R) Platinum 8481C CPU @ 2.70GHz and 32GB of memory.

### Q.5.2    Code for BEHAVIOR Evaluator and Computation Resources

The code for the BEHAVIOR evaluator is available on GitHub at `https://github.com/embodied-agent-interface/embodied-agent-interface/`. We provide easy-to-reproduce scripts for prompt generation and evaluation of each module: *goal interpretation*, *action sequencing*, *subgoal decomposition*, and *transition modeling*. The BEHAVIOR related experiments were conducted on a Windows 11 system equipped with an AMD Ryzen 7 7800X3D processor and 32GB of RAM.

### Q.5.3    Code for LLMs Implementations and Computation Resources

We provide convenient and easy-to-reproduce batch LLM inference by directly integrating our evaluation pipeline into the HELM [83] code base. In order to allow quick and easy access for all users with different development environments and GPU setups, we provide the option to use Together AI ‖ APIs for large open-source model inference. Users can easily set up their inference environment by following the instructions in `https://github.com/embodied-agent-interface/embodied-agent-interface`.

# R    Datasheets for EMBODIED AGENT INTERFACE (EAI)



**Motivation**



**For what purpose was the dataset created?** Was there a specific task in mind? Was there a specific gap that needed to be filled? Please provide a description.

Please refer to Appendix B.1.

---

‖ https://www.together.ai/

**Who created this dataset (e.g., which team, research group) and on behalf of which entity (e.g., company, institution, organization)?**

The authors created the dataset within the Stanford Vision and Learning Lab at Stanford University. It is created for public use and is not affiliated with or tied to any specific organization or institution.

**Who funded the creation of the dataset?** If there is an associated grant, please provide the name of the grantor and the grant name and number.

This work was in part supported by the Stanford Institute for Human-Centered Artificial Intelligence (HAI), NSF CCRI #2120095, AFOSR YIP FA9550-23-1-0127, ONR MURI N00014-22-1-2740, ONR YIP N00014-24-1-2117, Amazon, and Microsoft.

**Any other comments?**

No.

---

## Composition

**What do the instances that comprise the dataset represent (e.g., documents, photos, people, countries)?** Are there multiple types of instances (e.g., movies, users, and ratings; people and interactions between them; nodes and edges)? Please provide a description.

Each instance in the dataset represents a task with its corresponding ground truth plan. Specifically, each task contains the following data: (1) natural language task name, (2) natural language task instruction, (3) symbolic goal definition, (4) symbolic action trajectory, and (5) the transition models involved in the task. For tasks in the BEHAVIOR environment, the dataset also includes accompanying demo videos that showcase the execution of the ground truth action trajectories. Further details regarding the dataset format can be found in Appendix L.1.

**How many instances are there in total (of each type, if appropriate)?**

We have released a dataset containing task plans and trajectories for two environments: VirtualHome and BEHAVIOR . For VirtualHome , there are 338 task plans/trajectories with a total of 2,960 steps and 801 goal conditions. For BEHAVIOR , there are 100 task plans/trajectories with a total of 1,460 steps and 673 goal conditions. Additionally, we have annotated transition models for both environments. VirtualHome has 33 annotated transition models with 99 preconditions and 57 effects, while BEHAVIOR has 30 annotated transition models with 84 preconditions and 51 effects.

More detailed information can be found in Table 2 of the main paper. Further statistics on the entire dataset are available on the website `https://embodied-agent-interface.github.io/`. We will continue releasing more data and updating the information on the website in the future.

**Does the dataset contain all possible instances or is it a sample (not necessarily random) of instances from a larger set?** If the dataset is a sample, then what is the larger set? Is the sample representative of the larger set (e.g., geographic coverage)? If so, please describe how this representativeness was validated/verified. If it is not representative of the larger set, please describe why not (e.g., to cover a more diverse range of instances, because instances were withheld or unavailable).

The benchmark is built on top of two simulators VirtualHome  [5] and BEHAVIOR -100 [4]. To better evaluate the decision-making ability, we selectively focus on long-horizon tasks with complicated goals in VirtualHome and BEHAVIOR-100. We select these two simulators due to their length of task plans and the number of goal conditions, as detailed in Appendix M.1. Detailed statistics of the selected subset are discussed in Appendix L.2, where we show that the selected subset is very diverse in tasks and environments.

**What data does each instance consist of?** Raw data (e.g., unprocessed text or images) or features? In either case, please provide a description.

The benchmark contains task annotations along with an evaluator codebase for executing plans in the simulators `https://github.com/embodied-agent-interface/embodied-agent-interface/`. This codebase is created by modifying the existing simulator backbones VirtualHome and BEHAVIOR. For example, in the BEHAVIOR environment, we implemented a symbolic simulator. Detailed information about the evaluator implementations and the revisions made to the simulators to support the evaluation can be found in Appendix N.1 for the BEHAVIOR environment and Appendix N.2 for the VirtualHome environment. Using the provided simulator, users can visualize the plan execution process and capture screenshots. For tasks in the BEHAVIOR environment, the dataset also includes demo videos that demonstrate the execution of the ground truth action trajectories, complementing the annotated ground truth goals and plans.

**Is there a label or target associated with each instance?** If so, please provide a description.

Each instance is associated with a sequence of labels showing the ground truth action sequence.

**Is any information missing from individual instances?** If so, please provide a description, explaining why this information is missing (e.g. because it was unavailable). This does not include intentionally removed information but might include, e.g., redacted text.

All instances are complete.

**Are relationships between individual instances made explicit (e.g., users' movie ratings, social network links)?** If so, please describe how these relationships are made explicit.

Some instances may share the same task name or same goals, but different trajectories. It can be viewed as alternative decision-making trajectories to achieve the goal. It will be indicated by a unique identifier indicating the scene and the task.

**Are there recommended data splits (e.g., training, development/validation, testing)?** If so, please provide a description of these splits, explaining the rationale behind them.

The benchmark presented in this work is specifically designed for zero-shot testing, with the primary goal of assessing the out-of-the-box long-horizon embodied decision-making capabilities of LLMs.

**Are there any errors, sources of noise, or redundancies in the dataset?** If so, please provide a description.

The dataset was very carefully manually curated to mitigate any incidence of errors within the goal annotations, trajectory annotations, and transition model annotations. For VirtualHome , we cross-examine the goal annotations by executing alternative action trajectories and summarizing the overlapping final states. After that, we manually cleaned and removed any actions that were deemed unrelated. The annotation process, which utilized a human-interactive annotation program, is discussed in further detail in Appendix M.3. For the BEHAVIOR environment, we cross-examine the action sequence annotations by grounding them to multiple demonstration videos. This process is described in more detail in Appendix M.2.

**Is the dataset self-contained, or does it link to or otherwise rely on external resources (e.g., websites, tweets, other datasets)?** If it links to or relies on external resources, a) are there guarantees that they will exist, and remain constant, over time; b) are there official archival versions of the complete dataset (i.e., including the external resources as they existed at the time the dataset was created); c) are there any restrictions (e.g., licenses, fees) associated with any of the external resources that might apply to a future user? Please provide descriptions of all external resources and any restrictions associated with them, as well as links or other access points, as appropriate.

The entire dataset will be made publicly available on our project website https://embodied-agent-interface.github.io/. We will also provide a dockerized simulator evaluator tool for executing plans within the simulator environment. The text data will be released in JSON format and hosted on our website. Detailed information about the dataset format can be found in Appendix L.1. The benchmark will be released under the VirtualHome and BEHAVIOR licenses, which allow public use of the simulators for both research and commercial purposes.

**Does the dataset contain data that might be considered confidential (e.g., data that is protected by legal privilege or by doctor-patient confidentiality, data that includes the content of individuals' non-public communications)?** If so, please provide a description.

No.

**Does the dataset contain data that, if viewed directly, might be offensive, insulting, threatening, or might otherwise cause anxiety?** If so, please describe why.

No.

**Does the dataset relate to people?** If not, you may skip the remaining questions in this section.

No.

**Does the dataset identify any subpopulations (e.g., by age, gender)?** If so, please describe how these subpopulations are identified and provide a description of their respective distributions within the dataset.

No.

**Is it possible to identify individuals (i.e., one or more natural persons), either directly or indirectly (i.e., in combination with other data) from the dataset?** If so, please describe how.

No.

**Does the dataset contain data that might be considered sensitive in any way (e.g., data that reveals racial or ethnic origins, sexual orientations, religious beliefs, political opinions or union memberships, or locations;**

**financial or health data; biometric or genetic data; forms of government identification, such as social security numbers; criminal history)?** If so, please provide a description.

No.

**Any other comments?**

No.

---



**Collection Process**



**How was the data associated with each instance acquired?** Was the data directly observable (e.g., raw text, movie ratings), reported by subjects (e.g., survey responses), or indirectly inferred/derived from other data (e.g., part-of-speech tags, model-based guesses for age or language)? If data was reported by subjects or indirectly inferred/derived from other data, was the data validated/verified? If so, please describe how.

We annotate the data based on existing annotations from the VirtualHome and BEHAVIOR environments. To ensure the accuracy of the annotations, we execute the plans in the simulator and verify that the goals are satisfied.

**What mechanisms or procedures were used to collect the data (e.g., hardware apparatus or sensor, manual human curation, software program, software API)?** How were these mechanisms or procedures validated?

As mentioned above, the data is collected based on existing annotations from VirtualHome and BEHAVIOR . We supplement these annotations with additional manual annotations about goals, trajectories, transition models, and natural language task instructions, which are described in detail in Appendix M.

**If the dataset is a sample from a larger set, what was the sampling strategy (e.g., deterministic, probabilistic with specific sampling probabilities)?**

To better evaluate the decision-making ability, we selectively focus on long-horizon tasks with complicated goals in VirtualHome and BEHAVIOR-100. We select these two simulators due to their length of task plans and the number of goal conditions, as detailed in Appendix M.1. Detailed statistics of the selected subset is discussed in Appendix L.2, where we show that the selected subset is very diverse in tasks and environments.

**Who was involved in the data collection process (e.g., students, crowd workers, contractors) and how were they compensated (e.g., how much were crowd workers paid)?**

The annotations are purely done by the authors in this paper, which are expert researchers.

**Over what timeframe was the data collected? Does this timeframe match the creation timeframe of the data associated with the instances (e.g., the recent crawl of old news articles)?** If not, please describe the timeframe in which the data associated with the instances was created.

The newly added annotations are done in 2024. VirtualHome was created in 2018, and BEHAVIOR was created in 2022.

**Were any ethical review processes conducted (e.g., by an institutional review board)?** If so, please provide a description of these review processes, including the outcomes, as well as a link or other access point to any supporting documentation.

No.

**Does the dataset relate to people?** If not, you may skip the remaining questions in this section.

No.

**Did you collect the data from the individuals in question directly, or obtain it via third parties or other sources (e.g., websites)?**

N/A.

**Were the individuals in question notified about the data collection?** If so, please describe (or show with screenshots or other information) how notice was provided, and provide a link or other access point to, or otherwise reproduce, the exact language of the notification itself.

N/A.

**Did the individuals in question consent to the collection and use of their data?** If so, please describe (or show with screenshots or other information) how consent was requested and provided, and provide a link or other access point to, or otherwise reproduce, the exact language to which the individuals consented.

N/A.

**If consent was obtained, were the consenting individuals provided with a mechanism to revoke their consent in the future or for certain uses?** If so, please provide a description, as well as a link or other access point to the mechanism (if appropriate).

N/A.

**Has an analysis of the potential impact of the dataset and its use on data subjects (e.g., a data protection impact analysis) been conducted?** If so, please provide a description of this analysis, including the outcomes, as well as a link or other access point to any supporting documentation.

N/A.

**Any other comments?**

No.



**Preprocessing/cleaning/labeling**



**Was any preprocessing/cleaning/labeling of the data done (e.g., discretization or bucketing, tokenization, part-of-speech tagging, SIFT feature extraction, removal of instances, processing of missing values)?** If so, please provide a description. If not, you may skip the remainder of the questions in this section.

The annotations were curated by humans. We execute the trajectories in the simulators to validate the goals, trajectories, and transition models.

**Was the raw data saved in addition to the preprocessed/cleaned/labeled data (e.g., to support unanticipated future uses)?** If so, please provide a link or other access point to the "raw" data.

Human annotations of missing goals, trajectories, and transition models are created from scratch, with the process detailed in Appendix M. The raw demo videos used for annotation reference are publicly available from BEHAVIOR [4].

**Is the software used to preprocess/clean/label the instances available?** If so, please provide a link or other access point.

The code used to facilitate the dataset annotation process is available at `https://github.com/embodied-agent-interface/embodied-agent-interface/` and is described in detail in Appendix M.

**Any other comments?**

No.



**Uses**



**Has the dataset been used for any tasks already?** If so, please provide a description.

The dataset presented in this work has been used as a benchmark for evaluating the performance of 15 large language models (LLMs). More details about these LLMs can be found in Appendix O. Furthermore, the code for conducting zero-shot evaluation using our dataset will be made available on our project's GitHub repository.

**Is there a repository that links to any or all papers or systems that use the dataset?** If so, please provide a link or other access point.

It will be made public on our website once more papers start to use our dataset.

**What (other) tasks could the dataset be used for?** Is there anything about the composition of the dataset or the way it was collected and preprocessed/cleaned/labeled that might impact future uses? For example, is there anything that a future user might need to know to avoid uses that could result in unfair treatment of individuals or groups (e.g., stereotyping, quality of service issues) or other undesirable harms (e.g., financial harms, legal risks) If so, please provide a description. Is there anything a future user could do to mitigate these undesirable harms?

There may be potential to benchmark vision-language foundation models (VLMs) using our benchmark. However, we leave the exploration of these possibilities for future work.

**Is there anything about the composition of the dataset or the way it was collected and preprocessed/cleaned/labeled that might impact future uses?** For example, is there anything that a future user might need to know to avoid uses that could result in unfair treatment of individuals or groups (e.g., stereotyping, quality of service issues) or other undesirable harms (e.g., financial harms, legal risks) If so, please provide a description. Is there anything a future user could do to mitigate these undesirable harms?

No.

**Are there tasks for which the dataset should not be used?** If so, please provide a description.

No.

**Any other comments?**

No.



| Distribution |
| :---: |

**Will the dataset be distributed to third parties outside of the entity (e.g., company, institution, organization) on behalf of which the dataset was created?** If so, please provide a description.

The dataset will be made publicly available and can be used for both research and commercial purposes under the VirtualHome and BEHAVIOR licenses.

**How will the dataset be distributed (e.g., tarball on the website, API, GitHub)** Does the dataset have a digital object identifier (DOI)?

The dataset will be distributed as a JSON file that includes a unique identifier for each task goal, along with its associated action trajectory, natural language task name, natural language task instructions, symbolic goal definition, and the transition models involved in the task. Additionally, we will release the evaluator, which enables the automatic calculation of fine-grained systematic metrics. Further details regarding the dataset format can be found in Appendix L.1.

**When will the dataset be distributed?**

The full dataset will be made available upon the acceptance of the paper before the camera-ready deadline. We release it on our website.

**Will the dataset be distributed under a copyright or other intellectual property (IP) license, and/or under applicable terms of use (ToU)?** If so, please describe this license and/or ToU, and provide a link or other access point to, or otherwise reproduce, any relevant licensing terms or ToU, as well as any fees associated with these restrictions.

The dataset will be publicly released under the VirtualHome and BEHAVIOR licenses, which allow direct public use for both research and commercial purposes.

**Have any third parties imposed IP-based or other restrictions on the data associated with the instances?** If so, please describe these restrictions, and provide a link or other access point to, or otherwise reproduce, any relevant licensing terms, as well as any fees associated with these restrictions.

No.

**Do any export controls or other regulatory restrictions apply to the dataset or to individual instances?** If so, please describe these restrictions, and provide a link or other access point to, or otherwise reproduce, any supporting documentation.

No.

**Any other comments?**

No.



| Maintenance |
| :---: |

**Who will be supporting/hosting/maintaining the dataset?**

The authors of the paper will maintain the dataset, and pointers to the dataset will be hosted on the GitHub repository https://github.com/embodied-agent-interface/embodied-agent-interface/. The repository will also include the code for downloading the dataset and the evaluation tool.

**How can the owner/curator/manager of the dataset be contacted (e.g., email address)?**

Contact information for the authors is available on our website. We can also be reached through GitHub issues or via email for any inquiries or support related to the dataset and evaluation tool.

**Is there an erratum?** If so, please provide a link or other access point.

We will maintain an erratum on the GitHub repository to host any approved corrections or updates suggested by the authors or the broader video research community.

**Will the dataset be updated (e.g., to correct labeling errors, add new instances, delete instances)?** If so, please describe how often, by whom, and how updates will be communicated to users (e.g., mailing list, GitHub)?

Yes, we plan to host an erratum publicly on our GitHub. We also plan to expand the scale of the annotations, and any updates or extensions will be announced on our website and GitHub repository.

**If the dataset relates to people, are there applicable limits on the retention of the data associated with the instances (e.g., were individuals in question told that their data would be retained for a fixed period of time and then deleted)?** If so, please describe these limits and explain how they will be enforced.

No.

**Will older versions of the dataset continue to be supported/hosted/maintained?** If so, please describe how. If not, please describe how its obsolescence will be communicated to users.

N/A. There are no older versions at the current moment. All updates regarding the current version will be communicated via our website and Github.

**If others want to extend/augment/build on/contribute to the dataset, is there a mechanism for them to do so?** If so, please provide a description. Will these contributions be validated/verified? If so, please describe how. If not, why not? Is there a process for communicating/distributing these contributions to other users? If so, please provide a description.

Contributions to the dataset and evaluation tool will be made possible using standard open-source practices. Interested parties can submit pull requests to the relevant GitHub repository, which will be reviewed and incorporated by the authors.

**Any other comments?**

No.

# S    Impact, Limitations and Future Directions

## S.1    Broader Impact

The proposed EMBODIED AGENT INTERFACE and the comprehensive evaluation of Large Language Models (LLMs) for embodied decision-making have significant implications for the development and deployment of intelligent agents in various domains, including robotics, autonomous systems, and human-robot interaction.

Recent advancements in LLMs have revolutionized the field of artificial intelligence, particularly in the domains of digital and embodied agents. LLMs have demonstrated remarkable capabilities in natural language understanding, generation, and reasoning, enabling the development of sophisticated AI systems that can interact with humans and the environment in more natural and intuitive ways. The potential of LLMs extends beyond purely digital interactions, as embodied agents that can perceive, reason, and act within physical environments have also begun to benefit from the integration of LLMs. By combining the language understanding and generation capabilities of LLMs with the perceptual and motor skills of embodied agents, researchers aim to create more versatile and intelligent robots that can assist humans in various tasks.

However, despite the progress made in applying LLMs to embodied agents, significant challenges remain. Embodied agents require the ability to ground natural language instructions in the physical world, reason about object properties and relationships, and plan and execute actions in dynamic environments. While LLMs have shown promise in these areas, they exhibit inconsistencies in their interactions with the physical world, occasionally making correct decisions but frequently producing incorrect plans without reliable ways to control their performance. This inconsistency leads to doubts regarding their suitability and readiness for robotic tasks. Current evaluations of LLMs in embodied decision-making tasks often entangle various capabilities, making it difficult to understand why LLMs sometimes work and sometimes fail. Existing benchmarks oversimplify the problem by focusing only on high-level semantic failures, assuming that low-level physics can be automatically fulfilled. This assumption results in unrealistic plans, such as a robot heating food in a microwave without first ensuring the door is closed or the food is properly placed inside, which ignores crucial physical preconditions and state changes.

The standardization of goal specifications, modules, and interfaces through the EMBODIED AGENT INTERFACE framework enables a more systematic and rigorous evaluation of LLMs' capabilities in embodied decision-making tasks. This standardization facilitates the comparison of different approaches and architectures, promoting the development of more advanced and reliable LLM-based embodied agents. By identifying the strengths and weaknesses of current LLMs through fine-grained metrics and error analysis, researchers and practitioners can focus their efforts on addressing specific limitations and improving the overall performance of these agents.

Moreover, the insights gained from the evaluation of LLMs in embodied decision-making tasks can inform the design of future LLMs and training strategies. The identified challenges, such as the difficulty in grounding natural language instructions to environment-specific objects and states, the lack of reasoning abilities in handling preconditions and post-effects of actions, and the reporting bias in goal interpretation, highlight the need for more targeted training approaches. These findings can guide the development of LLMs that are better suited for embodied decision-making, potentially leading to more effective and efficient agents in real-world applications. The fine-grained evaluation results and the breakdown of LLM abilities into detailed tests can pinpoint the strengths and weaknesses in LLM interactions with the physical world, offering a more robotics-centered evaluation to uncover how much LLMs understand about the physical world and what knowledge can be improved or added for robotics.

The EMBODIED AGENT INTERFACE framework and the fine-grained evaluation results can also contribute to the development of safer and more trustworthy embodied agents. By providing a comprehensive understanding of LLMs' limitations and failure modes in embodied decision-making, this work can help in designing safeguards and fallback mechanisms to mitigate potential risks associated with the deployment of these agents in real-world scenarios. The fine-grained error analysis can inform the development of monitoring and intervention strategies to ensure the safe and reliable operation of LLM-based embodied agents.

Furthermore, the EMBODIED AGENT INTERFACE framework and the evaluation methodology can be extended to other domains beyond household tasks, such as industrial automation, healthcare robotics, and autonomous vehicles. The standardization of goal specifications, modules, and interfaces can facilitate the application of LLMs in these domains, enabling the development of more intelligent and adaptive agents that can understand and execute complex tasks based on natural language instructions.

However, it is essential to consider the potential negative impacts and ethical implications of deploying LLM-based embodied agents. The evaluation results highlight the need for careful consideration of the limitations and biases of these agents, particularly in safety-critical applications. It is crucial to establish guidelines and best practices for the responsible development and deployment of LLM-based embodied agents, ensuring transparency, accountability, and alignment with human values.

In conclusion, the EMBODIED AGENT INTERFACE framework and the comprehensive evaluation of LLMs for embodied decision-making have the potential to advance the field of intelligent agents and enable the development of more capable, reliable, and trustworthy embodied agents. By providing a standardized approach for evaluation and identifying key challenges and opportunities, this work can guide future research and development efforts in this area, ultimately leading to the realization of intelligent agents that can effectively understand and execute complex tasks in real-world environments. As LLMs continue to evolve and be integrated into embodied agents, it is essential to address the challenges and ethical considerations to ensure their responsible and beneficial deployment in various domains.

## S.2 Limitations

While we currently evaluate the capabilities and limitations of LLMs in embodied decision-making tasks, our approach has limitations as we abstract input environments using relational graphs of objects, which may not adequately represent the rich multimodal information, low-level physical properties, and dynamics of real-world environments. It does not cover the challenge of grounding natural language instructions in perceptual data and executing actions that require fine-grained control and precise manipulation. Moreover, our current evaluation primarily focuses on symbolic reasoning and decision-making, without directly incorporating sensory inputs or actuation outputs. While this approach allows us to isolate and assess specific cognitive capabilities, it neglects the crucial integration of perception and action, which is fundamental to embodied agents operating in physical environments. Realistic embodied decision-making requires seamless interplay between language understanding, sensory processing, and motor control, which our current framework does not fully capture.

## S.3 Potential Negative Social Impact

The development of embodied agents powered by large language models has significant potential social impacts that must be carefully considered. On one hand, these intelligent agents could revolutionize numerous domains and provide immense societal benefits. In healthcare, they could assist in elderly care, rehabilitation, and medical procedures. In education, they could serve as personalized companions and learning companions. In manufacturing and logistics, they could streamline operations, improve efficiency, and enhance workplace safety.

However, the deployment of such capable agents also raises concerns over privacy, security, and ethical implications. There are risks of these systems being exploited for malicious purposes or exhibiting harmful biases learned from training data. Additionally, their widespread adoption could disrupt certain job markets and exacerbate economic inequalities if not managed responsibly. As we strive to unlock the full capabilities of large language models for embodied decision-making, it is crucial to prioritize the development of robust safety measures, ethical frameworks, and regulatory guidelines. Engaging diverse stakeholders, including policymakers, domain experts, and the general public, will be vital in navigating the intricate societal impacts and ensuring these powerful technologies are harnessed for the greater benefit of humanity.

## S.4 Potential Future Directions

We outline the future directions to represent a roadmap for advancing embodied decision making with large foundation models.

- **Multimodal Grounding and Integration:** Extend the evaluation to incorporate multimodal inputs such as vision, audio, by integrating with Vision-Language Models (VLMs) and other multimodal models. This includes assessing LLMs' ability to ground language in rich sensory contexts, generate low-level control commands, and reason about visual information like object keypoints, grasp poses, and scene configurations. Additionally, explore the use of symbolic scene graph representations as input to LLMs, enabling more effective reasoning about object relationships, spatial configurations, and visual attributes.

- **Vision and Low-Level Control:** Evaluate LLMs in scenarios that require tight integration with vision systems and low-level control modules, such as tabletop object manipulation tasks. This could involve predicting object keypoints, generating pick-and-place grasp poses, or generating goal configurations for object arrangements.

- **Episodic Memory and Scene Graph Memory:** Investigate the integration of memory systems into LLMs, including episodic memory for storing and retrieving past experiences, state memory for maintaining internal environment representations, and scene graph memory structures. Evaluating memory-augmented LLMs can provide insights into leveraging past knowledge and rich environment representations for improved decision-making.

- **Physical and Geometric Reasoning:** Incorporate tasks that require physical and geometric reasoning capabilities, such as spatial planning, object manipulation, collision avoidance, stability prediction, and reasoning about object affordances, sizes, shapes, and attachments. Evaluate LLMs' potential for precise physical interactions and reasoning in real-world applications.

- **Navigation and Exploration:** Include navigation tasks and scenarios that require LLMs to understand navigational instructions, plan efficient routes, adapt to dynamic environments, and explore unknown object states, physical properties, and environmental conditions crucial for robust decision-making.

- **Forward, Backward, and Counterfactual Prediction:** Assess LLMs' ability to perform forward prediction (action $\rightarrow$ state change), backward prediction (goal state $\rightarrow$ necessary objects and subgoals), and counterfactual reasoning (hypothetical "what-if" scenarios) to evaluate their capability in multi-step planning and reasoning about the effects of actions or action sequences.

- **Dataset, Simulation, and Benchmark Diversity:** Develop new datasets, simulation environments, and integrate with existing robotics benchmarks to capture the complexity and diversity of real-world scenarios. This includes creating environments with richer physics simulations, more diverse object interactions, and more challenging task setups, as well as integrating with benchmarks focused on task and motion planning, manipulation, and navigation.

- **Fine-Tuning Decision Making Models:** Explore fine-tuning LLMs using the fine-grained error analysis and feedback from the evaluation framework, enabling them to learn from their mistakes and improve decision-making capabilities. Additionally, develop specialized models such as a decision making GPT, trained on trajectories of embodied decision-making tasks, to generalize to unseen trajectories and environments.

By addressing these directions, researchers and practitioners can overcome the current limitations, bridge the gap between language models and embodied agents, and unlock the full potential of LLMs in real-world, dynamic environments. Ultimately, these efforts will pave the way for more capable, adaptable, and trustworthy embodied agents that can seamlessly understand natural language instructions, reason about physical properties and constraints, and execute complex tasks with precision and robustness.

