# OpenReview forum: "Embodied Agent Interface: Benchmarking LLMs for Embodied Decision Making"
_NeurIPS.cc/2024/Datasets_and_Benchmarks_Track — NeurIPS 2024 Track Datasets and Benchmarks Oral_

### Official Review · Reviewer_hSVb · 2024-07-16
**A well-contributed, reproducible, and worthy-of-acceptance work.**

**Rating:** 8
**Confidence:** 4
**Correctness:** Reasonable.
**Clarity:** Yes.

**Review:**

The paper is well-structured, and the proposed interface is a significant step towards standardizing LLM evaluation in embodied decision-making. I recommend for acceptance.

**Strengths:**

1. Conducting a detailed evaluation of LLMs' capabilities in embodied control tasks is highly valuable, currently lacking, and greatly contributes to the community.
2. The analysis of LLM modules for decision making, error types, etc., is thorough, reasonable, and valuable. The evaluation of common LLMs is also comprehensive.
3. High reproducibility. As a submission for the benchmark track, usability and reproducibility are crucial for researchers. I noticed that the authors provided detailed information (implementation details, prompts, examples, etc.).

**Additional Feedback:**

N/A

**Documentation:**

Comprehensive, mainly in the appendix file.

**Limitations:**

The authors mentioned limitations and discussed them. Verification of VLMs is necessary but might not be urgent.

**Opportunities For Improvement:**

1. Lack of discussion and comparison about VLMs, as mentioned in the limitations section by the authors. However, this may not be a drawback and could be considered future work.
2. I understand the limitations on the length of the main text, but some key conclusions, such as those based on LLM evaluations, and discussions on current challenges and research-worthy issues of LLMs in embodied control applications should be placed in prominent sections of the main text.
3. Some minor details:
a. In lines 69 and 75, the terms "trajectory feasibility error" and "trajectory evaluation performance" are mentioned for the first time without any prior explanation. This may confuse readers and reduce readability.
b. The example of hallucination error in Figure 3 is not appropriate. The concept of hallucination is very broad, and many examples mentioned in Figure 3 can be attributed to LLM hallucinations. Thus, the authors might consider using more precise terms to describe these errors.

**Relation To Prior Work:**

Clear.

**Summary And Contributions:**

The paper provides a comprehensive evaluation of Large Language Models (LLMs) in the context of embodied decision-making. The authors propose a standardized interface, EMBODIED AGENT INTERFACE, to formalize tasks and input-output specifications for LLM-based modules. The interface unifies various decision-making tasks, incorporates four critical LLM-based modules, and introduces fine-grained evaluation metrics to better understand the strengths and weaknesses of LLMs in embodied AI systems.

---

> ### Author Rebuttal · Authors · 2024-08-17
>
> We deeply appreciate Reviewer hSVb for the insightful review. We are delighted that you recognize our work as **well-structured** and the significance of our work, as well as its **high reproducibility** and **comprehensive evaluation of LLM's capabilities** in embodied control tasks.
>
> We carefully considered your valuable feedback, which we address in detail below:
>
> ---
> **[Q1] Lack of discussion and comparison about VLMs, as mentioned in the limitations section by the authors. However, this may not be a drawback and could be considered future work.**
>
> We are happy that the reviewer brings up this concern, which we have in mind during our evaluation design and carefully examined with experiments. We appreciate the reviewer acknowledges this could be considered future work, and we also pinpoint this future work focusing on VLMs in Section 6 (Lines 308-L313).
>
> In this paper, we deliberately scope our **long-horizon decision-making evaluation** in the LLM setting, as it is currently **most applicable**. Existing VLMs are not designed for taking scene-level information from multiple rooms in a household environment. Recent works on benchmarking VLMs for embodied scenarios [1,2,3] **focus on perception** rather than long-horizon decision-making. Therefore, we focus on **long-horizon decision-making capability that bridges perception and control**, a common use of LLMs in embodied decision-making [4,5,6,7,8,9]. While many studies focus on short-horizon control-level tasks, we fill in the blanks here.
>
> - **Why we do not choose VLMs:** We experiment with VLMs' long-horizon decision-making.
> Comparing Llama-3 and LLaVA (with Llama-3 as the backbone for a fair comparison), VLMs consistently underperform LLMs. The gap reveals a crucial issue: **VLMs struggle with low performance, especially in complex scenarios with multiple objects** like BEHAVIOR and VirtualHome.
>     * **(Exp.0) `scene_graph` → LLMs → `planning_output`** is the original setting in our paper.
>     * **(Exp.1) `image` → VLMs → `planning_output`** is end-to-end, without intermediate scene graphs. VLMs perform significantly worse, **entangling perception and decision-making errors**, thus making it difficult to locate decision-making errors. Evaluating VLMs this way shows reduced overall performance but does not identify specific failures in abilities like goal interpretation or transition modeling.
>     * **(Exp.2) `image` + `scene_graph` →VLMs → `planning_output`**, where the ground truth scene graph is provided and the VLM is augmented with visual signals compared to the LLM, yet still underperforms. It suggests that **VLMs are not yet able to sufficiently use multimodal input to improve long-horizon decision-making**. VLMs' long-horizon reasoning abilities primarily stem from their language modeling components[11,12], but visual information often acts as a distractor rather than a helper due to its currently limited quality.
>     *  **VLMs (Exp.1, Exp.2) vs LLMs (Exp.0) on BEHAVIOR**
>         | | **Goal Interpretation ($F_1$)** | **Action Sequencing (Success Rate, SR %)** |
>         | :----- | :--------: | :-----: |
>         | **Llama** | 31.5 | 11.1 |
>         | **LLaVA** (w/o scene graphs) | 9.10 | 2.50 |
>         | **LLaVA** (w/ scene graphs) | 25.8 | 11.0 |
>     * As a result, we adopt a **decomposed evaluation approach** to (1) **accurately locate** current models' strengths and weaknesses to (2) **guide LLM/VLM improvements for embodied AI**. We break down abilities into key modules, providing specific feedback on each. This idea of decomposed evaluation and strategic LLM usage is **supported by many influential robotics research** [4,5,6,7,8,9,10].
> * **Why LLM eval is useful - insights from LLM eval can be applied to VLMs**:
> Our benchmark findings can guide future integrations with complex, real-world robotic systems. For example, our experiments reveal a pattern about adding broader context:
>     * If **(Exp.3) `scene graph` + `broader context` → LLMs → `improved planning output`**
>     * Then **(Exp.4) `image` + `scene graph` + `broader context` → VLMs → `improved planning output`**
>     * It suggests that methodological findings from our LLM eval can also be used to enhance VLM reasoning, potentially bridging the gap between abstract planning and visually grounded tasks in robotics.
>     *  **Insights from LLMs (Exp.3) can be applied to VLMs (Exp.4)**
>         | | **Goal Interpretation ($F_1$) Improvement** | **Action Sequencing (SR %) Improvement** |
>         |:-------------------------------------- | :-----: | :-----: |
>         | **Llama** | 31.5 → 31.9 | 11.1 → 13.9 |
>         | **LLaVA** (w/ scene graphs) | 25.8 → 25.9 | 11.0 → 12.3 |
> * **Validated by existing research**: Current approaches like CodeAsPolicies[4], Voyager[7], VIMA[10], and VoxPoser[9] typically use LLMs for planning, so it is our major focus. Such **standardized evaluation** of LLMs can (a) guide **LLM researchers** to improve to better support embodied AI; (b) provide **robotics researchers** with selection and modularization of different LLMs and their roles.
>
> [1] MMRo: Are MLLMs Eligible as the Brain for In-Home Robotics? arXiv24\
> [2] MFE-ETP: A Comprehensive Benchmark for MLLMs on Embodied Task Planning arXiv24\
> [3] OpenEQA: Embodied Question Answering in the Era of Foundation Models CVPR24\
> [4] Do As I Can, Not As I Say: Grounding Language in Robotic Affordances CoRL22\
> [5] Code as Policies: LM Programs for Embodied Control ICRA23\
> [6] Learning Adaptive Planning Representations with Natural Language Guidance ICLR24\
> [7] Voyager: An Open-Ended Embodied Agent with LLMs TMLR24\
> [8] SayPlan: Grounding LLMs using 3D Scene Graphs for Scalable Robot Task Planning CoRL23\
> [9] Voxposer: Composable 3d value maps for robotic manipulation with LLMs CoRL23\
> [10] VIMA: General Robot Manipulation with Multimodal Prompts ICML 23\
> [11] SimVLM: Simple VL Model Pretraining with Weak Supervision ICLR 22\
> [12] Unifying Vision-and-Language Tasks via Text Generation ICML 21

---

> > ### Author Rebuttal · Authors · 2024-08-17
> >
> > ---
> >
> > **[Q2] main text vs appendix: Necessary content missing in main part**
> >
> > Thank you for pointing out this. In the [revision](https://embodied-agent-eval.github.io/website/data/NeurIPS2024__Embodied_Agent_Interface_Evaluating_LLMs_for_Physical_Everyday_Decision_Making.pdf), we have moved Part of Appendix E to the main text. Will also summarize the key conclusions in each takeaway section and move them to the main text.
> >
> > ---
> > **[Q3] Writing fixes: explanation for trajectory feasibility error and trajectory evaluation performance**
> >
> > We appreciate the reviewer's observation regarding the terms "trajectory feasibility error" and "trajectory evaluation performance." We acknowledge that these terms were introduced without prior explanation, potentially causing confusion. We will address this in our revised version as follows:
> >
> >
> > * The explanation of Trajectory Feasibility Error, currently found in lines 146-150, will be moved earlier in the paper to provide better context. This section will detail our metric collection process and the types of errors included.
> > * For Trajectory Evaluation Performance, we will introduce a clear definition earlier in the paper, explaining that this term encompasses both "Goal SR" and "Execution SR". We will to elaborate on these components in the revised version, ensuring a more comprehensive understanding of our evaluation metrics. This restructuring aims to improve clarity by introducing these crucial concepts earlier and providing more detailed explanations of our evaluation methodology.
> >
> >
> > ---
> > **[Q3] Writing fixes: the concept of hallucination**
> >
> > Thank you for pointing out this. We will rename “hallucinations” to “object hallucinations” following [1] to indicate objects that cannot be grounded to the environment.
> >
> > [1] Rohrbach Anna, et al. "Object Hallucination in Image Captioning" EMNLP 2018

---

> > > ### Comment · Reviewer_hSVb · 2024-08-22
> > >
> > > Thank you for the detailed response. The authors have addressed most of my concerns in their response and revised version, and I believe that the presentation clarity has improved in the revised version. My score remains unchanged.

---

> > ### Author Response · Authors · 2024-08-24
> > **Thank you Reviewer hSVb for thorough review of our rebuttal**
> >
> > We appreciate your thorough review of our rebuttal. We are glad that our revised version has addressed most of your concerns. We appreciate your continued support for the rating of 8.
> >
> > We are happy to address any further concerns you may have, and sincerely appreciate your time and effort.

---

### Official Review · Reviewer_yhYD · 2024-07-22
**A Benchmark of LLMs for Embodied Agents**

**Rating:** 8
**Confidence:** 4

**Review:**

There is no doubt that this work demonstrates a high level of thoroughness and quality throughout, with few apparent weak points. In the supplementary material, the authors provide comprehensive specifications of the linear temporal logic formulas. They also provide detailed in-context prompts used for evaluating LLMs. They also provide in-depth analysis on their experimental findings. In particular, I believe the application of logical language to represent the goals, states and actions is novel and significant.

**Strengths:**

- This work employs unified linear temporal logic (LTL) language to describe the goals, subgoals, object states, and action sequences. I believe such LTL formulas can be easily extended to different embodied simulators.
- The benchmark is comprehensive and inclusive, with 15 different LLMs, four embodied tasks and two simulators.
- The findings have provide adequate insights on the shortcomings of applying LLMs for embodied agents.

**Additional Feedback:**

1. Maybe I missed something, how do you obtain the dataset annotations? Apart from the annotation details described in Appendix P, did you recruit crowdworkers to label the real demonstrations? How do you verify the quality of annotations?
2. Why do you ask the LLMs to generate the logical language in JSON format? How do you parse or post-process the outputs if they are not *strictly formatted* as JSON?

**Clarity:**

The paper is well written and clearly organized, although many technical details are put in the Appendix.

**Correctness:**

The claims made in the submission is correct. The evaluation methods and experiment design are appropriate and performed correctly.

**Documentation:**

There is sufficient detail on data collection and organization, availability and maintenance, and ethical and responsible use. The authors also provide the source code to support reproducibility.

**Ethics:**

No, there are no or only very minor ethics concerns.

**Limitations:**

The authors have provided a dedicated section to address the limitations and potential negative societal impact.

**Opportunities For Improvement:**

In Appendix G.3, there are duplicate statements in the BFS searching algorithm.
> For each subgoal $\phi_i$ in the subgoal sequence $\bar{\phi}$

**Relation To Prior Work:**

This work clearly discussed how it differs from previous contributions, with a couple of tables for comparison.

**Summary And Contributions:**

This work presents Embodied Agent Interface to evaluate large language models for embodied agents. The main contributions of Embodied Agent Interface include: (1) it employs linear temporal logic (LTL) language to describe the goals, subgoals, object states, and action sequences; (2) the authors define four distinct evaluatioin tasks to assess LLMs abilities, each involves varying levels of external module avaliability: goal interpretation, subgoal decomposition, action sequencing, and transition modeling; (3) it implements fine-grained evaluation metrics through both embodied simulation and logical verification.

The authors evaluate 15 LLMs on two simulators. The key findings outline potential directions that LLMs can be improved for embodied decision making, including but not limited to reasoning capability, long-horizontal goals, and reporting bias.

---

> ### Author Rebuttal · Authors · 2024-08-17
>
> We sincerely thank Reviewer yhYD for the thoughtful and encouraging feedback. We are pleased that you found our work to **be of high quality**, with thoroughness demonstrated across the method and analysis. We appreciate your recognition of our **novel application of LTL in describing (goals, states, actions)**, **the potential of our benchmark for the community**, and **fine-grained evaluation metrics and logical verification**.
>
> Please find your suggestions and additional feedback carefully addressed below:
>
> ---
>
> **[Q1] Duplicate statements in Appendix G.3**
>
> In the [revision](https://embodied-agent-eval.github.io/website/data/NeurIPS2024__Embodied_Agent_Interface_Evaluating_LLMs_for_Physical_Everyday_Decision_Making.pdf), we have carefully proofread and remove the duplicate statements.
>
> ---
>
> **[F1] Maybe I missed something, how do you obtain the dataset annotations? Apart from the annotation details described in Appendix P, did you recruit crowdworkers to label the real demonstrations? How do you verify the quality of annotations?**
>
> We appreciate the reviewer's question. Following your suggestions, we have added more annotation details in Appendix P Annotation Details of the [revision](https://embodied-agent-eval.github.io/website/data/NeurIPS2024__Embodied_Agent_Interface_Evaluating_LLMs_for_Physical_Everyday_Decision_Making.pdf).
> * Annotation Process:
>     * Four authors worked on the data annotation: two for VirtualHome and two for BEHAVIOR. We used real demonstrations from the BEHAVIOR-100 dataset, then we annotated trajectories, goals, action preconditions, and post-effects using our expert knowledge. We did not use crowdworkers as the annotators are expected to have **expert knowledge of LTL, transition models**, and familiar with **simulator-supported actions**. Annotators also worked on the simulator implementation (e.g., implementing actions and transition models in BEHAVIOR, detailed in Appendix Q.1) to **ensure the annotations were executable**.
>     * To control the data quality, we have a **two-round annotation** process: a) initial annotation and b) revision round based on initial results.
>     * We also have additional rounds if needed until consensus reached on each data point.
> * Quality Verification:
>     * We have a automated verification using simulators to check consistency and correctness
>     * Based on your suggestions, we have conducted an additional human evaluation for the data annotation quality. We follow VirtualHome to design a survey to ask annotators to score 1-5 for Annotation Accuracy, Annotation Coverage, and Human Preference. We recruit 20 annotators and ask each of them to verify annotations of 10 tasks.
>     * Annotation Quality Evaluation
>       |         | Mean Score | Weighted MSE |
>       | :---------------- | :-------: |  :---------------: |
>       | **Annotation Accuracy**    | 3.73   | 0.4062        |
>       | **Annotation Coverage**    | 4.07   | 1.8438        |
>       | **Human Preference**   | 4.27  | 0.8125       |
>    * Here, weighted Mean Squared Error (MSE) calculation assesses score variability across tasks and evaluators. For multi-evaluator tasks, we calculate the MSE per attribute by averaging squared differences between individual scores and the task's mean. We then compute a weighted mean of these MSEs across all tasks, with weights based on evaluation frequency. This method balances individual task variability with overall assessment consistency, accounting for varying numbers of evaluations per task.
>    * We also visualize the score distribution, attached in the [PDF](https://openreview.net/attachment?id=02nzMxgA8b&name=pdf).
>
> ---
>
> **[F2] Why do you ask the LLMs to generate the logical language in JSON format? How do you parse or post-process the outputs if they are not strictly formatted as JSON?**
>
> We are happy that the reviewer brings up this question, which we carefully examined with experiments. We opted for JSON output from LLMs in our evaluation for several reasons:
>
> * We report the format errors in Table 6, where **format parsing error rate is below 1%** for top models.
> * We aim to **minimize** the involvement of **instruction following** and **disentangle** the **decision making** and instruction following.
>     * Format errors reflect instruction-following ability rather than decision-making capability.
>     * Environment representation is a **structured representation**, JSON format has been well trained for structured representation.
> * We aim to work with **different simulators**, which have varying **grammars** and **formats**. Direct prompting for specific formats often results in punctuation errors.
>     * Example: VirtualHome's `[action_name] <object_name>` format.
>     * Observed erroneous variations: `[FIND] phone`, `FIND \<phone\>`, `FIND phone`.
>     * JSON output (e.g., `{"FIND": ["phone"]}`) eliminates these inconsistencies.
> * Ease of Post-processing:
>     * JSON format allows for simpler post-processing.
> * We performed experiments when making this design choice:
>     * We observed that LLM can handle json format very well. Taking BEHAVIOR subgoal decomposition as an example,  **1495 out of 1500 can be successfully parsed**.
>     * We found that plan output with json format has a **higher executable rate** than that with natural language format.
>     | Action Sequencing       | Executable Rate |
> | ----------------------- | --------------- |
> | Natural Language Output | 78.1            |
> | JSON Output             | 84.3            |
>
> We have these details in the [revision](https://embodied-agent-eval.github.io/website/data/NeurIPS2024__Embodied_Agent_Interface_Evaluating_LLMs_for_Physical_Everyday_Decision_Making.pdf).

---

> > ### Author Response · Authors · 2024-08-31
> > **Thank you Reviewer yhYD for the encouraging and thoughtful feedback**
> >
> > Dear Reviewer yhYD,
> >
> > Thank you again for your encouraging and constructive feedback.
> >
> > We have addressed all your concerns and conducted additional sets of experiments with more details in the revision. We are happy to address any further concerns you may have. Again, we sincerely appreciate your time and effort.
> >
> > Thank you,
> > Authors

---

### Official Review · Reviewer_tF6x · 2024-07-23
**A timely benchmark for LLMs in Robot Learning, but no vision is involved**

**Rating:** 7
**Confidence:** 4
**Correctness:** I did not notice any major mistakes.
**Clarity:** Yes, it is clear.

**Review:**

See "Strengths" and "Opportunities For Improvement".

**Strengths:**

This paper has several strengths:

- Utilizing LLMs in robot learning is a popular trend, yet there are no robust evaluation benchmarks available. The topic is significant, and the paper's timing is opportune.

- By breaking down embodied decision-making into four fundamental ability modules (Goal Interpretation, Subgoal Decomposition, Action Sequencing, and Transition Modeling), the paper allows for targeted evaluation and improvement of specific aspects of LLM performance.

- The framework provides fine-grained metrics and automatic error identification tools, going beyond simple success rates. This allows for a more nuanced understanding of LLM capabilities and limitations in embodied tasks.

- The study evaluates 15 different LLMs, including both open-source and proprietary models, providing a comprehensive overview of current capabilities.

- The paper provides in-depth analysis of different types of errors made by LLMs, offering valuable insights for future improvements in embodied AI systems.

**Additional Feedback:**

N/A

**Documentation:**

The links on their project website appeared to be incorrect at the time of review.

**Limitations:**

The authors have discussed some limitations in the paper.

**Opportunities For Improvement:**

- This evaluation framework presupposes that all elements (inputs, outputs, goals, states, etc.) can be described using text, which is not always true in many practical scenarios. Real-world embodied tasks often involve sensory inputs and physical actions that text alone cannot fully capture. In the context of robot learning, a more typical scenario involves tasks described using multi-modal elements (e.g., text descriptions, goal images, etc.), where the robot must formulate a plan based on a history of observations (typically images). Consequently, I anticipate that multi-modal LLMs (e.g., vision-language models) will be preferred in future robot learning research. This perspective leads me to question the utility of evaluating purely text-based LLMs.

- Additionally, by representing all elements in text, this paper overlooks various potential sources of error. For instance:
	- Errors stemming from perception and state estimation are not considered.
	- The paper assumes all actions are abstract, yet in the real world, these abstract actions (or skills) can fail, necessitating re-planning to mitigate the consequences of such failures.

- This benchmark necessitates that the environment can provide relational graphs of objects, a requirement not commonly met by many existing robot learning environments and challenging to annotate. Moreover, extracting these relational graphs in real-world settings is significantly more complex, presenting additional hurdles for practical implementation.

- While the use of Linear Temporal Logic (LTL) for goal specification simplifies verification processes, it may oversimplify certain aspects of real-world goals that are more challenging to formalize in logical terms. It's essential to consider whether there are concepts that LTL can describe which natural language cannot. If everything is described using formal languages like LTL and PDDL, one might question the necessity of employing LLMs instead of directly utilizing Task and Motion Planning (TAMP) systems. This raises fundamental questions about the specific advantages of LLMs in scenarios where structured formal planning could be more directly applicable.

- This benchmark employs a set of manually designed prompts to evaluate the LLMs. However, these prompts can be model-specific, potentially skewing the results. For instance, GPT might outperform Gemini with one prompt (Prompt A), while Gemini could excel with another (Prompt B). It's crucial to question whether the authors have considered the potential effects of such variability. Addressing this issue would help ensure that the evaluation is fair and representative of each model's capabilities across a range of scenarios, not just those tailored to specific prompts.

- The study focuses on only two simulators (BEHAVIOR and VirtualHome), which may not fully represent the diversity of real-world embodied tasks. This could limit the generalizability of the findings.

- The paper doesn't provide a comparison between LLM performance and human performance on these tasks, which could provide valuable context for interpreting the results.

**Relation To Prior Work:**

Yes

**Summary And Contributions:**

This paper introduces EMBODIED AGENT INTERFACE, a systematic evaluation framework for **benchmarking Large Language Models (LLMs) in embodied decision-making tasks**.

Basically, they formalize four critical ability modules in LLM-based embodied decision making: Goal Interpretation, Subgoal Decomposition, Action Sequencing, Transition Modeling. They also formalize the input-output specifications that LLMs can use to interface with other modules in the environment. The authors implement their framework on two embodied decision-making benchmarks: BEHAVIOR and VirtualHome, and evaluate 15 different LLMs. The paper also provides insights on the potential integration of LLM-based ability modules and their robustness through sensitivity analysis, modularized vs. pipeline-based experiments, and replanning.


The key contributions of this work include:

- Standardization of goal specifications using Linear Temporal Logic (LTL) formulas, allowing for both state-based and temporally extended goals.

- Unification of decision-making tasks through a standard interface and four fundamental ability modules: Goal Interpretation, Subgoal Decomposition, Action Sequencing, and Transition Modeling.

- Comprehensive fine-grained evaluation metrics and automatic error identification tools.

---

> ### Author Rebuttal · Authors · 2024-08-17
>
> We sincerely thank Reviewer tF6x for the comprehensive feedback. We appreciate your recognition of our work's **significance and timeliness in using LLMs for embodied decision-making tasks**, and positive remarks of **decision-making ablities** and **the fine-grained evaluation metrics**. We truly value your insights and have carefully addressed them:
>
> ---
> **[Q1] All inputs are text; the utility of evaluating purely text-based LLMs**
>
> We are happy that the reviewer brings up this concern, which we have in mind during our evaluation design and carefully examined with experiments. Also, as noted in Section 6 (Lines 308-L313), we acknowledge this limitation and pinpoint future work focusing on VLMs.
>
> We deliberately scope our **long-horizon decision-making evaluation** in the LLM setting as it is currently **most applicable**. Existing VLMs are not designed for taking scene-level information from multiple rooms in a household environment. Recent works on benchmarking VLMs for embodied scenarios [1,2,3] **focus on perception** rather than long-horizon decision-making. Therefore, we focus on **long-horizon decision-making capability that bridges perception and control**, a common use of LLMs in embodied decision-making [4,5,6,7,8,9]. While many studies focus on short-horizon control-level tasks, we fill in the blanks here.
>
> To explore VLMs' long-horizon decision-making, we experiment with VLMs with multimodal input:
> - **Why we do not choose VLMs:**
> Comparing Llama-3 and LLaVA (with Llama-3 as the backbone for a fair comparison), VLMs consistently underperform LLMs. The gap reveals a crucial issue: **VLMs struggle with low performance, especially in complex scenarios with multiple objects** like BEHAVIOR and VirtualHome.
>     * **(Exp.0) `scene_graph` → LLMs → `planning_output`** is the original setting in our paper.
>     * **(Exp.1) `image` → VLMs → `planning_output`** is end-to-end, without intermediate scene graphs. VLMs perform significantly worse, **entangling perception and decision-making errors**, thus making it difficult to locate decision-making errors. Evaluating VLMs this way shows reduced overall performance but does not identify specific failures in abilities like goal interpretation or transition modeling.
>     * **(Exp.2) `image` + `scene_graph` →VLMs → `planning_output`**, where the ground truth scene graph is provided and the VLM is augmented with visual signals compared to the LLM, yet still underperforms. It suggests that **VLMs are not yet able to sufficiently use multimodal input to improve long-horizon decision-making**. VLMs' long-horizon reasoning abilities primarily stem from their language modeling components[11,12], but visual information often acts as a distractor rather than a helper due to its currently limited quality.
>     *  **VLMs (Exp.1, Exp.2) vs LLMs (Exp.0) on BEHAVIOR**
>         | | **Goal Interpretation ($F_1$)** | **Action Sequencing (Success Rate, SR %)** |
>         | :----- | :--------: | :-----: |
>         | **Llama** | 31.5 | 11.1 |
>         | **LLaVA** (w/o scene graphs) | 9.10 | 2.50 |
>         | **LLaVA** (w/ scene graphs) | 25.8 | 11.0 |
>     * As a result, we adopt a **decomposed evaluation approach** to (1) **accurately locate** current models' strengths and weaknesses to (2) **guide LLM/VLM improvements for embodied AI**. We break down abilities into key modules, providing specific feedback on each. This idea of decomposed evaluation and strategic LLM usage is supported by many influential robotics research [4,5,6,7,8,9,10].
> * **Why LLM eval is useful - insights from LLM eval can be applied to VLMs**:
> Our benchmark findings can guide future integrations with complex, real-world robotic systems. For example, our experiments reveal a transferable pattern about adding broader context:
>     * If **(Exp.3) `scene_graph` + `broader_context` → LLMs → `improved_planning_output`**
>     * Then **(Exp.4) `image` + `scene_graph` + `broader_context` → VLMs → `improved_planning_output`**
>     * It suggests that methodological findings from our LLM eval can also be used to enhance VLM reasoning, potentially bridging the gap between abstract planning and visually grounded tasks in robotics.
>     *  **Insights from LLMs (Exp.3) can be applied to VLMs (Exp.4)**
>         | | **Goal Interpretation ($F_1$) Improvement** | **Action Sequencing (SR %) Improvement** |
>         |:-------------------------------------- | :-----: | :-----: |
>         | **Llama** | 31.5 → 31.9 | 11.1 → 13.9 |
>         | **LLaVA** (w/ scene graphs) | 25.8 → 25.9 | 11.0 → 12.3 |
> * **Validated by existing research**: Current approaches like CodeAsPolicies[4], Voyager[7], VIMA[10], and VoxPoser[9] typically use LLMs for planning, so it is our major focus.
> * **Our Goal**: Instead of matching the practical robot learning framework of doing certain tasks, we aim for **standardized evaluation** of LLMs to (a) guide **LLM researchers** to improve to better support embodied AI; (b) provide **robotics researchers** with selection and modularization of different LLMs and their roles.
>
> [1] MMRo: Are MLLMs Eligible as the Brain for In-Home Robotics? arXiv24\
> [2] MFE-ETP: A Comprehensive Evaluation Benchmark for MLLMs on Embodied Task Planning arXiv24\
> [3] OpenEQA: Embodied Question Answering in the Era of Foundation Models CVPR24\
> [4] Do As I Can, Not As I Say: Grounding Language in Robotic Affordances CoRL22\
> [5] Code as Policies: LM Programs for Embodied Control ICRA23\
> [6] Learning Adaptive Planning Representations with Natural Language Guidance ICLR24\
> [7] Voyager: An Open-Ended Embodied Agent with LLMs TMLR24\
> [8] SayPlan: Grounding LLMs using 3D Scene Graphs for Scalable Robot Task Planning CoRL23\
> [9] Voxposer: Composable 3d value maps for robotic manipulation with LLMs CoRL23\
> [10] VIMA: General Robot Manipulation with Multimodal Prompts ICML 23\
> [11] SimVLM: Simple Visual Language Model Pretraining with Weak Supervision" ICLR 22\
> [12] Unifying Vision-and-Language Tasks via Text Generation ICML 21

---

> > ### Author Rebuttal · Authors · 2024-08-17
> >
> > ---
> > **[Q2] Perception and state estimation errors**
> >
> > * Previous Exp.1, Exp.2, Exp.3 and Exp.4 are end-to-end settings. We add additional experiments using **VLM-generated scene graphs** and then input to LLMs. We focus on subgoal decomposition since it is highly sensitive to perception performance.
> >
> > * **(Exp.5) `Image` → VLMs → `scene graphs` → LLMs → `planning output` on BEHAVIOR**
> >
> >     | VLMs | **Scene Graph** |  | **Subgoal Decomposition** ||
> >     | :----- | :----- | :----- | :----- | :----- |
> >     | | **Object Error Rate** | **Predicate Error Rate** | **#step_generate** |**Success Rate**|
> >     | **Claude-3.5-Sonnet** | 0% | **8%** | **13** |**75.0%**|
> >     | **ChatGPT-4o**      | 0% | 20% | 8 |50.0%|
> >     | **LLaVA-Vicuna-13B**  | 0% | 83% | 6 |12.5%|
> >
> > * Results show significant performance decline with increased hallucination. We also visualize the typical perception errors in the attached PDF.
> >
> >
> > ---
> > **[Q2] Assumption on all actions are abstract**
> >
> > * We absolutely agree with the reviewer on the importance of analyzing such failure cases, which is precisely why **we provide replanning settings** in Lines 1382-1410 in the main paper, Appendix K and Table 11. This replanning settings can address two things: errors in predictions, and stochasticity in execution.
> > * Additionally, we add a new experiment to simulate stochastic actions with a certain failure probability, and allow replanning for three times. The experiments are done on GPT-4o for action sequencing task.
> > * Replanning for Stochastic Actions on BEHAVIOR (Exp.6)
> >     |        |             | Execution SR (%) | Goal SR (%) |
> >     | :----- | :----- | :----- | :----- |
> >     | $P_{fail}=0.05$ | **w/o Replanning**  | 60.0          | 50.0     |
> >     |              | **w/ Replanning** | 85.0 (↑25.0)       | 65.0 (↑15.0)    |
> >     | $P_{fail}=0.1$ | **w/o Replanning**  | 25.0         | 15.0    |
> >     |              | **w/ Replanning** | 70.0 (↑45.0)        | 55.0 (↑40.0)    |
> >     | $P_{fail}=0.2$ | **w/o Replanning**  | 10.0         | 5.0    |
> >     |              | **w/ Replanning** | 65.0 (↑55.0)        | 45.0 (↑40.0)    |
> > * We show that (1) replanning can be helpful; (2) **the gap between with and without replanning generally increases when the failure probability is larger**.
> >
> > ---
> >
> > **[Q3] Assumption on scene graphs: environment can provide relational graphs of objects**
> >
> > * Generation of relational graphs from real-world data is an active area with lots of researchers working on, such as Semantic-Geometric Scene Graph [1], Semantic SLAM [2], ConceptGraph [3], BEHAVIOR Vision Suite [4]. As this field progresses, the insights from our benchmark will become increasingly relevant for practical implementations.
> > * The relational graphs are widely used in robotics, which allows us to evaluate decision making capabilities directly, without the added complexity of perception tasks. This approach enables us to isolate and analyze the LLM's planning and reasoning skills specifically.
> > * The scene graphs we used in Behavior and VirtualHome are pretty simple, as listed in Appendix Table 17 and Appendix Table 18.
> >
> > [1] Kurenkov, et al. “Semantic and Geometric Modeling with Neural Message Passing in 3D Scene Graphs for Hierarchical Mechanical Search” ICRA 2021\
> > [2] Rosinol, et al. “Kimera: an Open-Source Library for Real-Time Metric-Semantic Localization and Mapping” ICRA 2020\
> > [3] Gu, et al. “ConceptGraphs: Open-Vocabulary 3D Scene Graphs for Perception and Planning” ICRA 2024\
> > [4] Ge, et al. “BEHAVIOR Vision Suite: Customizable Dataset Generation via Simulation” CVPR 2024\
> > [5] Agia, et al. “TASKOGRAPHY: Evaluating robot task planning over large 3D scene graphs” CoRL 2021\
> > [6] Li, et al. “Pre-Trained Language Models for Interactive Decision-Making” NeurIPS 2022\
> > [7] Lin, et al. “Text2motion: From natural language instructions to feasible plans” Autonomous Robots 2023\
> > [8] Rana, et al. "SayPlan: Grounding Large Language Models using 3D Scene Graphs for Scalable Robot Task Planning" CoRL 2023
> >
> > ---
> >
> > **[Q4] Assumption on LTL and formal language: oversimplification of LTL; whether there are concepts that LTL can describe which natural language cannot; necessity of employing LLMs instead of directly utilizing Task and Motion Planning (TAMP) systems. This raises fundamental questions about the specific advantages of LLMs in scenarios where structured formal planning could be more directly applicable**
> >
> > * The primary aim of our paper is to **evaluate** LLMs' capabilities in decision-making to help the robotics community to use LLMs selectively.
> > * We totally agree that the role of structured or symbolic representations in improving model performance remains a debating topic. However, **structured representations are valuable for model evaluation** as they offer several key advantages:
> >    * _Quantization_: Enable precise measurement and comparison of model outputs
> >    * _Error localization_: Pinpoint exact deviations from expected results
> >    * _Standardization_: Facilitate consistent cross-model comparisons
> >    * _Task-specific Assessment_: Allow for fine-grained evaluation criteria
> > * This difference is crucial for understanding our methodological choices. Instead of matching the practical robot learning framework of doing certain tasks, we aim for **standardized evaluation** for LLMs, which can
> >    * (a) guide **LLM researchers** for improvements to better support embodied AI;
> >    * (b) provide **robotics researchers** with selection and modularization of different LLMs and their roles.
> > * Compared to natural language, our approach of using formal language provides a rigorous method. Compared to TAMP systems, the search space of TAMP can become extremely large and using these systems require accurate operator definitions, and leveraging LLMs can help alleviate these issues by providing more flexible and adaptive planning capabilities.

---

> > ### Author Rebuttal · Authors · 2024-08-17
> >
> > ---
> > **[Q5] Prompt is mode-specific and fair comparison**
> >
> > * We are also glad for the reviewer pointing out our choice of using model-agnostic prompt design for evaluating LLMs, as it is a design choice we carefully considered and extensively discussed during our experiments. As detailed in **Appendix Section L.6 Prompt Analysis and Learned Lessons** (L2131-L2143), we follow the convention established by [1] and other works. It provides a **fair** and **scalable** evaluation setting for LLM/VLMs, aligned with the principle that large models should exhibit more **generalist** capabilities. We agree with this convention because our evaluation aims to improve LLMs for embodied agents. So performance should be assessed from the perspective of **embodied AI experts**, who may not prefer extensive and expert-level prompt engineering.
> > * Please note that we have thoroughly considered the potential effects of prompt variation on different large language models. The prompts we use result from many iterations of **empirical testing on all LLMs**, ensuring that all models can adequately understand the prompt and **no model would incur a format error rate of > 3.8%**.
> > * We provide **ablations of various aspects of prompt design** in Appendix Section L.6 Prompt Analysis and Learned Lessons (e.g. GPT-4o often does not require chain-of-thought explicitly to perform reasoning tasks effectively).
> > * Following your suggestion, we have added experiments on **tuning the model-specific prompt** for goal interpretation task on VirtualHome based on the general prompt, and test on "claude-3-opus-20240229", "gpt-4o-2024-05-13", "gemini-1.5-pro-preview-0409". We named the prompt in the following way:
> >     * 1) `base`, `general` and `general_improved` are the three versions of prompts we used in previous benchmark development. We started from `base`, and finalized on the final version of `general_improved` prompt, based on which all our results are reported in the paper.
> >     * 2) {model_name}\_{number} are the model-specific prompts we tuned for each {model_name} with TOP\_{number} performance gain.
> >     * We present **heatmap results in the attached [PDF](https://openreview.net/attachment?id=440TO7Rhvv&name=pdf)**. Our findings indicate that model-specific prompts do not lead to significant differences. For example, in the [figures](https://openreview.net/attachment?id=440TO7Rhvv&name=pdf), when Claude performs best with certain prompts, these prompts are also among the top performed prompts for other models. This suggests that prompt improvements can be generalized across large models, supporting our initial point.
> >
> > [1] Liang, Bommasani, Lee, et al. "Holistic Evaluation of Language Models" TMLR 2023
> >
> > ---
> >
> >
> > **[Q6] Only two simulators (BEHAVIOR and VirtualHome)**
> >
> > * We have a detailed explanation in **Appendix M.2 Simulator Comparison and Selection** (L1461-1508), showing a comparison with other simulators.
> > * **Evaluation Focus**: We are more interested in simulators with tasks that adequately challenge the decision making capabilities of LLMs, which requires a significant number of reasoning steps. Many simulators that only support control-level inputs, such as ALFRED, ManiSkill, RoboSuite, and Isaac Gym with much shorter horizons do not align with our goals.
> > * **Comprehensive Evaluation**: We have ensured that our evaluation interface is generic enough; however, not every simulator natively supports all of our evaluation in four ability modules, e.g. , Minecraft does not provide specific goals and tasks, which makes it less suitable for our study. Significant effort is required to modify some simulators for comprehensive evaluation. Therefore, we selected two representatives from the pool.
> >
> > ---
> >
> > **[Q7] LLM performance and human performance**
> >
> > Following your suggestions, we add human performance as below (%):
> >
> > |                  | Goal Interpretation: $F_1$ | Action Sequence: Goal SR  | Action Sequence: Exec. SR | Subgoal Decomposition: Goal SR | Subgoal Decomposition: Exec. SR | Transition Modeling: $F_1$ | Transition Modeling: Planner SR |
> > | ---------------- | ---------------------------- | ------------------------- | -------------------------- | ------------------------------- | -------------------------------- | ------------------------ | -------------------------------- |
> > | **GPT-4o**           | 37.6                         | 42.9                      | 57.1                       | 70.0                            | 90.0                             | 50.0                     | 100.0                            |
> > | **Human Annotation** | 80.6                         | 57.1                      | 85.7                       | 60.0                            | 80.0                             | 52.9                     | 66.7                             |
> >
> > Our findings indicate that Transition Modeling is particularly challenging for humans due to its requirement for complex logical reasoning. In contrast, Goal Interpretation shows the highest human performance as it requires smallest context window among tasks. It highlights that LLMs could excel in long-context based reasoning such as long-horizong logical reasoning and large-scale scene graph tracking.

---

> > > ### Comment · Reviewer_tF6x · 2024-08-26
> > >
> > > Thank you for your response and additional results. I have increased my rating.

---

> > > > ### Author Response · Authors · 2024-08-31
> > > > **Thank you Reviewer tF6x for increasing your rating**
> > > >
> > > > Thank you for reviewing our rebuttal and increasing your rating. We sincerely appreciate your time and effort.
> > > >
> > > > Thank you,
> > > > Authors

---

### Official Review · Reviewer_1xw9 · 2024-07-28
**Review Comments**

**Rating:** 9
**Confidence:** 4
**Correctness:** Yes
**Clarity:** This paper is well-written.

**Review:**

As indicated in the paper summary, this paper proposes a standardized interface for embodied agent, including the representation, the pipeline, the LLM-capable modules and evaluations that can insert into. Therefore, different LLMs can be fairly evaluated, and LLM’s performance specialized for decision making can be examined in a fine-grained and explicit way. These contributions make the benchmark comprehensive, and insightful findings can be summarized accordingly (see the introduction).

I am almost satisfied with this paper, but there are still minor concerns. For example, the evaluations in the paper examine more instruction-following abilities from the models rather than the ability to make decisions.

**Strengths:**

1. Standardization the interface for embodied decision making: Using LTL formulas to unify different goal specifications; defines four key ability modules that LLMs can be applied to (goal interpretation, subgoal decomposition, action sequencing, transition modeling). This kind of modularization enables fair comparisons and identification of LLM strengths and weaknesses across core embodied reasoning capabilities.

2. Comprehensive metrics and evaluation of models' capabilities in embodied tasks: Introduces fine-grained evaluation metrics that go beyond overall task success rate to measure specific error types (e.g. affordance errors, hallucination, wrong action order). This aids in pinpointing the limitations of current LLMs.

**Additional Feedback:**

N/A

**Documentation:**

The documentation is comprehensive.

**Ethics:**

There are no ethics concerns

**Limitations:**

The authors have addressed the limitations, and there is no potential negative societal impact of their work.

**Opportunities For Improvement:**

1. More details should be provided to make the paper easiler to follow: For example, it is unclear what's the difference between LTL Goal and subgoal trajectory. And what's the usage of state-action trajectory? Some basic knowledges should be mentioned in the main text or supplementary materials.
2. Is it fair to compare models that can accept visual inputs with purely text-based LLM models (GPT-4o vs. Mixtral)? The current evaluation seems to focus more on understanding instructions rather than making decisions. Additionally, the interface seems not include robot foundation models, such as RT-1 and RT-H. Is it possible to include these models as well?

**Relation To Prior Work:**

Yes. this paper has adequantly discussed the relations with the previous works.

**Summary And Contributions:**

This paper provided a comprehensive benchmark that assesses LLMs' capability in embodied decision making. It proposes an impressive interface that standardizes recent LLM-based embodied decision-making pipelines, including the formulations of various types of tasks and IO specifications, and summarizes 4 LLM-based modules that can be included in decision making. The evaluation metrics are much more fine-grained and purpose-oriented, than just a final success rate in common decision-making evaluations. Therefore, comprehensive assessments of LLM’s performance for different capabilities in making embodied decisions become feasible. Overall, this is a good benchmark papers that should be accepted in this track.

---

> ### Author Rebuttal · Authors · 2024-08-17
>
> We sincerely thank Reviewer 1xw9 for the insightful feedback! We are encouraged that you find **our paper well-written and easy to follow**, **our baselines appropriate**, and **our analysis detailed** while **opening up multiple areas for future research**! Please find your suggestions carefully addressed below:
>
> ---
>
> **[Q1] difference between LTL Goal and subgoal trajectory**
>
> Thank you for bringing up the question, which is a crucial aspect we carefully designed.
> - We've detailed their relationship in L121-129, with additional details in Appendix N.2, and Fig. 53 and Fig. 54 as examples. **Following your suggestions, we added more explanations in our [revision](https://embodied-agent-eval.github.io/website/data/NeurIPS2024__Embodied_Agent_Interface_Evaluating_LLMs_for_Physical_Everyday_Decision_Making.pdf)** in Sec 2.2.
> - LTL Goals:
>     - Definition: LTL Goals represent the final states, **associated with a task** (L106-115).
>     - Representation: In LTL formulas, capturing complex logic and/or temporal relationships (L116-120).
>     - Example _Bottling Fruits_:
>     ```
>     exists jar_n_01. (inside(strawberry.0, jar_n_01) and (not inside(peach.0, jar_n_01))) and exists jar_n_01. (inside(peach.0, jar_n_01) and (not inside(strawberry.0, jar_n_01))) and forall jar_n_01. (not open(jar_n_01))) and sliced(strawberry.0) and sliced(peach.0) and exists(jar_n_01, (inside(peach.0, jar_n_01) and (not inside(strawberry.0, jar_n_01))) and exists jar_n_01. (inside(peach.0, jar_n_01) and (not inside(strawberry.0, jar_n_01))) and forall jar_n_01.  (not open(jar_n_01)) and sliced(strawberry.0) and sliced(peach.0)
>     ```
> - Subgoal Trajectories:
>     - Definition: A sequence of intermediate objectives that guide the model towards achieving the LTL goal. It is an **intermediate representation proposed by LLMs**, and can serve as input to a planner to work with.
>     - Representation: A sequence of states, with no logics involved.
>     - Example _Bottling Fruits_:
>     ```
>     ontop(strawberry.0, countertop.84) and ontop(peach.0, countertop.84) then holds_rh(carving_knife.0) then sliced(strawberry.0) and sliced(peach.0) then inside(strawberry.0, jar.0) and not inside(peach.0, jar.0) then inside(peach.0, jar.1) and not inside(strawberry.0, jar.1) then not open(jar.0) and not open(jar.1)
>     ```
> - Key differences:
>     - LTL grammar formally describes the temporal ordering (L110-113).
>     - We use LTL grammar to describe final goals and subgoal trajectory, where final goals represented in LTL grammar can be abbreviated as LTL Goals (L123-126), and subgoals are a trajectory for the achievement of the LTL goal in time order (L127-129).
>     - The sum of all subgoals should lead to the LTL goal satisfaction.
>
> ---
>
> **[Q1] the usage of state-action trajectory**
>
> Thanks for the constructive suggestions. In Sec 2.2 (L121-129), we detail how state-action trajectories are used to verify LTL goal satisfaction, where we explain how LTL formula acts as a classifier, distinguishing successful trajectory that achieve the goal and those that fail.
> We further elaborate in Sec 2.4 and Sec 2.5 on how this LTL-based classification is used to calculate success rates for trajectory-based components: subgoal decomposition and action sequencing.
>
> Following your suggestion, in our [revision](https://embodied-agent-eval.github.io/website/data/NeurIPS2024__Embodied_Agent_Interface_Evaluating_LLMs_for_Physical_Everyday_Decision_Making.pdf), we enhanced Sec 2.2, Sec 2.4, and Sec 2.5 with additional walk-through examples for a clearer demonstration of our evaluation for complex, temporally-extended goals.
>
> ---
>
> **[Q2] fair comparison: models accept visual inputs vs text-based LLMs (GPT-4o vs. Mixtral)**
>
> * From an inference perspective, the comparison is designed to be fair, providing the same input to both model types, evaluating decision-making when given all available information, highlighting the core reasoning capabilities underlying the models.
>     * Decision-making abilities are key for foundation models to serve as generalist agents, regardless of training on textual or visual data.
> * From a training perspective, such comparison is meaningful to learn whether visual training data is helpful in long-horizon embodied decision making, which does not show much improvement in our results for long-horizon tasks.
> ---
>
> **[Q2] focus more on understanding instructions rather than making decisions**
>
> * We **minimize the involvement of instruction following** by using LTL based goals with a standardized JSON format (Appendix L.6, L1452-1454). Natural language task instruction is **only** used for goal interpretation. Other components focus on **decision making by operating on environment-grounded symbolic goals**, **not direct instructions**.
> * We use **Markov Decision Process (MDP)** to formulate evaluation tasks (Appendix F.2) to **distinguish from instruction following abilities**. Each task can be viewed as an approximation of the MDP probability (Appendix F.3):
>    * Action Sequencing: predict a sequence of actions given symbolic goals
>    * Subgoal Decomposition: predict a sequence of states given symbolic goals
>    * Transition Modeling: predict precondition and post-effect of the action
> * Pure instruction following is primarily evaluated in Goal Interpretation, typically JSON parsing errors (error rates below 1% in Table 6).
>
> ---
>
> **[Q2] why not include robot foundation models, such as RT-1 and RT-H**
>
> Vision-Language-Action models (VLAs) like RT-1, RT-2 and RT-H focus on short-horizon tasks in a limited single-view scene (e.g. "open pistachio jar", "move bowl away from cereal dispenser"), whereas our evaluations focus on complex, longer-horizon tasks in a large env scene (e.g. "cleaning_up_the_kitchen", which involves finding and using soap/rag to clean plates, cabinets and floor, storing oil in one cabinet and plates in another cabinet, and putting the fruits and vegetables in the fridge). So it is not applicable to evaluate VLAs.

---

> > ### Author Response · Authors · 2024-08-31
> > **Thank you Reviewer 1xw9 for the encouraging and insightful feedback**
> >
> > Dear Reviewer 1xw9,
> >
> > Thank you again for your encouraging and insightful feedback.
> >
> > We have addressed all your concerns and conducted additional sets of experiments with more details in the revision. We are happy to address any further concerns you may have. Again, we sincerely appreciate your time and effort.
> >
> > Thank you,
> > Authors

---

### Author Response · Authors · 2024-08-17
**General Response: Thank you for the valuable suggestions**

We would like to express our gratitude to the reviewers for their thorough evaluation of our work and their constructive feedback.

---

We are pleased that the reviewers recognized several key contributions of our work:
* The introduction of **a standardized interface** (Embodied Agent Interface) for evaluating LLMs in embodied decision-making tasks (tF6x, yhYD, hSVb)
* The development of **comprehensive** and **fine-grained** evaluation metrics that go beyond overall task rates and formalization of four critical ability modules (1xw9, tF6x, hSVb)
* The adaptation of **LTL** formulas to **unify** goal specifications, allowing for both **state-based** and **temporally extended goals** (1xw9, tF6x, yhYD)
* The high level of **thoroughness**, **quality**, and **reproducibility** of our work, including detailed specifications and prompts in the supplementary material (yhYD, hSVb)
* The implementation of **automatic error identification tools** (tF6x)
* **Comprehensive** evaluation over 15 LLMs (tF6x, yhYD)
* The potential for our framework to be extended to **different simulators** (yhYD)
* The **valuable insights** provided into the shortcomings of LLMs for embodied agents (yhYD, hSVb)

&nbsp;

---

We wanted to highlight our goal of providing **a SINGLE line to evaluate LLMs for embodied agents**:
* It allows researchers to **avoid simulator installation and execution debugging**, with an easy-accessible way to **evaluate without learning to use the simulator at all**.
* The input only requires LLM predictions, and automatically provides **output including trajectory execution results, error messages and categorization, fine-grained metrics and a single overall metric**.
* We have packaged the simulators into a PyPI package and a single Docker container, with details in our website, as included in the submission:
   * **PyPI packages**: https://pypi.org/project/behavior-eval/ , https://pypi.org/project/virtualhome-eval/
   * **Docker**: https://hub.docker.com/repository/docker/jameskrw/eval-embodied-agent/general
   * **Documentation**: https://embodied-agent-eval.readthedocs.io/en/latest/#
   * **Github**: https://github.com/embodied-agent-eval/embodied-agent-eval
   * **Leaderboard**: https://embodied-agent-eval.github.io/

---

We also sincerely appreciate the reviewers’ thoughtful feedback and concerns. Based on their comments, we have made a significant effort to address these issues:
1. **Justifications about the design choice of VLMs vs LLMs in long-horizon decision making** (tF6x, hSVb):
   + We scope our **long-horizon decision making evaluation** in the LLM setting as it is currently the **most applicable** setting.
   + (Exp.1) `image → VLMs → planning_output` with extreme low performance: Existing VLMs are not designed for taking scene-level information from multiple rooms in a household environment, especially the large complex scenarios like BEHAVIOR and VirtualHome.
   + (Exp.2) `image + scene_graph → VLMs → planning_output` with lower performance: VLMs' long-horizon reasoning abilities primarily stem from their LLM components. VLMs are not yet able to use multimodal input to improve long-horizon decision-making.
   + If (Exp.3) `scene_graph + broader_context → LLMs → improved_planning_output`, then (Exp.4) `image + scene_graph + broader_context → VLMs → improved_planning_output`: Methodological findings from our LLM evaluation can also be applied to enhance VLM decision making performance.
   + As a result, we choose LLMs for embodied decision making, which decomposes perception and reasoning. Our focus is on the **long-horizon decision-making** capability that bridges perception and control.
   + The impact includes (a) guiding **LLM researchers** to improve to better support embodied AI; (b) providing **robotics researchers** with selection and modularization of different LLMs and their roles.

2. **Justifications for the robustness of our evaluation** (tF6x):
   + We cited our replanning experiment in the original manuscript, and added a new replanning experiment to fix stochasticity in execution.

3. **Added human performance** (tF6x):
    + We added human performance for comparison with LLM performance.

4. **Added human evaluation for annotation quality verification** (yhYD):
    + We provided a detailed explanation of our dataset annotation process and quality verification methods.

5. **Improved writing** (1xw9, tF6x):
    + We elaborated on the differences between LTL Goals and subgoal trajectories with examples.
    + We clarified the usage of state-action trajectories.
    + We explained the feasibility of using scene graphs.
    + We explained our approach to prompt design and added an experiment to ensure fairness comparison across LLMs.

6. We added a **Empirical Finding Summary** section in the Appendix.

Once again, we are grateful for the valuable comments to improve our manuscript, and we will include updates in the next version of our work.

---

### Decision · Program_Chairs · 2024-09-26

**Decision:**

Accept (Oral)

**Comment:**

In this paper the authors propose the Embodied Agent Interface for evaluating LLMs in embodied decision making tasks. Specifically, the work goes beyond relying on metrics of overall task success rates. The joint consideration of four modules for decision making (goal interpretation, subgoal decomposition, action sequencing, and transition modeling), and the fine-grained evaluation of different types of errors are especially useful for advancing this research space in my opinion.

The reviewers have overall recognized the contribution of this work, and the authors have provided a comprehensive rebuttal to address the reviewer concerns. I consequently recommend accepting this paper into the program.

I encourage the authors to include the relevant parts of their rebuttal and the results shared (especially discussions with reviewer tF6x) into the main paper.